# HIGH-PROBABILITY CONVERGENCE FOR COMPOSITE AND DISTRIBUTED STOCHASTIC MINIMIZATION AND VARIATIONAL INEQUALITIES WITH HEAVY-TAILED NOISE

## ABSTRACT

High-probability analysis of stochastic first-order optimization methods under mild assumptions on the noise has been gaining a lot of attention in recent years. Typically, gradient clipping is one of the key algorithmic ingredients to derive good high-probability guarantees when the noise is heavy-tailed. However, if implemented naïvely, clipping can spoil the convergence of the popular methods for composite and distributed optimization (Prox-SGD/Parallel SGD) even in the absence of any noise. Due to this reason, many works on high-probability analysis consider only unconstrained non-distributed problems, and the existing results for composite/distributed problems do not include some important special cases (like strongly convex problems) and are not optimal. To address this issue, we propose new stochastic methods for composite and distributed optimization based on the clipping of stochastic gradient differences and prove tight high-probability convergence results (including nearly optimal ones) for the new methods. Using similar ideas, we also develop new methods for composite and distributed variational inequalities and analyze the high-probability convergence of these methods.

## 1 INTRODUCTION

Many recent works on stochastic optimization have the ultimate goal of bridging the theory and practice in machine learning. This is mostly reflected in the attempts at the theoretical analysis of optimization methods under weaker assumptions than the standard ones. Moreover, some phenomena cannot be explained using classical in-expectation convergence analysis (see the motivating example from (Gorbunov et al., 2020a)) that results in the growing interest in more accurate ways to the analysis of stochastic methods, for example, *high-probability convergence analysis*.

However, despite the significant attention to this topic (Nazin et al., 2019; Davis et al., 2021; Gorbunov et al., 2020a; 2022a; Cutkosky & Mehta, 2021; Sadiev et al., 2023; Nguyen et al., 2023b; Liu & Zhou, 2023; Liu et al., 2023), several important directions remain unexplored. In particular, all mentioned works either consider unconstrained problems or consider general composite/constrained minimization/variational inequality problems but have some noticeable limitations, such as bounded domain assumption, extra logarithmic factors in the complexity bounds, not optimal (not accelerated) convergence rates, or no analysis of (quasi-) strongly convex (monotone) case. The importance of composite/constrained formulations for the machine learning community can be justified in many ways. For example, composite optimization and distributed optimization have a lot of similarities, i.e., one can view a distributed optimization problem as a special composite optimization problem (Parikh & Boyd, 2014). Due to the large sizes of modern machine learning models and datasets, many important problems can be solved in a reasonable time only via distributed methods. Next, composite formulations are very useful for handling different regularizations popular in machine learning and statistics (Zou & Hastie, 2005; Shalev-Shwartz & Ben-David, 2014; Beck, 2017). Finally, variational inequalities are usually considered with constraints as well.

The discrepancy between the importance of composite/constrained formulations and the lack of high-probability convergence results in this setup can be partially explained as follows. SOTA high-probability convergence results are derived for the algorithms that use *gradient clipping* (Pascanu

et al., 2013), i.e., the clipping operator defined as $\texttt{clip}(x, \lambda) = \min\{1, \lambda/\|x\|\}x$ for $x \neq 0$ and $\texttt{clip}(0, \lambda) = 0$ with some clipping level $\lambda > 0$ is applied to the stochastic gradients. If $\lambda$ is too small, then naïve Proximal Gradient Descent with gradient clipping is not a fixed point method, i.e., the method escapes the solution even if it is initialized there (see a technical explanation in Section 2). This fact implies that one has either to increase the clipping level or to decrease the stepsize to converge to the exact solution asymptotically; the latter approach leads to a slower convergence rate. On the other hand, even in the unconstrained case, the existing results with acceleration/linear convergence are derived for the methods using decreasing clipping level (Gorbunov et al., 2020a; Sadiev et al., 2023). Therefore, new algorithms and analyses are required to handle this issue.

In this work, we close this gap by proposing new stochastic methods for composite and distributed problems via the clipping of *gradient differences* that converge to zero with high probability. This allows us to achieve the desirable acceleration and linear convergence. Before we move on to the presentation of the main contributions, we need to introduce the problem settings formally.

## 1.1 SETUP

**Notation.** The standard Euclidean norm of vector $x \in \mathbb{R}^d$ is denoted as $\|x\| = \sqrt{\langle x, x \rangle}$. $B_R(x) = \{y \in \mathbb{R}^d \mid \|y - x\| \leq R\}$ is the ball centered at $x$ with radius $R$. Bregman divergence w.r.t. function $f$ is denoted as $D_f(x, y) \stackrel{\text{def}}{=} f(x) - f(y) - \langle \nabla f(y), x - y \rangle$. In $\mathcal{O}(\cdot)$, we omit the numerical factors, and in $\widetilde{\mathcal{O}}(\cdot)$, we omit numerical and logarithmic factors. For natural $n \geq 1$ the set $\{1, 2, \ldots, n\}$ is denoted as $[n]$. Finally, we use $\mathbb{E}_\xi[\cdot]$ to denote the expectation w.r.t. the randomness coming from $\xi$.

**Considered problems.** The first class of problems we consider in this work is stochastic composite minimization problems:

$$\min_{x \in \mathbb{R}^d} \left\{ \Phi(x) = f(x) + \Psi(x) \right\}, \tag{1}$$

where $f(x) = \mathbb{E}_{\xi \sim \mathcal{D}}[f_\xi(x)]$ is a differentiable function satisfying some properties to be defined later and $\Psi(x)$ is a proper, closed, convex function (composite/regularization term). The examples of problem (1) arise in various applications, e.g., machine learning (Shalev-Shwartz & Ben-David, 2014), signal processing (Combettes & Pesquet, 2011), image processing (Luke, 2020). We also consider variational inequality problems, see Appendix C.

The distributed version of (1) has the following structure of $f$:

$$f(x) = \frac{1}{n} \sum_{i=1}^{n} \left\{ f_i(x) = \mathbb{E}_{\xi_i \sim \mathcal{D}_i}[f_{\xi_i}(x)] \right\}. \tag{2}$$

In this case, there are $n$ workers connected in a centralized way with some parameter server; worker $i$ can query some noisy information (stochastic gradients/estimates) about $f_i$.

**In-expectation and high-probability convergence.** In-expectation convergence guarantees provide the upper bounds on the number of iterations/oracle calls $\hat{K} = \hat{K}(\varepsilon)$ for a method needed to find point $x^{\hat{K}}$ such that $\mathbb{E}[\mathcal{C}(x^{\hat{K}})] \leq \varepsilon$ for given convergence criterion $\mathcal{C}(x)$ (e.g., $\mathcal{C}(x)$ can be $f(x) - f(x^*)$, $\|x - x^*\|^2$, $\|\nabla f(x)\|^2$) and given accuracy $\varepsilon > 0$. High-probability convergence guarantees give the upper bounds on the number of iterations/oracle calls $K = K(\varepsilon, \beta)$ for a method needed to find point $x$ such that $\mathbb{P}\{\mathcal{C}(x^K) \leq \varepsilon\} \geq 1 - \beta$, where $\beta \in (0, 1)$ is a confidence level. It is worth noting that Markov's inequality implies $\mathbb{P}\{\mathcal{C}(x^K) > \varepsilon\} < \mathbb{E}[\mathcal{C}(x^K)]/\varepsilon$, meaning that it is sufficient to take $K = \hat{K}(\beta\varepsilon) = \hat{K}$: $\mathbb{P}\{\mathcal{C}(x^{\hat{K}}) > \varepsilon\} < \mathbb{E}[\mathcal{C}(x^{\hat{K}})]/\varepsilon \leq \beta$. However, this typically leads to the polynomial dependence on $1/\beta$ that significantly spoils the complexity of the method when $\beta$ is small. Therefore, we focus on the high-probability convergence guarantees that depend on $1/\beta$ poly-logarithmically. Moreover, such high-probability results are more sensitive to the noise distribution (and, thus, more accurate) than in-expectation ones (Gorbunov et al., 2020a; Sadiev et al., 2023).

**Proximal operator.** We assume that function $\Psi(x)$ has a relatively simple structure such that one can efficiently compute *proximal operator*: $\text{prox}_{\gamma\Psi}(x) = \arg\min_{y \in \mathbb{R}^d}\{\gamma\Psi(y) + \frac{1}{2}\|y - x\|^2\}$. For the properties of the proximal operator and examples of functions $\Psi(x)$ such that $\text{prox}_{\gamma\Psi}(x)$ can be easily computed, we refer the reader to (Beck, 2017).

**Bounded central $\alpha$-th moment.** We consider the situation when $f_i$ and $F_i$ are accessible through the stochastic oracle calls. The stochastic estimates satisfy the following assumption.[1]

**Assumption 1.** *There exist some set $Q \subseteq \mathbb{R}^d$ and values $\sigma \geq 0$, $\alpha \in (1, 2]$ such that for all $x \in Q$ we have $\mathbb{E}_{\xi_i \sim \mathcal{D}_i}[\nabla f_{\xi_i}(x)] = \nabla f_i(x)$ and*

$$\mathbb{E}_{\xi_i \sim \mathcal{D}_i}[\|\nabla f_{\xi_i}(x) - \nabla f_i(x)\|^\alpha] \leq \sigma^\alpha. \tag{3}$$

For $\alpha = 2$, Assumption 1 reduces to the bounded variance assumption, and for $\alpha \in (1, 2)$ variance of the stochastic estimator can be unbounded, e.g., the noise can have Lévy $\alpha$-stable distribution (Zhang et al., 2020b), which is heavy-tailed.

**Assumptions on $f_i$.** We assume that functions $\{f_i\}_{i \in [n]}$ are $L$-smooth.

**Assumption 2.** *We assume that there exist some set $Q \subseteq \mathbb{R}^d$ and constant $L > 0$ such that for all $x, y \in Q, i \in [n]$ and for all $x^* \in \arg\min_{x \in \mathbb{R}^d} \Phi(x)$*

$$\|\nabla f_i(x) - \nabla f_i(y)\| \leq L\|x - y\|, \tag{4}$$
$$\|\nabla f_i(x) - \nabla f_i(x^*)\|^2 \leq 2L\left(f_i(x) - f_i(x^*) - \langle \nabla f_i(x^*), x - x^* \rangle\right). \tag{5}$$

As noted in Appendix B from (Sadiev et al., 2023), (5) is satisfied on the set $Q \neq \mathbb{R}^d$ if (4) holds on a slightly larger set in the case of $\Psi \equiv 0$, $n = 1$ (unconstrained single-node case). For simplicity, we assume that both (4) and (5) hold on $Q$. This is always the case for $L$-smooth functions on $Q = \mathbb{R}^d$ when $\Psi \equiv 0$, $n = 1$. In a more general situation, condition (5) can be viewed as an assumption on the structured non-convexity of $\{f_i\}_{i \in [n]}$. Finally, if $\{f_i\}_{i \in [n]}$ are convex and $L$-smooth on the whole domain of the problem (1), then Assumption 2 holds.

Next, for each particular result about the convergence of methods for (1), we make one of the following assumptions.

**Assumption 3.** *There exist some set $Q \subseteq \mathbb{R}^d$ and constant $\mu \geq 0$ such that $f$ is $\mu$-strongly convex:*

$$f(y) \geq f(x) + \langle \nabla f(x), y - x \rangle + \frac{\mu}{2}\|y - x\|^2 \quad \forall x, y \in Q. \tag{6}$$

*When $\mu = 0$, function $f$ is called convex on $Q$.*

This is a standard assumption for optimization literature (Nesterov et al., 2018). We also consider a relaxation of strong convexity.

**Assumption 4.** *There exist some set $Q \subseteq \mathbb{R}^d$ and constant $\mu \geq 0$ such that $f_1, \ldots, f_n$ are $(\mu, x^*)$-quasi-strongly convex for all $x^* \in \arg\min_{x \in \mathbb{R}^d} \Phi(x)$:*

$$f_i(x^*) \geq f_i(x) + \langle \nabla f_i(x), x^* - x \rangle + \frac{\mu}{2}\|x - x^*\|^2 \quad \forall x \in Q, \, i \in [n]. \tag{7}$$

Condition (7) is weaker than (6) and holds even for some non-convex functions (Necoara et al., 2019).

## 1.2 OUR CONTRIBUTIONS

• **Methods with clipping of gradient differences for distributed composite minimization.** We develop two stochastic methods for composite minimization problems – Proximal Clipped SGD with shifts (Prox-clipped-SGD-shift) and Proximal Clipped Similar Triangles Method with shifts (Prox-clipped-SSTM-shift). Instead of clipping stochastic gradients, these methods clip the difference between the stochastic gradients and the shifts that are updated on the fly. This trick allows us to use decreasing clipping levels, and, as a result, we derive the first accelerated high-probability convergence rates and tight high-probability convergence rates for the non-accelerated method in the

---

[1]Following (Sadiev et al., 2023), we consider all assumptions only on some bounded set $Q \subseteq \mathbb{R}^d$; the diameter of $Q$ depends on the starting point. We emphasize that we do not assume boundedness of the domain of the original problem. Instead, we prove via induction that the iterates of the considered methods stay in some ball around the solution with high probability (see the details in Section 3). Thus, it is sufficient for us to assume everything just on this ball, though our analysis remains unchanged if we introduce all assumptions on the whole domain.

**Table 1:** Summary of known and new high-probability complexity results for solving (non-) composite (non-) distributed smooth optimization problem (1). Column "Setup" indicates the assumptions made in addition to Assumptions 1 and 2. All assumptions are made only on some ball around the solution with radius $\sim R \geq \|x^0 - x^*\|$. Complexity is the number of stochastic oracle calls (per worker) needed for a method to guarantee that $\mathbb{P}\{\text{Metric} \leq \varepsilon\} \geq 1 - \beta$ for some $\varepsilon > 0$, $\beta \in (0, 1]$ and "Metric" is taken from the corresponding column. Numerical and logarithmic factors are omitted for simplicity. Column "C?" shows whether the problem (1) is composite, "D?" indicates whether the problem (1) is distributed. Notation: $L$ = Lipschitz constant; $\sigma$ = parameter from Assumption 1; $R$ = any upper bound on $\|x^0 - x^*\|$; $\zeta_* = \sqrt{\frac{1}{n} \sum_{i=1}^n \|\nabla f_i(x^*)\|^2}$; $\widehat{R}^2 = R \left(3R + L^{-1}(2\eta\sigma + \|\nabla f(x^0)\|)\right)$ for some $\eta > 0$ (for the result from (Nguyen et al., 2023a); one can show that $\widehat{R}^2 = \Theta(R^2 + R\zeta_*/L)$ when $n = 1$, see the discussion after Theorem 2.3); $\mu$ = (quasi-)strong convexity parameter. The results of this paper are highlighted in blue.

| Setup | Method | Metric | Complexity | C? | D? |
|---|---|---|---|---|---|
| As. 3 $(\mu = 0)$ | clipped-SGD (Sadiev et al., 2023) | $f(\overline{x}^K) - f(x^*)$ | $\max\left\{\frac{LR^2}{\varepsilon}, \left(\frac{\sigma R}{\varepsilon}\right)^{\frac{\alpha}{\alpha-1}}\right\}$ | ✗ | ✗ |
| | clipped-SSTM (Sadiev et al., 2023) | $f(y^K) - f(x^*)$ | $\max\left\{\sqrt{\frac{LR^2}{\varepsilon}}, \left(\frac{\sigma R}{\varepsilon}\right)^{\frac{\alpha}{\alpha-1}}\right\}$ | ✗ | ✗ |
| | Clipped-SMD [1],[2] (Nguyen et al., 2023a) | $\Phi(\overline{x}^K) - \Phi(x^*)$ | $\max\left\{\frac{L\widehat{R}^2}{\varepsilon}, \left(\frac{\sigma R}{\varepsilon}\right)^{\frac{\alpha}{\alpha-1}}\right\}$ | ✓ | ✗ |
| | Clipped-ASMD [1] (Nguyen et al., 2023a) | $\Phi(y^K) - \Phi(x^*)$ | $\max\left\{\sqrt{\frac{LR^2}{\varepsilon}}, \left(\frac{\sigma R}{\varepsilon}\right)^{\frac{\alpha}{\alpha-1}}\right\}$ | ✓✗[3] | ✗ |
| | DProx-clipped-SGD-shift Theorem 2.3 | $\Phi(\overline{x}^K) - \Phi(x^*)$ | $\max\left\{\frac{LR^2}{\varepsilon}, \frac{R\zeta_*}{\sqrt{n}\varepsilon}, \frac{1}{n}\left(\frac{\sigma R}{\varepsilon}\right)^{\frac{\alpha}{\alpha-1}}\right\}$ | ✓ | ✓ |
| | DProx-clipped-SSTM-shift Theorem 2.4 | $\Phi(y^K) - \Phi(x^*)$ | $\max\left\{\sqrt{\frac{LR^2}{\varepsilon}}, \sqrt{\frac{R\zeta_*}{\sqrt{n}\varepsilon}}, \frac{1}{n}\left(\frac{\sigma R}{\varepsilon}\right)^{\frac{\alpha}{\alpha-1}}\right\}$ | ✓ | ✓ |
| As. 4 $(\mu > 0)$ | clipped-SGD (Sadiev et al., 2023) | $\|x^K - x^*\|^2$ | $\max\left\{\frac{L}{\mu}, \left(\frac{\sigma^2}{\mu^2\varepsilon}\right)^{\frac{\alpha}{2(\alpha-1)}}\right\}$ | ✗ | ✗ |
| | DProx-clipped-SGD-shift Theorem 2.2 | $\|x^K - x^*\|^2$ | $\max\left\{\frac{L}{\mu}, \frac{1}{n}\left(\frac{\sigma^2}{\mu^2\varepsilon}\right)^{\frac{\alpha}{2(\alpha-1)}}\right\}$ | ✓ | ✓ |

[1] All assumptions are made on the whole domain.
[2] The authors additionally assume that for a chosen point $\widehat{x}$ from the domain and for $\eta > 0$ one can compute an estimate $\widehat{g}$ such that $\mathbb{P}\{\|\widehat{g} - \nabla f(\widehat{x})\| > \eta\sigma\} \leq \epsilon$. Such an estimate can be found using geometric median computed over $\mathcal{O}(\ln \epsilon^{-1})$ samples (Minsker, 2015).
[3] The authors assume that $\nabla f(x^*) = 0$, which is not true for general composite optimization.

quasi-strongly convex case. We also generalize the proposed methods to the distributed case (DProx-clipped-SGD-shift and DProx-clipped-SSTM-shift) and prove that they benefit from parallelization. To the best of our knowledge, our results are the first showing linear speed-up under Assumption 1.

• **Methods with clipping of gradient differences for distributed composite VIPs.** We also apply the proposed trick to the methods for variational inequalities. In particular, we propose DProx-clipped-SGDA-shifts and DProx-clipped-SEG-shifts and rigorously analyze their high-probability convergence. As in the minimization case, the proposed methods have provable benefits from parallelization.

• **Tight convergence rates.** As a separate contribution, we highlight the tightness of our analysis: in the known special cases ($\Psi \equiv 0$ and/or $n = 1$), the derived complexity bounds either recover or outperform previously known ones (see Table 1 and also Table 2 in the appendix). Moreover, in certain regimes, the results have optimal (up to logarithms) dependencies on $\varepsilon$. This is achieved under quite general assumptions.

## 1.3 CLOSELY RELATED WORK

We discuss closely related work here and defer additional discussion to Appendix A.

**High-probability bounds for unconstrained convex problems.** Standard high-probability convergence results are obtained under the so-called light-tails assumption (sub-Gaussian noise) (Nemirovski et al., 2009; Juditsky et al., 2011; Ghadimi & Lan, 2012). The first work addressing this limitation is (Nazin et al., 2019), where the authors derive the first high-probability complexity bounds for the case of minimization on a bounded set under bounded variance assumption. In the unconstrained case, these results are extended and accelerated by Gorbunov et al. (2020a) for smooth convex and strongly convex minimization problems. Gorbunov et al. (2021) tightens them and generalizes to the case of problems with Hölder-continuous gradients and Gorbunov et al. (2022a) derives high-probability convergence rates in the case of VIPs. Sadiev et al. (2023) relaxes the assumption of bounded variance to Assumption 1 for all problem classes mentioned above, and the results under the same assumption are also derived for clipped-SGD (without acceleration) by Nguyen et al. (2023b) in the convex and non-convex cases.

**High-probability bounds for composite convex problems.** Nazin et al. (2019) propose a truncated version of Mirror Descent for convex and strongly convex composite problems and prove non-accelerated rates of convergence under bounded variance and *bounded domain* assumptions. Accelerated results under bounded variance assumption for strongly convex composite problems are proven by Davis et al. (2021), who propose an approach based on robust distance estimation. Since this approach requires solving some auxiliary problem at each iteration of the method, the complexity bound from Davis et al. (2021) contains extra logarithmic factors independent of the confidence level. Finally, in their very recent work, Nguyen et al. (2023a) prove high-probability convergence for Clipped Stochastic Mirror Descent (Clipped-SMD) for *convex* composite problems. Moreover, the authors also propose Accelerated Clipped-SMD (Clipped-ASMD) and show that the algorithm is indeed accelerated *but only under the additional assumption that $\nabla f(x^*) = 0$.*

## 2 MAIN RESULTS FOR COMPOSITE DISTRIBUTED MINIMIZATION PROBLEMS

In this section, we consider problem (1) and methods for it.

**Failure of the naïve approach.** For simplicity, consider a non-stochastic case with strongly convex $f(x)$, $n = 1$. The standard deterministic first-order method for solving problems like (1) is Proximal Gradient Descent (Prox-GD) (Combettes & Pesquet, 2011; Nesterov, 2013): $x^{k+1} = \text{prox}_{\gamma\Psi}(x^k - \gamma\nabla f(x^k))$. Due to the good interplay between the structure of the problem, properties of the proximal operator, and the structure of the method, Prox-GD has the same (linear) convergence rate as GD for minimization of $f(x)$. One of the key reasons for that is that any solution $x^*$ of problem (1) satisfies $x^* = \text{prox}_{\gamma\Psi}(x^* - \gamma\nabla f(x^*))$, i.e., the solutions of (1) are fixed points of Prox-GD (and vice versa), which is equivalent to $-\nabla f(x^*) \in \partial\Psi(x^*)$, where $\partial\Psi(x^*)$ is a subdifferential of $\Psi$ at $x^*$. However, if we apply gradient clipping to Prox-GD naïvely

$$x^{k+1} = \text{prox}_{\gamma\Psi}\left(x^k - \gamma\texttt{clip}(\nabla f(x^k), \lambda)\right), \tag{8}$$

then the method loses a fixed point property if $\|\nabla f(x^*)\| > \lambda$, because in this case, $-\texttt{clip}(\nabla f(x^*), \lambda)$ does not necessarily belongs to $\partial\Psi(x^*)$ and $x^* \neq \text{prox}_{\gamma\Psi}(x^* - \gamma\texttt{clip}(\nabla f(x^*), \lambda))$ in general. Therefore, for such $\lambda$, one has to decrease the stepsize $\gamma$ to achieve any accuracy of the solution. This approach slows down the convergence making it sublinear even without any stochasticity in the gradients. To avoid this issue, it is necessary to set $\lambda$ large enough. This strategy works in the deterministic case but becomes problematic for a stochastic version of the method from (8):

$$x^{k+1} = \text{prox}_{\gamma\Psi}\left(x^k - \gamma\texttt{clip}(\nabla f_{\xi^k}(x^k), \lambda_k)\right), \tag{9}$$

where $\xi^k$ is sampled independently from previous iterations. The problem comes from the fact the existing analysis in the unconstrained case (which is a special case of the composite case) requires taking decreasing $\lambda_k$ (Gorbunov et al., 2021; Sadiev et al., 2023) that contradicts the requirement that clipping level has to be large enough. Therefore, more fundamental algorithmic changes are needed.

**Non-implementable solution.** Let us reformulate the issue: (i) to handle the heavy-tailed noise, we want to use decreasing clipping level $\lambda_k$, (ii) but the method should also converge linearly without the noise, i.e., when $\nabla f_{\xi^k}(x^k) = \mathbb{E}_{\xi^k}[\nabla f_{\xi^k}(x^k)] = \nabla f(x^k)$. In other words, *the expectation of the vector that is clipped in the method should converge to zero with the same rate as $\lambda_k$.* The method should converge, i.e., with high probability, we should have $\nabla f(x^k) \to \nabla f(x^*)$. These observations lead us to the following purely theoretical algorithm that we call Prox-clipped-SGD-star[2]:

$$x^{k+1} = \text{prox}_{\gamma\Psi}\left(x^k - \gamma\widetilde{g}^k\right), \quad \text{where } \widetilde{g}^k = \nabla f(x^*) + \texttt{clip}\left(\nabla f_{\xi^k}(x^k) - \nabla f(x^*), \lambda_k\right). \tag{10}$$

The method is non-implementable since $\nabla f(x^*)$ is unknown in advance. Nevertheless, as we explain in the next subsection, the method is useful in designing and analyzing implementable versions. The following theorem gives the complexity of Prox-clipped-SGD-star.

---

[2]The idea behind and the name of this method is inspired by SGD-star proposed by Gorbunov et al. (2020b); Hanzely & Richtárik (2019).

**Theorem 2.1.** *Let* $n = 1$ *and Assumptions [1], [2], and [4] with* $\mu > 0$ *hold for* $Q = B_{2R}(x^*)$, $R \geq \|x^0 - x^*\|$, *for some[3]* $x^* \in \arg\min_{x \in \mathbb{R}^d} \Phi(x)$. *Assume that* $K \geq 1$, $\beta \in (0, 1)$, $A = \ln \frac{4(K+1)}{\beta}$,

$$0 < \gamma = \mathcal{O}\left(\min\left\{\frac{1}{LA}, \frac{\ln(B_K)}{\mu(K+1)}\right\}\right), \quad B_K = \Theta\left(\max\left\{2, \frac{(K+1)^{2(\alpha-1)/\alpha}\mu^2 R^2}{\sigma^2 A^{2(\alpha-1)/\alpha}\ln^2(B_K)}\right\}\right),$$

$$\lambda_k = \Theta\left(\frac{\exp(-\gamma\mu(1+k/2))R}{\gamma A}\right).$$

*Then to guarantee* $\|x^K - x^*\|^2 \leq \varepsilon$ *with probability* $\geq 1 - \beta$ Prox-clipped-SGD-star *requires*

$$\widetilde{\mathcal{O}}\left(\max\left\{\frac{L}{\mu}, \left(\frac{\sigma^2}{\mu^2\varepsilon}\right)^{\frac{\alpha}{2(\alpha-1)}}\right\}\right) \quad \text{iterations/oracle calls.} \tag{11}$$

*Sketch of the proof.* Following Gorbunov et al. (2020a); Sadiev et al. (2023), we prove by induction[4] that $\|x^k - x^*\|^2 \leq 2\exp(-\gamma\mu k)R^2$ with high probability. This and $L$-smoothness imply that $\|\nabla f(x^k) - \nabla f(x^*)\| \sim \exp(-\gamma\mu k/2)$ and $\|\nabla f(x^k) - \nabla f(x^*)\| \leq \lambda_k/2$ with high probability. These facts allow us to properly clip the heavy-tailed noise without sacrificing the convergence rate. See the complete formulation of Theorem 2.1 and the full proof in Appendix D. $\qquad\square$

The above complexity bound for Prox-clipped-SGD-star coincides with the known one for clipped-SGD for the unconstrained problems under the same assumptions (Sadiev et al., 2023) – similarly as the complexity of Prox-GD coincides with the complexity of GD for unconstrained smooth problems.

**Prox-clipped-SGD-shift.** As mentioned before, the key limitation of Prox-clipped-SGD-star is that it explicitly uses shift $\nabla f(x^*)$, which is not known in advance. Therefore, guided by the literature on variance reduction and communication compression (Gorbunov et al., 2020b; Gower et al., 2020; Mishchenko et al., 2019), it is natural to approximate $\nabla f(x^*)$ via shifts $h^k$. This leads us to a new method called Prox-clipped-SGD-shift: as before $x^{k+1} = \text{prox}_{\gamma\Psi}\left(x^k - \gamma\widetilde{g}^k\right)$ but now

$$\widetilde{g}^k = h^k + \hat{\Delta}^k, \quad h^{k+1} = h^k + \nu\hat{\Delta}^k, \quad \hat{\Delta}^k = \texttt{clip}\left(\nabla f_{\xi^k}(x^k) - h^k, \lambda_k\right), \tag{12}$$

where $\nu > 0$ is a stepsize for learning shifts. Similar shifts are proposed by Mishchenko et al. (2019) in the context of distributed optimization with communication compression. Since Prox-clipped-SGD-shift is a special case of its distributed variant, we continue our discussion with the distributed version of the method.

**Distributed Prox-clipped-SGD-shift.** We propose a generalization of Prox-clipped-SGD-shift to the distributed case (2) called Distributed Prox-clipped-SGD-shift (DProx-clipped-SGD-shift):

$$x^{k+1} = \text{prox}_{\gamma\Psi}\left(x^k - \gamma\widetilde{g}^k\right), \quad \text{where} \quad \widetilde{g}^k = \frac{1}{n}\sum_{i=1}^{n}\widetilde{g}_i^k, \quad \widetilde{g}_i^k = h_i^k + \hat{\Delta}_i^k, \tag{13}$$

$$h_i^{k+1} = h_i^k + \nu\hat{\Delta}_i^k, \quad \hat{\Delta}_i^k = \texttt{clip}\left(\nabla f_{\xi_i^k}(x^k) - h_i^k, \lambda_k\right), \tag{14}$$

where $\xi_1^k, \ldots, \xi_n^k$ are sampled independently from each other and previous steps. In this method, worker $i$ updates the shift $h_i^k$ and sends clipped vector $\hat{\Delta}_i^k$ to the server. Since $\widetilde{g}^k = h^k + \frac{1}{n}\sum_{i=1}^{n}\hat{\Delta}_i^k$ and $h^{k+1} = h^k + \frac{\nu}{n}\sum_{i=1}^{n}\hat{\Delta}_i^k$, where $h^k = \frac{1}{n}\sum_{i=1}^{n}h_i^k$, workers do not need to send $h_i^k$ to the server for $k > 0$. We notice that even when $\Psi \equiv 0$, i.e., the problem is unconstrained, individual gradients $\{\nabla f_i(x^*)\}_{i \in [n]}$ of the clients' function at the solution of problem (1) are not necessary zero, though their sum equals to zero. However, if applied without any shifts to the local (stochastic) gradients, then, similarly to the case of non-distributed Prox-GD (8), the clipping operation also breaks the fixed point property, since $\frac{1}{n}\sum_{i=1}^{n}\texttt{clip}(\nabla f_i(x^*), \lambda) \neq 0$ for small values of $\lambda$. This highlights the importance of the shifts for distributed unconstrained case.

For the proposed method, we derive the following result.

---

[3]If all of our results, one can use any solution $x^*$, e.g., one can take $x^*$ being a projection of $x^*$ on the solution set.

[4]We use the induction to apply Bernstein's inequality for the estimation of the sums appearing due to the stochasticity of the gradients. We refer to Section 3 for the details.

**Theorem 2.2** (Convergence of DProx-clipped-SGD-shift: quasi-strongly convex case). *Let $K \geq 1$, $\beta \in (0, 1)$, $A = \ln \frac{48n(K+1)}{\beta}$. Let Assumptions 1, 2, and 4 with $\mu > 0$ hold for $Q = B_{3n\sqrt{2}R}(x^*)$, where $R \geq \|x^0 - x^*\|^2$. Assume that $\zeta_* = \sqrt{\frac{1}{n} \sum_{i=1}^{n} \|\nabla f_i(x^*)\|^2}$,*

$$\nu = \Theta\left(\frac{1}{A}\right), \quad 0 < \gamma = \mathcal{O}\left(\min\left\{\frac{1}{LA}, \frac{\sqrt{n}R}{A\zeta_*}, \frac{\ln(B_K)}{\mu(K+1)}\right\}\right),$$

$$B_K = \Theta\left(\max\left\{2, \frac{(K+1)^{2(\alpha-1)/\alpha}\mu^2 n^{2(\alpha-1)/\alpha}R^2}{\sigma^2 A^{2(\alpha-1)/\alpha}\ln^2(B_K)}\right\}\right), \quad \lambda_k = \Theta\left(\frac{n\exp(-\gamma\mu(1 + k/2))R}{\gamma A}\right).$$

*Then to guarantee $\|x^K - x^*\|^2 \leq \varepsilon$ with probability $\geq 1 - \beta$ DProx-clipped-SGD-shift requires*

$$\widetilde{\mathcal{O}}\left(\max\left\{\frac{L}{\mu}, \frac{\zeta_*}{\sqrt{n}\mu R}, \frac{1}{n}\left(\frac{\sigma^2}{\mu^2\varepsilon}\right)^{\frac{\alpha}{2(\alpha-1)}}\right\}\right) \quad \textit{iterations/oracle calls per worker.} \quad (15)$$

*Sketch of the proof.* The proof follows similar steps to the proof of Theorem 2.1 up the change of the Lyapunov function: by induction, we prove that $V_k \leq 2\exp(-\gamma\mu k)V$ with high probability, where $V_k = \|x^k - x^*\|^2 + \frac{C^2\gamma^2 A^2}{n}\sum_{i=1}^{n}\|h_i^k - \nabla f_i(x^*)\|^2$. The choice of the Lyapunov function reflects the importance of the "quality" of shifts $\{h_i^k\}_{i\in[n]}$, i.e., their proximity to $\{\nabla f_i(x^*)\}_{i\in[n]}$. Moreover, we increase the clipping level $n$ times to balance the bias and variance of $\widetilde{g}^k$; see Appendix B. This allows us to reduce the last term in the complexity bound $n$ times. See the complete formulation of Theorem 2.2 and the full proof in Appendix E. $\quad\square$

The next theorem gives the convergence result in the convex case.

**Theorem 2.3** (Convergence of DProx-clipped-SGD-shift: convex case). *Let $K \geq 1$, $\beta \in (0, 1)$, $A = \ln \frac{48n(K+1)}{\beta}$. Let Assumptions 1, 2, and 3 with $\mu = 0$ hold for $Q = B_{\sqrt{2}R}(x^*)$, where $R \geq \|x^0 - x^*\|$. Assume that $\nu = 0$, $\zeta_* = \sqrt{\frac{1}{n}\sum_{i=1}^{n}\|\nabla f_i(x^*)\|^2}$,*

$$0 < \gamma = \mathcal{O}\left(\min\left\{\frac{1}{LA}, \frac{\sqrt{n}R}{A\zeta_*}, \frac{n^{(\alpha-1)/\alpha}R}{\sigma K^{1/\alpha}A^{(\alpha-1)/\alpha}}\right\}\right), \quad \lambda_k = \lambda = \Theta\left(\frac{nR}{\gamma A}\right).$$

*Then to guarantee $\Phi(\bar{x}^K) - \Phi(x^*) \leq \varepsilon$ for $\bar{x}^K = \frac{1}{K+1}\sum_{k=0}^{K}x^k$ with probability $\geq 1 - \beta$ DProx-clipped-SGD-shift requires*

$$\widetilde{\mathcal{O}}\left(\max\left\{\frac{LR^2}{\varepsilon}, \frac{R\zeta_*}{\sqrt{n}\varepsilon}, \frac{1}{n}\left(\frac{\sigma R}{\varepsilon}\right)^{\frac{\alpha}{\alpha-1}}\right\}\right) \quad \textit{iterations/oracle calls per worker.} \quad (16)$$

**Discussion of the results for DProx-clipped-SGD-shift.** Up to the difference between $V$ and $\|x^0 - x^*\|^2$, in the single-node case, the derived results coincide with ones known for clipped-SGD in the unconstrained case (Sadiev et al., 2023). In the composite non-distributed case ($n = 1$), the result of Theorem 2.2 is the first known of its type, and Theorem 2.3 recovers (up to logarithmic factors) the result from (Nguyen et al., 2023a) for a version of Stochastic Mirror Descent with gradient clipping (Clipped-SMD), see Table 1. Indeed, parameter $\widehat{R}^2 = R(3R + L^{-1}(2\eta\sigma + \|\nabla f(x^0)\|))$ for some $\eta > 0$ from the result by Nguyen et al. (2023a) equals $\Theta(\Theta(R^2 + R\zeta_*/L))$, when $\eta$ is sufficiently small (otherwise $\widehat{R}$ can be worse than $\Theta(R^2 + R\zeta_*/L)$), which can be seen from the following inequalities following smoothness: $\|\nabla f(x^0)\| \leq \|\nabla f(x^*)\| + \|\nabla f(x^0) - \nabla f(x^*)\| \leq \|\nabla f(x^*)\| + L\|x^0 - x^*\|$ and $\|\nabla f(x^*)\| \leq \|\nabla f(x^0)\| + \|\nabla f(x^0) - \nabla f(x^*)\| \leq \|\nabla f(x^0)\| + L\|x^0 - x^*\|$. Since in this work we do not focus on the logarithmic factors, we do not show them in the main text and provide the complete expressions in the appendix. Nguyen et al. (2023a) has better dependencies on the parameters under logarithms than our results. We conjecture that adjusting the proof technique from (Nguyen et al., 2023a) one can improve the logarithmic factors in our results as well.

It is worth mentioning that shifts are not needed in the convex case because the method does not have fast enough convergence, which makes it work with a constant clipping level, i.e., the method in the convex case requires less tight gradient estimates and is more robust to the bias than in strongly

convex. In the quasi-strongly convex case, the shifts' stepsize is chosen as $\nu \sim \Theta(1/A)$ and it does not explicitly affect the rate since $\gamma\mu = \Theta(1/A)$, see the details in Section 3 and Appendix E.

Next, as expected for a distributed method, the terms in the complexity bounds related to the noise improve with the growth of $n$. More precisely, the terms depending on the noise level $\sigma$ are proportional to $1/n$, i.e., our results show so-called linear speed-up in the complexity – a desirable feature for a stochastic distributed method. This aspect highlights the benefits of parallelization. To the best of our knowledge, the results for the distributed methods proposed in our work are the only existing ones under Assumption 1 (even if we take into account the in-expectation convergence results). In the special case of $\alpha = 2$, our results match (up to logarithmic factors) the SOTA ones from (Gorbunov et al., 2021) since parallelization with linear speed-up follows for free under the bounded variance assumption, if the clipping is applied after averaging as it should be in the parallelized version of methods from (Gorbunov et al., 2021) to keep the analysis from (Gorbunov et al., 2021) unchanged. Indeed, when $\{\nabla f_{\xi_i}(x)\}_{i\in[n]}$ are independent stochastic gradients satisfying Assumption 1 with parameters $\sigma > 0$ and $\alpha = 2$, then $\frac{1}{n}\sum_{i\in[n]}\nabla f_{\xi_i}(x)$ also satisfies Assumption 1 with parameters $\sigma/\sqrt{n}$ and $\alpha = 2$. However, when $\alpha < 2$ achieving linear speed-up is not that straightforward. If $\{\nabla f_{\xi_i}(x)\}_{i\in[n]}$ are independent stochastic gradients satisfying Assumption 1 with parameters $\sigma > 0$ and $\alpha < 2$, then the existing results (Wang et al., 2021, Lemma 7) give a weaker guarantee: $\frac{1}{n}\sum_{i\in[n]}\nabla f_{\xi_i}(x)$ satisfies Assumption 1 with parameters $\frac{2^{2-\alpha}d^{\frac{1}{\alpha}-\frac{1}{2}}\sigma}{n^{\frac{\alpha-1}{\alpha}}}$, which is *dimension dependent*, and the same $\alpha$. Therefore, if one applies this result to the known ones from (Sadiev et al., 2023; Nguyen et al., 2023a), then the resulting complexity will have an extra factor of $d^{\frac{1}{\alpha-1}-\frac{\alpha}{2(\alpha-1)}}$ in the term that depends on $\sigma$. For large-scale or even medium-scale heavy-tailed problems, this factor can be huge, e.g., when $d = 1000$ and $\alpha = \frac{7}{6}$, this factor is $1000^{6-\frac{7}{3}} > 1000^3 = 10^9$.

To avoid these issues, we apply gradient clipping on the workers and then average clipped vectors, not vice versa. This is also partially motivated by the popularity of gradient clipping for ensuring differential privacy guarantees (Abadi et al., 2016; Chen et al., 2020) in Federated Learning (Konečný et al., 2016; Kairouz et al., 2021). Therefore, the proposed distributed methods can be useful for differential privacy as well, though we do not study this aspect in our work.

**Acceleration.** Next, we propose a distributed version of clipped Stochastic Similar Triangles Method (Gorbunov et al., 2020a; Gasnikov & Nesterov, 2016) for composite problems (DProx-clipped-SSTM-shift): $x^0 = y^0 = z^0$, $A_0 = \alpha_0 = 0$, $\alpha_{k+1} = \frac{k+2}{2aL}$, $A_{k+1} = A_k + \alpha_{k+1}$ and

$$x^{k+1} = \frac{A_k y^k + \alpha_{k+1} z^k}{A_{k+1}}, \quad z^{k+1} = \mathrm{prox}_{\alpha_{k+1}\Psi}\left(z^k - \alpha_{k+1}\widetilde{g}(x^{k+1})\right), \tag{17}$$

$$\widetilde{g}(x^{k+1}) = \frac{1}{n}\sum_{i=1}^{n}\widetilde{g}_i(x^{k+1}), \quad \widetilde{g}_i(x^{k+1}) = h_i^k + \hat{\Delta}_i^k, \tag{18}$$

$$h_i^{k+1} = h_i^k + \nu_k\hat{\Delta}_i^k, \quad \hat{\Delta}_i^k = \mathtt{clip}\left(\nabla f_{\xi_i^k}(x^{k+1}) - h_i^k, \lambda_k\right), \tag{19}$$

$$y^{k+1} = \frac{A_k y^k + \alpha_{k+1} z^{k+1}}{A_{k+1}} \tag{20}$$

where $\xi_1^k, \ldots, \xi_n^k$ are sampled independently from each other and previous steps. For the proposed method, we derive the following result.

**Theorem 2.4** (Convergence of DProx-clipped-SSTM-shift). *Let Assumptions 1, 2, and 3 with* $\mu = 0$ *hold for* $Q = B_{5\sqrt{2}nR}(x^*)$, *where* $R \geq \|x^0 - x^*\|^2$. *Let* $\zeta_* = \sqrt{\frac{1}{n}\sum_{i=1}^{n}\|\nabla f_i(x^*)\|^2}$, $C = \Theta(A/\sqrt{n})$, $K_0 = \Theta(A^2)$, *where* $K \geq 1$, $\beta \in (0, 1)$, $A = \ln\frac{10nK}{\beta}$. *Assume that*

$$\nu_k = \begin{cases} \frac{2k+5}{(k+3)^2}, & \text{if } k > K_0, \\ \frac{(k+2)^2}{C^2(K_0+2)^2 n}, & \text{if } k \leq K_0, \end{cases} \quad a = \Theta\left(\max\left\{2, \frac{A^4}{n}, \frac{A^3\zeta_*}{L\sqrt{n}R}, \frac{\sigma K^{(\alpha+1)/\alpha}A^{(\alpha-1)/\alpha}}{LRn^{\alpha-1/\alpha}}\right\}\right),$$

$$\lambda_k = \Theta\left(\frac{nR}{\alpha_{k+1}A}\right).$$

*Then to guarantee $\Phi(y^K) - \Phi(x^*) \leq \varepsilon$ with probability $\geq 1 - \beta$* DProx-clipped-SSTM-shift *requires*

$$\widetilde{\mathcal{O}}\left(\max\left\{\sqrt{\frac{LR^2}{\varepsilon}}, \sqrt{\frac{R\zeta_*}{\sqrt{n}\varepsilon}}, \frac{1}{n}\left(\frac{\sigma R}{\varepsilon}\right)^{\frac{\alpha}{\alpha-1}}\right\}\right) \qquad \text{iterations/oracle calls per worker.} \qquad (21)$$

*Sketch of the proof.* The proof of this result resembles the proof for clipped-SSTM from (Sadiev et al., 2023) but has some noticeable differences. In addition to handling the extra technical challenges appearing due to the composite structure (e.g., one cannot apply some useful formulas like $z^k - z^{k+1} = \alpha_{k+1}\widetilde{g}(x^{k+1})$ that hold in the unconstrained case), we use a non-standard potential function $M_k$ defined as $M_k = \|z^k - x^*\|^2 + (C^2\alpha_{K_0+1}^2/n)\sum_{i=1}^n \|h_i^k - \nabla f_i(x^*)\|^2$ for $k \leq K_0$ and $M_k = \|z^k - x^*\|^2 + \frac{C^2\alpha_{k+1}^2}{n}\sum_{i=1}^n \|h_i^k - \nabla f_i(x^*)\|^2$ for $k > K_0$. We elaborate on this and provide the complete proof in Appendix F. $\qquad\square$

When $n = 1$, the derived result has optimal dependence on $\varepsilon$ (up to logarithmic factors) (Nemirovskij & Yudin, 1983; Zhang et al., 2020b). In contrast to the result from (Nguyen et al., 2023a), we do not assume that $\nabla f(x^*) = 0$. Moreover, as DProx-clipped-SGD-shift, DProx-clipped-SSTM-shift benefits from parallelization since the second term in (21) is proportional $1/n$. When $n$ is sufficiently large, the effect of acceleration can become significant even for large $\sigma$. In Appendix F.2, we also provide the convergence results for the restarted version of DProx-clipped-SSTM-shift assuming additionally that $f$ is strongly convex and one can compute starting shifts $h_i^0$ as $\nabla f_i(x^0)$.

## 3 ON THE PROOFS STRUCTURE

In this section, we elaborate on the proofs structure of our results and highlight additional challenges appearing due to the presence of the composite term and distributed nature of the methods. The proof of each result consist of two parts: optimization/descent lemma and the analysis of the sums appearing due to the stochasticity and biasedness of the updates (due to the clipping). In the first part, we usually follow some standard analysis of corresponding deterministic method without clipping and separate the stochastic part from the deterministic one (though for DProx-clipped-SSTM-shift we use quite non-standard Lyapunov function, which can be interesting on its own). For example, in the analysis[5] of DProx-clipped-SGD-shift under Assumption 4, we prove the following inequality:

$$V_{K+1} \leq (1-\gamma\mu)^{K+1}V_0 + \frac{2\gamma}{n}\sum_{k=0}^K\sum_{i=1}^n(1-\gamma\mu)^{K-k}\langle x^k - x^* - \gamma(\nabla f(x^k) - \nabla f(x^*)), \omega_{i,k}\rangle$$

$$+ \frac{\gamma^2}{n^2}\sum_{k=0}^K\sum_{i=1}^n(1-\gamma\mu)^{K-k}\|\omega_{i,k}\|^2 + \gamma^2\sum_{k=0}^K(1-\gamma\mu)^{K-k}\|\omega_k\|^2,$$

where $V_k = \|x^k - x^*\|^2 + \frac{C^2\gamma^2A^2}{n}\sum_{i=1}^n\|h_i^k - \nabla f_i(x^*)\|^2$ for some numerical constant $C > 0$ and vectors $\omega_{i,k} = \nabla f_i(x^k) - \widetilde{g}_i^k$ represent the discrepancy between the full gradients and their estimates. Moreover, to use this inequality for some $K = T \geq 0$ we need to show that $\{x^k\}_{k=0}^T$ belong to the set where the assumptions hold (in this particular case, to $B_{3n\sqrt{2}R}(x^*)$) with high probability. We do it always by induction. More precisely, we prove that $\mathbb{P}\{E_k\} \geq 1 - k\beta/(K+1)$ for the probability event $E_k$ defined as follows: inequalities $V_t \leq 4\exp(-\gamma\mu t)R^2$ and $\left\|\frac{\gamma}{n}\sum_{i=1}^{r-1}\omega_{i,t-1}^u\right\| \leq \exp(-\gamma\mu(t-1)/2)\sqrt{R^2/2}$ hold for $t = 0, 1, \ldots, k$ and $r = 1, 2, \ldots, n$ simultaneously, where $\omega_{i,t}^u = \mathbb{E}_{\xi_i^t}[\widetilde{g}_i^t] - \widetilde{g}_i^t$ and $\mathbb{E}_{\xi_i^t}[\cdot]$ denotes an expectation w.r.t. $\xi_i^t$. To prove this, we use Bernstein inequality for martingale difference (see Lemma B.1). However, to apply Bernstein inequality we need to circumvent multiple technical difficulties related to the estimation of the norm of the clipped vector (that involves derivations related to the shifts $\{h_i^k\}_{i\in[n]}$), proper choice of the clipping level to control the bias and variance and achieve desired linear speed-up (see Lemma B.3 and the following discussion). Moreover, when $n > 1$ (distributed case), we also need to apply additional induction over clients to estimate sums like ⑥ from (265).

---

[5]In the appendix, we analyze this case in the generality of variational inequalities. Here we provide a simplified version for minimization.

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

CONTENTS

## A   EXTRA RELATED WORK

**Non-convex case.**   Li & Orabona (2020) analyze the high-probability convergence rate of SGD for finding first-order stationary points for smooth non-convex unconstrained problems. The first high-probability result under Assumption 1 for the same class of functions is derived by Cutkosky & Mehta (2021). However, the result of Cutkosky & Mehta (2021) relies on the additional assumption that the gradients are bounded. Sadiev et al. (2023) remove the bounded gradient assumption but derive a slightly worse rate. Nguyen et al. (2023b) improve the result and achieve the same rate as in (Cutkosky & Mehta, 2021) without assuming boundedness of the gradients. It is worth mentioning that Cutkosky & Mehta (2021); Sadiev et al. (2023); Nguyen et al. (2023b) derive their main results for the methods that use gradient clipping.

**Gradient clipping**   is a very useful algorithmic tool in the training of deep neural networks (Pascanu et al., 2013; Goodfellow et al., 2016). Gradient clipping also has some good theoretical properties, e.g., it can be useful for minimization of $(L_0, L_1)$-smooth functions (Zhang et al., 2020a), in differential privacy (Abadi et al., 2016), Byzantine-robustness (Karimireddy et al., 2021). Moreover, as we already mentioned, almost all existing high-probability results that do not rely on the light-tailed noise assumption are derived for the methods with clipping. Recently, Sadiev et al. (2023) theoretically showed that SGD has worse high-probability convergence than clipped-SGD even when the noise in the gradient has bounded variance.

# B    AUXILIARY AND TECHNICAL RESULTS

**Bernstein inequality.**    In the final stages of our proofs, we need to estimate certain sums of random variables. The main tool that we use to handle such sums is *Bernstein inequality for martingale differences* (Bennett, 1962; Dzhaparidze & Van Zanten, 2001; Freedman et al., 1975).

**Lemma B.1.** *Let the sequence of random variables $\{X_i\}_{i \geq 1}$ form a martingale difference sequence, i.e. $\mathbb{E}\left[X_i \mid X_{i-1}, \ldots, X_1\right] = 0$ for all $i \geq 1$. Assume that conditional variances $\sigma_i^2 \overset{def}{=} \mathbb{E}\left[X_i^2 \mid X_{i-1}, \ldots, X_1\right]$ exist and are bounded and also assume that there exists deterministic constant $c > 0$ such that $|X_i| \leq c$ almost surely for all $i \geq 1$. Then for all $b > 0$, $G > 0$ and $n \geq 1$*

$$\mathbb{P}\left\{\left|\sum_{i=1}^{n} X_i\right| > b \text{ and } \sum_{i=1}^{n} \sigma_i^2 \leq G\right\} \leq 2\exp\left(-\frac{b^2}{2G + {}^{2cb}/_3}\right). \tag{22}$$

**Impact of clipping on the bias and variance.**    The following lemma also helps to handle the aforementioned sums of random variables.

**Lemma B.2** (Lemma 5.1 from Sadiev et al. (2023)). *Let $X$ be a random vector in $\mathbb{R}^d$ and $\widetilde{X} = \text{clip}(X, \lambda)$. Then, $\|\widetilde{X} - \mathbb{E}[\widetilde{X}]\| \leq 2\lambda$. Moreover, if for some $\sigma \geq 0$ and $\alpha \in (1, 2]$ we have $\mathbb{E}[X] = x \in \mathbb{R}^d$, $\mathbb{E}[\|X - x\|^\alpha] \leq \sigma^\alpha$, and $\|x\| \leq {}^\lambda/_2$, then*

$$\left\|\mathbb{E}[\widetilde{X}] - x\right\| \leq \frac{2^\alpha \sigma^\alpha}{\lambda^{\alpha-1}}, \tag{23}$$

$$\mathbb{E}\left[\left\|\widetilde{X} - \mathbb{E}[\widetilde{X}]\right\|^2\right] \leq 18\lambda^{2-\alpha}\sigma^\alpha. \tag{24}$$

**Intuition behind the choice of clipping level in the distributed case.**    To better illustrate why we increase clipping level $n$ times, we prove the following lemma.

**Lemma B.3.** *Let $X_1, X_2, \ldots, X_n$ be independent random vectors in $\mathbb{R}^d$ and $\widetilde{X}_i = \text{clip}(X_i, \lambda)$ for all $i \in [n]$. Then, for $\widetilde{X} = \frac{1}{n}\sum_{i=1}^{n} \widetilde{X}_i$ we have $\|\widetilde{X} - \mathbb{E}[\widetilde{X}]\| \leq 2\lambda$. Moreover, if for some $\sigma \geq 0$ and $\alpha \in (1, 2]$ we have $\mathbb{E}[X_i] = x_i \in \mathbb{R}^d$, $\mathbb{E}[\|X_i - x_i\|^\alpha] \leq \sigma^\alpha$, and $\|x_i\| \leq {}^\lambda/_2$ for all $i \in [n]$, then for $x = \frac{1}{n}\sum_{i=1}^{n} x_i$ the following inequalities hold*

$$\left\|\mathbb{E}[\widetilde{X}] - x\right\| \leq \frac{2^\alpha \sigma^\alpha}{\lambda^{\alpha-1}}, \tag{25}$$

$$\mathbb{E}\left[\left\|\widetilde{X} - \mathbb{E}[\widetilde{X}]\right\|^2\right] \leq \frac{18\lambda^{2-\alpha}\sigma^\alpha}{n}. \tag{26}$$

*Proof.* From Lemma B.2 we have for all $i \in [n]$ that $\|\widetilde{X}_i - \mathbb{E}[\widetilde{X}_i]\| \leq 2\lambda$ and

$$\left\|\mathbb{E}[\widetilde{X}_i] - x_i\right\| \leq \frac{2^\alpha \sigma^\alpha}{\lambda^{\alpha-1}}, \tag{27}$$

$$\mathbb{E}\left[\left\|\widetilde{X}_i - \mathbb{E}[\widetilde{X}_i]\right\|^2\right] \leq 18\lambda^{2-\alpha}\sigma^\alpha. \tag{28}$$

Jensen's inequality implies

$$\left\|\widetilde{X} - \mathbb{E}[\widetilde{X}]\right\| = \left\|\frac{1}{n}\sum_{i=1}^{n}\left(\widetilde{X}_i - \mathbb{E}[\widetilde{X}_i]\right)\right\| \leq \frac{1}{n}\sum_{i=1}^{n}\left\|\widetilde{X}_i - \mathbb{E}[\widetilde{X}_i]\right\| \leq 2\lambda,$$

$$\left\|\mathbb{E}[\widetilde{X}] - x\right\| = \left\|\frac{1}{n}\sum_{i=1}^{n}\left(\mathbb{E}[\widetilde{X}_i] - x_i\right)\right\| \leq \frac{1}{n}\sum_{i=1}^{n}\left\|\mathbb{E}[\widetilde{X}_i] - x_i\right\| \overset{(27)}{\leq} \frac{2^\alpha \sigma^\alpha}{\lambda^{\alpha-1}}.$$

Finally, using the independence of $\widetilde{X}_1, \ldots, \widetilde{X}_n$, we derive

$$\mathbb{E}\left[\left\|\widetilde{X} - \mathbb{E}[\widetilde{X}]\right\|^2\right] = \mathbb{E}\left[\left\|\frac{1}{n}\sum_{i=1}^{n}\left(\widetilde{X}_i - \mathbb{E}[\widetilde{X}_i]\right)\right\|^2\right] = \frac{1}{n^2}\sum_{i=1}^{n}\mathbb{E}\left[\left\|\widetilde{X}_i - \mathbb{E}[\widetilde{X}_i]\right\|^2\right]$$

$$\overset{(28)}{\leq} \frac{18\lambda^{2-\alpha}\sigma^\alpha}{n}$$

that concludes the proof. □

From (25)-(26), we see that number of workers $n$ appears differently in the bound on bias and variance. However, if we replace $\lambda$ with $n\lambda$, then both bounds will transform to (23)-(24) respectively with $\sigma^\alpha = \sigma^\alpha/n^{\alpha-1}$ (in other words, bias and variance will have the same dependence on $n$). These observations hint that the complexity bounds for distributed methods should be similar to the ones proven for non-distributed methods (in the unconstrained case) by Sadiev et al. (2023) up to the replacement of $\sigma^\alpha$ with $\sigma^\alpha/n^{\alpha-1}$. Nevertheless, our analysis of the distributed case does not rely on Lemma B.3 and has some important differences with the single-node case (even when $\Psi \equiv 0$).

**Useful inequality related to prox-operator.**    In the analysis of DProx-clipped-SGDA-shift, we use the following standard result.

**Lemma B.4** (Theorem 6.39 (iii) from (Beck, 2017))**.** *Let $\Psi$ be a proper lower semicontinuous convex function and $x^+ = \mathrm{prox}_{\gamma\Psi}(x)$. Then for all $y \in \mathbb{R}^d$ the following inequality holds:*

$$\langle x^+ - x, y - x^+ \rangle \geq \gamma \left( \Psi(x^+) - \Psi(y) \right).$$

Table 2: Summary of known and new high-probability complexity results for solving (non-) composite (non-) distributed variational inequality problem (29). Column "Setup" indicates the assumptions made in addition to Assumptions 1. All assumptions are made only on some ball around the solution with radius $\sim R \geq \|x^0 - x^*\|$ (for the results from (Sadiev et al., 2023)) or radius $\sim \sqrt{V}$ (Theorems C.1 and C.2). Complexity is the number of stochastic oracle calls(per worker) needed for a method to guarantee that $\mathbb{P}\{\text{Metric} \leq \varepsilon\} \geq 1 - \beta$ for some $\varepsilon > 0$, $\beta \in (0, 1]$ and "Metric" is taken from the corresponding column. Numerical and logarithmic factors are omitted for simplicity. Column "C?" shows whether the problem (1) is composite, "D?" indicates whether the problem (1) is distributed. Notation: $\widetilde{x}^K_{\text{avg}} = \frac{1}{K+1}\sum_{k=0}^{K}\widetilde{x}^k$ (for SEG-type methods), $x^K_{\text{avg}} = \frac{1}{K+1}\sum_{k=0}^{K}x^k$ (for SGDA-type methods); $L$ = Lipschitz constant; $\sigma$ = parameter from Assumption 1; $R$ = any upper bound on $\|x^0 - x^*\|$ (for the results from (Sadiev et al., 2023)); $V$ = any upper bound on $\|x^0 - x^*\|^2 + \frac{409600\gamma^2 \ln^2 \frac{48n(K+1)}{\beta}}{n^2}\sum_{i=1}^{n}\|F_i(x^*)\|^2$ (for the results of this paper); $\mu$ = quasi-strong monotonicity parameter; $\ell$ = star-cocoercivity parameter. The results of this paper are highlighted in blue.

| Setup | Method | Metric | Complexity | C? | D? |
|---|---|---|---|---|---|
| As. 6 & 7 | clipped-SEG (Sadiev et al., 2023) | $\text{Gap}_R(\widetilde{x}^K_{\text{avg}})$ | $\max\left\{\frac{LR^2}{\varepsilon}, \left(\frac{\sigma R}{\varepsilon}\right)^{\frac{\alpha}{\alpha-1}}\right\}$ | ✗ | ✗ |
| | DProx-clipped-SEG-shift Theorem H.1 | $\text{Gap}_{\sqrt{V}}(\widetilde{x}^K_{\text{avg}})$ | $\max\left\{\frac{LV}{\varepsilon}, \frac{1}{n}\left(\frac{\sigma\sqrt{V}}{\varepsilon}\right)^{\frac{\alpha}{\alpha-1}}\right\}$ | ✓ | ✓ |
| As. 6 & 8 | clipped-SEG (Sadiev et al., 2023) | $\|x^k - x^*\|^2$ | $\max\left\{\frac{L}{\mu}, \left(\frac{\sigma^2}{\mu^2\varepsilon}\right)^{\frac{\alpha}{2(\alpha-1)}}\right\}$ | ✗ | ✗ |
| | DProx-clipped-SEG-shift Theorem H.2 | $\|x^k - x^*\|^2$ | $\max\left\{\frac{L}{\mu}, \frac{1}{n}\left(\frac{\sigma^2}{\mu^2\varepsilon}\right)^{\frac{\alpha}{2(\alpha-1)}}\right\}$ | ✓ | ✓ |
| As. 7 & 9 & 10 | clipped-SGDA (Sadiev et al., 2023) | $\text{Gap}_R(x^K_{\text{avg}})$ | $\max\left\{\frac{\ell R^2}{\varepsilon}, \left(\frac{\sigma R}{\varepsilon}\right)^{\frac{\alpha}{\alpha-1}}\right\}$ | ✗ | ✗ |
| | DProx-clipped-SGDA-shift Theorem G.1 | $\text{Gap}_{\sqrt{V}}(x^K_{\text{avg}})$ | $\max\left\{\frac{\ell V}{\varepsilon}, \frac{1}{n}\left(\frac{\sigma\sqrt{V}}{\varepsilon}\right)^{\frac{\alpha}{\alpha-1}}\right\}$ | ✓ | ✓ |
| As. 8 & 9 | clipped-SGDA (Sadiev et al., 2023) | $\|x^K - x^*\|^2$ | $\max\left\{\frac{\ell}{\mu}, \left(\frac{\sigma^2}{\mu^2\varepsilon}\right)^{\frac{\alpha}{2(\alpha-1)}}\right\}$ | ✗ | ✗ |
| | DProx-clipped-SGDA-shift Theorem G.2 | $\|x^K - x^*\|^2$ | $\max\left\{\frac{\ell}{\mu}, \frac{1}{n}\left(\frac{\sigma^2}{\mu^2\varepsilon}\right)^{\frac{\alpha}{2(\alpha-1)}}\right\}$ | ✓ | ✓ |

# C   COMPOSITE DISTRIBUTED VARIATIONAL INEQUALITIES

In this section, we provide an overview of the obtained results for variational inequalities.

## C.1   SETUP

In addition to the minimization problems, we also consider stochastic composite variational inequality problems (VIPs):

$$\text{find } x^* \in \mathbb{R}^d \text{ such that } \langle F(x^*), x - x^* \rangle + \Psi(x) - \Psi(x^*) \geq 0, \tag{29}$$

where the assumptions on operator $F(x) = \mathbb{E}_{\xi \sim \mathcal{D}}[F_\xi(x)] : \mathbb{R}^d \to \mathbb{R}^d$ will be specified later and, as in the case of minimization, $\Psi(x)$ is a proper, closed, convex function. When $f(x)$ is convex problem (1) is a special case of (29) with $F(x) = \nabla f(x)$. For the examples of problems of type (29), we refer to (Alacaoglu & Malitsky, 2022; Beznosikov et al., 2023).

The distributed version of (29) has the following structure of $F$:

$$F(x) = \frac{1}{n}\sum_{i=1}^{n}\left\{F_i(x) = \mathbb{E}_{\xi_i \sim \mathcal{D}_i}[F_{\xi_i}(x)]\right\}. \tag{30}$$

In this case, there are $n$ workers connected in a centralized way with some parameter server; worker $i$ can query some noisy information (stochastic gradients/estimates) about $F_i$.

## C.2   ASSUMPTIONS

**Bounded central $\alpha$-th moment.**   We consider the situation when $F_i$ are accessible through the stochastic oracle calls. The stochastic estimates satisfy the following assumption.[6]

**Assumption 5.** *There exist some set $Q \subseteq \mathbb{R}^d$ and values $\sigma \geq 0$, $\alpha \in (1, 2]$ such that for all $x \in Q$ we have $\mathbb{E}_{\xi_i \sim \mathcal{D}_i}[F_{\xi_i}(x)] = F_i(x)$ and*

$$\mathbb{E}_{\xi_i \sim \mathcal{D}_i}[\|F_{\xi_i}(x) - F_i(x)\|^\alpha] \leq \sigma^\alpha. \tag{31}$$

---

[6]Following (Sadiev et al., 2023), we consider all assumptions only on some bounded set $Q \subseteq \mathbb{R}^d$; the diameter of $Q$ depends on the starting point.

**Assumptions on $F_i$.** We make standard assumptions on $\{F_i\}_{i \in [n]}$. The first one is Lipschitzness.

**Assumption 6.** *There exist some set $Q \subseteq \mathbb{R}^d$ such that operators $F_i$ are $L$-Lipschitz:*

$$\|F_i(x) - F_i(y)\| \leq L\|x - y\| \quad \forall x, y \in Q, \ i \in [n]. \tag{32}$$

Next, for each particular result, we make one or two of the following assumptions.

**Assumption 7.** *There exist some set $Q \subseteq \mathbb{R}^d$ such that $F$ is monotone on $Q$:*

$$\langle F(x) - F(y), x - y \rangle \geq 0 \quad \forall x, y \in Q. \tag{33}$$

**Assumption 8.** *There exist some set $Q \subseteq \mathbb{R}^d$ such that $F$ is $(\mu, x^*)$-quasi strongly monotone on $Q$ for some $\mu \geq 0$ and any solution $x^*$ of (29):*

$$\langle F(x) - F(x^*), x - x^* \rangle \geq \mu\|x - x^*\|^2, \quad \forall x \in Q. \tag{34}$$

**Assumption 9.** *There exist some set $Q \subseteq \mathbb{R}^d$ such that $\{F_i\}_{i \in [n]}$ are $(\ell, x^*)$-star-cocoercive on $Q$ for some $\ell > 0$ and any solution $x^*$ of (29):*

$$\|F_i(x) - F_i(x^*)\|^2 \leq \ell\langle F_i(x) - F_i(x^*), x - x^* \rangle, \quad \forall x \in Q, \ i \in [n]. \tag{35}$$

**Assumption 10.** *There exist some set $Q \subseteq \mathbb{R}^d$ such that $F$ is $\ell$-cocoercive on $Q$ for some $\ell > 0$:*

$$\|F(x) - F(y)\|^2 \leq \ell\langle F(x) - F(y), x - y \rangle, \quad \forall x, y \in Q. \tag{36}$$

Assumption 7 is a standard assumption for the literature on VIPs. Quasi-strong monotonicity (Mertikopoulos & Zhou, 2019; Song et al., 2020; Loizou et al., 2021) is weaker than standard strong monotonicity[7] and star-cocoercivity is weaker than standard cocoercivity (Assumption 10), which implies monotonicity and Lipschitzness but not vice versa. Both conditions (34) and (35) imply neither monotonicity nor Lipschitzness (Loizou et al., 2021).

## C.3 DProx-clipped-SGDA-shift

For composite variational inequalities, we start with Distributed Prox-clipped-SGDA-shift (DProx-clipped-SGDA-shift) that is defined in (13)-(14) with the following change: $\hat{\Delta}_i^k = \texttt{clip}\left(F_{\xi_i^k}(x^k) - h_i^k, \lambda_k\right)$, where $\xi_1^k, \ldots, \xi_n^k$ are sampled independently from each other and previous steps. For the proposed method, we derive the following result.

**Theorem C.1** (Convergence of DProx-clipped-SGDA-shift). *Let $K \geq 1$, $\beta \in (0, 1)$, $A = \ln\frac{48n(K+1)}{\beta}$, $V \geq \|x^0 - x^*\|^2 + \frac{25600\gamma^2 A^2}{n^2}\sum_{i=1}^n \|F_i(x^*)\|^2$.*
***Case 1.*** *Let Assumptions 1, 8 with $\mu > 0$, and 9 hold for $Q = B_{3\sqrt{V}}(x^*)$. Assume that $0 < \nu = \mathcal{O}(1/\sqrt{n}A)$, $0 < \gamma = \mathcal{O}\left(\min\{1/\sqrt{n}A\mu, 1/\ell A, \ln(B_K)/\mu(K+1)\}\right)$, $B_K = \Theta\left(\max\{2, (K+1)^{2(\alpha-1)/\alpha}\mu^2 n^{2(\alpha-1)/\alpha}V/\sigma^2 A^{2(\alpha-1)/\alpha}\ln^2(B_K)\}\right)$, $\lambda_k = \Theta(n\exp(-\gamma\mu(1+k/2))\sqrt{V}/\gamma A)$.*
***Case 2.*** *Let Assumptions 1, 7, and 9 hold for $Q = B_{3\sqrt{V}}(x^*)$. Assume that $\nu = 0$, $0 < \gamma = \mathcal{O}(\min\{1/\ell A, n^{(\alpha-1)/\alpha}\sqrt{V}/\sigma K^{1/\alpha}A^{(\alpha-1)/\alpha}\})$, $\lambda_k = \lambda = \Theta(n\sqrt{V}/\gamma A)$.*
*Then to guarantee $\|x^K - x^*\|^2 \leq \varepsilon$ in **Case 1** and $\texttt{Gap}_{\sqrt{V}}(x_{avg}^K) = \max_{y \in B_{\sqrt{V}}(x^*)}\left\{\langle F(y), x_{avg}^K - y \rangle + \Psi(x_{avg}^K) - \Psi(y)\right\} \leq \varepsilon$ in **Case 2** with $x_{avg}^K = \frac{1}{K+1}\sum_{k=0}^K x^k$ with probability $\geq 1 - \beta$ DProx-clipped-SGDA-shift requires*

$$\textbf{Case 1:} \quad \widetilde{\mathcal{O}}\left(\max\left\{\frac{\ell}{\mu}, \frac{1}{n}\left(\frac{\sigma^2}{\mu^2\varepsilon}\right)^{\frac{\alpha}{2(\alpha-1)}}\right\}\right) \quad \textit{iterations/oracle calls per worker,} \tag{37}$$

$$\textbf{Case 2:} \quad \widetilde{\mathcal{O}}\left(\max\left\{\frac{\ell V}{\varepsilon}, \frac{1}{n}\left(\frac{\sigma\sqrt{V}}{\varepsilon}\right)^{\frac{\alpha}{\alpha-1}}\right\}\right) \quad \textit{iterations/oracle calls per worker.} \tag{38}$$

As in the case of minimization, in the single-node case, the derived results coincide with ones known for clipped-SGD in the unconstrained case (Sadiev et al., 2023) Up to the difference between $V$ and $\|x^0 - x^*\|^2$. In the distributed case, we also observe the benefits of parallelization.

---

[7]Operator $F$ is called $\mu$-strongly monotone on $Q$ if $\langle F(x) - F(y), x - y \rangle \geq \mu\|x - y\|^2$.

## C.4 DPROX-CLIPPED-SEG-SHIFT

Finally, we propose a distributed version of clipped-SEG for composite VIPs (DProx-clipped-SEG-shift):

$$\widetilde{x}^k = \text{prox}_{\gamma\Psi}\left(x^k - \gamma\widetilde{g}^k\right), \ \ \widetilde{g}^k = \frac{1}{n}\sum_{i=1}^n \widetilde{g}_i^k, \ \ \widetilde{g}_i^k = \widetilde{h}_i^k + \tilde{\Delta}_i^k, \ \ \widetilde{h}_i^{k+1} = \widetilde{h}_i^k + \nu\tilde{\Delta}_i^k \quad (39)$$

$$x^{k+1} = \text{prox}_{\gamma\Psi}\left(x^k - \gamma\widehat{g}^k\right), \ \ \widehat{g}^k = \frac{1}{n}\sum_{i=1}^n \widehat{g}_i^k, \ \ \widehat{g}_i^k = \widehat{h}_i^k + \hat{\Delta}_i^k, \ \ \widehat{h}_i^{k+1} = \widehat{h}_i^k + \nu\hat{\Delta}_i^k \quad (40)$$

where $\tilde{\Delta}_i^k = \text{clip}(F_{\xi_{1,i}^k}(x^k) - \widetilde{h}_i^k, \lambda_k)$, $\hat{\Delta}_i^k = \text{clip}(F_{\xi_{2,i}^k}(\widetilde{x}^k) - \widehat{h}_i^k, \lambda_k)$ and $\xi_{1,1}^k, \ldots, \xi_{1,n}^k, \xi_{2,1}^k, \ldots, \xi_{2,n}^k$ are sampled independently from each other and previous steps. For the proposed method, we derive the following result.

**Theorem C.2** (Convergence of DProx-clipped-SEG-shift). *Let $K \geq 1$, $\beta \in (0,1)$, $A = \ln\frac{48n(K+1)}{\beta}$, $V \geq \|x^0 - x^*\|^2 + \frac{409600\gamma^2 A^2}{n^2}\sum_{i=1}^n\|F_i(x^*)\|^2$.*
***Case 1.*** *Let Assumptions 1, 6, and 8 with $\mu > 0$ hold for $Q = B_{3\sqrt{V}}(x^*)$. Assume that $\nu = \gamma\mu$, $0 < \gamma = \mathcal{O}\left(\min\{1/\mu A^2, 1/L, \sqrt{n}/LA, \ln(B_K)/\mu(K+1)\}\right)$, $B_K = \Theta\left(\max\{2, (K+1)^{2(\alpha-1)/\alpha}\mu^2 n^{2(\alpha-1)/\alpha}V/\sigma^2 A^{2(\alpha-1)/\alpha}\ln^2(B_K)\}\right)$, $\lambda_k = \Theta(n\exp(-\gamma\mu(1+k/4))\sqrt{V}/\gamma A)$.*
***Case 2.*** *Let Assumptions 1, 6, and 7 hold for $Q = B_{4n\sqrt{V}}(x^*)$. Assume that $\nu = 0$, $0 < \gamma = \mathcal{O}(\min\{1/LA, n^{(\alpha-1)/\alpha}\sqrt{V}/\sigma K^{1/\alpha}A^{(\alpha-1)/\alpha}\})$, $\lambda_k = \lambda = \Theta(n\sqrt{V}/\gamma A)$.*
*Then to guarantee $\|x^K - x^*\|^2 \leq \varepsilon$ in **Case 1** and $\text{Gap}_{\sqrt{V}}(\widetilde{x}_{avg}^K) = \max_{y\in B_{\sqrt{V}}(x^*)}\left\{\langle F(y), \widetilde{x}_{avg}^K - y\rangle + \Psi(\widetilde{x}_{avg}^K) - \Psi(y)\right\} \leq \varepsilon$ in **Case 2** with $\widetilde{x}_{avg}^K = \frac{1}{K+1}\sum_{k=0}^K \widetilde{x}^k$ with probability $\geq 1 - \beta$ DProx-clipped-SEG-shift requires*

$$\textbf{Case 1:} \quad \widetilde{\mathcal{O}}\left(\max\left\{\frac{L}{\mu}, \frac{1}{n}\left(\frac{\sigma^2}{\mu^2\varepsilon}\right)^{\frac{\alpha}{2(\alpha-1)}}\right\}\right) \qquad \textit{iterations/oracle calls per worker,} \quad (41)$$

$$\textbf{Case 2:} \quad \widetilde{\mathcal{O}}\left(\max\left\{\frac{LV}{\varepsilon}, \frac{1}{n}\left(\frac{\sigma\sqrt{V}}{\varepsilon}\right)^{\frac{\alpha}{\alpha-1}}\right\}\right) \qquad \textit{iterations/oracle calls per worker.} \quad (42)$$

The main properties of the above result are similar to the ones of the result for DProx-clipped-SGDA-shift. The only difference is that the methods (DProx-clipped-SGDA/SEG-shift) are analyzed for different classes of problems and, thus, complement each other. According to the known lower bounds, our upper bound (41) has optimal dependence on $\varepsilon$ up to logarithmic factors.

# D    MISSING PROOFS FOR Prox-clipped-SGD-star

This section provides the complete formulations of our results for Prox-clipped-SGD-star and rigorous proofs. We start with the following result – a generalization of Lemma E.7 from (Sadiev et al., 2023) to the composite distributed problems.

**Lemma D.1.** *Consider differentiable function $f : \mathbb{R}^d \to \mathbb{R}$ having a finite-sum structure* (2)*. If $f$ satisfies Assumption 4 on some set $Q$ with parameter $\mu$ and $D_f(x, x^*) \geq 0$ for all[8] $x \in Q$, then operator $F(x) = \nabla f(x)$ satisfies Assumption 8 on $Q$ with parameter $\mu/2$. If $f_1, \ldots, f_n$ satisfy Assumption 2 and 4 with $\mu = 0$ on some set $Q$, then operator $F(x) = \nabla f(x)$ satisfies Assumption 9 on $Q$ with $\ell = 2L$.*

*Proof.* Let Assumption 4 hold on some set $Q$ and $D_f(x, x^*) \geq 0$ for all $x \in Q$. Then, averaging inequalities (7), we get that for all $x \in Q$

$$f(x^*) \geq f(x) + \langle \nabla f(x), x^* - x \rangle + \frac{\mu}{2}\|x - x^*\|^2,$$

implying for $F(x) = \nabla f(x)$ that

$$\langle F(x) - F(x^*), x - x^* \rangle \quad \geq \quad D_f(x, x^*) + \frac{\mu}{2}\|x - x^*\|^2 \geq \frac{\mu}{2}\|x - x^*\|^2,$$

meaning that Assumption 8 is satisfied with parameter $\mu/2$.

It remains to show the second part of the lemma. Let Assumptions 2 and 4 with $\mu = 0$ hold on some set $Q$. We need to show that operators $F_i(x) = \nabla f_i(x)$, $i = 1, \ldots, n$ satisfy Assumption 9 on $Q$ with $\ell = 2L$. Guided by (Gorbunov et al., 2022b, Lemma C.6) and (Sadiev et al., 2023, Lemma E.7), we derive

$$
\begin{aligned}
\left\| x - x^* - \frac{1}{L}(F_i(x) - F_i(x^*)) \right\|^2 &= \|x - x^*\|^2 - \frac{2}{L}\langle x - x^*, F_i(x) - F_i(x^*) \rangle \\
&\quad + \frac{1}{L^2}\|F_i(x) - F_i(x^*)\|^2 \qquad (43) \\
&= \|x - x^*\|^2 - \frac{2}{L}\langle x - x^*, \nabla f_i(x) - \nabla f_i(x^*) \rangle \\
&\quad + \frac{1}{L^2}\|\nabla f_i(x) - \nabla f_i(x^*)\|^2 \\
&\overset{(5)}{\leq} \|x - x^*\|^2 - \frac{2}{L}\langle x - x^*, \nabla f_i(x) \rangle \\
&\quad + \frac{2}{L}(f_i(x) - f_i(x^*)) \\
&\overset{(7)}{\leq} \|x - x^*\|^2. \qquad (44)
\end{aligned}
$$

From (43) and (44) we get

$$\|x - x^*\|^2 - \frac{2}{L}\langle x - x^*, F_i(x) - F_i(x^*) \rangle + \frac{1}{L^2}\|F_i(x) - F_i(x^*)\|^2 \leq \|x - x^*\|^2$$

that is equivalent to (35) with $\ell = 2L$. $\qquad\qquad\square$

Therefore, for smooth quasi-strongly convex $f$ such that $D_f(x, x^*) \geq 0$ ($n = 1$) we can consider operator $F(x) = \nabla f(x)$ and VI formulation instead. In this case, the method is equivalent to Prox-clipped-SGDA-star:

$$x^{k+1} = \text{prox}_{\gamma\Psi}\left(x^k - \gamma\widetilde{g}^k\right), \quad \widetilde{g}^k = F(x^*) + \text{clip}\left(F_{\xi^k}(x^k) - F(x^*), \lambda_k\right)$$
$$\widehat{g}^k = \text{clip}\left(F_{\xi^k}(x^k) - F(x^*), \lambda_k\right).$$

The following lemma is the main "optimization" part of the analysis of Prox-clipped-SGDA-star.

---

[8]For example $D_f(x, x^*) \geq 0$ when $f$ is convex or when $\Psi(x) = 0$. We notice that Assumption 2 implies $D_f(x, x^*) \geq 0$ since the right-hand side of (5) equals $D_f(x, x^*)$ after averaging.

**Lemma D.2.** *Let $n = 1$, Assumptions 8, 9 hold for $Q = B_{2R}(x^*)$, where $R \geq R_0 \stackrel{def}{=} \|x^0 - x^*\|$, and $0 < \gamma \leq 1/\ell$. If $x^k$ lies in $B_{2R}(x^*)$ for all $k = 0, 1, \ldots, K$ for some $K \geq 0$, then the iterates produced by* Prox-clipped-SGDA-star *satisfy*

$$
\begin{aligned}
\|x^{K+1} - x^*\|^2 &\leq (1 - \gamma\mu)^{K+1}\|x^0 - x^*\|^2 + \gamma^2 \sum_{k=0}^{K}(1 - \gamma\mu)^{K-k}\|\omega_k\|^2 \\
&\quad + 2\gamma \sum_{k=0}^{K}(1 - \gamma\mu)^{K-k}\langle x^k - x^* - \gamma(F(x^k) - F(x^*)), \omega_k \rangle,
\end{aligned}
\tag{45}
$$

$$
\omega_k \stackrel{def}{=} F(x^k) - F(x^*) - \widehat{g}^k.
\tag{46}
$$

*Proof.* Using the update rule of Prox-clipped-SGDA-star, we obtain

$$
\begin{aligned}
\|x^{k+1} - x^*\|^2 &= \|\operatorname{prox}_{\gamma\Psi}\left(x^k - \gamma\widetilde{g}^k\right) - \operatorname{prox}_{\gamma\Psi}\left(x^* - \gamma F(x^*)\right)\|^2 \\
&\leq \|x^k - x^* - \gamma(\widetilde{g}^k - F(x^*))\|^2 \\
&= \|x^k - x^*\|^2 - 2\gamma\langle x^k - x^*, \widehat{g}^k \rangle + \gamma^2\|\widehat{g}^k\|^2 \\
&\stackrel{(46)}{=} \|x^k - x^*\|^2 - 2\gamma\langle x^k - x^*, F(x^k) - F(x^*)\rangle - 2\gamma^2\langle F(x^k) - F(x^*), \omega_k \rangle \\
&\quad + 2\gamma\langle x^k - x^*, \omega_k \rangle + \gamma^2\|F(x^k) - F(x^*)\|^2 + \gamma^2\|\omega_k\|^2 \\
&\stackrel{(35)}{\leq} \|x^k - x^*\|^2 + 2\gamma\langle x^k - x^*, \omega_k \rangle - 2\gamma^2\langle F(x^k) - F(x^*), \omega_k \rangle \\
&\quad - 2\gamma\left(1 - \frac{\gamma\ell}{2}\right)\langle x^k - x^*, F(x^k) - F(x^*)\rangle + \gamma^2\|\omega_k\|^2 \\
&\stackrel{(34), \gamma \leq \frac{1}{\ell}}{\leq} \|x^k - x^*\|^2 + 2\gamma\langle x^k - x^* - \gamma(F(x^k) - F(x^*)), \omega_k \rangle \\
&\quad - 2\gamma\mu\left(1 - \frac{\gamma\ell}{2}\right)\|x^k - x^*\|^2 + \gamma^2\|\omega_k\|^2 \\
&\stackrel{\gamma \leq \frac{1}{\ell}}{\leq} (1 - \gamma\mu)\|x^k - x^*\|^2 + 2\gamma\langle x^k - x^* - \gamma(F(x^k) - F(x^*)), \omega_k \rangle + \gamma^2\|\omega_k\|^2.
\end{aligned}
$$

Unrolling the recurrence, we obtain (45). $\qquad\square$

**Theorem D.1.** *Let $n = 1$, Assumptions 8, 9, hold for $Q = B_{2R}(x^*) = \{x \in \mathbb{R}^d \mid \|x - x^*\| \leq 2R\}$ for any $x \in B_{2R}(x^*)$, where $R \geq \|x^0 - x^*\|$, and*

$$
0 < \gamma \leq \min\left\{\frac{1}{400\ell\ln\frac{4(K+1)}{\beta}}, \frac{\ln(B_K)}{\mu(K+1)}\right\},
\tag{47}
$$

$$
B_K = \max\left\{2, \frac{(K+1)^{\frac{2\alpha-1}{\alpha}}\mu^2 R^2}{4 \cdot 10^{\frac{1}{\alpha}}120^{\frac{2(\alpha-1)}{\alpha}}\sigma^2\ln^{\frac{2(\alpha-1)}{\alpha}}\left(\frac{4(K+1)}{\beta}\right)\ln^2(B_K)}\right\}
\tag{48}
$$

$$
= \mathcal{O}\left(\max\left\{2, \frac{K^{\frac{2\alpha-1}{\alpha}}\mu^2 R^2}{\sigma^2\ln^{\frac{2(\alpha-1)}{\alpha}}\left(\frac{K}{\beta}\right)\ln^2\left(\max\left\{2, \frac{K^{\frac{2\alpha-1}{\alpha}}\mu^2 R^2}{\sigma^2\ln^{\frac{2(\alpha-1)}{\alpha}}\left(\frac{K}{\beta}\right)}\right\}\right)}\right\}\right),
\tag{49}
$$

$$
\lambda_k = \frac{\exp(-\gamma\mu(1 + k/2))R}{120\gamma\ln\frac{4(K+1)}{\beta}},
\tag{50}
$$

*for some $K \geq 0$ and $\beta \in (0, 1]$ such that $\ln\frac{4(K+1)}{\beta} \geq 1$. Then, after $K$ iterations the iterates produced by* Prox-clipped-SGDA-star *with probability at least $1 - \beta$ satisfy*

$$
\|x^{K+1} - x^*\|^2 \leq 2\exp(-\gamma\mu(K+1))R^2.
\tag{51}
$$

*In particular, when $\gamma$ equals the minimum from (47), then the iterates produced by* Prox-clipped-SGDA-star *after $K$ iterations with probability at least $1 - \beta$ satisfy*

$$R_K^2 = \mathcal{O}\left(\max\left\{R^2 \exp\left(-\frac{\mu K}{\ell \ln \frac{K}{\beta}}\right), \frac{\sigma^2 \ln^{\frac{2(\alpha-1)}{\alpha}}\left(\frac{K}{\beta}\right) \ln^2\left(\max\left\{2, \frac{K^{\frac{2\alpha-1}{\alpha}}\mu^2 R^2}{\sigma^2 \ln^{\frac{2(\alpha-1)}{\alpha}}\left(\frac{K}{\beta}\right)}\right\}\right)}{K^{\frac{2\alpha-1}{\alpha}}\mu^2}\right\}\right), \quad (52)$$

*meaning that to achieve $R_K^2 = \|x^K - x^*\|^2 \le \varepsilon$ with probability at least $1 - \beta$* Prox-clipped-SGDA-star *requires*

$$K = \mathcal{O}\left(\frac{\ell}{\mu}\ln\left(\frac{R^2}{\varepsilon}\right)\ln\left(\frac{\ell}{\mu\beta}\ln\frac{R^2}{\varepsilon}\right), \left(\frac{\sigma^2}{\mu^2\varepsilon}\right)^{\frac{\alpha}{2\alpha-1}}\ln\left(\frac{1}{\beta}\left(\frac{\sigma^2}{\mu^2\varepsilon}\right)^{\frac{\alpha}{2\alpha-1}}\right)\ln^{\frac{\alpha}{\alpha-1}}\left(B_\varepsilon\right)\right) \quad (53)$$

*iterations/oracle calls, where*

$$B_\varepsilon = \max\left\{2, \frac{R^2}{\varepsilon \ln\left(\frac{1}{\beta}\left(\frac{\sigma^2}{\mu^2\varepsilon}\right)^{\frac{\alpha}{2\alpha-1}}\right)}\right\}.$$

*Proof.* Let $R_k = \|x^k - x^*\|$ for all $k \ge 0$. Our proof is induction-based: by induction, we show that the iterates of the method stay in some ball around the solution with high probability. To formulate the statement rigorously, we introduce probability event $E_k$ for each $k = 0, 1, \ldots, K+1$ as follows: inequalities

$$R_t^2 \le 2\exp(-\gamma\mu t)R^2 \quad (54)$$

hold for $t = 0, 1, \ldots, k$ simultaneously. We will prove by induction that $\mathbb{P}\{E_k\} \ge 1 - {}^{k\beta}/(K+1)$ for all $k = 0, 1, \ldots, K+1$. The base of the induction follows immediately by the definition of $R$. Next, assume that for $k = T - 1 \le K$ the statement holds: $\mathbb{P}\{E_{T-1}\} \ge 1 - {}^{(T-1)\beta}/(K+1)$. Given this, we need to prove $\mathbb{P}\{E_T\} \ge 1 - {}^{T\beta}/(K+1)$. Since $R_t^2 \le 2\exp(-\gamma\mu t)R^2 \le 2R^2$, we have $x^t \in B_{2R}(x^*)$ for $t = 0, 1, \ldots, T - 1$, where operator $F$ is $\ell$-star-cocoercive. Thus, $E_{T-1}$ implies

$$\|F(x^t) - F(x^*)\| \quad \le \quad \ell\|x^t - x^*\| \overset{(54)}{\le} \sqrt{2}\ell\exp(-\gamma\mu t/2)R \overset{(47),(50)}{\le} \frac{\lambda_t}{2} \quad (55)$$

and

$$\|\omega_t\|^2 \quad \le \quad 2\|F(x^t) - F(x^*)\|^2 + 2\|\widehat{g}^t\|^2 \overset{(55)}{\le} \frac{5}{2}\lambda_t^2 \overset{(50)}{\le} \frac{\exp(-\gamma\mu t)R^2}{4\gamma^2} \quad (56)$$

for all $t = 0, 1, \ldots, T - 1$, where we use that $\|a + b\|^2 \le 2\|a\|^2 + 2\|b\|^2$ holding for all $a, b \in \mathbb{R}^d$. This means that we can apply Lemma D.2 and $(1 - \gamma\mu)^T \le \exp(-\gamma\mu T)$: $E_{T-1}$ implies

$$R_T^2 \quad \le \quad \exp(-\gamma\mu T)R^2 + 2\gamma\sum_{t=0}^{T-1}(1-\gamma\mu)^{T-1-t}\langle x^t - x^* - \gamma(F(x^t) - F(x^*)), \omega_t\rangle$$

$$+\gamma^2\sum_{t=0}^{T-1}(1-\gamma\mu)^{T-1-t}\|\omega_t\|^2.$$

Before we proceed, we introduce a new notation:

$$\eta_t = \begin{cases} \underbrace{x^t - x^* - \gamma(F(x^t) - F(x^*))}_{\hat{\eta}_t}, & \text{if } \|\hat{\eta}_t\| \le \sqrt{2}(1+\gamma\ell)\exp(-\gamma\mu t/2)R, \\ 0, & \text{otherwise}, \end{cases} \quad (57)$$

for $t = 0, 1, \ldots, T - 1$. Random vectors $\{\eta_t\}_{t=0}^T$ are bounded almost surely:

$$\|\eta_t\| \le \sqrt{2}(1 + \gamma\ell)\exp(-\gamma\mu t/2)R \quad (58)$$

for all $t = 0, 1, \ldots, T-1$. We also notice that $E_{T-1}$ implies $\|F(x^t) - F(x^*)\| \leq \sqrt{2}\ell \exp(-\gamma\mu t/2)R$ (due to (55)) and

$$
\begin{aligned}
\|x^t - x^* - \gamma(F(x^t) - F(x^*))\| &\leq \|x^t - x^*\| + \gamma\|F(x^t) - F(x^*)\| \\
&\overset{(55)}{\leq} \sqrt{2}(1 + \gamma\ell)\exp(-\gamma\mu t/2)R
\end{aligned}
$$

for $t = 0, 1, \ldots, T-1$. Therefore, $E_{T-1}$ implies $\eta_t = x^t - x^* - \gamma(F(x^t) - F(x^*))$ for all $t = 0, 1, \ldots, T-1$ and from $E_{T-1}$ it follows that

$$
\begin{aligned}
R_T^2 &\leq \exp(-\gamma\mu T)R^2 + 2\gamma\sum_{t=0}^{T-1}(1-\gamma\mu)^{T-1-t}\langle\eta_t, \omega_t\rangle \\
&\quad + \gamma^2\sum_{t=0}^{T-1}(1-\gamma\mu)^{T-1-t}\|\omega_t\|^2.
\end{aligned}
$$

For convenience, we define unbiased and biased parts of $\omega_t$:

$$
\omega_t^u \overset{\text{def}}{=} \mathbb{E}_{\xi^t}\left[\widehat{g}^t\right] - \widehat{g}^t, \quad \omega_t^b \overset{\text{def}}{=} F(x^t) - F(x^*) - \mathbb{E}_{\xi^t}\left[\widehat{g}^t\right], \tag{59}
$$

for all $t = 0, \ldots, T-1$. By definition we have $\omega_t = \omega_t^u + \omega_t^b$ for all $t = 0, \ldots, T-1$. Therefore, $E_{T-1}$ implies

$$
\begin{aligned}
R_T^2 &\leq \exp(-\gamma\mu T)R^2 + \underbrace{2\gamma\sum_{t=0}^{T-1}(1-\gamma\mu)^{T-1-t}\langle\eta_t, \omega_t^u\rangle}_{\text{①}} \\
&\quad + \underbrace{2\gamma\sum_{t=0}^{T-1}(1-\gamma\mu)^{T-1-t}\langle\eta_t, \omega_t^b\rangle}_{\text{②}} + \underbrace{2\gamma^2\sum_{t=0}^{T-1}(1-\gamma\mu)^{T-1-t}\mathbb{E}_{\xi^t}\left[\|\omega_t^u\|^2\right]}_{\text{③}} \\
&\quad + \underbrace{2\gamma^2\sum_{t=0}^{T-1}(1-\gamma\mu)^{T-1-t}\left(\|\omega_t^u\|^2 - \mathbb{E}_{\xi^t}\left[\|\omega_t^u\|^2\right]\right)}_{\text{④}} \\
&\quad + \underbrace{2\gamma^2\sum_{t=0}^{T-1}(1-\gamma\mu)^{T-1-t}\|\omega_t^b\|^2}_{\text{⑤}}. \tag{60}
\end{aligned}
$$

where we also use inequality $\|a + b\|^2 \leq 2\|a\|^2 + 2\|b\|^2$ holding for all $a, b \in \mathbb{R}^d$ to upper bound $\|\omega_t\|^2$. To derive high-probability bounds for ①, ②, ③, ④, ⑤ we need to establish several useful inequalities related to $\omega_{i,t}^u, \omega_{i,t}^b$. First, by definition of clipping

$$
\|\omega_t^u\| \leq 2\lambda_t. \tag{61}
$$

Next, $E_{T-1}$ implies that $\|F(x^t) - F(x^*)\| \leq \lambda_t/2$ for all $t = 0, 1, \ldots, T-1$ (see (55)). Therefore, from Lemma B.2 we also have that $E_{T-1}$ implies

$$
\left\|\omega_t^b\right\| \leq \frac{2^\alpha\sigma^\alpha}{\lambda_t^{\alpha-1}}, \tag{62}
$$

$$
\mathbb{E}_{\xi^t}\left[\left\|\omega_t^b\right\|^2\right] \leq 18\lambda_t^{2-\alpha}\sigma^\alpha, \tag{63}
$$

$$
\mathbb{E}_{\xi^t}\left[\left\|\omega_t^u\right\|^2\right] \leq 18\lambda_t^{2-\alpha}\sigma^\alpha, \tag{64}
$$

for all $t = 0, 1, \ldots, T-1$.

**Upper bound for ①.** To estimate this sum, we will use Bernstein's inequality. The summands have conditional expectations equal to zero:

$$
\mathbb{E}_{\xi^t}\left[2\gamma(1-\gamma\mu)^{T-1-t}\langle\eta_t, \omega_t^u\rangle\right] = 0.
$$

Next, the summands are bounded:

$$
\begin{aligned}
|2\gamma(1-\gamma\mu)^{T-1-t}\langle\eta_t,\omega_t^u\rangle| &\leq 2\gamma\exp(-\gamma\mu(T-1-t))\|\eta_t\|\cdot\|\omega_t^u\| \\
&\overset{(58),(61)}{\leq} 4\sqrt{2}\gamma(1+\gamma\ell)\exp(-\gamma\mu(T-1-t/2))R\lambda_t \\
&\overset{(47),(50)}{\leq} \frac{\exp(-\gamma\mu T)R^2}{5\ln\frac{4(K+1)}{\beta}} \overset{\text{def}}{=} c.
\end{aligned}
\tag{65}
$$

Finally, conditional variances $\sigma_t^2 \overset{\text{def}}{=} \mathbb{E}_{\xi^t}\left[4\gamma^2(1-\gamma\mu)^{2T-2-2t}\langle\eta_t,\omega_t^u\rangle^2\right]$ of the summands are bounded:

$$
\begin{aligned}
\sigma_t^2 &\leq \mathbb{E}_{\xi^t}\left[4\gamma^2\exp(-\gamma\mu(2T-2-2t))\|\eta_t\|^2\cdot\|\omega_t^u\|^2\right] \\
&\overset{(58)}{\leq} 8\gamma^2(1+\gamma\ell)^2\exp(-\gamma\mu(2T-2-t))R^2\mathbb{E}_{\xi^t}\left[\|\omega_t^u\|^2\right] \\
&\overset{(47)}{\leq} 10\gamma^2\exp(-\gamma\mu(2T-t))R^2\mathbb{E}_{\xi^t}\left[\|\omega_t^u\|^2\right].
\end{aligned}
\tag{66}
$$

Applying Bernstein's inequality (Lemma B.1) with $X_t = 2\gamma(1-\gamma\mu)^{T-1-t}\langle\eta_t,\omega_t^u\rangle$, constant $c$ defined in (65), $b = \frac{1}{5}\exp(-\gamma\mu T)R^2$, $G = \frac{\exp(-2\gamma\mu T)R^4}{150\ln\frac{4(K+1)}{\beta}}$, we get

$$
\begin{aligned}
\mathbb{P}\left\{|①| > \frac{1}{5}\exp(-\gamma\mu T)R^2 \text{ and } \sum_{t=0}^{T-1}\sigma_t^2 \leq \frac{\exp(-2\gamma\mu T)R^4}{150\ln\frac{4(K+1)}{\beta}}\right\} &\leq 2\exp\left(-\frac{b^2}{2F+2cb/3}\right) \\
&= \frac{\beta}{2(K+1)}.
\end{aligned}
$$

The above is equivalent to $\mathbb{P}\{E_①\} \geq 1 - \frac{\beta}{2(K+1)}$ for

$$
E_① = \left\{\text{either} \quad \sum_{t=0}^{T-1}\sigma_t^2 > \frac{\exp(-2\gamma\mu T)R^4}{150\ln\frac{4(K+1)}{\beta}} \quad \text{or} \quad |①| \leq \frac{1}{5}\exp(-\gamma\mu T)R^2\right\}.
\tag{67}
$$

Moreover, $E_{T-1}$ implies

$$
\begin{aligned}
\sum_{t=0}^{T-1}\sigma_t^2 &\overset{(66)}{\leq} 10\gamma^2\exp(-2\gamma\mu T)R^2\sum_{t=0}^{T-1}\frac{\mathbb{E}_{\xi^t}\left[\|\omega_t^u\|^2\right]}{\exp(-\gamma\mu t)} \\
&\overset{(64),T\leq K+1}{\lesssim} 180\gamma^2\exp(-2\gamma\mu T)R^2\sigma^2\sum_{t=0}^{K}\frac{\lambda_t^{2-\alpha}}{\exp(-\gamma\mu t)} \\
&\overset{(50)}{\leq} \frac{180\gamma^\alpha\exp(-2\gamma\mu T)R^{4-\alpha}\sigma^\alpha(K+1)\exp(\frac{\gamma\mu\alpha K}{2})}{120^{2-\alpha}\ln^{2-\alpha}\frac{4(K+1)}{\beta}} \\
&\overset{(47)}{\leq} \frac{\exp(-2\gamma\mu T)R^4}{150\ln\frac{4(K+1)}{\beta}}.
\end{aligned}
\tag{68}
$$

**Upper bound for ②.** Probability event $E_{T-1}$ implies

$$
\begin{aligned}
② &\leq 2\gamma\exp(-\gamma\mu(T-1))\sum_{t=0}^{T-1}\frac{\|\eta_t\|\cdot\|\omega_t^b\|}{\exp(-\gamma\mu t)} \\
&\overset{(58),(62)}{\leq} 2^{1+\alpha}\sqrt{2}\gamma(1+\gamma\ell)\exp(-\gamma\mu(T-1))R\sigma^\alpha\sum_{t=0}^{T-1}\frac{1}{\lambda_t^{\alpha-1}\exp(-\gamma\mu t/2)} \\
&\overset{(50),T\leq K+1}{\leq} \frac{2^{1+\alpha}120^{\alpha-1}\sqrt{2}\gamma^\alpha\sigma^\alpha R^{2-\alpha}(1+\gamma\ell)\exp(-\gamma\mu(T-1))(K+1)\exp\left(\frac{\gamma\mu\alpha K}{2}\right)}{\ln^{1-\alpha}\frac{4(K+1)}{\beta}} \\
&\overset{(47)}{\leq} \frac{1}{5}\exp(-\gamma\mu T)R^2.
\end{aligned}
\tag{69}
$$

**Upper bound for ③.** Probability event $E_{T-1}$ implies

$$
\begin{aligned}
③ \quad &= \quad 2\gamma^2 \exp(-\gamma\mu(T-1)) \sum_{t=0}^{T-1} \frac{\mathbb{E}_{\xi^t}\left[\|\omega_t^u\|^2\right]}{\exp(-\gamma\mu t)} \\
&\overset{(64)}{\leq} \quad 36\gamma^2 \exp(-\gamma\mu(T-1))\sigma^\alpha \sum_{t=0}^{T-1} \frac{\lambda_t^{2-\alpha}}{\exp(-\gamma\mu t)} \\
&\overset{(50), T \leq K+1}{\leq} \quad \frac{36\gamma^\alpha R^{2-\alpha} \exp(-\gamma\mu(T-1))\sigma^\alpha (K+1) \exp(\frac{\gamma\mu\alpha K}{2})}{120^{2-\alpha} \ln^{2-\alpha} \frac{4(K+1)}{\beta}} \\
&\overset{(47)}{\leq} \quad \frac{1}{5} \exp(-\gamma\mu T) R^2.
\end{aligned}
\tag{70}
$$

**Upper bound for ④.** To estimate this sum, we will use Bernstein's inequality. The summands have conditional expectations equal to zero:

$$
2\gamma^2 (1-\gamma\mu)^{T-1-t} \mathbb{E}_{\xi^t}\left[\|\omega_t^u\|^2 - \mathbb{E}_{\xi^t}\left[\|\omega_t^u\|^2\right]\right] = 0.
$$

Next, the summands are bounded:

$$
\begin{aligned}
2\gamma^2 (1-\gamma\mu)^{T-1-t} \left|\|\omega_t^u\|^2 - \mathbb{E}_{\xi^t}\left[\|\omega_t^u\|^2\right]\right| \quad &\overset{(61)}{\leq} \quad \frac{16\gamma^2 \exp(-\gamma\mu T)\lambda_t^2}{\exp(-\gamma\mu(t+1))} \\
&\overset{(50)}{\leq} \quad \frac{\exp(-\gamma\mu T)R^2}{5\ln\frac{4(K+1)}{\beta}} \\
&\overset{\text{def}}{=} \quad c.
\end{aligned}
\tag{71}
$$

Finally, conditional variances

$$
\widetilde{\sigma}_t^2 \overset{\text{def}}{=} \mathbb{E}_{\xi^t}\left[4\gamma^4 (1-\gamma\mu)^{2T-2-2t}\left|\|\omega_t^u\|^2 - \mathbb{E}_{\xi^t}\left[\|\omega_t^u\|^2\right]\right|^2\right]
$$

of the summands are bounded:

$$
\begin{aligned}
\widetilde{\sigma}_t^2 \quad &\overset{(71)}{\leq} \quad \frac{2\gamma^2 \exp(-2\gamma\mu T)R^2}{5\exp(-\gamma\mu(1+t))\ln\frac{4(K+1)}{\beta}} \mathbb{E}_{\xi^t}\left[\left|\|\omega_t^u\|^2 - \mathbb{E}_{\xi^t}\left[\|\omega_t^u\|^2\right]\right|\right] \\
&\leq \quad \frac{4\gamma^2 \exp(-2\gamma\mu T)R^2}{5\exp(-\gamma\mu(1+t))\ln\frac{4(K+1)}{\beta}} \mathbb{E}_{\xi^t}\left[\|\omega_t^u\|^2\right].
\end{aligned}
\tag{72}
$$

Applying Bernstein's inequality (Lemma B.1) with $X_t = 2\gamma^2 (1-\gamma\mu)^{T-1-t}\left(\|\omega_t^u\|^2 - \mathbb{E}_{\xi^t}\left[\|\omega_t^u\|^2\right]\right)$, constant $c$ defined in (71), $b = \frac{1}{5}\exp(-\gamma\mu T)R^2$, $G = \frac{\exp(-2\gamma\mu T)R^4}{150\ln\frac{4(K+1)}{\beta}}$, we get:

$$
\begin{aligned}
\mathbb{P}\left\{|④| > \frac{1}{5}\exp(-\gamma\mu T)R^2 \text{ and } \sum_{l=0}^{T-1} \widetilde{\sigma}_t^2 \leq \frac{\exp(-2\gamma\mu T)R^4}{150\ln\frac{4(K+1)}{\beta}}\right\} \quad &\leq \quad 2\exp\left(-\frac{b^2}{2G + \frac{2cb}{3}}\right) \\
&= \quad \frac{\beta}{2(K+1)}.
\end{aligned}
$$

The above is equivalent to $\mathbb{P}\{E_④\} \geq 1 - \frac{\beta}{2(K+1)}$ for

$$
E_④ = \left\{\text{either} \quad \sum_{t=0}^{T-1} \widetilde{\sigma}_t^2 > \frac{\exp(-2\gamma\mu T)R^4}{150\ln\frac{4(K+1)}{\beta}} \quad \text{or} \quad |④| \leq \frac{1}{5}\exp(-\gamma\mu T)R^2\right\}.
\tag{73}
$$

Moreover, $E_{T-1}$ implies

$$
\begin{aligned}
\sum_{l=0}^{T-1}\widetilde{\sigma}_t^2 &\overset{(72)}{\leq} \frac{4\gamma^2\exp(-\gamma\mu(2T-1))R^2}{5\ln\frac{4(K+1)}{\beta}}\sum_{t=0}^{T-1}\frac{\mathbb{E}_{\xi^t}\left[\|\omega_l^u\|^2\right]}{\exp(-\gamma\mu t)} \\
&\overset{(64),T\leq K+1}{\leq} \frac{72\gamma^2\exp(-\gamma\mu(2T-1))R^2\sigma^\alpha}{5\ln\frac{4(K+1)}{\beta}}\sum_{t=0}^{K}\frac{\lambda_t^{2-\alpha}}{\exp(-\gamma\mu t)} \\
&\overset{(50)}{\leq} \frac{72\gamma^\alpha\exp(-\gamma\mu(2T-1))R^{4-\alpha}\sigma^\alpha(K+1)\exp(\frac{\gamma\mu\alpha K}{2})}{5\cdot 120^{2-\alpha}\ln^{3-\alpha}\frac{4(K+1)}{\beta}} \\
&\overset{(47)}{\leq} \frac{\exp(-2\gamma\mu T)R^4}{150\ln\frac{4(K+1)}{\beta}}.
\end{aligned}
\tag{74}
$$

**Upper bound for ⑤.** Probability event $E_{T-1}$ implies

$$
\begin{aligned}
⑤ &= 2\gamma^2\sum_{t=0}^{T-1}\exp(-\gamma\mu(T-1-t))\|\omega_t^b\|^2 \\
&\overset{(62)}{\leq} 2\cdot 2^{2\alpha}\gamma^2\sigma^{2\alpha}\exp(-\gamma\mu(T-1))\sum_{t=0}^{T-1}\frac{1}{\lambda_t^{2\alpha-2}\exp(-\gamma\mu t)} \\
&\overset{(50),T\leq K+1}{\leq} \frac{2\cdot 2^{2\alpha}120^{2\alpha-2}\gamma^{2\alpha}\sigma^{2\alpha}\exp(-\gamma\mu(T-3))\ln^{2\alpha-2}\frac{4(K+1)}{\beta}}{R^{2\alpha-2}}\sum_{t=0}^{K}\exp\left(\gamma\mu\alpha t\right) \\
&\leq \frac{2\cdot 2^{2\alpha}120^{2\alpha-2}\gamma^{2\alpha}\sigma^{2\alpha}\exp(-\gamma\mu(T-3))\ln^{2\alpha-2}\frac{4(K+1)}{\beta}(K+1)\exp(\gamma\mu\alpha K)}{R^{2\alpha-2}} \\
&\overset{(47)}{\leq} \frac{1}{5}\exp(-\gamma\mu T)R^2.
\end{aligned}
\tag{75}
$$

That is, we derive the upper bounds for ①, ②, ③, ④, ⑤. More precisely, $E_{T-1}$ implies

$$
R_T^2 \overset{(60)}{\leq} \exp(-\gamma\mu T)R^2 + ① + ② + ③ + ④ + ⑤,
$$

$$
② \overset{(69)}{\leq} \frac{1}{5}\exp(-\gamma\mu T)R^2,\quad ③\overset{(70)}{\leq}\frac{1}{5}\exp(-\gamma\mu T)R^2,\quad ⑤\overset{(75)}{\leq}\frac{1}{5}\exp(-\gamma\mu T)R^2,
$$

$$
\sum_{t=0}^{T-1}\sigma_t^2 \overset{(68)}{\leq} \frac{\exp(-2\gamma\mu T)R^4}{150\ln\frac{4(K+1)}{\beta}},\quad \sum_{t=0}^{T-1}\widetilde{\sigma}_t^2 \overset{(74)}{\leq} \frac{\exp(-2\gamma\mu T)R^4}{150\ln\frac{4(K+1)}{\beta}}.
$$

In addition, we also establish (see (67), (73) and our induction assumption)

$$
\mathbb{P}\{E_{T-1}\} \geq 1 - \frac{(T-1)\beta}{K+1},
$$

$$
\mathbb{P}\{E_①\} \geq 1 - \frac{\beta}{2(K+1)},\quad \mathbb{P}\{E_④\}\geq 1 - \frac{\beta}{2(K+1)}.
$$

where

$$
\begin{aligned}
E_① &= \left\{\text{either}\quad \sum_{t=0}^{T-1}\sigma_t^2 > \frac{\exp(-2\gamma\mu T)R^4}{150\ln\frac{4(K+1)}{\beta}}\quad\text{or}\quad |①|\leq\frac{1}{5}\exp(-\gamma\mu T)R^2\right\}, \\
E_④ &= \left\{\text{either}\quad \sum_{t=0}^{T-1}\widetilde{\sigma}_t^2 > \frac{\exp(-2\gamma\mu T)R^4}{150\ln\frac{4(K+1)}{\beta}}\quad\text{or}\quad |④|\leq\frac{1}{5}\exp(-\gamma\mu T)R^2\right\}.
\end{aligned}
$$

Therefore, probability event $E_{T-1}\cap E_①\cap E_④$ implies

$$
\begin{aligned}
R_T^2 &\overset{(60)}{\leq} \exp(-\gamma\mu T)R^2 + ① + ② + ③ + ④ + ⑤ \\
&\leq 2\exp(-\gamma\mu T)R^2,
\end{aligned}
$$

which is equivalent to (54) for $t = T$. Moreover,

$$\mathbb{P}\{E_T\} \geq \mathbb{P}\{E_{T-1} \cap E_① \cap E_④\} = 1 - \mathbb{P}\{\overline{E}_{T-1} \cup \overline{E}_① \cup \overline{E}_④\} \geq 1 - \frac{T\beta}{K+1}.$$

In other words, we showed that $\mathbb{P}\{E_k\} \geq 1 - {}^{k\beta}/(K+1)$ for all $k = 0, 1, \ldots, K+1$. For $k = K+1$ we have that with probability at least $1 - \beta$

$$\|x^{K+1} - x^*\|^2 \leq 2\exp(-\gamma\mu(K+1))R^2.$$

Finally, if

$$\gamma = \min\left\{\frac{1}{400\ell \ln\frac{4(K+1)}{\beta}}, \frac{\ln(B_K)}{\mu(K+1)}\right\},$$

$$B_K = \max\left\{2, \frac{(K+1)^{\frac{2\alpha-1}{\alpha}}\mu^2 R^2}{4 \cdot 10^{\frac{1}{\alpha}}120^{\frac{2(\alpha-1)}{\alpha}}\sigma^2 \ln^{\frac{2(\alpha-1)}{\alpha}}\left(\frac{4(K+1)}{\beta}\right)\ln^2(B_K)}\right\}$$

$$= \mathcal{O}\left(\max\left\{2, \frac{K^{\frac{2\alpha-1}{\alpha}}\mu^2 R^2}{\sigma^2 \ln^{\frac{2(\alpha-1)}{\alpha}}\left(\frac{K}{\beta}\right)\ln^2\left(\max\left\{2, \frac{K^{\frac{2\alpha-1}{\alpha}}\mu^2 R^2}{\sigma^2 \ln^{\frac{2(\alpha-1)}{\alpha}}\left(\frac{K}{\beta}\right)}\right\}\right)}\right\}\right)$$

then with probability at least $1 - \beta$

$$\|x^{K+1} - x^*\|^2 \leq 2\exp(-\gamma\mu(K+1))R^2$$

$$= 2R^2\max\left\{\exp\left(-\frac{\mu(K+1)}{400\ell \ln\frac{4(K+1)}{\beta}}\right), \frac{1}{B_K}\right\}$$

$$= \mathcal{O}\left(\max\left\{R^2\exp\left(-\frac{\mu K}{\ell \ln\frac{K}{\beta}}\right), \frac{\sigma^2\left(\frac{K}{\beta}\right)\ln^2\left(\max\left\{2, \frac{K^{\frac{2\alpha-1}{\alpha}}\mu^2 R^2}{\sigma^2 \ln^{\frac{2(\alpha-1)}{\alpha}}\left(\frac{K}{\beta}\right)}\right\}\right)}{\ln^{\frac{2(1-\alpha)}{\alpha}}K^{\frac{2\alpha-1}{\alpha}}\mu^2}\right\}\right).$$

To get $\|x^{K+1} - x^*\|^2 \leq \varepsilon$ with probability at least $1 - \beta$, $K$ should be

$$K = \mathcal{O}\left(\frac{\ell}{\mu}\ln\left(\frac{R^2}{\varepsilon}\right)\ln\left(\frac{\ell}{\mu\beta}\ln\frac{R^2}{\varepsilon}\right), \left(\frac{\sigma^2}{\mu^2\varepsilon}\right)^{\frac{\alpha}{2\alpha-1}}\ln\left(\frac{1}{\beta}\left(\frac{\sigma^2}{\mu^2\varepsilon}\right)^{\frac{\alpha}{2\alpha-1}}\right)\ln^{\frac{\alpha}{\alpha-1}}(B_\varepsilon)\right),$$

where

$$B_\varepsilon = \max\left\{2, \frac{R^2}{\varepsilon \ln\left(\frac{1}{\beta}\left(\frac{\sigma^2}{\mu^2\varepsilon}\right)^{\frac{\alpha}{2\alpha-1}}\right)}\right\}.$$

$\square$

# E    MISSING PROOFS FOR DProx-clipped-SGD-shift

In this section, we give the complete formulations of our results for DProx-clipped-SGD-shift and rigorous proofs. For the readers' convenience, the method's update rule is repeated below:

$$x^{k+1} = \text{prox}_{\gamma\Psi}\left(x^k - \gamma\widetilde{g}^k\right), \quad \text{where } \widetilde{g}^k = \frac{1}{n}\sum_{i=1}^{n}\widetilde{g}_i^k, \quad \widetilde{g}_i^k = h_i^k + \hat{\Delta}_i^k,$$

$$h_i^{k+1} = h_i^k + \nu\hat{\Delta}_i^k, \quad \hat{\Delta}_i^k = \texttt{clip}\left(\nabla f_{\xi_i^k}(x^k) - h_i^k, \lambda_k\right).$$

**Lemma E.1.** *Let Assumptions 2 and 3 with $\mu = 0$ hold on $Q = B_{3n\sqrt{V}}(x^*)$, where $V \geq \|x^0 - x^*\|^2 + \frac{36864\gamma^2\ln^2\frac{48n(K+1)}{\beta}}{n^2}\sum_{i=1}^{n}\|\nabla f_i(x^*)\|^2$, and let stepsize $\gamma$ satisfy $\gamma \leq \frac{1}{L}$. If $x^k \in Q$ for all $k = 0, 1, \ldots, K+1$, $K \geq 0$, then after $K$ iterations of DProx-clipped-SGD-shift we have*

$$2\gamma\left(\Phi(\overline{x}^{K+1}) - \Phi(x^*)\right) \leq \frac{\|x^0 - x^*\|^2 - \|x^{K+1} - x^*\|^2}{K+1}$$

$$-\frac{2\gamma}{K+1}\sum_{k=0}^{K}\langle\omega_k, \hat{x}^k - x^*\rangle + \frac{2\gamma^2}{K+1}\sum_{k=0}^{K}\|\omega_k\|^2, \quad (76)$$

$$\overline{x}^{K+1} \stackrel{def}{=} \frac{1}{K+1}\sum_{k=0}^{K}x^{k+1}, \quad (77)$$

$$\hat{x}^k \stackrel{def}{=} \text{prox}_{\gamma\Psi}\left(x^k - \gamma\nabla f(x^k)\right), \quad (78)$$

$$\omega_k \stackrel{def}{=} \nabla f(x^k) - \widetilde{g}^k. \quad (79)$$

*Proof.* Using Lemma C.2 from (Khaled et al., 2020) with $p = x^{k+1}$, $y = x^k - \gamma\widetilde{g}^k$, $x = x^k$, we derive for all $k = 0, 1, \ldots, K$ that

$$2\gamma\left(\Phi(x^{k+1}) - \Phi(x^*)\right) \leq \|x^k - x^*\|^2 - \|x^{k+1} - x^*\|^2 - 2\gamma\langle\widetilde{g}^k - \nabla f(x^k), x^{k+1} - x^*\rangle.$$

Next, we obtain the following inequality

$$-2\gamma\langle\widetilde{g}^k - \nabla f(x^k), x^{k+1} - x^*\rangle = -2\gamma\langle\widetilde{g}^k - \nabla f(x^k), \hat{x}^k - x^*\rangle + 2\gamma\langle\widetilde{g}^k - \nabla f(x^k), \hat{x}^k - x^{k+1}\rangle$$

$$\stackrel{(79)}{\leq} -2\gamma\langle\omega_k, \hat{x}^k - x^*\rangle + 2\gamma\|\widetilde{g}^k - \nabla f(x^k)\| \cdot \|\hat{x}^k - x^{k+1}\|$$

$$\stackrel{(78)}{=} -2\gamma\langle\omega_k, \hat{x}^k - x^*\rangle + 2\gamma\|\widetilde{g}^k - \nabla f(x^k)\|$$

$$\cdot \|\text{prox}_{\gamma\Psi}\left(x^k - \gamma\nabla f(x^k)\right) - \text{prox}_{\gamma\Psi}\left(x^k - \gamma\widetilde{g}^k\right)\|$$

$$\stackrel{(79)}{\leq} -2\gamma\langle\omega_k, \hat{x}^k - x^*\rangle + 2\gamma^2\|\omega_k\|^2.$$

Putting all together we get

$$2\gamma\left(\Phi(x^{k+1}) - \Phi(x^*)\right) \leq \|x^k - x^*\|^2 - \|x^{k+1} - x^*\|^2 - 2\gamma\langle\omega_k, \hat{x}^k - x^*\rangle + 2\gamma^2\|\omega_k\|^2.$$

Summing up the above inequalities for $k = 0, 1, \ldots, K$, we get

$$\frac{2\gamma}{K+1}\sum_{k=0}^{K}\left(\Phi(x^{k+1}) - \Phi(x^*)\right) \leq \frac{1}{K+1}\sum_{k=0}^{K}\left(\|x^k - x^*\|^2 - \|x^{k+1} - x^*\|^2\right)$$

$$-\frac{2\gamma}{K+1}\sum_{k=0}^{K}\langle\omega_k, \hat{x}^k - x^*\rangle + \frac{2\gamma^2}{K+1}\sum_{k=0}^{K}\|\omega_k\|^2$$

$$= \frac{\|x^0 - x^*\|^2 - \|x^{K+1} - x^*\|^2}{K+1} - \frac{2\gamma}{K+1}\sum_{k=0}^{K}\langle\omega_k, \hat{x}^k - x^*\rangle$$

$$+ \frac{2\gamma^2}{K+1}\sum_{k=0}^{K}\|\omega_k\|^2.$$

Finally, we use the definition of $\overline{x}^K$ and Jensen's inequality and get the result.    □

**Theorem E.1.** *Let Assumptions 2 and 3 with $\mu = 0$ hold on $Q = B_{3n\sqrt{V}}(x^*)$, where $V \geq \|x^0 - x^*\|^2 + \frac{36864\gamma^2 \ln^2 \frac{48n(K+1)}{\beta}}{n^2} \sum_{i=1}^{n} \|\nabla f_i(x^*)\|^2$, and $\nu = 0$, $h_1^0 = \ldots = h_n^0 = 0$,*

$$\gamma \leq \min \left\{ \frac{1}{360 L \ln \frac{48n(K+1)}{\beta}}, \frac{R\sqrt{n}}{192 A\zeta_*}, \frac{\sqrt{V} n^{\frac{\alpha-1}{\alpha}}}{27^{\frac{1}{\alpha}} \cdot 48\sigma K^{\frac{1}{\alpha}} \left( \ln \frac{48n(K+1)}{\beta} \right)^{\frac{\alpha-1}{\alpha}}} \right\}, \quad (80)$$

$$\lambda_k = \lambda = \frac{n\sqrt{V}}{48\gamma \ln \frac{48n(K+1)}{\beta}}, \quad (81)$$

*for some $\zeta_* = \sqrt{\frac{1}{n} \sum_{i=1}^{n} \|\nabla f_i(x^*)\|^2}$, $K + 1 > 0$ and $\beta \in (0, 1]$. Then, after $K + 1$ iterations of* DProx-clipped-SGD-shift *the iterates with probability at least $1 - \beta$ satisfy*

$$\Phi(\overline{x}^{K+1}) - \Phi(x^*) \leq \frac{V}{\gamma(K+1)} \quad and \quad \{x^k\}_{k=0}^{K+1} \subseteq B_{3n\sqrt{V}}(x^*). \quad (82)$$

*In particular, we have $V \leq 2R^2$, and when $\gamma$ equals the minimum from (80), then the iterates produced by* DProx-clipped-SGD-shift *after $K + 1$ iterations with probability at least $1 - \beta$ satisfy*

$$\Phi(\overline{x}^{K+1}) - \Phi(x^*) = \mathcal{O}\left( \max \left\{ \frac{LR^2 \ln \frac{nK}{\beta}}{K}, \frac{R\zeta_* \ln \frac{nK}{\beta}}{\sqrt{n}K}, \frac{\sigma R \ln^{\frac{\alpha-1}{\alpha}} \frac{nK}{\beta}}{n^{\frac{\alpha-1}{\alpha}} K^{\frac{\alpha-1}{\alpha}}} \right\} \right), \quad (83)$$

*meaning that to achieve $\Phi(\overline{x}^{K+1}) - \Phi(x^*) \leq \varepsilon$ with probability at least $1 - \beta$* DProx-clipped-SGD-shift *requires*

$$K = \mathcal{O}\left( \max \left\{ \frac{LR^2}{\varepsilon} \ln \frac{nLR^2}{\varepsilon}, \frac{R\zeta_*}{\sqrt{n}\varepsilon} \ln \frac{\sqrt{n}R\zeta_*}{\varepsilon}, \left( \frac{\sigma\sqrt{V}}{\varepsilon n^{\frac{\alpha-1}{\alpha}}} \right)^{\frac{\alpha}{\alpha-1}} \ln \left( \frac{1}{\beta} \left( \frac{\sigma\sqrt{V}}{\varepsilon} \right)^{\frac{\alpha}{\alpha-1}} \right) \right\} \right) \quad (84)$$

*iterations/oracle calls.*

*Proof.* The key idea behind the proof is similar to the one used in (Gorbunov et al., 2022a; Sadiev et al., 2023): we prove by induction that the iterates do not leave some ball and the sums decrease as $1/K+1$. To formulate the statement rigorously, we introduce probability event $E_k$ for each $k = 0, 1, \ldots, K+1$ as follows: inequalities

$$\underbrace{\|x^0 - x^*\|^2 - 2\gamma \sum_{l=0}^{t-1} \langle \omega_l, \hat{x}^l - x^* \rangle + 2\gamma^2 \sum_{l=0}^{t-1} \|\omega_l\|^2}_{A_t} \leq 2V, \quad (85)$$

$$\left\| \frac{\gamma}{n} \sum_{i=1}^{r-1} \omega_{i,t-1}^u \right\| \leq \frac{\sqrt{V}}{2} \quad (86)$$

hold for $t = 0, 1, \ldots, k$ and $r = 1, 2, \ldots, n$ simultaneously, where

$$\omega_l = \omega_l^u + \omega_l^b, \quad (87)$$

$$\omega_l^u \stackrel{\text{def}}{=} \frac{1}{n} \sum_{i=1}^{n} \omega_{i,l}^u, \quad \omega_l^b \stackrel{\text{def}}{=} \frac{1}{n} \sum_{i=1}^{n} \omega_{i,l}^b, \quad (88)$$

$$\omega_{i,l}^u \stackrel{\text{def}}{=} \mathbb{E}_{\xi_i^l} \left[ \widetilde{g}_i^l \right] - \widetilde{g}_i^l, \quad \omega_{i,l}^b \stackrel{\text{def}}{=} \nabla f(x^l) - \mathbb{E}_{\xi_i^l} \left[ \widetilde{g}_i^l \right] \quad \forall i \in [n]. \quad (89)$$

We will prove by induction that $\mathbb{P}\{E_k\} \geq 1 - k\beta/(K+1)$ for all $k = 0, 1, \ldots, K+1$. The base of induction follows immediately: $\|x^0 - x^*\|^2 \leq V < 2V$ and for $k = 0$ we have $\|\frac{\gamma}{n} \sum_{i=1}^{r-1} \omega_{i,k-1}^u\| = 0$ since $\omega_{i,-1}^u = 0$. Next, we assume that the statement holds for $k = T - 1 \leq K$, i.e., $\mathbb{P}\{E_{T-1}\} \geq 1 - (T-1)\beta/(K+1)$. Let us show that it also holds for $k = T$, i.e., $\mathbb{P}\{E_T\} \geq 1 - T\beta/(K+1)$.

To proceed, we need to show that $E_{T-1}$ implies $\|x^t - x^*\| \le 3n\sqrt{V}$ for all $t = 0, 1, \ldots, T$. First, for $t = 0, 1, \ldots, T - 1$ probability event $E_{T-1}$ implies (in view of, $\Phi(\overline{x}^t) - \Phi(x^*) \ge 0$)

$$\|x^t - x^*\|^2 \overset{(76)}{\le} A_t \overset{(85)}{\le} 2V. \tag{90}$$

Next, by definition of $V$ we have

$$\|\nabla f(x^*)\| = \sqrt{\|\nabla f(x^*)\|^2} \le \sqrt{\sum_{i=1}^n \|\nabla f_i(x^*)\|^2} \le \frac{n\sqrt{V}}{192\gamma \ln \frac{48n(K+1)}{\beta}}. \tag{91}$$

Then, for $t = T$ we have that $E_{T-1}$ implies

$$
\begin{aligned}
\|x^T - x^*\| &= \|\operatorname{prox}_{\gamma\Psi}(x^k - \gamma\widetilde{g}^k) - \operatorname{prox}_{\gamma\Psi}(x^* - \gamma\nabla f(x^*))\| \\
&\le \|x^k - \gamma\widetilde{g}^k - x^* + \gamma\nabla f(x^*)\| \le \|x^k - x^*\| + \gamma\|\widetilde{g}^k\| + \gamma\|\nabla f(x^*)\| \\
&\overset{(90),(91)}{\le} \left(\sqrt{2} + \frac{n}{192\ln \frac{48n(K+1)}{\beta}}\right)\sqrt{V} + \gamma\lambda \overset{(81)}{\le} 3n\sqrt{V}.
\end{aligned}
$$

This means that $E_{T-1}$ implies $x^t \in B_{3n\sqrt{V}}(x^*)$ for $t = 0, 1, \ldots, T$ and we can apply Lemma E.1: $E_{T-1}$ implies

$$
\begin{aligned}
2\gamma\left(\Phi(\overline{x}^T) - \Phi(x^*)\right) &\le \frac{\|x^0 - x^*\|^2 - \|x^T - x^*\|^2}{T} \\
&\quad - \frac{2\gamma}{T}\sum_{l=0}^{T-1}\langle\omega_l, \hat{x}^l - x^*\rangle + \frac{2\gamma^2}{T}\sum_{l=0}^{T-1}\|\omega_l\|^2 \\
&\le \frac{A_T}{T}. \tag{92}
\end{aligned}
$$

Before we proceed, we introduce a new notation:

$$\eta_t = \begin{cases} \hat{x}^t - x^*, & \text{if } \|\hat{x}^t - x^*\| \le 2\sqrt{V}, \\ 0, & \text{otherwise,} \end{cases}$$

for all $t = 0, 1, \ldots, T - 1$. Random vectors $\{\eta_t\}_{t=0}^T$ are bounded almost surely:

$$\|\eta_t\| \le 2\sqrt{V}. \tag{93}$$

for all $t = 0, 1, \ldots, T - 1$. In addition, $E_{T-1}$ implies for all $t = 0, 1, \ldots, T - 1$ that

$$
\begin{aligned}
\|\hat{x}^t - x^*\| &= \|\operatorname{prox}_{\gamma\Psi}\left(x^t - \gamma\nabla f(x^t)\right) - \operatorname{prox}_{\gamma\Psi}\left(x^* - \gamma\nabla f(x^*)\right)\| \\
&\le \|x^t - x^* - \gamma(\nabla f(x^t) - \nabla f(x^*))\| \\
&\le \|x^t - x^*\| + \gamma\|\nabla f(x^t) - \nabla f(x^*)\| \\
&\overset{(4)}{\le} (1 + L\gamma)\|x^t - x^*\| \overset{(80)}{\le} \frac{361}{360}\|x^t - x^*\| \overset{(90)}{\le} 2\sqrt{V}.
\end{aligned}
$$

meaning that $\eta_t = \hat{x}^t - x^*$ follows from $E_{T-1}$ for all $t = 0, 1, \ldots, T - 1$. Thus, $E_{T-1}$ implies

$$
\begin{aligned}
A_T &\overset{(85)}{=} \|x^0 - x^*\|^2 - 2\gamma\sum_{l=0}^{T-1}\langle\omega_l, \hat{x}^l - x^*\rangle + 2\gamma^2\sum_{l=0}^{T-1}\|\omega_l\|^2 \\
&\le V - 2\gamma\sum_{l=0}^{T-1}\langle\omega_l, \eta_l\rangle + 2\gamma^2\sum_{l=0}^{T-1}\|\omega_l\|^2. \tag{94}
\end{aligned}
$$

Using the notation from (87)-(89), we can rewrite $\|\omega_l\|^2$ as

$$
\begin{aligned}
\|\omega_l\|^2 &\le 2\|\omega_l^u\|^2 + 2\|\omega_l^b\|^2 = \frac{2}{n^2}\left\|\sum_{i=1}^n \omega_{i,l}^u\right\|^2 + 2\|\omega_l^b\|^2 \\
&= \frac{2}{n^2}\sum_{i=1}^n \|\omega_{i,l}^u\|^2 + \frac{4}{n^2}\sum_{j=2}^n\left\langle\sum_{i=1}^{j-1}\omega_{i,l}^u, \omega_{j,l}^u\right\rangle + 2\|\omega_l^b\|^2. \tag{95}
\end{aligned}
$$

Putting all together, we obtain that $E_{T-1}$ implies

$$
\begin{aligned}
A_T \;\leq\; & V - \underbrace{\frac{2\gamma}{n}\sum_{l=0}^{T-1}\sum_{i=1}^{n}\langle\omega_{i,l}^u,\eta_l\rangle}_{\text{①}} - \underbrace{2\gamma\sum_{l=0}^{T-1}\langle\omega_l^b,\eta_l\rangle}_{\text{②}} + \underbrace{\frac{4\gamma^2}{n^2}\sum_{l=0}^{T-1}\sum_{i=1}^{n}\left(\|\omega_{i,l}^u\|^2 - \mathbb{E}_{\xi_i^l}\left[\|\omega_{i,l}^u\|^2\right]\right)}_{\text{③}} \\
& + \underbrace{\frac{4\gamma^2}{n^2}\sum_{l=0}^{T-1}\sum_{i=1}^{n}\mathbb{E}_{\xi_i^l}\left[\|\omega_{i,l}^u\|^2\right]}_{\text{④}} + \underbrace{4\gamma^2\sum_{l=0}^{T-1}\|\omega_l^b\|^2}_{\text{⑤}} + \underbrace{\frac{8\gamma^2}{n^2}\sum_{l=0}^{T-1}\sum_{j=2}^{n}\left\langle\sum_{i=1}^{j-1}\omega_{i,l}^u,\omega_{j,l}^u\right\rangle}_{\text{⑥}}. \quad (96)
\end{aligned}
$$

To finish the proof, it remains to estimate ①, ②, ③, ④, ⑤, ⑥ with high probability. More precisely, the goal is to prove that ① + ② + ③ + ④ + ⑤ + ⑥ ≤ $V$ with high probability. Before we proceed, we need to derive several useful inequalities related to $\omega_{i,l}^u, \omega_l^b$. First of all, we have

$$
\|\omega_{i,l}^u\| \leq 2\lambda \tag{97}
$$

by definition of the clipping operator. Next, probability event $E_{T-1}$ implies

$$
\begin{aligned}
\|\nabla f_i(x^t)\| \;\leq\; & \|\nabla f_i(x^t) - \nabla f_i(x^*)\| + \|\nabla f_i(x^*)\| \overset{(4)}{\leq} L\|x^t - x^*\| + \sqrt{\sum_{i=1}^{n}\|\nabla f_i(x^*)\|^2} \\
\leq\; & \sqrt{2}L\sqrt{V} + \frac{n\sqrt{V}}{192\gamma\ln\frac{48n(K+1)}{\beta}} \leq \frac{n\sqrt{V}}{96\gamma\ln\frac{48n(K+1)}{\beta}} \leq \frac{\lambda}{2}. \quad (98)
\end{aligned}
$$

for $t = 0, 1, \ldots, T-1$ and $i \in [n]$. Therefore, Lemma B.2 and $E_{T-1}$ imply

$$
\|\omega_l^b\| \leq \frac{1}{n}\sum_{i=1}^{n}\|\omega_{i,l}^b\| \leq \frac{2^\alpha\sigma^\alpha}{\lambda^{\alpha-1}}, \tag{99}
$$

$$
\mathbb{E}_{\xi_i^l}\left[\|\omega_{i,l}^u\|^2\right] \leq 18\lambda^{2-\alpha}\sigma^\alpha, \tag{100}
$$

for all $l = 0, 1, \ldots, T-1$ and $i \in [n]$.

**Upper bound for ①.** To estimate this sum, we will use Bernstein's inequality. The summands have conditional expectations equal to zero:

$$
\mathbb{E}_{\xi_i^l}\left[-\frac{2\gamma}{n}\langle\omega_{i,l}^u,\eta_l\rangle\right] = -\frac{2\gamma}{n}\left\langle\eta_l,\mathbb{E}_{\xi_i^l}[\omega_{i,l}^u]\right\rangle = 0.
$$

Moreover, for all $l = 0, \ldots, T-1$ random vectors $\{\omega_{i,l}^u\}_{i=1}^n$ are independent. Thus, sequence $\left\{-\frac{2\gamma}{n}\langle\eta_l,\omega_{i,l}^u\rangle\right\}_{l,i=0,1}^{T-1,n}$ is a martingale difference sequence. Next, the summands are bounded:

$$
\left|\frac{2\gamma}{n}\langle\omega_{i,l}^u,\eta_l\rangle\right| \leq \frac{2\gamma}{n}\|\omega_{i,l}^u\|\cdot\|\eta_l\| \overset{(93),(97)}{\leq} \frac{8\gamma\lambda\sqrt{V}}{n} \overset{(81)}{=} \frac{V}{6\ln\frac{48n(K+1)}{\beta}} \overset{\text{def}}{=} c. \tag{101}
$$

Finally, conditional variances $\sigma_{i,l}^2 \overset{\text{def}}{=} \mathbb{E}_{\xi_i^l}[\frac{4\gamma^2}{n^2}\langle\omega_{i,l}^u,\eta_l\rangle^2]$ of the summands are bounded:

$$
\sigma_{i,t}^2 \leq \mathbb{E}_{\xi_i^t}\left[\frac{4\gamma^2}{n^2}\|\omega_{i,t}^u\|^2\cdot\|\eta_t\|^2\right] \overset{(93)}{\leq} \frac{16\gamma^2 V}{n^2}\mathbb{E}_{\xi_i^t}\left[\|\omega_{i,t}^u\|^2\right]. \tag{102}
$$

Applying Bernstein's inequality (Lemma B.1) with $X_{i,l} = -\frac{2\gamma}{n}\langle\eta_l,\omega_{i,l}^u\rangle$, constant $c$ defined in (101), $b = \frac{V}{6}$, $G = \frac{V^2}{216\ln\frac{48n(K+1)}{\beta}}$, we get

$$
\mathbb{P}\left\{|\text{①}| > \frac{V}{6} \quad\text{and}\quad \sum_{l=0}^{T-1}\sum_{i=1}^{n}\sigma_{i,l}^2 \leq \frac{V^2}{216\ln\frac{48n(K+1)}{\beta}}\right\} \leq 2\exp\left(-\frac{b^2}{2G + 2cb/24n}\right) = \frac{\beta}{24n(K+1)}.
$$

The above is equivalent to

$$\mathbb{P}\left\{E_{①}\right\} \geq 1 - \frac{\beta}{24n(K+1)}, \quad \text{for} \quad E_{①} = \left\{\text{either} \quad \sum_{l=0}^{T-1}\sum_{i=1}^{n} \sigma_{i,l}^2 > \frac{V^2}{216 \ln \frac{48n(K+1)}{\beta}} \quad \text{or} \quad |①| \leq \frac{V}{6}\right\}.$$

(103)

Moreover, $E_{T-1}$ implies

$$\sum_{l=0}^{T-1}\sum_{i=1}^{n} \sigma_{i,l}^2 \overset{(102)}{\leq} \frac{16\gamma^2 V}{n^2}\sum_{l=0}^{T-1}\sum_{i=1}^{n}\mathbb{E}_{\xi_i^l}\left[\|\omega_{i,l}^u\|^2\right] \overset{(100)}{\leq} \frac{288\gamma^2 V \sigma^\alpha T \lambda^{2-\alpha}}{n}$$

$$\overset{(81)}{=} \frac{48^\alpha \sqrt{V}^{4-\alpha}\sigma^\alpha T \gamma^\alpha}{8n^{\alpha-1}\ln^{2-\alpha}\frac{48n(K+1)}{\beta}} \overset{(80)}{\leq} \frac{V^2}{216 \ln \frac{48n(K+1)}{\beta}}.$$

(104)

**Upper bound for ②.** Probability event $E_{T-1}$ implies

$$② = -2\gamma \sum_{l=0}^{T-1}\langle \omega_l^b, \eta_l\rangle \leq 2\gamma \sum_{l=0}^{T-1}\|\omega_l^b\| \cdot \|\eta_l\| \overset{(93),(99)}{\leq} \frac{4 \cdot 2^\alpha \gamma \sigma^\alpha T \sqrt{V}}{\lambda^{\alpha-1}}$$

$$\overset{(81)}{=} \frac{96^\alpha}{12} \cdot \frac{\sigma^\alpha T \sqrt{V}^{2-\alpha}\gamma^\alpha}{n^{\alpha-1}\ln^{1-\alpha}\frac{48n(K+1)}{\beta}} \overset{(80)}{\leq} \frac{V}{6}.$$

(105)

**Upper bound for ③.** To estimate this sum, we will use Bernstein's inequality. The summands have conditional expectations equal to zero:

$$\mathbb{E}_{\xi_i^l}\left[\frac{4\gamma^2}{n^2}\left(\|\omega_{i,l}^u\|^2 - \mathbb{E}_{\xi_i^l}\left[\|\omega_{i,l}^u\|^2\right]\right)\right] = 0.$$

Moreover, for all $l = 0, \ldots, T-1$ random vectors $\{\omega_{i,l}^u\}_{i=1}^n$ are independent. Thus, sequence $\left\{\frac{4\gamma^2}{n^2}\left(\|\omega_{i,l}^u\|^2 - \mathbb{E}_{\xi_i^l}\left[\|\omega_{i,l}^u\|^2\right]\right)\right\}_{l,i=0,1}^{T-1,n}$ is a martingale difference sequence. Next, the summands are bounded:

$$\left|\frac{4\gamma^2}{n^2}\left(\|\omega_{i,l}^u\|^2 - \mathbb{E}_{\xi_i^l}\left[\|\omega_{i,l}^u\|^2\right]\right)\right| \leq \frac{4\gamma^2}{n^2}\left(\|\omega_{i,l}^u\|^2 + \mathbb{E}_{\xi_i^l}\left[\|\omega_{i,l}^u\|^2\right]\right)$$

$$\overset{(97)}{\leq} \frac{32\gamma^2\lambda^2}{n^2} \overset{(81)}{=} \frac{V}{72 \ln^2 \frac{48n(K+1)}{\beta}}$$

$$\leq \frac{V}{12 \ln \frac{48n(K+1)}{\beta}} \overset{\text{def}}{=} c.$$

(106)

Finally, conditional variances

$$\widetilde{\sigma}_{i,t}^2 \overset{\text{def}}{=} \mathbb{E}_{\xi_i^t}\left[\frac{16\gamma^4}{n^4}\left(\|\omega_{i,t}^u\|^2 - \mathbb{E}_{\xi_i^t}\left[\|\omega_{i,t}^u\|^2\right]\right)^2\right]$$

of the summands are bounded:

$$\widetilde{\sigma}_{i,t}^2 \overset{(106)}{\leq} \frac{V}{12 \ln \frac{48n(K+1)}{\beta}}\mathbb{E}_{\xi_i^t}\left[\frac{4\gamma^2}{n^2}\left|\|\omega_{i,t}^u\|^2 - \mathbb{E}_{\xi_i^t}\left[\|\omega_{i,t}^u\|^2\right]\right|\right]$$

$$\leq \frac{\gamma^2 V}{3n^2 \ln \frac{48n(K+1)}{\beta}}\mathbb{E}_{\xi_i^t}\left[\|\omega_{i,t}^u\|^2\right].$$

(107)

Applying Bernstein's inequality (Lemma B.1) with $X_{i,l} = \frac{4\gamma^2}{n^2}\left(\|\omega_{i,l}^u\|^2 - \mathbb{E}_{\xi_i^l}\left[\|\omega_{i,l}^u\|^2\right]\right)$, constant $c$ defined in (106), $b = \frac{V}{12}$, $G = \frac{V^2}{864 \ln \frac{48n(K+1)}{\beta}}$, we get

$$\mathbb{P}\left\{|③| > \frac{V}{12} \quad \text{and} \quad \sum_{l=0}^{T-1}\sum_{i=1}^{n}\widetilde{\sigma}_{i,l}^2 \leq \frac{V^2}{864 \ln \frac{48n(K+1)}{\beta}}\right\} \leq 2\exp\left(-\frac{b^2}{2G + 2cb/3}\right) = \frac{\beta}{24n(K+1)}.$$

The above is equivalent to

$$\mathbb{P}\left\{E_{③}\right\} \geq 1 - \frac{\beta}{24n(K+1)}, \quad \text{for} \quad E_{③} = \left\{ \text{either} \quad \sum_{l=0}^{T-1}\sum_{i=1}^{n} \widetilde{\sigma}_{i,l}^2 > \frac{V^2}{864 \ln \frac{48n(K+1)}{\beta}} \quad \text{or} \quad |③| \leq \frac{V}{12} \right\}.$$
(108)

Moreover, $E_{T-1}$ implies

$$\sum_{l=0}^{T-1}\sum_{i=1}^{n} \widetilde{\sigma}_{i,l}^2 \overset{(107)}{\leq} \frac{\gamma^2 V}{3n^2 \ln \frac{48n(K+1)}{\beta}} \sum_{l=0}^{T-1}\sum_{i=1}^{n} \mathbb{E}_{\xi_i^l}\left[\|\omega_{i,l}^u\|^2\right] \overset{(100)}{\leq} \frac{18\gamma^2 V \lambda^{2-\alpha}\sigma^\alpha T}{3n \ln \frac{48n(K+1)}{\beta}}$$

$$\overset{(81)}{=} \frac{48^\alpha \cdot 6}{48^2} \cdot \frac{\sigma^\alpha T \sqrt{V}^{4-\alpha}\gamma^\alpha}{n^{\alpha-1}\ln^{3-\alpha}\frac{48n(K+1)}{\beta}} \overset{(80)}{\leq} \frac{V^2}{864 \ln \frac{48n(K+1)}{\beta}}.$$
(109)

**Upper bound for ④.** Probability event $E_{T-1}$ implies

$$④ = \frac{4\gamma^2}{n^2}\sum_{l=0}^{T-1}\sum_{i=1}^{n}\mathbb{E}_{\xi_i^l}\left[\|\omega_{i,l}^u\|^2\right] \overset{(100)}{\leq} \frac{72\gamma^2\lambda^{2-\alpha}\sigma^\alpha T}{n} \overset{(81)}{=} \frac{48^\alpha \gamma^\alpha \sigma^\alpha T \sqrt{V}^{2-\alpha}}{32n^{\alpha-1}\ln^{2-\alpha}\frac{48n(K+1)}{\beta}} \overset{(80)}{\leq} \frac{V}{12}. \quad (110)$$

**Upper bound for ⑤.** Probability event $E_{T-1}$ implies

$$⑤ = 4\gamma^2 \sum_{l=0}^{T-1}\left\|\omega_l^b\right\|^2 \overset{(99)}{\leq} \frac{4^{(\alpha+1)}\sigma^{2\alpha}T\gamma^2}{\lambda^{2(\alpha-1)}} \overset{(81)}{=} \frac{9216^\alpha}{576} \cdot \frac{\sigma^{2\alpha}T\gamma^{2\alpha}\sqrt{V}^{2(1-\alpha)}}{n^{2(1-\alpha)}\ln^{2(1-\alpha)}\frac{48n(K+1)}{\beta}} \overset{(80)}{\leq} \frac{V}{6}. \quad (111)$$

**Upper bounds for ⑥.** This sum requires a more refined analysis. We introduce new vectors:

$$\delta_j^l = \begin{cases} \frac{\gamma}{n}\sum_{i=1}^{j-1}\omega_{i,l}^u, & \text{if } \left\|\frac{\gamma}{n}\sum_{i=1}^{j-1}\omega_{i,l}^u\right\| \leq \frac{\sqrt{V}}{2}, \\ 0, & \text{otherwise}, \end{cases}$$
(112)

for all $j \in [n]$ and $l = 0, \ldots, T-1$. Then, by definition

$$\|\delta_j^l\| \leq \frac{\sqrt{V}}{2}$$
(113)

and

$$⑥ = \underbrace{\frac{8\gamma}{n}\sum_{l=0}^{T-1}\sum_{j=2}^{n}\left\langle \delta_j^l, \omega_{j,l}^u \right\rangle}_{⑥'} + \frac{8\gamma}{n}\sum_{l=0}^{T-1}\sum_{j=2}^{n}\left\langle \frac{\gamma}{n}\sum_{i=1}^{j-1}\omega_{i,l}^u - \delta_j^l, \omega_{j,l}^u \right\rangle. \quad (114)$$

We also note here that $E_{T-1}$ implies

$$\frac{8\gamma}{n}\sum_{l=0}^{T-1}\sum_{j=2}^{n}\left\langle \frac{\gamma}{n}\sum_{i=1}^{j-1}\omega_{i,l}^u - \delta_j^l, \omega_{j,l}^u \right\rangle = \frac{8\gamma}{n}\sum_{j=2}^{n}\left\langle \frac{\gamma}{n}\sum_{i=1}^{j-1}\omega_{i,T-1}^u - \delta_j^{T-1}, \omega_{j,T-1}^u \right\rangle. \quad (115)$$

**Upper bound for ⑥'.** To estimate this sum, we will use Bernstein's inequality. The summands have conditional expectations equal to zero:

$$\mathbb{E}_{\xi_j^l}\left[\frac{8\gamma}{n}\left\langle \delta_j^l, \omega_{j,l}^u \right\rangle\right] = \frac{8\gamma}{n}\left\langle \delta_j^l, \mathbb{E}_{\xi_j^l}[\omega_{j,l}^u] \right\rangle = 0.$$

Moreover, for all $l = 0, \ldots, T-1$ random vectors $\{\omega_{i,l}^u\}_{i=1}^n$ are independent. Thus, sequence $\left\{\frac{8\gamma}{n}\left\langle \delta_j^l, \omega_{j,l}^u \right\rangle\right\}_{l,j=0,2}^{T-1,n}$ is a martingale difference sequence. Next, the summands are bounded:

$$\left|\frac{8\gamma}{n}\left\langle \delta_j^l, \omega_{j,l}^u \right\rangle\right| \leq \frac{8\gamma}{n}\left\|\delta_j^l\right\| \cdot \left\|\omega_{j,l}^u\right\| \overset{(113),(97)}{\leq} \frac{8\gamma}{n} \cdot \frac{\sqrt{V}}{2} \cdot 2\lambda = \frac{V}{6 \ln \frac{48n(K+1)}{\beta}} \overset{\text{def}}{=} c. \quad (116)$$

Finally, conditional variances $(\sigma'_{j,l})^2 \stackrel{\text{def}}{=} \mathbb{E}_{\xi_j^l}\left[\frac{64\gamma^2}{n^2}\langle\delta_j^l,\omega_{j,l}^u\rangle^2\right]$ of the summands are bounded:

$$(\sigma'_{j,l})^2 \le \mathbb{E}_{\xi_j^l}\left[\frac{64\gamma^2}{n^2}\|\delta_j^l\|^2\cdot\|\omega_{j,l}^u\|^2\right] \stackrel{(113)}{\le} \frac{16\gamma^2 V}{n^2}\mathbb{E}_{\xi_j^l}\left[\|\omega_{j,l}^u\|^2\right]. \tag{117}$$

Applying Bernstein's inequality (Lemma B.1) with $X_{i,l}=\frac{8\gamma}{n}\left\langle\delta_j^l,\omega_{j,l}^u\right\rangle$, constant $c$ defined in (116), $b=\frac{V}{6}$, $G=\frac{V^2}{216\ln\frac{48n(K+1)}{\beta}}$, we get

$$\mathbb{P}\left\{|\text{\textcircled{6}}'|>\frac{V}{6}\text{ and }\sum_{l=0}^{T-1}\sum_{j=2}^{n}(\sigma'_{i,l})^2\le\frac{V^2}{216\ln\frac{48n(K+1)}{\beta}}\right\}\le 2\exp\left(-\frac{b^2}{2G+{}^{2cb}\!/_3}\right)=\frac{\beta}{24n(K+1)}.$$

The above is equivalent to

$$\mathbb{P}\{E_{\text{\textcircled{6}}'}\}\ge 1-\frac{\beta}{24n(K+1)},\ \text{ for }\ E_{\text{\textcircled{6}}'}=\left\{\text{either }\sum_{l=0}^{T-1}\sum_{j=2}^{n}(\sigma'_{i,l})^2>\frac{V^2}{216\ln\frac{48n(K+1)}{\beta}}\text{ or }|\text{\textcircled{6}}'|\le\frac{V}{6}\right\}. \tag{118}$$

Moreover, $E_{T-1}$ implies

$$\sum_{l=0}^{T-1}\sum_{j=2}^{n}(\sigma'_{j,l})^2 \stackrel{(117)}{\le} \frac{16\gamma^2 V}{n^2}\sum_{l=0}^{T-1}\sum_{j=2}^{n}\mathbb{E}_{\xi_j^l}\left[\|\omega_{j,l}^u\|^2\right] \stackrel{(100),T\le K+1}{\le} \frac{288(K+1)\gamma^2 V\lambda^{2-\alpha}\sigma^\alpha}{n}$$

$$\stackrel{(81)}{\le} \frac{288(K+1)\gamma^\alpha\sigma^\alpha V^{2-\frac{\alpha}{2}}}{48^{2-\alpha}n^{\alpha-1}\ln^{2-\alpha}\frac{48n(K+1)}{\beta}} \stackrel{(80)}{\le} \frac{V^2}{216\ln\frac{48n(K+1)}{\beta}}. \tag{119}$$

That is, we derive the upper bounds for ①, ②, ③, ④, ⑤, ⑥. More precisely, $E_{T-1}$ implies

$$A_T \stackrel{(96)}{\le} V+\text{①}+\text{②}+\text{③}+\text{④}+\text{⑤}+\text{⑥},$$

$$\text{⑥}\stackrel{(114)}{=}\text{⑥}'+\frac{8\gamma}{n}\sum_{j=2}^{n}\left\langle\frac{\gamma}{n}\sum_{i=1}^{j-1}\omega_{i,T-1}^u-\delta_j^{T-1},\omega_{j,T-1}^u\right\rangle,$$

$$\text{②}\stackrel{(105)}{\le}\frac{V}{6},\quad\text{④}\stackrel{(110)}{\le}\frac{V}{12},\quad\text{⑤}\stackrel{(111)}{\le}\frac{V}{6},$$

$$\sum_{t=0}^{T-1}\sigma_t^2\stackrel{(104)}{\le}\frac{V^2}{216\ln\frac{48n(K+1)}{\beta}},\quad\sum_{t=0}^{T-1}\widetilde{\sigma}_t^2\stackrel{(109)}{\le}\frac{V^2}{864\ln\frac{48n(K+1)}{\beta}},$$

$$\sum_{l=0}^{T-1}\sum_{j=2}^{n}(\sigma'_{j,l})^2\stackrel{(119)}{\le}\frac{V^2}{216\ln\frac{48n(K+1)}{\beta}}.$$

In addition, we also establish (see (103), (108), (118) and our induction assumption)

$$\mathbb{P}\{E_{T-1}\}\ge 1-\frac{(T-1)\beta}{K+1},$$

$$\mathbb{P}\{E_{\text{①}}\}\ge 1-\frac{\beta}{24n(K+1)},\quad\mathbb{P}\{E_{\text{③}}\}\ge 1-\frac{\beta}{24n(K+1)},\quad\mathbb{P}\{E_{\text{⑥}'}\}\ge 1-\frac{\beta}{24n(K+1)},$$

where

$$E_{\text{①}} = \left\{\text{either }\sum_{l=0}^{T-1}\sigma_l^2>\frac{V^2}{216\ln\frac{48n(K+1)}{\beta}}\quad\text{or}\quad|\text{①}|\le\frac{V}{6}\right\},$$

$$E_{\text{③}} = \left\{\text{either }\sum_{l=0}^{T-1}\widetilde{\sigma}_l^2>\frac{V^2}{864\ln\frac{48n(K+1)}{\beta}}\quad\text{or}\quad|\text{③}|\le\frac{V}{12}\right\},$$

$$E_{\text{⑥}'} = \left\{\text{either }\sum_{t=0}^{T-1}\sum_{j=2}^{n}(\sigma'_{j,l})^2>\frac{V^2}{216\ln\frac{48n(K+1)}{\beta}}\quad\text{or}\quad|\text{⑥}'|\le\frac{V}{6}\right\}.$$

Therefore, probability event $E_{T-1} \cap E_① \cap E_③ \cap E_{⑥'}$ implies

$$
\begin{aligned}
A_T \quad \leq \quad & V + \frac{V}{6} + \frac{V}{6} + \frac{V}{12} + \frac{V}{12} + \frac{V}{6} + \frac{V}{6} \\
& + \frac{8\gamma}{n} \sum_{j=2}^{n} \left\langle \frac{\gamma}{n} \sum_{i=1}^{j-1} \omega_{i,T-1}^u - \delta_j^{T-1}, \omega_{j,T-1}^u \right\rangle \\
\leq \quad & 2V + \frac{8\gamma}{n} \sum_{j=2}^{n} \left\langle \frac{\gamma}{n} \sum_{i=1}^{j-1} \omega_{i,T-1}^u - \delta_j^{T-1}, \omega_{j,T-1}^u \right\rangle
\end{aligned}
\tag{120}
$$

for $t = T$.

In the final part of the proof, we will show that $\frac{\gamma}{n} \sum_{i=1}^{j-1} \omega_{i,T-1}^u = \delta_j^{T-1}$ with high probability. In particular, we consider probability event $\widetilde{E}_{T-1,j}$ defined as follows: inequalities

$$
\left\| \frac{\gamma}{n} \sum_{i=1}^{r-1} \omega_{i,T-1}^u \right\| \leq \frac{\sqrt{V}}{2}
$$

hold for $r = 2, \ldots, j$ simultaneously. We want to show that $\mathbb{P}\{E_{T-1} \cap \widetilde{E}_{T-1,j}\} \geq 1 - \frac{(T-1)\beta}{K+1} - \frac{j\beta}{8n(K+1)}$ for all $j = 2, \ldots, n$. For $j = 2$ the statement is trivial since

$$
\left\| \frac{\gamma}{n} \omega_{1,T-1}^u \right\| \overset{(97)}{\leq} \frac{2\gamma\lambda}{n} \leq \frac{\sqrt{V}}{2}.
$$

Next, we assume that the statement holds for some $j = m - 1 < n$, i.e., $\mathbb{P}\{E_{T-1} \cap \widetilde{E}_{T-1,m-1}\} \geq 1 - \frac{(T-1)\beta}{K+1} - \frac{(m-1)\beta}{8n(K+1)}$. Our goal is to prove that $\mathbb{P}\{E_{T-1} \cap \widetilde{E}_{T-1,m}\} \geq 1 - \frac{(T-1)\beta}{K+1} - \frac{m\beta}{8n(K+1)}$. We have

$$
\begin{aligned}
\left\| \frac{\gamma}{n} \sum_{i=1}^{m-1} \omega_{i,T-1}^u \right\| \quad = \quad & \sqrt{ \frac{\gamma^2}{n^2} \left\| \sum_{i=1}^{m-1} \omega_{i,T-1}^u \right\|^2 } \\
= \quad & \sqrt{ \frac{\gamma^2}{n^2} \sum_{i=1}^{m-1} \|\omega_{i,T-1}^u\|^2 + \frac{2\gamma}{n} \sum_{i=1}^{m-1} \left\langle \frac{\gamma}{n} \sum_{r=1}^{i-1} \omega_{r,T-1}^u, \omega_{i,T-1}^u \right\rangle } \\
\leq \quad & \sqrt{ \frac{\gamma^2}{n^2} \sum_{l=0}^{T-1} \sum_{i=1}^{m-1} \|\omega_{i,l}^u\|^2 + \frac{2\gamma}{n} \sum_{i=1}^{m-1} \left\langle \frac{\gamma}{n} \sum_{r=1}^{i-1} \omega_{r,T-1}^u, \omega_{i,T-1}^u \right\rangle }.
\end{aligned}
$$

Next, we introduce a new notation:

$$
\rho_{i,T-1}' = \begin{cases} \frac{\gamma}{n} \sum_{r=1}^{i-1} \omega_{r,T-1}^u, & \text{if } \left\| \frac{\gamma}{n} \sum_{r=1}^{i-1} \omega_{r,T-1}^u \right\| \leq \frac{\sqrt{V}}{2}, \\ 0, & \text{otherwise} \end{cases}
$$

for $i = 1, \ldots, m - 1$. By definition, we have

$$
\|\rho_{i,T-1}'\| \leq \frac{\sqrt{V}}{2}
\tag{121}
$$

for $i = 1, \ldots, m - 1$. Moreover, $\widetilde{E}_{T-1,m-1}$ implies $\rho_{i,T-1}' = \frac{\gamma}{n} \sum_{r=1}^{i-1} \omega_{r,T-1}^u$ for $i = 1, \ldots, m - 1$ and

$$
\left\| \frac{\gamma}{n} \sum_{i=1}^{m-1} \omega_{i,l}^u \right\| \quad \leq \quad \sqrt{③ + ④ + ⑦},
$$

where

$$\text{⑦} = \frac{2\gamma}{n} \sum_{i=1}^{m-1} \left\langle \rho'_{i,T-1}, \omega^u_{i,T-1} \right\rangle.$$

It remains to estimate ⑦.

**Upper bound for ⑦.** To estimate this sum, we will use Bernstein's inequality. The summands have conditional expectations equal to zero:

$$\mathbb{E}_{\xi_i^{T-1}} \left[ \frac{2\gamma}{n} \langle \rho'_{i,T-1}, \omega^u_{i,T-1} \rangle \right] = \frac{2\gamma}{n} \left\langle \rho'_{i,T-1}, \mathbb{E}_{\xi_i^{T-1}}[\omega^u_{i,T-1}] \right\rangle = 0,$$

since random vectors $\{\omega^u_{i,T-1}\}_{i=1}^n$ are independent. Thus, sequence $\left\{ \frac{2\gamma}{n} \langle \rho'_{i,T-1}, \omega^u_{i,T-1} \rangle \right\}_{i=1}^{m-1}$ is a martingale difference sequence. Next, the summands are bounded:

$$\left| \frac{2\gamma}{n} \langle \rho'_{i,T-1}, \omega^u_{i,T-1} \rangle \right| \le \frac{2\gamma}{n} \|\rho'_{i,T-1}\| \cdot \|\omega^u_{i,T-1}\| \overset{(121),(97)}{\le} \frac{\gamma}{n} \sqrt{V} \lambda \overset{(81)}{=} \frac{V}{24 \ln \frac{48n(K+1)}{\beta}} \overset{\text{def}}{=} c. \quad (122)$$

Finally, conditional variances $(\widetilde{\sigma}'_{i,T-1})^2 \overset{\text{def}}{=} \mathbb{E}_{\xi_i^{T-1}} \left[ \frac{4\gamma^2}{n^2} \langle \rho'_{i,T-1}, \omega^u_{i,T-1} \rangle^2 \right]$ of the summands are bounded:

$$(\widetilde{\sigma}'_{i,T-1})^2 \le \mathbb{E}_{\xi_i^{T-1}} \left[ \frac{4\gamma^2}{n^2} \|\rho'_{i,T-1}\|^2 \cdot \|\omega^u_{i,T-1}\|^2 \right] \overset{(121)}{\le} \frac{\gamma^2 V}{n^2} \mathbb{E}_{\xi_i^{T-1}} \left[ \|\omega^u_{i,T-1}\|^2 \right]. \quad (123)$$

Applying Bernstein's inequality (Lemma B.1) with $X_i = \frac{2\gamma}{n} \langle \rho'_{i,T-1}, \omega^u_{i,T-1} \rangle$, constant $c$ defined in (122), $b = \frac{V}{24}$, $G = \frac{V^2}{3456 \ln \frac{48n(K+1)}{\beta}}$, we get

$$\mathbb{P} \left\{ |\text{⑦}| > \frac{V}{24} \text{ and } \sum_{i=1}^{m-1} (\widetilde{\sigma}'_{i,T-1})^2 \le \frac{V^2}{3456 \ln \frac{48n(K+1)}{\beta}} \right\} \le 2 \exp \left( -\frac{b^2}{2G + 2cb/3} \right) = \frac{\beta}{24n(K+1)}.$$

The above is equivalent to

$$\mathbb{P}\{E_{\text{⑦}}\} \ge 1 - \frac{\beta}{24n(K+1)}, \quad \text{for } E_{\text{⑦}} = \left\{ \text{either } \sum_{i=1}^{m-1} (\widetilde{\sigma}'_{i,T-1})^2 > \frac{V^2}{3456 \ln \frac{48n(K+1)}{\beta}} \text{ or } |\text{⑦}| \le \frac{V}{24} \right\}. \quad (124)$$

Moreover, $E_{T-1}$ implies

$$\sum_{i=1}^{m-1} (\widetilde{\sigma}'_{i,T-1})^2 \overset{(123)}{\le} \frac{\gamma^2 V}{n^2} \sum_{i=1}^{n} \mathbb{E}_{\xi_i^{T-1}} \left[ \|\omega^u_{i,T-1}\|^2 \right] \overset{(100)}{\le} \frac{18\gamma^2 V \lambda^{2-\alpha} \sigma^\alpha}{n}$$

$$\overset{(81)}{\le} \frac{18\gamma^\alpha \sigma^\alpha V^{2-\frac{\alpha}{2}}}{48^{2-\alpha} n^{\alpha-1} \ln^{2-\alpha} \frac{48n(K+1)}{\beta}} \overset{(80)}{\le} \frac{V^2}{3456 \ln \frac{48n(K+1)}{\beta}}. \quad (125)$$

Putting all together we get that $E_{T-1} \cap \widetilde{E}_{T-1,m-1}$ implies

$$\left\| \frac{\gamma}{n} \sum_{i=1}^{m-1} \omega^u_{i,T-1} \right\| \le \sqrt{\text{③} + \text{④} + \text{⑦}}, \quad \text{④} \overset{(110)}{\le} \frac{V}{6},$$

$$\sum_{l=0}^{T-1} \sum_{i=1}^{n} \widetilde{\sigma}_{i,l}^2 \overset{(109)}{\le} \frac{V^2}{216 \ln \frac{48n(K+1)}{\beta}}, \quad \sum_{i=1}^{m-1} (\widetilde{\sigma}'_{i,T-1})^2 \le \frac{V^2}{3456 \ln \frac{48n(K+1)}{\beta}}.$$

In addition, we also establish (see (108), (124) and our induction assumption)

$$\mathbb{P}\{E_{T-1} \cap \widetilde{E}_{T-1,m-1}\} \ge 1 - \frac{(T-1)\beta}{K+1} - \frac{(m-1)\beta}{8n(K+1)},$$

$$\mathbb{P}\{E_{\text{③}}\} \ge 1 - \frac{\beta}{24n(K+1)}, \quad \mathbb{P}\{E_{\text{⑦}}\} \ge 1 - \frac{\beta}{24n(K+1)}$$

where

$$
\begin{aligned}
E_{③} &= \left\{ \text{either} \sum_{l=0}^{T-1} \sum_{i=1}^{n} \widetilde{\sigma}_{i,l}^2 > \frac{V^2}{864 \ln \frac{48n(K+1)}{\beta}} \text{ or } |③| \leq \frac{V}{12} \right\}, \\
E_{⑦} &= \left\{ \text{either} \sum_{i=1}^{m-1} (\widetilde{\sigma}_{i,T-1}')^2 > \frac{V^2}{3456 \ln \frac{48n(K+1)}{\beta}} \text{ or } |⑦| \leq \frac{V}{24} \right\}
\end{aligned}
$$

Therefore, probability event $E_{T-1} \cap \widetilde{E}_{T-1,m-1} \cap E_{③} \cap E_{⑦}$ implies

$$
\left\| \frac{\gamma}{n} \sum_{i=1}^{m-1} \omega_{i,T-1}^u \right\| \leq \sqrt{\frac{V}{12} + \frac{V}{12} + \frac{V}{24}} \leq \frac{\sqrt{V}}{2}.
$$

This implies $\widetilde{E}_{T-1,m}$ and

$$
\begin{aligned}
\mathbb{P}\{E_{T-1} \cap \widetilde{E}_{T-1,m}\} &\geq \mathbb{P}\{E_{T-1} \cap \widetilde{E}_{T-1,m-1} \cap E_{③} \cap E_{⑦}\} \\
&= 1 - \mathbb{P}\left\{ \overline{E_{T-1} \cap \widetilde{E}_{T-1,m-1}} \cup \overline{E}_{③} \cup \overline{E}_{⑦} \right\} \\
&\geq 1 - \frac{(T-1)\beta}{K+1} - \frac{m\beta}{8n(K+1)}.
\end{aligned}
$$

Therefore, for all $m = 2, \ldots, n$ the statement holds and, in particular, $\mathbb{P}\{E_{T-1} \cap \widetilde{E}_{T-1,n}\} \geq 1 - \frac{(T-1)\beta}{K+1} - \frac{\beta}{8(K+1)}$. Taking into account (120), we conclude that $E_{T-1} \cap \widetilde{E}_{T-1,n} \cap E_{①} \cap E_{③} \cap E_{⑥'}$ implies

$$
A_T \leq 2V
$$

that is equivalent to (85) for $t = T$. Moreover,

$$
\begin{aligned}
\mathbb{P}\{E_T\} &\geq \mathbb{P}\left\{ E_{T-1} \cap \widetilde{E}_{T-1,n} \cap E_{①} \cap E_{③} \cap E_{⑥'} \right\} \\
&= 1 - \mathbb{P}\left\{ \overline{E_{T-1} \cap \widetilde{E}_n} \cup \overline{E}_{①} \cup \overline{E}_{③} \cup \overline{E}_{⑥'} \right\} \\
&= 1 - \frac{(T-1)\beta}{K+1} - \frac{\beta}{8(K+1)} - 3 \cdot \frac{\beta}{24n(K+1)} = 1 - \frac{T\beta}{K+1}.
\end{aligned}
$$

In other words, we showed that $\mathbb{P}\{E_k\} \geq 1 - k\beta/(K+1)$ for all $k = 0, 1, \ldots, K+1$. For $k = K+1$ we have that with probability at least $1 - \beta$

$$
\Phi(\overline{x}^{K+1}) - \Phi(x^*) \overset{(92),(85)}{\leq} \frac{V}{\gamma(K+1)}.
$$

Finally, if

$$
\gamma \leq \min \left\{ \frac{1}{360L \ln \frac{48n(K+1)}{\beta}}, \frac{n^{\frac{\alpha-1}{\alpha}} \sqrt{V}}{27^{\frac{1}{\alpha}} \cdot 48\sigma K^{\frac{1}{\alpha}} \left( \ln \frac{48n(K+1)}{\beta} \right)^{\frac{\alpha-1}{\alpha}}} \right\},
$$

then with probability at least $1 - \beta$

$$
\begin{aligned}
\Phi(\overline{x}^{K+1}) - \Phi(x^*) &\leq \frac{V}{\gamma(K+1)} \\
&= \max \left\{ \frac{360LV \ln \frac{48n(K+1)}{\beta}}{K+1}, \frac{48 \cdot 27^{\frac{1}{\alpha}} \sigma \sqrt{V} K^{\frac{1}{\alpha}} \left( \ln \frac{48n(K+1)}{\beta} \right)^{\frac{\alpha-1}{\alpha}}}{n^{\frac{\alpha-1}{\alpha}}(K+1)} \right\} \\
&= \mathcal{O}\left( \max \left\{ \frac{LV \ln \frac{nK}{\beta}}{K}, \frac{\sigma \sqrt{V} \ln^{\frac{\alpha-1}{\alpha}} \frac{nK}{\beta}}{n^{\frac{\alpha-1}{\alpha}} K^{\frac{\alpha-1}{\alpha}}} \right\} \right).
\end{aligned}
$$

To get $\Phi(\overline{x}^{K+1}) - \Phi(x^*) \leq \varepsilon$ with probability at least $1 - \beta$ it is sufficient to choose $K$ such that both terms in the maximum above are $\mathcal{O}(\varepsilon)$. This leads to

$$K = \mathcal{O}\left(\max\left\{\frac{LV}{\varepsilon}\ln\frac{LV}{\varepsilon\beta}, \left(\frac{\sigma\sqrt{V}}{\varepsilon n^{\frac{\alpha-1}{\alpha}}}\right)^{\frac{\alpha}{\alpha-1}}\ln\left(\frac{1}{\beta}\left(\frac{\sigma\sqrt{V}}{\varepsilon}\right)^{\frac{\alpha}{\alpha-1}}\right)\right\}\right),$$

which concludes the proof. $\qquad\square$

In view of Lemma D.1, the result in the quasi-strongly convex case for DProx-clipped-SGD-shift follows from our result for DProx-clipped-SGDA-shift.

## F   MISSING PROOFS FOR DProx-clipped-SSTM-shift

In this section, we provide the complete formulations of our results for DProx-clipped-SSTM-shift and proofs. For the readers' convenience, the method's update rule is repeated below: $x^0 = y^0 = z^0$, $A_0 = \alpha_0 = 0$, $\alpha_{k+1} = \frac{k+2}{2aL}$, $A_{k+1} = A_k + \alpha_{k+1}$ and

$$x^{k+1} = \frac{A_k y^k + \alpha_{k+1} z^k}{A_{k+1}}, \quad z^{k+1} = \operatorname{prox}_{\alpha_{k+1}\Psi}\left(z^k - \alpha_{k+1}\widetilde{g}(x^{k+1})\right),$$

$$\widetilde{g}(x^{k+1}) = \frac{1}{n}\sum_{i=1}^n \widetilde{g}_i(x^{k+1}), \ \ \widetilde{g}_i(x^{k+1}) = h_i^k + \hat{\Delta}_i^k,$$

$$h_i^{k+1} = h_i^k + \nu_k \hat{\Delta}_i^k, \quad \hat{\Delta}_i^k = \texttt{clip}\left(\nabla f_{\xi_i^k}(x^{k+1}) - h_i^k, \lambda_k\right),$$

$$y^{k+1} = \frac{A_k y^k + \alpha_{k+1} z^{k+1}}{A_{k+1}}$$

where $\xi_1^k, \dots, \xi_n^k$ are sampled independently from each other and previous steps.

### F.1   CONVEX CASE

The following lemma is the main "optimization" part of the analysis of DProx-clipped-SSTM-shift.

**Lemma F.1.** *Let Assumptions 1, 2 and 3($\mu = 0$) hold on $Q = B_{5n\sqrt{M}}(x^*)$, where $M \geq \|x^0 - x^*\|^2 + C^2\alpha_{K_0+1}^2\frac{1}{n}\sum_{i=1}^n \|\nabla f_i(x^*)\|^2$, where $C > 0$, and $a \geq 0$. Let $x^k$, $y^k$, $z^k$ lie in $B_{5n\sqrt{M}}(x^*)$ for all $k = 0, 1, \dots, K$ for some $K \geq 0$. Additionally, let parameters of DProx-clipped-SSTM-shift satisfy*

$$a \geq \max\left\{2, \frac{7}{6}C^2\right\}, \quad K_0 = \left\lceil\frac{3}{2}C^2 n\right\rceil; \tag{126}$$

$$\nu_k = \begin{cases} \frac{(k+2)^2}{C^2(K_0+2)^2 n}, & \text{if } k < K_0; \\ \frac{2k+5}{(k+3)^2}, & \text{if } k \geq K_0; \end{cases} \tag{127}$$

*then the iterates produced by DProx-clipped-SSTM-shift satisfy*

$$
\begin{aligned}
A_K(\Phi(y^K) - \Phi(x^*)) &\leq \frac{1}{2}M_0 - \frac{1}{2}M_K + \sum_{k=0}^{K-1}\alpha_{k+1}\langle\omega_{k+1}, x^* - z^k\rangle + \sum_{k=0}^{K-1}\alpha_{k+1}^2\|\omega_{k+1}\|^2 \\
&\quad + \sum_{k=0}^{K-1}\sum_{i=1}^n\frac{\alpha_{k+1}^2}{n^2}\|\omega_{i,k+1}\|^2,
\end{aligned}
\tag{128}
$$

*where Lyapunov function $M_k$ is defined as follows*

$$M_k = \|z^k - x^*\|^2 + C^2\widetilde{\alpha}_{k+1}^2\frac{1}{n}\sum_{i=1}^n \|h_i^k - h_i^*\|^2, \tag{129}$$

*where*

$$\widetilde{\alpha}_{k+1} = \begin{cases} \alpha_{K_0+1} & \text{if } k < K_0; \\ \alpha_{k+1} & \text{if } k \geq K_0; \end{cases} \tag{130}$$

*and $\omega_{k+1}$ is defined as follows*

$$\omega_{i,k+1} \stackrel{def}{=} \widetilde{g}_i(x^{k+1}) - \nabla f_i(x^{k+1}), \quad \omega_{k+1} \stackrel{def}{=} \frac{1}{n}\sum_{i=1}^n \omega_{i,k+1}. \tag{131}$$

*Proof.* By optimality condition for the problem (17), we have for any $z \in B_{3\sqrt{M}}(x^*)$

$$
\begin{aligned}
\alpha_{k+1} \left\langle \widetilde{g}(x^{k+1}), z^k - z \right\rangle \quad \leq \quad & \alpha_{k+1}(\Psi(z) - \Psi(z^{k+1})) + \alpha_{k+1} \left\langle \widetilde{g}(x^{k+1}), z^k - z^{k+1} \right\rangle \\
& + \frac{1}{2}\|z^k - z\|^2 - \frac{1}{2}\|z^{k+1} - z\|^2 - \frac{1}{2}\|z^{k+1} - z^k\|^2 \\
\overset{(131)}{\leq} \quad & \alpha_{k+1}(\Psi(z) - \Psi(z^{k+1})) + \alpha_{k+1} \left\langle \omega_{k+1}, z^k - z^{k+1} \right\rangle \\
& + \alpha_{k+1} \left\langle \nabla f(x^{k+1}), z^k - z^{k+1} \right\rangle \\
& + \frac{1}{2}\|z^k - z\|^2 - \frac{1}{2}\|z^{k+1} - z\|^2 - \frac{1}{2}\|z^{k+1} - z^k\|^2
\end{aligned}
$$

Using $A_{k+1}(y^{k+1} - x^{k+1}) = \alpha_{k+1}(z^{k+1} - z^k)$, we get

$$
\begin{aligned}
\alpha_{k+1} \left\langle \widetilde{g}(x^{k+1}), z^k - z \right\rangle \quad \leq \quad & \alpha_{k+1}(\Psi(z) - \Psi(z^{k+1})) + \alpha_{k+1} \left\langle \omega_{k+1}, z^k - z^{k+1} \right\rangle \\
& + A_{k+1} \left\langle \nabla f(x^{k+1}), x^{k+1} - y^{k+1} \right\rangle \\
& + \frac{1}{2}\|z^k - z\|^2 - \frac{1}{2}\|z^{k+1} - z\|^2 - \frac{1}{2}\|z^{k+1} - z^k\|^2 \\
\overset{(*)}{\leq} \quad & \alpha_{k+1}(\Psi(z) - \Psi(z^{k+1})) + \alpha_{k+1} \left\langle \omega_{k+1}, z^k - z^{k+1} \right\rangle \\
& + A_{k+1} \left( f(x^{k+1}) - f(y^{k+1}) + \frac{L}{2}\|y^{k+1} - x^{k+1}\|^2 \right) \\
& + \frac{1}{2}\|z^k - z\|^2 - \frac{1}{2}\|z^{k+1} - z\|^2 - \frac{1}{2}\|z^{k+1} - z^k\|^2 \\
= \quad & \alpha_{k+1}(\Psi(z) - \Psi(z^{k+1})) + \alpha_{k+1} \left\langle \omega_{k+1}, z^k - z^{k+1} \right\rangle \\
& + A_{k+1} \left( f(x^{k+1}) - f(y^{k+1}) \right) + \frac{\alpha_{k+1}^2 L}{2 A_{k+1}}\|z^{k+1} - z^k\|^2 \\
& + \frac{1}{2}\|z^k - z\|^2 - \frac{1}{2}\|z^{k+1} - z\|^2 - \frac{1}{2}\|z^{k+1} - z^k\|^2 \\
= \quad & \alpha_{k+1}(\Psi(z) - \Psi(z^{k+1})) + \alpha_{k+1} \left\langle \omega_{k+1}, z^k - z^{k+1} \right\rangle \\
& + A_{k+1} \left( f(x^{k+1}) - f(y^{k+1}) \right) + \frac{1}{2}\|z^k - z\|^2 - \frac{1}{2}\|z^{k+1} - z\|^2 \\
& - \frac{1}{2} \left( 1 - \frac{\alpha_{k+1}^2 L}{A_{k+1}} \right) \|z^{k+1} - z^k\|^2
\end{aligned}
$$

where in $(*)$ $L$-smoothness of $f$ was used. Using Young's inequality, we have

$$
\begin{aligned}
\alpha_{k+1} \left\langle \widetilde{g}(x^{k+1}), z^k - z \right\rangle \quad \leq \quad & \alpha_{k+1}(\Psi(z) - \Psi(z^{k+1})) + \alpha_{k+1} \frac{D}{2}\|\omega_{k+1}\|^2 + \frac{\alpha_{k+1}}{2D}\|z^k - z^{k+1}\|^2 \\
& + A_{k+1} \left( f(x^{k+1}) - f(y^{k+1}) \right) + \frac{1}{2}\|z^k - z\|^2 - \frac{1}{2}\|z^{k+1} - z\|^2 \\
& - \frac{1}{2} \left( 1 - \frac{\alpha_{k+1}^2 L}{A_{k+1}} \right) \|z^{k+1} - z^k\|^2 \\
\overset{D=2\alpha_{k+1}}{=} \quad & \alpha_{k+1}(\Psi(z) - \Psi(z^{k+1})) + \alpha_{k+1}^2\|\omega_{k+1}\|^2 + \frac{1}{4}\|z^k - z^{k+1}\|^2 \\
& + A_{k+1} \left( f(x^{k+1}) - f(y^{k+1}) \right) + \frac{1}{2}\|z^k - z\|^2 - \frac{1}{2}\|z^{k+1} - z\|^2 \\
& - \frac{1}{2} \left( 1 - \frac{\alpha_{k+1}^2 L}{A_{k+1}} \right) \|z^{k+1} - z^k\|^2
\end{aligned}
$$

Now, by $a \geq 2$, we have $\frac{1}{2} - \frac{\alpha_{k+1}^2 L}{A_{k+1}} \geq 0$ and

$$
\begin{aligned}
\alpha_{k+1} \left\langle \widetilde{g}(x^{k+1}), z^k - z \right\rangle \leq \ & \alpha_{k+1}(\Psi(z) - \Psi(z^{k+1})) + \alpha_{k+1}^2 \|\omega_{k+1}\|^2 \\
& + A_{k+1} \left( f(x^{k+1}) - f(y^{k+1}) \right) + \frac{1}{2}\|z^k - z\|^2 - \frac{1}{2}\|z^{k+1} - z\|^2 \quad (132) \\
& - \frac{1}{2}\left( \frac{1}{2} - \frac{\alpha_{k+1}^2 L}{A_{k+1}} \right) \|z^{k+1} - z^k\|^2 \\
\leq \ & \alpha_{k+1}(\Psi(z) - \Psi(z^{k+1})) + \alpha_{k+1}^2 \|\omega_{k+1}\|^2 + A_{k+1} \left( f(x^{k+1}) - f(y^{k+1}) \right) \\
& + \frac{1}{2}\|z^k - z\|^2 - \frac{1}{2}\|z^{k+1} - z\|^2. \quad (133)
\end{aligned}
$$

To continue the proof, we have to mention that

$$
\begin{aligned}
\left\langle \widetilde{g}(x^{k+1}), y^k - x^{k+1} \right\rangle \overset{(131)}{=} \ & \left\langle \nabla f(x^{k+1}), y^k - x^{k+1} \right\rangle + \left\langle \omega_{k+1}, y^k - x^{k+1} \right\rangle \\
\leq \ & f(y^k) - f(x^{k+1}) + \left\langle \omega_{k+1}, y^k - x^{k+1} \right\rangle, \quad (134)
\end{aligned}
$$

where in the last inequality we used convexity of $f$. Also, by convexity of $\Psi$ and definition of $y^{k+1}$, we have

$$
\begin{aligned}
\Psi(y^{k+1}) \ = \ & \Psi \left( \frac{A_k}{A_{k+1}} y^k + \frac{\alpha_{k+1}}{A_{k+1}} z^{k+1} \right) \leq \frac{A_k}{A_{k+1}} \Psi(y^k) + \frac{\alpha_{k+1}}{A_{k+1}} \Psi(z^{k+1}); \\
-\alpha_{k+1}\Psi(z^{k+1}) \ \leq \ & -A_{k+1}\Psi(y^{k+1}) + A_k \Psi(y^k). \quad (135)
\end{aligned}
$$

Thus, we acquire

$$
\begin{aligned}
\alpha_{k+1} \left\langle \widetilde{g}(x^{k+1}), x^{k+1} - z \right\rangle \ = \ & \alpha_{k+1} \left\langle \widetilde{g}(x^{k+1}), x^{k+1} - z^k \right\rangle + \alpha_{k+1} \left\langle \widetilde{g}(x^{k+1}), z^k - z \right\rangle \\
= \ & A_k \left\langle \widetilde{g}(x^{k+1}), y^k - x^{k+1} \right\rangle + \alpha_{k+1} \left\langle \widetilde{g}(x^{k+1}), z^k - z \right\rangle
\end{aligned}
$$

where the last equation is true due to that $\alpha_{k+1}(x^{k+1} - z^k) = A_k(y^k - x^{k+1})$. By (132), (134), we get

$$
\begin{aligned}
\alpha_{k+1} \left\langle \widetilde{g}(x^{k+1}), x^{k+1} - z \right\rangle \ \leq \ & A_k(f(y^k) - f(x^{k+1})) + A_k \left\langle \omega_{k+1}, y^k - x^{k+1} \right\rangle \\
& + \alpha_{k+1}(\Psi(z) - \Psi(z^{k+1})) + A_{k+1} \left( f(x^{k+1}) - f(y^{k+1}) \right) \\
& + \alpha_{k+1}^2 \|\omega_{k+1}\|^2 + \frac{1}{2}\|z^k - z\|^2 - \frac{1}{2}\|z^{k+1} - z\|^2 \\
\overset{(135)}{\leq} \ & A_k(f(y^k) - f(x^{k+1})) + A_k \left\langle \omega_{k+1}, y^k - x^{k+1} \right\rangle \\
& + \alpha_{k+1}\Psi(z) - A_{k+1}\Psi(y^{k+1}) + A_k \Psi(y^k) + A_{k+1} \left( f(x^{k+1}) - f(y^{k+1}) \right) \\
& + \alpha_{k+1}^2 \|\omega_{k+1}\|^2 + \frac{1}{2}\|z^k - z\|^2 - \frac{1}{2}\|z^{k+1} - z\|^2.
\end{aligned}
$$

By definition of function $\Phi(\cdot)$ (1), we have

$$
\begin{aligned}
\alpha_{k+1} \left\langle \widetilde{g}(x^{k+1}), x^{k+1} - z \right\rangle \ \leq \ & A_k \Phi(y^k) - A_{k+1}\Phi(y^{k+1}) + A_k \left\langle \omega_{k+1}, y^k - x^{k+1} \right\rangle \\
& + \alpha_{k+1}\Psi(z) + (A_{k+1} - A_k)f(x^{k+1}) \\
& + \alpha_{k+1}^2 \|\omega_{k+1}\|^2 + \frac{1}{2}\|z^k - z\|^2 - \frac{1}{2}\|z^{k+1} - z\|^2 \\
\overset{(**)}{=} \ & A_k \Phi(y^k) - A_{k+1}\Phi(y^{k+1}) + \alpha_{k+1} \left\langle \omega_{k+1}, x^{k+1} - z^k \right\rangle \\
& + \alpha_{k+1}\Psi(z) + \alpha_{k+1} f(x^{k+1}) \\
& + \alpha_{k+1}^2 \|\omega_{k+1}\|^2 + \frac{1}{2}\|z^k - z\|^2 - \frac{1}{2}\|z^{k+1} - z\|^2
\end{aligned}
$$

where in $(**)$ we used $\alpha_{k+1}(x^{k+1} - z^k) = A_k(y^k - x^{k+1})$ and $A_{k+1} = A_k + \alpha_{k+1}$. Making a small rearrangement, we derive

$$
\begin{aligned}
A_{k+1}\Phi(y^{k+1}) - A_k\Phi(y^k) \quad \leq \quad & \frac{1}{2}\|z^k - z\|^2 - \frac{1}{2}\|z^{k+1} - z\|^2 + \alpha_{k+1}\Psi(z) \\
& + \alpha_{k+1}f(x^{k+1}) + \alpha_{k+1}\left\langle \widetilde{g}(x^{k+1}), z - x^{k+1}\right\rangle \\
& + \alpha_{k+1}\left\langle \omega_{k+1}, x^{k+1} - z^k\right\rangle + \alpha_{k+1}^2\|\omega_{k+1}\|^2 \\
\overset{(131)}{=} \quad & \frac{1}{2}\|z^k - z\|^2 - \frac{1}{2}\|z^{k+1} - z\|^2 + \alpha_{k+1}\Psi(z) \\
& + \alpha_{k+1}\frac{1}{n}\sum_{i=1}^{n}f_i(x^{k+1}) + \alpha_{k+1}\left\langle \frac{1}{n}\sum_{i=1}^{n}\nabla f_i(x^{k+1}), z - x^{k+1}\right\rangle \\
& + \alpha_{k+1}\left\langle \omega_{k+1}, z - x^{k+1}\right\rangle + \alpha_{k+1}\left\langle \omega_{k+1}, x^{k+1} - z^k\right\rangle + \alpha_{k+1}^2\|\omega_{k+1}\|^2 \\
\leq \quad & \frac{1}{2}\|z^k - z\|^2 - \frac{1}{2}\|z^{k+1} - z\|^2 + \alpha_{k+1}\Psi(z) + \alpha_{k+1}f(z) \\
& + \alpha_{k+1}\left\langle \omega_{k+1}, z - z^k\right\rangle + \alpha_{k+1}^2\|\omega_{k+1}\|^2 \\
& - \frac{\alpha_{k+1}}{2Ln}\sum_{i=1}^{n}\|\nabla f_i(x^{k+1}) - \nabla f_i(z)\|^2,
\end{aligned}
\tag{136}
$$

where in the last inequality we used $L$-smoothness and convexity of each $f_i$. Now we consider the sequences of $h_i^k$, produced by the method, for any $i \in [n]$. Denoting $h_i^* = \nabla f_i(x^*)$ and , we have

$$
\begin{aligned}
\|h_i^{k+1} - h_i^*\|^2 \quad \overset{(19)}{=} \quad & \|h_i^k - h_i^*\|^2 + 2\nu_k\left\langle \hat{\Delta}_i^k, h_i^k - h_i^*\right\rangle + \nu_k^2\|\hat{\Delta}_i^k\|^2 \\
= \quad & \|h_i^k - h_i^*\|^2 + 2\nu_k\left\langle \widetilde{g}_i(x^{k+1}) - h_i^k, h_i^k - h_i^*\right\rangle + \nu_k^2\|\widetilde{g}_i(x^{k+1}) - h_i^k\|^2 \\
\overset{\nu_k \leq 1}{\leq} \quad & \|h_i^k - h_i^*\|^2 + 2\nu_k\left\langle \widetilde{g}_i(x^{k+1}) - h_i^k, h_i^k - h_i^*\right\rangle + \nu_k\|\widetilde{g}_i(x^{k+1}) - h_i^k\|^2 \\
= \quad & \|h_i^k - h_i^*\|^2 + \nu_k\left\langle \widetilde{g}_i(x^{k+1}) - h_i^k, \widetilde{g}_i(x^{k+1}) + h_i^k - 2h_i^*\right\rangle \\
\leq \quad & (1 - \nu_k)\|h_i^k - h_i^*\|^2 + \nu_k\|\widetilde{g}_i(x^{k+1}) - h_i^*\|^2 \\
\leq \quad & (1 - \nu_k)\|h_i^k - h_i^*\|^2 + 2\nu_k\|\widetilde{g}_i(x^{k+1}) - \nabla f_i(x^{k+1})\|^2 + 2\nu_k\|\nabla f_i(x^{k+1}) - h_i^*\|^2 \\
\overset{(131)}{=} \quad & (1 - \nu_k)\|h_i^k - h_i^*\|^2 + 2\nu_k\|\omega_{i,k+1}\|^2 + 2\nu_k\|\nabla f_i(x^{k+1}) - \nabla f_i(x^\star)\|^2.
\end{aligned}
\tag{137}
$$

Summing up (137) by $i$ from 1 to $n$, we obtain

$$
\begin{aligned}
\frac{1}{n}\sum_{i=1}^{n}\|h_i^{k+1} - h_i^*\|^2 \quad \leq \quad & (1 - \nu_k)\frac{1}{n}\sum_{i=1}^{n}\|h_i^k - h_i^*\|^2 + \frac{2\nu_k}{n}\sum_{i=1}^{n}\|\omega_{i,k+1}\|^2 \\
& + \frac{2\nu_k}{n}\sum_{i=1}^{n}\|\nabla f_i(x^{k+1}) - \nabla f_i(x^\star)\|^2.
\end{aligned}
\tag{138}
$$

Combining inequality (136), where we take $z = x^*$, and inequality (138) multiplied by $\frac{1}{2}C^2\widetilde{\alpha}_{k+2}^2$, we get

$$
\begin{aligned}
A_{k+1}\left(\Phi(y^{k+1}) - \Phi(x^*)\right) \quad \leq \quad & A_k\left(\Phi(y^k) - \Phi(x^*)\right) + \frac{1}{2}\|z^k - x^*\|^2 + \frac{1}{2}C^2\widetilde{\alpha}_{k+1}^2\frac{1}{n}\sum_{i=1}^{n}\|h_i^k - h_i^*\|^2 \\
& - \frac{1}{2}\|z^{k+1} - x^*\|^2 - \frac{1}{2}C^2\widetilde{\alpha}_{k+2}^2\frac{1}{n}\sum_{i=1}^{n}\|h_i^{k+1} - h_i^*\|^2 \\
& + \frac{1}{2}(1 - \nu_k)C^2\widetilde{\alpha}_{k+2}^2\frac{1}{n}\sum_{i=1}^{n}\|h_i^k - h_i^*\|^2 - \frac{1}{2}C^2\widetilde{\alpha}_{k+1}^2\frac{1}{n}\sum_{i=1}^{n}\|h_i^k - h_i^*\|^2 \\
& + \alpha_{k+1}\left\langle \omega_{k+1}, x^* - z^k\right\rangle + \alpha_{k+1}^2\|\omega_{k+1}\|^2 + \frac{1}{2}C^2\widetilde{\alpha}_{k+2}^2\frac{2\nu_k}{n}\sum_{i=1}^{n}\|\omega_{i,k+1}\|^2 \\
& - \left(\frac{\alpha_{k+1}}{2Ln} - \frac{1}{n}\nu_kC^2\widetilde{\alpha}_{k+2}^2\right)\sum_{i=1}^{n}\|\nabla f_i(x^{k+1}) - \nabla f_i(z)\|^2.
\end{aligned}
$$

By the selection of parameters (126), (127) and definition of Lyapunov function $M_k$ (129), we have

$$
\begin{aligned}
A_{k+1}\left(\Phi(y^{k+1}) - \Phi(x^*)\right) \leq\ & A_k\left(\Phi(y^k) - \Phi(x^*)\right) + \frac{1}{2}M_k - \frac{1}{2}M_{k+1} \\
& + \alpha_{k+1}\left\langle \omega_{k+1}, x^* - z^k \right\rangle + \alpha_{k+1}^2 \|\omega_{k+1}\|^2 + \frac{\alpha_{k+1}^2}{n^2}\sum_{i=1}^{n}\|\omega_{i,k+1}\|^2.
\end{aligned}
$$

Summing up the previous inequality by $k$ from $0$ to $K - 1$, we finish the proof.

$\square$

**Theorem F.1.** *Let Assumptions 1, 2 and 3($\mu = 0$) hold on $Q = B_{5n\sqrt{M}}(x^*)$, where $M \geq \|x^0 - x^*\|^2 + C^2\alpha_{K_0+1}^2\frac{1}{n}\sum_{i=1}^{n}\|\nabla f_i(x^*)\|^2$, where $C = \frac{864}{n}\ln\frac{10nK}{\beta}$, and $a \geq 0$, and*

$$
a \geq \max\left\{2, \frac{8\cdot 3^5\cdot 72^4}{n}\ln^4\frac{10nK}{\beta}, \frac{18\cdot 6^5\sigma K^{\frac{1}{\alpha}}(K+1)}{\sqrt{M}Ln^{\frac{\alpha-1}{\alpha}}}\ln^{\frac{\alpha-1}{\alpha}}\frac{10nK}{\beta}\right\}, \tag{139}
$$

$$
\lambda_k = \frac{n\sqrt{M}}{72\widetilde{\alpha}_{k+1}\ln\frac{10nK}{\beta}}, \tag{140}
$$

*for some $K \geq K_0 = \left\lceil \frac{3}{2}C^2 n \right\rceil > 0$ and $\beta \in (0,1]$ such that $\ln\frac{10nK}{\beta} \geq 1$. Then, after $K$ iterations of* DProx-clipped-SSTM-shift *the following inequality holds with probability at least $1 - \beta$*

$$
\Phi(y^K) - \Phi(x^*) \leq \frac{6aLM}{K(K+3)} \quad and \quad \{x^k\}_{k=0}^{K+1}, \{z^k\}_{k=0}^{K}, \{y^k\}_{k=0}^{K} \subseteq B_{2\sqrt{M}}(x^*). \tag{141}
$$

*In particular, when parameter $a$ equals the maximum from (139), then after $K$ iterations of* DProx-clipped-SSTM-shift, *we have with probability at least $1 - \beta$*

$$
\Phi(y^K) - \Phi(x^*) = \mathcal{O}\left(\max\left\{\frac{LM}{K^2}, \frac{LM\ln^4\frac{nK}{\beta}}{nK^2}, \frac{\sigma\sqrt{M}\ln^{\frac{\alpha-1}{\alpha}}\frac{nK}{\beta}}{n^{\frac{\alpha-1}{\alpha}}K^{\frac{\alpha-1}{\alpha}}}\right\}\right), \tag{142}
$$

*i.e. achieve $\Phi(y^K) - \Phi(x^*) \leq \varepsilon$ with probability at least $1 - \beta$* DProx-clipped-SSTM-shift *requires*

$$
K = \mathcal{O}\left(\max\left\{\sqrt{\frac{LM}{\varepsilon}}, \sqrt{\frac{LM}{\varepsilon n}}\ln^2\frac{nLM}{\varepsilon\beta}, \frac{1}{n}\left(\frac{\sigma\sqrt{M}}{\varepsilon}\right)^{\frac{\alpha}{\alpha-1}}\ln\frac{\sigma\sqrt{M}}{\varepsilon\beta}\right\}\right) \tag{143}
$$

*iterations/oracle calls per worker.*

*Proof.* The key idea behind the proof is similar to the one used in (Gorbunov et al., 2021; Sadiev et al., 2023). We prove by induction that the iterates do not leave some ball and $\Phi(y^K) - \Phi(y^*)$ decreases as $\sim 1/K(K+3)$

Firstly, we denote $R_k = \|z^k - x^*\|$, $\widetilde{R}_0 = R_0$, $\widetilde{R}_{k+1} = \max\{\widetilde{R}_k, R_{k+1}\}$ for all $k \geq 0$, and now we show by induction that for all $k \geq 0$ the iterates $x^{k+1}, z^k, y^k$ lie in $B_{\widetilde{R}_k}(x^*)$. The induction base is trivial since $y^0 = z^0$, $\widetilde{R}_0 = R_0$, and $x^1 = \frac{A_0 y^0 + \alpha_1 z^0}{A_1} = z^0$. Next, we assume this statement is true for some $l \geq 1$: $x^l, z^{l-1}, y^{l-1} \in B_{\widetilde{R}_{l-1}}(x^*)$. According to definitions of $R_l$ and $\widetilde{R}_l$, we obtain $z^l \in B_{R_l}(x^*) \subseteq B_{\widetilde{R}_l}(x^*)$. Due to that $y^l$ is a convex combination of $y^{l-1} \in B_{\widetilde{R}_{l-1}}(x^*) \subseteq B_{\widetilde{R}_l}(x^*)$, $z^l \in B_{\widetilde{R}_l}(x^*)$ and $B_{\widetilde{R}_l}(x^*)$ is a convex set, we have that $y^l \in B_{\widetilde{R}_l}(x^*)$. Finally, since $x^{l+1}$ is a convex combination of $y^l$ and $z^l$, we conclude $x^{l+1}$ lies in $B_{\widetilde{R}_l}(x^*)$ as well.

Now to formulate the statement rigorously, we introduce probability event $E_k$ for each for each $k = 0, \ldots, K$ as follows: inequalities

$$\underbrace{2 \sum_{l=0}^{t-1} \alpha_{l+1} \langle \omega_{l+1}, x^* - z^l \rangle + 2 \sum_{l=0}^{t-1} \alpha_{l+1}^2 \|\omega_{l+1}\|^2 + 2 \sum_{l=0}^{t-1} \sum_{i=1}^{n} \frac{\alpha_{l+1}^2}{n^2} \|\omega_{i,l+1}\|^2 \leq M}_{B_t}, \quad (144)$$

$$R_t \leq \sqrt{M_t} \leq 2\sqrt{M}, \quad (145)$$

$$\left\| \frac{\alpha_t}{n} \sum_{i=1}^{r} \omega_{i,t}^u \right\| \leq \frac{M}{2} \quad (146)$$

hold for $t = 0, 1, \ldots, k$ and $r = 1, 2 \ldots, n$ simultaneously, where

$$\omega_{l+1} = \omega_{l+1}^u + \omega_{l+1}^b, \quad (147)$$

$$\omega_{l+1}^u \overset{\text{def}}{=} \frac{1}{n} \sum_{i=1}^{n} \omega_{i,l+1}^u, \quad \omega_{l+1}^b \overset{\text{def}}{=} \frac{1}{n} \sum_{i=1}^{n} \omega_{i,l+1}^b, \quad (148)$$

$$\omega_{i,l+1}^u \overset{\text{def}}{=} \widetilde{g}_i(x^{l+1}) - \mathbb{E}_{\xi_i^l} \left[ \widetilde{g}_i(x^{l+1}) \right], \quad \omega_{i,k+1}^b \overset{\text{def}}{=} \mathbb{E}_{\xi_i^k} \left[ \widetilde{g}_i(x^{k+1}) \right] - \nabla f_i(x^{k+1}), \quad \forall i \in [n]. \quad (149)$$

We want to show via induction $\widetilde{R}_l \leq 5n\sqrt{M}$ with high probability, which allows us to apply the result of Lemma F.1 and Bernstein's inequality to estimate the stochastic part of the upper-bound. After that, we will prove by induction that $\mathbb{P}\{E_k\} \geq 1 - k\beta/K$ for all $k = 0, 1, \ldots, K$. The base induction follows immediately: the left-hand side of (144) equals zero and $M \geq M_0$ by definition, and for $k = 0$ we have $\left\| \frac{\alpha_0}{n} \sum_{i=1}^{r} \omega_{i,0}^u \right\| = 0$, since $\alpha_0 = 0$. Next we assume that the statement holds for some $k = T - 1 \leq K - 1$: $\mathbb{P}\{E_{T-1}\} \geq 1 - (T-1)\beta/K$. Let us show that $\mathbb{P}\{E_T\} \geq 1 - T\beta/K$.

To proceed, we need to show that probability event $E_{T-1}$ implies that $\widetilde{R}_t \leq 2\sqrt{M}$ for all $t = 0, 1, \ldots, T$. The base is already proven. Next we assume that $\widetilde{R}_t \leq 2\sqrt{M}$ for all $t = 0, 1, \ldots, t'$ for some $t' < T$. Then for all $t = 0, 1, \ldots, t'$

$$
\begin{aligned}
\|z^t - x^\star\| &= \| \operatorname{prox}_{\alpha_t \Psi} \left( z^{t-1} - \alpha_t \widetilde{g}(x^t) \right) - \operatorname{prox}_{\alpha_t \Psi} \left( x^\star - \alpha_t \nabla f(x^\star) \right) \| \\
&\leq \| z^{t-1} - x^* - \alpha_t \left( \widetilde{g}(x^t) - \nabla f(x^\star) \right) \| \\
&\leq \| z^{t-1} - x^* \| + \alpha_t \| \widetilde{g}(x^t) - h^{t-1} \| + \alpha_t \| h^{t-1} - h^* \| \\
&\leq \left( 1 + \frac{1}{C} \right) \sqrt{ \| z^{t-1} - x^* \|^2 + C^2 \widetilde{\alpha}_t^2 \frac{1}{n} \sum_{i=1}^{n} \| h_i^{t-1} - h_i^* \|^2 } + \alpha_t \lambda_{t-1} \\
&\leq 2\sqrt{M_{t-1}} + \alpha_t \lambda_{t-1} \overset{(140),(145)}{\leq} 4\sqrt{M} + n\sqrt{M} \leq 5n\sqrt{M}.
\end{aligned}
$$

This means that $x^t, z^t, y^t \in B_{5n\sqrt{M}}(x^*)$ for $t = 0, 1, \ldots, t'$ and we can apply Lemma F.1: $E_{T-1}$ implies

$$
\begin{aligned}
A_{t'} \left( \Phi(y^{t'}) - \Phi(x^*) \right) &\leq \frac{1}{2} M_0 - \frac{1}{2} M_{t'} + \sum_{l=0}^{t'-1} \alpha_{l+1} \langle \omega_{l+1}, x^* - z^l \rangle + \sum_{l=0}^{t'-1} \alpha_{l+1}^2 \|\omega_{l+1}\|^2 \\
&\quad + \sum_{k=0}^{t'-1} \sum_{i=1}^{n} \frac{\alpha_{l+1}^2}{n^2} \|\omega_{i,k+1}\|^2 \\
&\leq \frac{1}{2} M_0 - \frac{1}{2} M_{t'} + B_{t'} \leq \frac{3}{2} M \quad (150)
\end{aligned}
$$

that gives

$$M_{t'} \leq M_0 + M \leq 2M.$$

That is, we showed that $E_{T-1}$ implies $x^t, z^t, y^t \in B_{2\sqrt{M}}(x^*)$ and

$$\Phi(y^t) - \Phi(x^*) \overset{(144),(150)}{\leq} \frac{\frac{1}{2} M_0 - \frac{1}{2} M_t + M}{A_t} \leq \frac{3M}{2A_t} = \frac{6aLM}{t(t+3)}. \quad (151)$$

for all $t = 0, 1, \ldots, T$. Before we proceed, we introduce a new notation:

$$\eta_t = \begin{cases} x^* - z^t, & \text{if } \|x^* - z^t\| \le 2\sqrt{M}, \\ 0, & \text{otherwise}, \end{cases}$$

for all $t = 0, 1, \ldots, T$. Random vectors $\{\eta_t\}_{t=0}^T$ are bounded almost surely:

$$\|\eta_t\| \le 2\sqrt{M} \tag{152}$$

for all $t = 0, 1, \ldots, T$. In addition, $\eta_t = x^* - z^t$ follows from $E_{T-1}$ for all $t = 0, 1, \ldots, T$ and, thus, $E_{T-1}$ implies

$$\begin{aligned} B_T &= 2 \sum_{k=0}^{T-1} \alpha_{k+1} \langle \omega_{k+1}, x^* - z^k \rangle + 2 \sum_{k=0}^{T-1} \alpha_{k+1}^2 \|\omega_{k+1}\|^2 + 2 \sum_{k=0}^{T-1} \sum_{i=1}^{n} \frac{\alpha_{k+1}^2}{n^2} \|\omega_{i,k+1}\|^2 \\ &= 2 \sum_{k=0}^{T-1} \alpha_{k+1} \langle \omega_{k+1}, \eta_k \rangle + 2 \sum_{k=0}^{T-1} \alpha_{k+1}^2 \|\omega_{k+1}\|^2 + 2 \sum_{k=0}^{T-1} \sum_{i=1}^{n} \frac{\alpha_{k+1}^2}{n^2} \|\omega_{i,k+1}\|^2. \end{aligned} \tag{153}$$

Using the notation from (147)-(149), we can rewrite $\|\omega_{k+1}\|^2$ and $\|\omega_{i,k+1}\|^2$ as

$$\|\omega_{k+1}\|^2 \le \frac{2}{n^2} \sum_{i=1}^{n} \|\omega_{i,k+1}^u\|^2 + \frac{4}{n^2} \sum_{j=2}^{n} \left\langle \sum_{i=1}^{j-1} \omega_{i,k+1}^u, \omega_{j,k+1}^u \right\rangle + 2\|\omega_{k+1}^b\|^2. \tag{154}$$

Putting all together, we obtain that $E_{T-1}$ implies

$$\begin{aligned} B_T \le \ & \underbrace{2 \sum_{k=0}^{T-1} \sum_{i=1}^{n} \frac{\alpha_{k+1}}{n} \left\langle \omega_{i,k+1}^u, \eta_k \right\rangle}_{①} + \underbrace{2 \sum_{k=0}^{T-1} \sum_{i=1}^{n} \frac{\alpha_{k+1}}{n} \left\langle \omega_{i,k+1}^b, \eta_k \right\rangle}_{②} \\ & + \underbrace{8 \sum_{k=0}^{T-1} \sum_{i=1}^{n} \frac{\alpha_{k+1}^2}{n^2} \left( \left\|\omega_{i,k+1}^u\right\|^2 - \mathbb{E}_{\xi_i^k} \left[ \left\|\omega_{i,k+1}^u\right\|^2 \right] \right)}_{③} \\ & + \underbrace{8 \sum_{k=0}^{T-1} \sum_{i=1}^{n} \frac{\alpha_{k+1}^2}{n^2} \mathbb{E}_{\xi^k} \left[ \left\|\omega_{i,k+1}^u\right\|^2 \right]}_{④} + \underbrace{8 \sum_{k=0}^{T-1} \sum_{i=1}^{n} \frac{\alpha_{k+1}^2}{n} \left\|\omega_{i,k+1}^b\right\|^2}_{⑤} \\ & + \underbrace{8 \sum_{k=0}^{T-1} \sum_{j=2}^{n} \frac{\alpha_{k+1}^2}{n^2} \left\langle \sum_{i=1}^{j-1} \omega_{i,k+1}^u, \omega_{j,k+1}^u \right\rangle}_{⑥}. \end{aligned} \tag{155}$$

To finish the proof, it remains to estimate ①, ②, ③, ④, ⑤, ⑥ with high probability. More precisely, the goal to prove that $① + ② + ③ + ④ + ⑤ + ⑥ \le M$ with high probability. Before we proceed, we need to derive several useful inequalities related to $\omega_{i,k+1}^u$, $\omega_{i,k+1}^b$. First of all, we have

$$\|\omega_{i,k+1}^u\| \le 2\lambda_k. \tag{156}$$

by definition of the clipping operator. Next, probability event $E_{T-1}$ implies that for $t = 0$ we have $x^1 = x^0$ and

$$\begin{aligned} \|\nabla f_i(x^1) - h_i^0\| &\le & \|\nabla f_i(x^0) - \nabla f_i(x^\star)\| + \|h_i^0 - h_i^*\| \\ &\overset{\text{smooth}}{\le} & L\|x^0 - x^*\| + \frac{\sqrt{n}}{C\widetilde{\alpha}_1} \sqrt{C^2 \widetilde{\alpha}_1^2 \frac{1}{n} \sum_{i=1}^{n} \|h_i^0 - h_i^*\|^2} \\ &\le & \left( \frac{2(K_0 + 2)}{a\widetilde{\alpha}_1} + \frac{\sqrt{n}}{C\widetilde{\alpha}_1} \right) \sqrt{M} \\ &\overset{(139),(140)}{\le} & \frac{\lambda_0}{2}. \end{aligned} \tag{157}$$

Next, for $t = 1, \ldots, T - 1$ event $E_{T-1}$ implies

$$
\begin{aligned}
\|\nabla f_i(x^{t+1}) - h_i^t\| \quad &\leq \quad \|\nabla f_i(x^{t+1}) - \nabla f_i(y^t)\| + \|\nabla f_i(y^t) - \nabla f_i(x^\star)\| + \|h_i^t - h_i^*\| \\
&\leq \quad L\|x^{t+1} - y^t\| + \sqrt{2L\left(f_i(y^t) - f_i(x^*) - \langle \nabla f_i(x^\star), y^t - x^\star \rangle\right)} \\
&\overset{(*)}{\leq} \quad L\|x^{t+1} - y^t\| + \sqrt{2nL\left(\Phi(y^t) - \Phi(x^*)\right)} + \sqrt{\sum_{i=1}^{n} \|h_i^t - h_i^*\|^2} \quad (158) \\
&\overset{(151)}{\leq} \quad \frac{L\alpha_{t+1}}{A_t}\|x^{t+1} - z^t\| + \sqrt{\frac{12anL^2 M}{t(t+3)}} + \frac{\sqrt{n}}{C\widetilde{\alpha}_{t+1}}\sqrt{\frac{1}{n}\sum_{i=1}^{n}\|h_i^t - h_i^*\|^2} \\
&\leq \quad \frac{4L\sqrt{M}\alpha_{t+1}}{A_t} + \sqrt{\frac{12anL^2 M}{t(t+3)}} + \frac{\sqrt{n}}{C\widetilde{\alpha}_{t+1}}\sqrt{M_t} \\
&\overset{(140)}{\leq} \quad \frac{\lambda_t}{2}\left(\frac{8 \cdot 72 L\alpha_{t+1}\widetilde{\alpha}_{t+1}\ln\frac{10nK}{\beta}}{nA_t} + 2\sqrt{\frac{12 \cdot 72^2 a L^2 \widetilde{\alpha}_{t+1}^2 \ln^2 \frac{10nK}{\beta}}{nt(t+3)}}\right) \\
&\quad + \frac{\lambda_t}{2} \cdot \frac{288}{C\sqrt{n}} \ln \frac{10nK}{\beta} \\
&\leq \quad \frac{\lambda_t}{2} \cdot \frac{576 a L^2 \max\{K_0 + 2, t + 2\}(t+2)\ln\frac{10nK}{\beta}}{a^2 L^2 t(t+3)n} \\
&\quad + \frac{\lambda_t}{2} \cdot \sqrt{\frac{12 a L^2 \max\{(K_0 + 2)^2, (t+2)^2\} 72^2 \ln^2 \frac{10nK}{\beta}}{na^2 L^2 t(t+3)}} \\
&\quad + \frac{\lambda_t}{2} \cdot \frac{288}{C\sqrt{n}} \ln \frac{10nK}{\beta} \\
&\leq \quad \frac{\lambda_t}{2} \cdot \frac{9}{a} \max\{(K_0 + 2), 2\} \frac{72}{n} \ln \frac{10nK}{\beta} \\
&\quad + \frac{\lambda_t}{2} \sqrt{\frac{3}{a} \max\{(K_0 + 2)^2, 9\} \frac{72^2}{n} \ln^2 \frac{10nK}{\beta}} \\
&\quad + \frac{\lambda_t}{2} \cdot \frac{288}{C\sqrt{n}} \ln \frac{10nK}{\beta} \overset{(139)}{\leq} \frac{\lambda_t}{2}, \quad (159)
\end{aligned}
$$

where in $(*)$ we use $-\langle \frac{1}{n}\sum_{i=1}^{n}\nabla f_i(x^\star), y^t - x^\star \rangle \leq \Psi(y^t) - \Psi(x^\star)$, and in the last row we use $\frac{(t+2)^2}{t(t+3)} \leq \frac{9}{4}$ for all $t \geq 1$ and $C \geq 12 \cdot 72 \ln \frac{10nK}{\beta}$.

Therefore, Lemma B.2 and $E_{T-1}$ imply

$$
\begin{aligned}
\|\omega_{i,k+1}^b\| &\leq \frac{2^\alpha \sigma^\alpha}{\lambda_k^{\alpha-1}}, \quad (160) \\
\mathbb{E}_{\xi_i^k}\left[\|\omega_{i,k+1}^u\|^2\right] &\leq 18\lambda_k^{2-\alpha}\sigma^\alpha. \quad (161)
\end{aligned}
$$

**Upper bound for ①.** To estimate this sum, we will use Bernstein's inequality. The summands have conditional equal to zero, since $\mathbb{E}_{\xi_i^k}[\omega_{i,k+1}^u] = 0$:

$$
\mathbb{E}_{\xi_i^k}\left[\frac{\alpha_{k+1}}{n}\langle \omega_{i,k+1}^u, \eta_k \rangle\right] = 0.
$$

Moreover, for all $k = 0, \ldots, T-1$ random vectors $\left\{\omega_{i,k+1}^u\right\}_{k=0}^{T-1}$ are independent. Thus, sequence $\left\{2\frac{\alpha_{k+1}}{n}\left\langle \omega_{i,k+1}^u, \eta_k\right\rangle\right\}_{k=0}^{T-1}$ is a martingale difference sequence. Next, the summands are bounded:

$$
\begin{aligned}
\left|2\frac{\alpha_{k+1}}{n}\left\langle \omega_{i,k+1}^u, \eta_k\right\rangle\right| &\leq 2\frac{\alpha_{k+1}}{n}\|\omega_{i,k+1}^u\| \cdot \|\eta_k\| \overset{(156)}{\leq} 4\frac{\alpha_{k+1}}{n}\lambda_k\sqrt{M} \\
&\overset{(140)}{=} \frac{4n\alpha_{k+1}\sqrt{M}}{72n\widetilde{\alpha}_{k+1}\ln\frac{10nK}{\beta}} \leq \frac{\sqrt{M}}{6\ln\frac{10nK}{\beta}} \overset{\text{def}}{=} c.
\end{aligned}
$$
(162)

Finally, conditional variances $\sigma_{i,k}^2 \overset{\text{def}}{=} \mathbb{E}_{\xi_i^k}\left[4\frac{\alpha_{k+1}^2}{n^2}\left\langle \omega_{i,k+1}^u, \eta_k\right\rangle^2\right]$ of the summands are bounded::

$$
\sigma_{i,k}^2 \leq \mathbb{E}_{\xi_i^k}\left[4\frac{\alpha_{k+1}^2}{n^2}\|\omega_{i,k+1}^u\|^2 \cdot \|\eta_k\|^2\right] \leq 16\frac{\alpha_{k+1}^2}{n^2}M\mathbb{E}_{\xi^t}\left[\|\omega_{i,k+1}^u\|^2\right].
$$
(163)

Applying Bernstein's inequality (Lemma B.1) with $X_{i,k} = 2\frac{\alpha_{k+1}}{n}\left\langle \omega_{i,k+1}^u, \eta_k\right\rangle$, parameter $c$ as in (162), $b = \frac{M}{6}$, $G = \frac{M^2}{6^3\ln\frac{10nK}{\beta}}$:

$$
\mathbb{P}\left\{|①| > \frac{M}{6} \quad\text{and}\quad \sum_{k=0}^{T-1}\sum_{i=1}^n \sigma_{i,k}^2 \leq \frac{M^2}{6^3\ln\frac{10nK}{\beta}}\right\} \leq 2\exp\left(-\frac{b^2}{2G + 2cb/3}\right) = \frac{\beta}{5nK}.
$$

The above is equivalent to

$$
\mathbb{P}\left\{E_①\right\} \geq 1 - \frac{\beta}{5nK}, \quad\text{for}\quad E_① = \left\{\text{either}\quad \sum_{k=0}^{T-1}\sum_{i=1}^n \sigma_{i,k}^2 > \frac{M^2}{6^3\ln\frac{10nK}{\beta}} \quad\text{or}\quad |①| \leq \frac{M}{6}\right\}.
$$
(164)

Moreover, $E_{T-1}$ implies that

$$
\begin{aligned}
\sum_{k=0}^{T-1}\sum_{i=1}^n \sigma_{i,k}^2 &\overset{(163)}{\leq} 16M\sum_{k=0}^{T-1}\sum_{i=1}^n \frac{\alpha_{k+1}^2}{n^2}\mathbb{E}_{\xi^t}\left[\|\omega_{i,k+1}^u\|^2\right] \overset{(161)}{\leq} 288\sigma^\alpha M\sum_{k=0}^{T-1}\sum_{i=1}^n \frac{\alpha_{k+1}^2}{n^2}\lambda_k^{2-\alpha} \\
&\overset{(140)}{\leq} \frac{288\sigma^\alpha M^{2-\alpha/2}}{72^{2-\alpha}\ln^{2-\alpha}\frac{10nK}{\beta}}\sum_{k=0}^{T-1}\frac{\alpha_{k+1}^2}{n^{\alpha-1}\widetilde{\alpha}_{k+1}^{2-\alpha}} \leq \frac{288\sigma^\alpha M^{2-\alpha/2}}{72^{2-\alpha}\ln^{2-\alpha}\frac{10nK}{\beta}}\sum_{k=0}^{T-1}\frac{\alpha_{k+1}^\alpha}{n^{\alpha-1}} \\
&\leq \frac{288\sigma^\alpha M^{2-\alpha/2}}{n^{\alpha-1}72^{2-\alpha}\cdot 2^\alpha a^\alpha L^\alpha\ln^{2-\alpha}\frac{10nK}{\beta}}\sum_{k=0}^{T-1}(k+2)^\alpha \\
&\leq \frac{1}{a^\alpha}\cdot\frac{144\sigma^\alpha M^{2-\alpha/2}T(T+1)^\alpha}{n^{\alpha-1}L^\alpha\ln^{2-\alpha}\frac{10nK}{\beta}} \overset{(139)}{\leq} \frac{M^2}{6^3\ln\frac{10nK}{\beta}}.
\end{aligned}
$$
(165)

**Upper bound for ②.** Probability event $E_{T-1}$ implies

$$
\begin{aligned}
② &\leq 2\sum_{k=0}^{T-1}\sum_{i=1}^n \frac{\alpha_{k+1}}{n}\|\omega_{i,k+1}^b\| \cdot \|\eta_k\| \overset{(160)}{\leq} 4\sqrt{M}\cdot 2^\alpha\sigma^\alpha\sum_{k=0}^{T-1}\frac{\alpha_{k+1}}{\lambda_k^{\alpha-1}} \\
&\overset{(140)}{\leq} \frac{16\cdot 72^{\alpha-1}M^{1-\alpha/2}\sigma^\alpha}{n^{\alpha-1}}\ln^{\alpha-1}\frac{10nK}{\beta}\sum_{k=0}^{T-1}\max\left\{\alpha_{k+1}\widetilde{\alpha}_{k+1}^{\alpha-1}, \alpha_{k+1}^\alpha\right\} \\
&\leq \frac{16\cdot 72^{\alpha-1}\sigma^\alpha M^{1-\alpha/2}}{2^\alpha a^\alpha L^\alpha}\ln^{\alpha-1}\frac{10nK}{\beta}\sum_{t=0}^{T-1}\max\left\{(K_0+2)(k+2)^{\alpha-1}, (k+2)^\alpha\right\} \\
&\overset{T,K_0\leq K}{\leq} \frac{1}{a^\alpha}\cdot\frac{12\cdot 16\cdot 72^{\alpha-1}\sigma^\alpha M^{1-\alpha/2}K(K+1)^\alpha}{4^\alpha L^\alpha}\ln^{\alpha-1}\frac{10nK}{\beta} \\
&\overset{(139)}{\leq} \frac{M}{6}.
\end{aligned}
$$
(166)

**Upper bound for ③.** To estimate this sum, we will use Bernstein's inequality. The summands have conditional expectations equal to zero:

$$\frac{8\alpha_{k+1}^2}{n^2}\mathbb{E}_{\xi_i^k}\left[\left\|\omega_{i,k+1}^u\right\|^2 - \mathbb{E}_{\xi_i^k}\left[\left\|\omega_{i,k+1}^u\right\|^2\right]\right] = 0.$$

Moreover, for all $k = 0, \ldots, T-1$ random vectors $\left\{\omega_{i,k+1}^u\right\}_{i=1}^n$ are independent. Thus, sequence $\left\{\frac{8\alpha_{k+1}^2}{n^2}\left(\left\|\omega_{i,k+1}^u\right\|^2 - \mathbb{E}_{\xi_i^k}\left[\left\|\omega_{i,k+1}^u\right\|^2\right]\right)\right\}_{k,i=0,1}^{T-1,n}$ is a martingale difference sequence. Next, the summands are bounded:

$$\left|\frac{8\alpha_{k+1}^2}{n^2}\left(\left\|\omega_{i,k+1}^u\right\|^2 - \mathbb{E}_{\xi_i^k}\left[\left\|\omega_{i,k+1}^u\right\|^2\right]\right)\right| \leq \frac{8\alpha_{k+1}^2}{n^2}\left(\left\|\omega_{i,k+1}^u\right\|^2 + \mathbb{E}_{\xi_i^k}\left[\left\|\omega_{i,k+1}^u\right\|^2\right]\right)$$

$$\overset{(156)}{\leq} \frac{64\alpha_{k+1}^2\lambda_k^2}{n^2} \overset{(140)}{\leq} \frac{M}{9\ln\frac{10nK}{\beta}} \overset{\text{def}}{=} c. \qquad (167)$$

Finally, conditional variances

$$\widetilde{\sigma}_{i,k}^2 \overset{\text{def}}{=} \mathbb{E}_{\xi_i^k}\left[\frac{64\alpha_{k+1}^4}{n^4}\left(\left\|\theta_{t+1}^u\right\|^2 - \mathbb{E}_{\xi_i^k}\left[\left\|\omega_{k+1}^u\right\|^2\right]\right)^2\right]$$

of the summands are bounded:

$$\widetilde{\sigma}_{i,k}^2 \overset{(167)}{\leq} \frac{8\alpha_{k+1}^2 M}{9n^2\ln\frac{10nK}{\beta}}\mathbb{E}_{\xi_i^k}\left[\left|\left\|\omega_{i,k+1}^u\right\|^2 - \mathbb{E}_{\xi_i^k}\left[\left\|\omega_{i,k+1}^u\right\|^2\right]\right|\right]$$

$$\leq \frac{16\alpha_{k+1}^2 M}{9n^2}\mathbb{E}_{\xi_i^k}\left[\left\|\omega_{i,k+1}^u\right\|^2\right]. \qquad (168)$$

Applying Bernstein's inequality (Lemma B.1) with $\widetilde{X}_{i,k} = \frac{8\alpha_{k+1}^2}{n^2}\left(\left\|\omega_{k+1}^u\right\|^2 - \mathbb{E}_{\xi_i^k}\left[\left\|\omega_{i,k+1}^u\right\|^2\right]\right)$, parameter $c$ defined in (167), $b = \frac{M}{9}$, $G = \frac{M^2}{6\cdot 9^2\ln\frac{10nK}{\beta}}$:

$$\mathbb{P}\left\{|③| > \frac{M}{9} \quad\text{and}\quad \sum_{k=0}^{T-1}\sum_{i=1}^n\widetilde{\sigma}_{i,k}^2 \leq \frac{M^2}{6\cdot 9^2\ln\frac{10nK}{\beta}}\right\} \leq 2\exp\left(-\frac{b^2}{2G + 2cb/3}\right) = \frac{\beta}{5nK}.$$

The above is equivalent to

$$\mathbb{P}\left\{E_③\right\} \geq 1 - \frac{\beta}{5nK}, \quad\text{for}\quad E_③ = \left\{\text{either} \quad \sum_{k=0}^{T-1}\sum_{i=1}^n\widetilde{\sigma}_{i,k}^2 > \frac{M^2}{6\cdot 9^2\ln\frac{10nK}{\beta}} \quad\text{or}\quad |③| \leq \frac{M}{9}\right\}. \qquad (169)$$

Moreover, $E_{T-1}$ implies

$$\sum_{k=0}^{T-1}\sum_{i=1}^n\widetilde{\sigma}_{i,k}^2 \overset{(168)}{\leq} \frac{16}{9}M\sum_{k=0}^{T-1}\sum_{i=1}^n\frac{\alpha_{k+1}^2}{n^2}\mathbb{E}_{\xi_i^k}\left[\left\|\omega_{k+1}^u\right\|^2\right] \overset{(165)}{\leq} \frac{M^2}{6\cdot 9^2\ln\frac{10nK}{\beta}}. \qquad (170)$$

**Upper bound for ④.** Probability event $E_{T-1}$ implies

$$④ = 8\sum_{k=0}^{T-1}\sum_{i=1}^n\frac{\alpha_{k+1}^2}{n^2}\mathbb{E}_{\xi_i^k}\left[\left\|\omega_{i,k+1}^u\right\|^2\right] \leq \frac{1}{M}\cdot 8M\sum_{k=0}^{T-1}\sum_{i=1}^n\frac{\alpha_{k+1}^2}{n^2}\mathbb{E}_{\xi^t}\left[\left\|\omega_{i,k+1}^u\right\|^2\right]$$

$$\overset{(165)}{\leq} \frac{M}{6^3\ln\frac{10nK}{\beta}} \leq \frac{M}{9}. \qquad (171)$$

**Upper bound for ⑤.** Probability event $E_{T-1}$ implies

$$
\begin{aligned}
⑤ &= 8 \sum_{k=0}^{T-1} \sum_{i=1}^{n} \frac{\alpha_{k+1}^2}{n} \left\| \omega_{i,k+1}^b \right\|^2 \leq 2^{2\alpha+3} \sigma^{2\alpha} \sum_{k=0}^{T-1} \frac{\alpha_{k+1}^2}{\lambda_k^{2\alpha-2}} \\
&\overset{(140)}{=} \frac{2^{2\alpha+3} \cdot 72^{2\alpha-2} \sigma^{2\alpha} \ln^{2\alpha-2} \frac{10nK}{\beta}}{n^{2\alpha-2} M^{\alpha-1}} \sum_{k=0}^{T-1} \max \left\{ \alpha_{k+1}^2 \widetilde{\alpha}_{k+1}^{2\alpha-2}, \alpha_{k+1}^2 \right\} \\
&= \frac{2^{2\alpha+3} \cdot 72^{2\alpha-2} \sigma^{2\alpha} \ln^{2\alpha-2} \frac{10nK}{\beta}}{2^{2\alpha} a^{2\alpha} n^{2\alpha-2} L^{2\alpha} M^{\alpha-1}} \sum_{k=0}^{T-1} \max \left\{ (k+2)^2, (K_0+2)^{2\alpha-2}(k+2)^2 \right\} \\
&\leq \frac{1}{a^{2\alpha}} \cdot \frac{8 \cdot 72^{2\alpha-2} \sigma^{2\alpha} K(K+1)^{2\alpha} \ln^{2\alpha-2} \frac{10nK}{\beta}}{n^{2\alpha-2} L^{2\alpha} M^{\alpha-1}} \overset{(139)}{\leq} \frac{M}{6}.
\end{aligned}
\tag{172}
$$

**Upper bound for ⑥.** This sum requires more refined analysis. We introduce a new vector:

$$
\chi_j^k = \begin{cases} \frac{\alpha_{k+1}}{n} \sum_{i=1}^{j-1} \omega_{i,k+1}^u, & \text{if } \left\| \frac{\alpha_{k+1}}{n} \sum_{i=1}^{j-1} \omega_{i,k+1}^u \right\| \leq \frac{\sqrt{M}}{2}, \\ 0, & \text{otherwise}, \end{cases}
\tag{173}
$$

Then, by definition

$$
\| \chi_j^k \| \leq \frac{\sqrt{V}}{2}
\tag{174}
$$

and

$$
⑥ = \underbrace{8 \sum_{k=0}^{T-1} \sum_{j=2}^{n} \frac{\alpha_{k+1}}{n} \left\langle \chi_j^k, \omega_{j,k+1}^u \right\rangle}_{⑥'} + 8 \sum_{k=0}^{T-1} \sum_{j=2}^{n} \left\langle \frac{\alpha_{k+1}}{n} \sum_{i=1}^{j-1} \omega_{i,k+1}^u - \chi_j^k, \omega_{j,k+1}^u \right\rangle.
\tag{175}
$$

We also note here that $E_{T-1}$ implies

$$
8 \sum_{k=0}^{T-1} \sum_{j=2}^{n} \left\langle \frac{\alpha_{k+1}}{n} \sum_{i=1}^{j-1} \omega_{i,k+1}^u - \chi_j^k, \omega_{j,k+1}^u \right\rangle = 8 \sum_{j=2}^{n} \left\langle \frac{\alpha_T}{n} \sum_{i=1}^{j-1} \omega_{i,T}^u - \chi_j^{T-1}, \omega_{j,T}^u \right\rangle.
\tag{176}
$$

**Upper bound for ⑥'.** To estimate this sum, we will use Bernstein's inequality. The summands have conditional expectations equal to zero:

$$
\mathbb{E}_{\xi_j^k} \left[ \frac{8\alpha_{k+1}}{n} \langle \chi_j^k, \omega_{j,k+1}^u \rangle \right] = \frac{8\alpha_{k+1}}{n} \langle \chi_j^k, \mathbb{E}_{\xi_j^k} [\omega_{j,k+1}^u] \rangle = 0.
$$

Moreover, for all $k = 0, \ldots, T-1$ random vectors $\{\omega_{i,l}^u\}_{i=1}^n$ are independent. Thus, sequence $\left\{ \frac{8\alpha_{k+1}}{n} \langle \chi_j^k, \omega_{j,k+1}^u \rangle \right\}_{k,j=0,2}^{T-1,n}$ is a martingale difference sequence. Next, the summands are bounded:

$$
\left| \frac{8\alpha_{k+1}}{n} \langle \chi_j^k, \omega_{j,k+1}^u \rangle \right| \leq \frac{8\alpha_{k+1}}{n} \| \chi_j^k \| \| \omega_{j,k+1}^u \| \overset{(156),(174)}{\leq} \frac{8\alpha_{k+1}}{n} \cdot \frac{\sqrt{M}}{2} \cdot 2\lambda_k \leq \frac{M}{6 \ln \frac{10nK}{\beta}} \overset{\text{def}}{=} c. \tag{177}
$$

Finally, conditional variances

$$
\hat{\sigma}_{j,k}^2 \overset{\text{def}}{=} \mathbb{E}_{\xi_j^k} \left[ \frac{64\alpha_{k+1}^2}{n^2} \langle \chi_k^j, \omega_{j,k+1}^u \rangle^2 \right]
$$

of summands are bounded:

$$
\hat{\sigma}_{j,k}^2 \leq \frac{64\alpha_{k+1}^2}{n^2} \mathbb{E}_{\xi_j^k} \left[ \| \chi_j^k \|^2 \| \omega_{k+1}^u \|^2 \right] \leq \frac{16\alpha_{k+1}^2 M}{n^2} \mathbb{E}_{\xi_j^k} \left[ \| \omega_{j,k+1}^u \|^2 \right].
\tag{178}
$$

Applying Bernstein's inequality (Lemma B.1) with $X_{j,k} = \frac{8\alpha_{k+1}}{n}\left\langle \chi_j^k, \omega_{j,k+1}^u \right\rangle$, constant $c$ defined in (177), $b = \frac{M}{6}$, $G = \frac{M^2}{6^3 \ln \frac{10nK}{\beta}}$, we get

$$\mathbb{P}\left\{|⑥'| > \frac{M}{6} \text{ and } \sum_{k=0}^{T-1}\sum_{j=2}^{n}\hat{\sigma}_{i,k}^2 \le \frac{M^2}{6^3 \ln \frac{10nK}{\beta}}\right\} \le 2\exp\left(-\frac{b^2}{2G + {2cb}/{3}}\right) = \frac{\beta}{5nK}.$$

The above is equivalent to

$$\mathbb{P}\{E_{⑥'}\} \ge 1 - \frac{\beta}{5nK}, \text{ for } E_{⑥'} = \left\{\text{either } \sum_{k=0}^{T-1}\sum_{j=2}^{n}\hat{\sigma}_{i,l}^2 > \frac{M^2}{6^3 \ln \frac{10nK}{\beta}} \quad \text{or} \quad |⑥'| \; \frac{M}{6}\right\}. \quad (179)$$

Moreover, $E_{T-1}$ implies

$$\sum_{k=0}^{T-1}\sum_{j=2}^{n}\hat{\sigma}_{i,k}^2 \overset{(178)}{\le} 16M\sum_{k=0}^{T-1}\sum_{j=1}^{n}\frac{\alpha_{k+1}^2}{n^2}\mathbb{E}_{\xi_j^k}\left[\|\omega_{j,k+1}^u\|^2\right] \overset{(165)}{\le} \frac{M^2}{6^3 \ln \frac{10nK}{\beta}} \quad (180)$$

That is, we derive the upper bounds for ①, ②, ③, ④, ⑤, ⑥. More precisely, $E_{T-1}$ implies

$$B_T \overset{(155)}{\le} ① + ② + ③ + ④ + ⑤ + ⑥,$$

$$⑥ \overset{(175),(176)}{=} ⑥' + 8\sum_{j=2}^{n}\left\langle \frac{\alpha_T}{n}\sum_{i=1}^{j-1}\omega_{i,T}^u - \chi_j^{T-1}, \omega_{j,T}^u \right\rangle,$$

$$② \overset{(166)}{\le} \frac{M}{6}, \quad ④ \overset{(171)}{\le} \frac{M}{9}, \quad ⑤ \overset{(172)}{\le} \frac{M}{6},$$

$$\sum_{k=0}^{T-1}\sum_{i=1}^{n}\sigma_{i,k}^2 \overset{(165)}{\le} \frac{M^2}{6^3 \ln \frac{10nK}{\beta}}, \quad \sum_{k=0}^{T-1}\sum_{i=1}^{n}\widetilde{\sigma}_{i,k}^2 \overset{(170)}{\le} \frac{M^2}{6 \cdot 9^2 \ln \frac{10nK}{\beta}},$$

$$\sum_{k=0}^{T-1}\sum_{j=2}^{n}\hat{\sigma}_{i,k}^2 \overset{(180)}{\le} \frac{M^2}{6^3 \ln \frac{10nK}{\beta}}.$$

In addition, we also establish (see (164), (169), (179) and our induction assumption):

$$\mathbb{P}\{E_{T-1}\} \ge 1 - \frac{(T-1)\beta}{K},$$

$$\mathbb{P}\{E_①\} \ge 1 - \frac{\beta}{5nK}, \quad \mathbb{P}\{E_③\} \ge 1 - \frac{\beta}{5nK}, \quad \mathbb{P}\{E_{⑥'}\} \ge 1 - \frac{\beta}{5nK},$$

where

$$E_① = \left\{\text{either } \sum_{k=0}^{T-1}\sum_{i=1}^{n}\sigma_{i,k}^2 > \frac{M^2}{6^3 \ln \frac{10nK}{\beta}} \quad \text{or} \quad |①| \le \frac{M}{6}\right\}.$$

$$E_③ = \left\{\text{either } \sum_{k=0}^{T-1}\sum_{i=1}^{n}\widetilde{\sigma}_{i,k}^2 > \frac{M^2}{6 \cdot 9^2 \ln \frac{10nK}{\beta}} \quad \text{or} \quad |③| \le \frac{M}{9}\right\},$$

$$E_{⑥'} = \left\{\text{either } \sum_{k=0}^{T-1}\sum_{j=2}^{n}\hat{\sigma}_{i,l}^2 > \frac{M^2}{6^3 \ln \frac{10nK}{\beta}} \quad \text{or} \quad |⑥'| \le \frac{M}{6}\right\}.$$

Therefore, probability event $E_{T-1} \cap E_{①} \cap E_{③} \cap E_{⑥'}$ implies

$$
\begin{aligned}
B_T &\leq \frac{M}{6} + \frac{M}{6} + \frac{M}{9} + \frac{M}{9} + \frac{M}{6} + \frac{M}{6} \\
&\quad + 8 \sum_{k=0}^{T-1} \sum_{j=2}^{n} \left\langle \frac{\alpha_{k+1}}{n} \sum_{i=1}^{j-1} \omega_{i,k+1}^u - \chi_j^k, \omega_{j,k+1}^u \right\rangle \\
&\leq M + 8 \sum_{j=2}^{n} \left\langle \frac{\alpha_T}{n} \sum_{i=1}^{j-1} \omega_{i,T}^u - \chi_j^{T-1}, \omega_{j,T}^u \right\rangle.
\end{aligned}
\tag{181}
$$

In the final part of the proof, we will show that $\frac{\alpha_{k+1}}{n} \sum_{i=1}^{j-1} \omega_{i,k+1}^u = \chi_j^k$ with high probability. In particular, we consider probability event $\widetilde{E}_{T-1,j}$ defined as follows: inequalities

$$
\left\| \frac{\alpha_T}{n} \sum_{i=1}^{r-1} \omega_{i,T}^u \right\| \leq \frac{\sqrt{M}}{2}
\tag{182}
$$

hold for $r = 2, \dots, j$ simultaneously. We want to show that $\mathbb{P}\{E_{T-1} \cap \widetilde{E}_{T-1,j}\} \geq 1 - \frac{(T-1)\beta}{K} - \frac{2j\beta}{5nK}$ for all $j = 2, \dots, n$. For $j = 2$ the statement is trivial since

$$
\left\| \frac{\alpha_T}{n} \omega_{1,T}^u \right\| \overset{(156)}{\leq} \frac{2\alpha_T \lambda_{T-1}}{n} \leq \frac{\sqrt{M}}{2}.
$$

Next, we assume that the statement holds for some $j = m - 1 < n$, i.e., $\mathbb{P}\{E_{T-1} \cap \widetilde{E}_{T-1,m-1}\} \geq 1 - \frac{(T-1)\beta}{K+1} - \frac{2(m-1)\beta}{5n(K+1)}$. Our goal is to prove that $\mathbb{P}\{E_{T-1} \cap \widetilde{E}_{T-1,m}\} \geq 1 - \frac{(T-1)\beta}{K} - \frac{2m\beta}{5nK}$. First, we consider $\left\| \frac{\alpha_T}{n} \sum_{i=1}^{m-1} \omega_{i,T}^u \right\|$:

$$
\begin{aligned}
\left\| \frac{\alpha_T}{n} \sum_{i=1}^{m-1} \omega_{i,T}^u \right\| &= \sqrt{\frac{\alpha_T^2}{n^2} \left\| \sum_{i=1}^{m-1} \omega_{i,T}^u \right\|^2} \\
&= \sqrt{\frac{\alpha_T^2}{n^2} \sum_{i=1}^{m-1} \|\omega_{i,T}^u\|^2 + \frac{2\alpha_T}{n} \sum_{i=1}^{m-1} \left\langle \frac{\alpha_T}{n} \sum_{r=1}^{i-1} \omega_{r,T}^u, \omega_{i,T}^u \right\rangle} \\
&\leq \sqrt{\sum_{k=0}^{T-1} \sum_{i=1}^{m-1} \frac{\alpha_{k+1}^2}{n^2} \|\omega_{i,k+1}^u\|^2 + \frac{2\alpha_T}{n} \sum_{i=1}^{m-1} \left\langle \frac{\alpha_T}{n} \sum_{r=1}^{i-1} \omega_{r,T}^u, \omega_{i,T}^u \right\rangle}.
\end{aligned}
$$

Next, we introduce a new notation:

$$
\rho_{i,T-1} = \begin{cases} \frac{\alpha_T}{n} \sum_{r=1}^{i-1} \omega_{r,T}^u, & \text{if } \left\| \frac{\alpha_T}{n} \sum_{r=1}^{i-1} \omega_{r,T}^u \right\| \leq \frac{\sqrt{M}}{2}, \\ 0, & \text{otherwise} \end{cases}
$$

for $i = 1, \dots, m - 1$ a. By definition, we have

$$
\|\rho_{i,T-1}\| \leq \frac{\sqrt{M}}{2}
\tag{183}
$$

for $i = 1, \dots, m - 1$. Moreover, $\widetilde{E}_{m-1}$ implies $\rho_{i,T-1} = \frac{\alpha_T}{n} \sum_{r=1}^{i-1} \omega_{r,T}^u$ for $i = 1, \dots, m - 1$ and

$$
\left\| \frac{\alpha_T}{n} \sum_{i=1}^{m-1} \omega_{i,T}^u \right\| \leq \sqrt{③ + ④ + ⑦},
$$

where

$$
⑦ = \frac{2\alpha_T}{n} \sum_{i=1}^{m-1} \left\langle \rho_{i,T-1}, \omega_{i,T}^u \right\rangle.
$$

It remains to estimate ⑦.

**Upper bound for** ⑦ . To estimate this sum, we will use Bernstein's inequality. The summands have conditional expectations equal to zero:

$$\mathbb{E}_{\xi_i^{T-1}} \left[ \frac{2\alpha_T}{n} \langle \rho_{i,T-1}, \omega_{i,T}^u \rangle \right] = \frac{2\alpha_T}{n} \left\langle \rho_{i,T-1}, \mathbb{E}_{\xi_i^{T-1}}[\omega_{i,T}^u] \right\rangle = 0,$$

since random vectors $\{\omega_{i,T}^u\}_{i=1}^n$ are independent. Thus, sequence $\left\{ \frac{2\alpha_T}{n} \langle \rho_{i,T-1}, \omega_{i,T}^u \rangle \right\}_{i=1}^{m-1}$ is a martingale difference sequence. Next, the summands are bounded:

$$\left| \frac{2\alpha_T}{n} \langle \rho_{i,T-1}, \omega_{i,T}^u \rangle \right| \le \frac{2\alpha_T}{n} \|\rho_{i,T-1}\| \cdot \|\omega_{i,T}^u\| \overset{(156),(183)}{\le} \frac{2\alpha_T}{n} \sqrt{M} \lambda_{T-1} \overset{(140)}{=} \frac{M}{36 \ln \frac{10nK}{\beta}} \overset{\text{def}}{=} c. \quad (184)$$

Finally, conditional variances $\bar{\sigma}_{i,T-1}^2 \overset{\text{def}}{=} \mathbb{E}_{\xi_i^{T-1}} \left[ \frac{4\alpha_T^2}{n^2} \langle \rho_{i,T-1}, \omega_{i,T}^u \rangle^2 \right]$ of the summands are bounded:

$$\bar{\sigma}_{i,T-1}^2 \le \mathbb{E}_{\xi_i^{T-1}} \left[ \frac{4\alpha_T^2}{n^2} \|\rho_{i,T-1}\|^2 \cdot \|\omega_{i,T}^u\|^2 \right] \overset{(183)}{\le} \frac{\alpha_T^2 M}{n^2} \mathbb{E}_{\xi_i^{T-1}} \left[ \|\omega_{i,T}^u\|^2 \right]. \quad (185)$$

Applying Bernstein's inequality (Lemma B.1) with $X_i = \frac{2\alpha_T}{n} \langle \rho_{i,T-1}, \omega_{i,T}^u \rangle$, constant $c$ defined in (184), $b = \frac{V}{36}$, $G = \frac{M^2}{6^5 \ln \frac{10nK}{\beta}}$, we get

$$\mathbb{P} \left\{ |⑦| > \frac{M}{36} \text{ and } \sum_{i=1}^{m-1} \bar{\sigma}_{i,T-1}^2 \le \frac{M^2}{6^5 \ln \frac{10nK}{\beta}} \right\} \le 2 \exp \left( -\frac{b^2}{2G + {2cb}/{3}} \right) = \frac{\beta}{5nK}.$$

The above is equivalent to

$$\mathbb{P}\{E_⑦\} \ge 1 - \frac{\beta}{5nK}, \text{ for } E_⑦ = \left\{ \text{either } \sum_{i=1}^{m-1} \bar{\sigma}_{i,T-1}^2 > \frac{M^2}{6^5 \ln \frac{10nK}{\beta}} \text{ or } |⑦| \le \frac{M}{36} \right\}. \quad (186)$$

Moreover, $E_{T-1}$ implies

$$
\begin{aligned}
\sum_{i=1}^{m-1} \bar{\sigma}_{i,T-1}^2 &\overset{(185)}{\le} \quad \sum_{i=1}^{m-1} \frac{\alpha_T^2 M}{n^2} \mathbb{E}_{\xi_i^{T-1}} \left[ \|\omega_{i,T}^u\|^2 \right] \overset{(161)}{\le} 18(m-1) \frac{\alpha_T^2}{n^2} M \sigma^\alpha \lambda_{T-1}^{2-\alpha} \\
&\overset{(140)}{\le} \quad 18(m-1) \frac{\alpha_T^2}{n^2} \cdot \frac{n^{2-\alpha} M^{2-\alpha/2}}{72^{2-\alpha} \widetilde{\alpha}_T^{2-\alpha}} \ln^{\alpha-2} \frac{10nK}{\beta} \\
&\overset{m \le n, T \le K}{\le} \quad 18 \cdot 6^5 \frac{\alpha_K^2}{n^{\alpha-1}} \left( \frac{\sigma}{\sqrt{M}} \right)^\alpha \ln^{\alpha-1} \frac{10nK}{\beta} \cdot \frac{M^2}{6^5 \ln \frac{10nK}{\beta}} \\
&\le \quad \frac{1}{a^\alpha} \frac{18 \cdot 6^5}{2^\alpha} \frac{1}{n^{\alpha-1}} \left( \frac{\sigma}{L\sqrt{M}} \right)^\alpha (K+1)^\alpha \ln^{\alpha-1} \frac{10nK}{\beta} \cdot \frac{M^2}{6^5 \ln \frac{10nK}{\beta}} \\
&\overset{(139)}{\le} \quad \frac{M^2}{6^5 \ln \frac{10nK}{\beta}}. 
\end{aligned}
\quad (187)
$$

Putting all together we get that $E_{T-1} \cap \widetilde{E}_{T-1,m-1}$ implies

$$\left\| \frac{\alpha_T}{n} \sum_{i=1}^{m-1} \omega_{i,T}^u \right\| \le \sqrt{③ + ④ + ⑦}, \quad ④ \overset{(171)}{\le} \frac{M}{9},$$

$$\sum_{k=0}^{T-1} \sum_{i=1}^n \widetilde{\sigma}_{i,k}^2 \overset{(170)}{\le} \frac{M^2}{6 \cdot 9^2 \ln \frac{10nK}{\beta}}, \quad \sum_{i=1}^{m-1} \bar{\sigma}_{i,T-1}^2 \overset{(187)}{\le} \frac{M^2}{6^5 \ln \frac{10nK}{\beta}}$$

In addition, we also establish (see and our induction assumption):

$$\mathbb{P}\{E_{T-1} \cap \widetilde{E}_{T-1,m-1}\} \ge 1 - \frac{(T-1)\beta}{K+1} - \frac{2(m-1)\beta}{5nK},$$

$$\mathbb{P}\{E_③\} \ge 1 - \frac{\beta}{5nK}, \quad \mathbb{P}\{E_⑦\} \ge 1 - \frac{\beta}{5nK},$$

where

$$E_\text{③} = \left\{ \text{either } \sum_{k=0}^{T-1} \sum_{i=1}^{n} \widetilde{\sigma}_{i,k}^2 > \frac{M^2}{6 \cdot 9^2 \ln \frac{10nK}{\beta}} \text{ or } |\text{④}| \le \frac{M}{9} \right\},$$

$$E_\text{⑦} = \left\{ \text{either } \sum_{i=1}^{m-1} \bar{\sigma}_{i,k}^2 > \frac{M^2}{6^5 \ln \frac{10nK}{\beta}} \text{ or } |\text{⑦}| \le \frac{M}{36} \right\}$$

Therefore, probability event $E_{T-1} \cap \widetilde{E}_{T-1,m-1} \cap E_\text{③} \cap E_\text{⑦}$ implies

$$\left\| \frac{\alpha_T}{n} \sum_{i=1}^{m-1} \omega_{i,T}^u \right\| \le \sqrt{\frac{M}{9} + \frac{M}{9} + \frac{M}{36}} \le \frac{\sqrt{M}}{2}.$$

This implies $\widetilde{E}_{T-1,m}$ and

$$
\begin{aligned}
\mathbb{P}\{ E_{T-1} \cap \widetilde{E}_{T-1,m} \} &\ge \mathbb{P}\{ E_{T-1} \cap \widetilde{E}_{T-1,m-1} \cap E_\text{③} \cap E_\text{⑦} \} \\
&= 1 - \mathbb{P}\left\{ \overline{E_{T-1} \cap \widetilde{E}_{T-1,m-1}} \cup \overline{E}_\text{③} \cup \overline{E}_\text{⑦} \right\} \\
&\ge 1 - \frac{(T-1)\beta}{K} - \frac{2m\beta}{5nK}.
\end{aligned}
$$

Therefore, for all $m = 2, \ldots, n$ the statement holds and, in particular, $\mathbb{P}\{ E_{T-1} \cap \widetilde{E}_{T-1,n} \} \ge 1 - \frac{(T-1)\beta}{K} - \frac{2\beta}{5K}$. Taking into account (181), we conclude that $E_{T-1} \cap \widetilde{E}_{T-1,n} \cap E_\text{①} \cap E_\text{③} \cap E_\text{⑥'} \cap E_\text{⑦}$ implies

$$B_T \le M$$

that is equivalent to (144) for $t = T$. Moreover,

$$
\begin{aligned}
\mathbb{P}\{ E_T \} &\ge \mathbb{P}\left\{ E_{T-1} \cap \widetilde{E}_{T-1,n} \cap E_\text{①} \cap E_\text{③} \cap E_\text{⑥'} \cap E_\text{⑦} \right\} \\
&= 1 - \mathbb{P}\left\{ \overline{E_{T-1} \cap \widetilde{E}_n} \cup \overline{E}_\text{①} \cup \overline{E}_\text{③} \cup \overline{E}_\text{⑥'} \cup \overline{E}_\text{⑦} \right\} \\
&= 1 - \frac{(T-1)\beta}{K} - \frac{2\beta}{5K} - 3 \cdot \frac{\beta}{5nK} \ge 1 - \frac{T\beta}{K}.
\end{aligned}
$$

Finally, if

$$a = \max \left\{ 2, \frac{8 \cdot 3^5 \cdot 72^4}{n} \ln^4 \frac{10nK}{\beta}, \frac{18 \cdot 6^5 \sigma K^{\frac{1}{\alpha}}(K+1)}{\sqrt{M} L n^{\frac{\alpha-1}{\alpha}}} \ln^{\frac{\alpha-1}{\alpha}} \frac{10nK}{\beta} \right\},$$

then with probability at least $1 - \beta$

$$
\begin{aligned}
\Phi(y^K) - \Phi(x^*) &\le \frac{6aLM}{K(K+3)} \\
&= \max \left\{ \frac{12LM}{K(K+3)}, \frac{162 \cdot 72^5 LM}{nK(K+3)} \ln^4 \frac{10nK}{\beta}, \frac{3 \cdot 6^7 \sigma \frac{K+1}{K+3}}{\sqrt{M}(Kn)^{\frac{\alpha-1}{\alpha}}} \ln^{\frac{\alpha-1}{\alpha}} \frac{10nK}{\beta} \right\} \\
&= \mathcal{O}\left( \max \left\{ \frac{LM}{K^2}, \frac{LM \ln^4 \frac{nK}{\beta}}{nK^2}, \frac{\sigma \sqrt{M} \ln^{\frac{\alpha-1}{\alpha}} \frac{nK}{\beta}}{n^{\frac{\alpha-1}{\alpha}} K^{\frac{\alpha-1}{\alpha}}} \right\} \right).
\end{aligned}
$$

To get $\Phi(y^K) - \Phi(x^*) \le \varepsilon$ wit probability $1 - \beta$, $K$ should be

$$\mathcal{O}\left( \max \left\{ \sqrt{\frac{LM}{\varepsilon}}, \sqrt{\frac{LM}{\varepsilon n}} \ln^2 \frac{nLM}{\varepsilon \beta}, \frac{1}{n} \left( \frac{\sigma \sqrt{M}}{\varepsilon} \right)^{\frac{\alpha}{\alpha-1}} \ln \frac{\sigma \sqrt{M}}{\varepsilon \beta} \right\} \right)$$

that concludes the proof. $\qquad \square$

## F.2 STRONGLY CONVEX CASE

In this section, we provide the complete formulation of our result for R-DProx-clipped-SSTM-shift (a restarted version for DProx-clipped-SSTM-shift) and proofs. We should mention that the results for DProx-clipped-SSTM-shift), Theorem F.1 and Lemma F.1, can be proven in the same way if we assume that $h_i^0 = \nabla f_i(x^0)$ and $M \geq \|x^0 - x^*\|^2 + C^2 \alpha_{K_0+1}^2 \frac{1}{n} \sum_{i=1}^n \|h_i^0 - \nabla f_i(x^*)\|^2$.

For the readers' convenience, the method's update rule is repeated below:

---

**Algorithm 1** Restarted DProx-clipped-SSTM-shift (R-DProx-clipped-SSTM-shift)

---

**Input:** starting point $x^0$, number of restarts $\tau$, number of steps of DProx-clipped-SSTM-shift between restarts $\{K_t\}_{t=1}^\tau$, stepsize parameters $\{a_t\}_{t=1}^\tau$, clipping levels $\{\lambda_k^1\}_{k=0}^{K_1-1}$, $\{\lambda_k^2\}_{k=0}^{K_2-1}$, ..., $\{\lambda_k^\tau\}_{k=0}^{K_\tau-1}$, smoothness constant $L$, the constant $\{N_t\}_{t=1}^\tau$.

1: $\hat{x}^0 = x^0$
2: **for** $t = 1, \ldots, \tau$ **do**
3:     Run DProx-clipped-SSTM-shift for $K_t$ iterations with stepsize parameter $a_t$, clipping levels $\{\lambda_k^t\}_{k=0}^{K_t-1}$, and starting point $\hat{x}^{t-1}$. Define the output of DProx-clipped-SSTM-shift by $\hat{x}^t$.
4: **end for**
**Output:** $\hat{x}^\tau$

---

**Theorem F.2.** *Let Assumptions 1, 2, 3 with $\mu > 0$ hold for $Q = B_{5n\sqrt{M}}(x^*)$, where $M \geq \|x^0 - x^*\|^2 + C_t^2 \alpha_{N_1+1}^2 \frac{1}{n} \sum_{i=1}^n \|h_i^0 - \nabla f_i(x^*)\|^2$ and R-DProx-clipped-SSTM-shift runs DProx-clipped-SSTM-shift $\tau$ times. Let*

$$K_t = \left\lceil \max \left\{ \sqrt{\frac{24LM_{t-1}}{\varepsilon_t}}, 2 \cdot 10^{15} \sqrt{\frac{LM_{t-1}}{n\varepsilon_t}} \ln \frac{2 \cdot 10^{16} n \sqrt{LM_{t-1}} \tau}{\sqrt{\varepsilon_t}\beta}, \right. \right.$$

$$\frac{1}{n} \left( \frac{6^8 \sigma \sqrt{M_{t-1}}}{\varepsilon_t} \right)^{\frac{\alpha}{\alpha-1}} \ln \left( \frac{10\tau}{\beta} \left( \frac{6^8 \sigma \sqrt{M_{t-1}}}{\varepsilon_t} \right)^{\frac{\alpha}{\alpha-1}} \right)$$

$$\left. \left. \frac{1}{n^{\frac{5\alpha-1}{\alpha-1}}} \left( \frac{16 \cdot 10^{24} \sigma \sqrt{M_{t-1}}}{\varepsilon_t} \right)^{\frac{\alpha}{\alpha-1}} \ln^{\frac{7\alpha-1}{\alpha-1}} \left( \frac{10\tau}{\beta} \left( \frac{16 \cdot 10^{24} \sigma \sqrt{M_{t-1}}}{\varepsilon_t} \right)^{\frac{\alpha}{\alpha-1}} \right) \right\} \right\rceil, \quad (188)$$

$$\varepsilon_t = \frac{\mu M_{t-1}}{4}, \quad M_{t-1} = \frac{M}{2^{(t-1)}}, \quad \tau = \left\lceil \log_2 \frac{\mu M}{2\varepsilon} \right\rceil, \quad \ln \frac{10 n K_t \tau}{\beta} \geq 1, \quad (189)$$

$$a \geq \max \left\{ 2, \frac{8 \cdot 3^5 \cdot 72^4}{n} \ln^4 \frac{10 n K_t}{\beta}, \frac{18 \cdot 6^5 \sigma K_t^{\frac{1}{\alpha}} (K_t + 1)}{\sqrt{M_t} L n^{\frac{\alpha-1}{\alpha}}} \ln^{\frac{\alpha-1}{\alpha}} \frac{10 n K_t}{\beta} \right\}, \quad (190)$$

$$\lambda_k^t = \frac{n \sqrt{M_t}}{72 \widetilde{\alpha}_{k+1}^t \ln \frac{10 n K_t}{\beta}}, \quad (191)$$

*for $t = 1, \ldots, \tau$, where $C_t = \frac{864}{n} \ln \frac{10 n K_t}{\beta}$, $N_t = \left\lceil \frac{3}{2} C_t^2 n \right\rceil > 0$. Then to achieve $\Phi(\hat{x}^\tau) - \Phi(x^*) \leq \varepsilon$ with probability at least $1 - \beta$ R-DProx-clipped-SSTM-shift requires*

$$\mathcal{O} \left( \max \left\{ \sqrt{\frac{L}{\mu}} \ln \left( \frac{\mu M}{\varepsilon} \right), \sqrt{\frac{L}{n\mu}} \ln \left( \frac{\mu M}{\varepsilon} \right) \ln^5 \left( \frac{\sqrt{L}}{\sqrt{\mu}\beta} \ln \left( \frac{\mu M}{\varepsilon} \right) \right), \right. \right.$$

$$\frac{1}{n} \left( \frac{\sigma^2}{\mu\varepsilon} \right)^{\frac{\alpha}{2(\alpha-1)}} \ln \left( \frac{1}{\beta} \left( \frac{\sigma^2}{\mu\varepsilon} \right)^{\frac{\alpha}{2(\alpha-1)}} \ln \left( \frac{\mu M}{\varepsilon} \right) \right),$$

$$\left. \left. \frac{1}{n^{\frac{5\alpha-1}{\alpha-1}}} \left( \frac{\sigma^2}{\mu\varepsilon} \right)^{\frac{\alpha}{2(\alpha-1)}} \ln^{\frac{7\alpha-1}{\alpha-1}} \left( \frac{1}{\beta} \left( \frac{\sigma^2}{\mu\varepsilon} \right)^{\frac{\alpha}{2(\alpha-1)}} \ln \left( \frac{\mu M}{\varepsilon} \right) \right) \right\} \right) \quad (192)$$

*iterations/oracle calls per worker. Moreover, with probability $\geq 1 - \beta$ the iterates of R-DProx-clipped-SSTM-shift at stage $t$ stay in the ball $B_{2\sqrt{M_{t-1}}}(x^*)$.*

*Proof.* The key idea behind the proof is similar to the one used in (Gorbunov et al., 2021; Sadiev et al., 2023). We prove by induction that for any $t = 1, \ldots, \tau$ with probability at least $1 - t\beta/\tau$ inequalities

$$\Phi(\hat{x}^l) - \Phi(x^*) \le \varepsilon_l, \quad \hat{M}_l \le M_l = \frac{M}{2^l} \tag{193}$$

hold for $l = 1, \ldots, t$ simultaneously. We recall the Lyapunov function is determined as

$$\hat{M}_l = \|\hat{x}^l - x^*\|^2 + C_l^2 (\alpha_{N_l+1}^l)^2 \frac{1}{n} \sum_{i=1}^n \|\nabla f_i(\hat{x}^l) - \nabla f_i(x^*)\|^2 \overset{(4)}{\le} \underbrace{(1 + C_l^2 (\alpha_{N_l+1}^l)^2 L^2)}_{\overset{\text{def}}{=} G_l} \|\hat{x}^l - x^*\|^2,$$

where by definition of $C_l, \alpha_{N_l+1}^l$, we can estimate $G_l$

$$G_l \le 2 \max \left\{ 1, \frac{9 \cdot 864^6}{4n^4} \ln^6 \frac{10nK_{l+1}\tau}{\beta} \right\} \tag{194}$$

Now, we prove the base of the induction. Theorem F.1 implies that with probability at least $1 - \beta/\tau$

$$
\begin{aligned}
G_1(\Phi(\hat{x}^1) - \Phi(x^*)) &\le G_1 \frac{6a_1 LR^2}{K_1(K_1+3)} \overset{(190)}{=} 2 \max \left\{ 1, \frac{9 \cdot 864^6}{4n^4} \ln^6 \frac{10nK_1\tau}{\beta} \right\} \\
&\quad \times \max \left\{ \frac{12LM}{K_1(K_1+3)}, \frac{162 \cdot 72^5 LM}{nK_1(K_1+3)} \ln^4 \frac{10nK_1\tau}{\beta}, \frac{3 \cdot 6^7 \sigma^{\frac{K_1+1}{K_1+3}}}{\sqrt{M}(K_1 n)^{\frac{\alpha-1}{\alpha}}} \ln^{\frac{\alpha-1}{\alpha}} \frac{10nK_1\tau}{\beta} \right\} \\
&\le \max \left\{ \frac{24LM}{K_1^2}, \frac{162 \cdot 72^5 \cdot 9 \cdot 864^6 LM}{nK_1^2} \ln^{10} \frac{10nK_1\tau}{\beta}, \right. \\
&\qquad \left. \frac{6^8 \sigma \sqrt{M} \ln^{\frac{\alpha-1}{\alpha}} \frac{10nK_1\tau}{\beta}}{(nK_1)^{\frac{\alpha-1}{\alpha}}}, \frac{81 \cdot 5184^6 \cdot \sigma \sqrt{M} \ln^{\frac{7\alpha-1}{\alpha}} \frac{10nK_1\tau}{\beta}}{n^{\frac{5\alpha-1}{\alpha}} K_1^{\frac{\alpha-1}{\alpha}}} \right\} \\
&\overset{(188)}{\le} \varepsilon_1 = \frac{\mu M}{4}
\end{aligned}
$$

and, due to the strong convexity,

$$\hat{M}_1 \le G_1 \|\hat{x}^1 - x^*\|^2 \le \frac{2G_1(\Phi(\hat{x}^1) - \Phi(x^*))}{\mu} \le \frac{M}{2} = M_1.$$

The base of the induction is proven. Now, assume that the statement holds for some $t = T < \tau$, i.e., with probability at least $1 - T\beta/\tau$ inequalities

$$\Phi(\hat{x}^l) - \Phi(x^*) \le \varepsilon_l, \quad \hat{M}_l \le M_l = \frac{M}{2^l} \tag{195}$$

hold for $l = 1, \ldots, T$ simultaneously. In particular, with probability at least $1 - T\beta/\tau$ we have $\hat{M}_T \le M_T$. Applying Theorem F.1 and using union bound for probability events, we get that with probability at least $1 - (T+1)\beta/\tau$

$$
\begin{aligned}
G_{T+1}(\Phi(\hat{x}^{T+1}) - \Phi(x^*)) &\le G_{T+1} \frac{6a_{T+1} LM_T^2}{K_{T+1}(K_{T+1}+3)} \overset{(190)}{=} 2 \max \left\{ 1, \frac{9 \cdot 864^6}{4n^4} \ln^6 \frac{10nK_{T+1}\tau}{\beta} \right\} \\
&\quad \times \max \left\{ \frac{12LM_T}{K_{T+1}(K_{T+1}+3)}, \frac{162 \cdot 72^5 LM_{T+1}}{nK_{T+1}(K_{T+1}+3)} \ln^4 \frac{10nK_{T+1}\tau}{\beta}, \right. \\
&\qquad \left. \frac{3 \cdot 6^7 \sigma^{\frac{K_{T+1}+1}{K_{T+1}+3}}}{\sqrt{M}(K_{T+1} n)^{\frac{\alpha-1}{\alpha}}} \ln^{\frac{\alpha-1}{\alpha}} \frac{10nK_{T+1}\tau}{\beta} \right\} \\
&\le \max \left\{ \frac{24LM_T}{K_{T+1}^2}, \frac{162 \cdot 72^5 \cdot 9 \cdot 864^6 LM_{T+1}}{nK_{T+1}^2} \ln^{10} \frac{10nK_{T+1}\tau}{\beta}, \right. \\
&\qquad \left. \frac{6^8 \sigma \sqrt{M_T} \ln^{\frac{\alpha-1}{\alpha}} \frac{10nK_{T+1}\tau}{\beta}}{(nK_{T+1})^{\frac{\alpha-1}{\alpha}}}, \frac{81 \cdot 5184^6 \cdot \sigma \sqrt{M_T} \ln^{\frac{7\alpha-1}{\alpha}} \frac{10nK_T\tau}{\beta}}{n^{\frac{5\alpha-1}{\alpha}} K_{T+1}^{\frac{\alpha-1}{\alpha}}} \right\} \\
&\overset{(188)}{\le} \varepsilon_{T+1} = \frac{\mu M_T}{4}
\end{aligned}
$$

and, due to the strong convexity,

$$\hat{M}_{T+1} \le G_{T+1}\|\hat{x}^{T+1} - x^*\|^2 \le \frac{2G_{T+1}(\Phi(\hat{x}^{T+1}) - \Phi(x^*))}{\mu} \le \frac{M_T}{2} = M_{T+1}.$$

Thus, we finished the inductive part of the proof. In particular, with probability at least $1 - \beta$ inequalities

$$\Phi(\hat{x}^l) - \Phi(x^*) \le \varepsilon_l, \quad \hat{M}_l \le M_l = \frac{M}{2^l}$$

hold for $l = 1, \dots, \tau$ simultaneously, which gives for $l = \tau$ that with probability at least $1 - \beta$

$$\Phi(\hat{x}^l) - \Phi(x^*) \le \varepsilon_\tau = \frac{\mu M_{\tau-1}}{4} = \frac{\mu M}{2^{\tau+1}} \overset{(189)}{\le} \varepsilon.$$

It remains to calculate the overall number of oracle calls during all runs of clipped-SSTM. We have

$$
\begin{aligned}
\sum_{t=1}^{\tau} K_t &= \mathcal{O}\Bigg( \sum_{t=1}^{\tau} \max\Bigg\{ \sqrt{\frac{LM_{t-1}}{\varepsilon_t}}, \sqrt{\frac{LM_{t-1}^2}{n\varepsilon_t}} \ln^5\left( \frac{n\sqrt{LM_{t-1}^2\tau}}{\sqrt{\varepsilon_t}\beta} \right), \\
&\qquad \frac{1}{n}\left( \frac{\sigma\sqrt{M_{t-1}}}{\varepsilon_t} \right)^{\frac{\alpha}{\alpha-1}} \ln\left( \frac{\tau}{\beta}\left( \frac{\sigma\sqrt{M_{t-1}}}{\varepsilon_t} \right)^{\frac{\alpha}{\alpha-1}} \right), \\
&\qquad \frac{1}{n^{\frac{7\alpha-1}{\alpha-1}}}\left( \frac{\sigma\sqrt{M_{t-1}}}{\varepsilon_t} \right)^{\frac{\alpha}{\alpha-1}} \ln^{\frac{5\alpha-1}{\alpha-1}}\left( \frac{\tau}{\beta}\left( \frac{\sigma\sqrt{M_{t-1}}}{\varepsilon_t} \right)^{\frac{\alpha}{\alpha-1}} \right) \Bigg\} \Bigg) \\
&= \mathcal{O}\Bigg( \sum_{t=1}^{\tau} \max\Bigg\{ \sqrt{\frac{L}{\mu}}, \sqrt{\frac{L}{n\mu}}\ln\left( \frac{n\sqrt{L}\tau}{\sqrt{\mu}\beta} \right), \frac{1}{n}\left( \frac{\sigma}{\mu\sqrt{M_{t-1}}} \right)^{\frac{\alpha}{\alpha-1}} \ln\left( \frac{\tau}{\beta}\left( \frac{\sigma}{\mu\sqrt{M_{t-1}}} \right)^{\frac{\alpha}{\alpha-1}} \right), \\
&\qquad \frac{1}{n^{\frac{5\alpha-1}{\alpha-1}}}\left( \frac{\sigma}{\mu\sqrt{M_{t-1}}} \right)^{\frac{\alpha}{\alpha-1}} \ln^{\frac{7\alpha-1}{\alpha-1}}\left( \frac{\tau}{\beta}\left( \frac{\sigma}{\mu\sqrt{M_{t-1}}} \right)^{\frac{\alpha}{\alpha-1}} \right) \Bigg\} \Bigg) \\
&= \mathcal{O}\Bigg( \max\Bigg\{ \tau\sqrt{\frac{L}{\mu}}, \tau\sqrt{\frac{L}{n\mu}}\ln\left( \frac{n\sqrt{L}\tau}{\sqrt{\mu}\beta} \right), \frac{1}{n}\sum_{t=1}^{\tau}\left( \frac{\sigma\cdot 2^{t/2}}{\mu\sqrt{M}} \right)^{\frac{\alpha}{\alpha-1}} \ln\left( \frac{\tau}{\beta}\left( \frac{\sigma\cdot 2^{t/2}}{\mu\sqrt{M}} \right)^{\frac{\alpha}{\alpha-1}} \right), \\
&\qquad \frac{1}{n^{\frac{5\alpha-1}{\alpha-1}}}\sum_{t=1}^{\tau}\left( \frac{\sigma\cdot 2^{t/2}}{\mu R} \right)^{\frac{\alpha}{\alpha-1}} \ln^{\frac{7\alpha-1}{\alpha-1}}\left( \frac{\tau}{\beta}\left( \frac{\sigma\cdot 2^{t/2}}{\mu\sqrt{M}} \right)^{\frac{\alpha}{\alpha-1}} \right) \Bigg\} \Bigg) \\
&= \mathcal{O}\Bigg( \max\Bigg\{ \sqrt{\frac{L}{\mu}}\ln\left( \frac{\mu M}{\varepsilon} \right), \sqrt{\frac{L}{n\mu}}\ln\left( \frac{\mu M}{\varepsilon} \right)\ln^5\left( \frac{n\sqrt{L}}{\sqrt{\mu}\beta}\ln\left( \frac{\mu M}{\varepsilon} \right) \right), \\
&\qquad \frac{1}{n}\left( \frac{\sigma}{\mu\sqrt{M}} \right)^{\frac{\alpha}{\alpha-1}} \ln\left( \frac{\tau}{\beta}\left( \frac{\sigma\cdot 2^{\tau/2}}{\mu\sqrt{M}} \right)^{\frac{\alpha}{\alpha-1}} \right)\sum_{t=1}^{\tau}2^{\frac{\alpha t}{2(\alpha-1)}}, \\
&\qquad \frac{1}{n^{\frac{5\alpha-1}{\alpha-1}}}\left( \frac{\sigma}{\mu\sqrt{M}} \right)^{\frac{\alpha}{\alpha-1}} \ln^{\frac{7\alpha-1}{\alpha-1}}\left( \frac{\tau}{\beta}\left( \frac{\sigma\cdot 2^{\tau/2}}{\mu\sqrt{M}} \right)^{\frac{\alpha}{\alpha-1}} \right)\sum_{t=1}^{\tau}2^{\frac{\alpha t}{2(\alpha-1)}} \Bigg\} \Bigg) \\
&= \mathcal{O}\Bigg( \max\Bigg\{ \sqrt{\frac{L}{\mu}}\ln\left( \frac{\mu M}{\varepsilon} \right), \sqrt{\frac{L}{n\mu}}\ln\left( \frac{\mu M}{\varepsilon} \right)\ln\left( \frac{\sqrt{L}}{\sqrt{\mu}\beta}\ln\left( \frac{\mu R^2}{\varepsilon} \right) \right), \\
&\qquad \frac{1}{n}\left( \frac{\sigma}{\mu\sqrt{M}} \right)^{\frac{\alpha}{\alpha-1}} \ln\left( \frac{\tau}{\beta}\left( \frac{\sigma}{\mu\sqrt{M}} \right)^{\frac{\alpha}{\alpha-1}}\cdot 2^{\frac{\alpha}{2(\alpha-1)}} \right)2^{\frac{\alpha\tau}{2(\alpha-1)}}, \\
&\qquad \frac{1}{n^{\frac{5\alpha-1}{\alpha-1}}}\left( \frac{\sigma}{\mu\sqrt{M}} \right)^{\frac{\alpha}{\alpha-1}} \ln^{\frac{7\alpha-1}{\alpha-1}}\left( \frac{\tau}{\beta}\left( \frac{\sigma}{\mu\sqrt{M}} \right)^{\frac{\alpha}{\alpha-1}}\cdot 2^{\frac{\alpha}{2(\alpha-1)}} \right)2^{\frac{\alpha\tau}{2(\alpha-1)}} \Bigg\} \Bigg).
\end{aligned}
$$

Thus, we have

$$
\begin{aligned}
\sum_{t=1}^{\tau} K_t \;=\; \mathcal{O}\Bigg( \max\Bigg\{ & \sqrt{\frac{L}{\mu}} \ln\left(\frac{\mu M}{\varepsilon}\right), \sqrt{\frac{L}{n\mu}} \ln\left(\frac{\mu M}{\varepsilon}\right) \ln^5\left(\frac{\sqrt{L}}{\sqrt{\mu}\beta} \ln\left(\frac{\mu M}{\varepsilon}\right)\right), \\
& \frac{1}{n}\left(\frac{\sigma^2}{\mu\varepsilon}\right)^{\frac{\alpha}{2(\alpha-1)}} \ln\left(\frac{1}{\beta}\left(\frac{\sigma^2}{\mu\varepsilon}\right)^{\frac{\alpha}{2(\alpha-1)}} \ln\left(\frac{\mu M}{\varepsilon}\right)\right), \\
& \frac{1}{n^{\frac{5\alpha-1}{\alpha-1}}}\left(\frac{\sigma^2}{\mu\varepsilon}\right)^{\frac{\alpha}{2(\alpha-1)}} \ln^{\frac{7\alpha-1}{\alpha-1}}\left(\frac{1}{\beta}\left(\frac{\sigma^2}{\mu\varepsilon}\right)^{\frac{\alpha}{2(\alpha-1)}} \ln\left(\frac{\mu M}{\varepsilon}\right)\right) \Bigg\} \Bigg),
\end{aligned}
$$

which concludes the proof. $\square$

# G  MISSING PROOFS FOR DProx-clipped-SGDA-shift

## G.1  COCOERCIVE CASE

In this section, we give the complete formulations of our results for DProx-clipped-SGDA-shift and rigorous proofs. For the readers' convenience, the method's update rule is repeated below:

$$x^{k+1} = \text{prox}_{\gamma\Psi}\left(x^k - \gamma\widetilde{g}^k\right), \quad \text{where } \widetilde{g}^k = \frac{1}{n}\sum_{i=1}^n \widetilde{g}_i^k, \ \widetilde{g}_i^k = h_i^k + \hat{\Delta}_i^k,$$

$$h_i^{k+1} = h_i^k + \nu\hat{\Delta}_i^k, \quad \hat{\Delta}_i^k = \text{clip}\left(F_{\xi_i^k}(x^k) - h_i^k, \lambda_k\right).$$

**Lemma G.1.** *Let Assumptions 7, 9 and 10 hold for $Q = B_{3\sqrt{V}}(x^*)$, where $V \geq \|x^0 - x^*\|^2 + \frac{25600\gamma^2 \ln^2 \frac{48n(K+1)}{\beta}}{n^2}\sum_{i=1}^n \|F_i(x^*)\|^2$ and $0 < \gamma \leq 1/\ell$. If $x^k$ lies in $B_{3\sqrt{V}}(x^*)$ for all $k = 0, 1, \ldots, K-1$ for some $K \geq 0$, then for all $u \in B_{3\sqrt{V}}(x^*)$ the iterates produced by DProx-clipped-SGDA-shift satisfy*

$$\langle F(u), x_{avg}^K - u\rangle + \Psi(x_{avg}^K) - \Psi(u) \leq \frac{\|x^0 - u\|^2 - \|x^K - u\|^2}{2\gamma K} + \frac{\gamma}{K}\sum_{k=0}^{K-1}\|\omega_k\|^2$$

$$+ \frac{1}{K}\sum_{k=0}^{K-1}\langle x^k - u, \omega_k\rangle, \quad (196)$$

$$x_{avg}^K \stackrel{def}{=} \frac{1}{K}\sum_{k=0}^{K-1} x^{k+1}, \quad (197)$$

$$\omega_k \stackrel{def}{=} F(x^k) - \widetilde{g}^k. \quad (198)$$

*Proof.* The proof of this lemma follows the proof of Theorem D.3 from (Beznosikov et al., 2023). For completeness, we provide here the full proof. We start with the application of Lemma B.4 with $x^+ = x^{k+1}$, $x = x^k - \gamma g^k$, and $y = u$ for arbitrary $u \in B_{3\sqrt{V}}(x^*)$:

$$\langle x^{k+1} - x^k + \gamma\widetilde{g}^k, u - x^{k+1}\rangle \geq \gamma\left(\Psi(x^{k+1}) - \Psi(u)\right).$$

Rearranging the terms, we get

$$2\gamma\left(\Psi(x^{k+1}) - \Psi(u)\right) \leq 2\gamma\langle\widetilde{g}^k, u - x^k\rangle + 2\langle x^{k+1} - x^k, u - x^k\rangle$$
$$+ 2\langle x^{k+1} - x^k + \gamma\widetilde{g}^k, x^k - x^{k+1}\rangle$$

implying

$$2\gamma\left(\langle F(x^k), x^k - u\rangle + \Psi(x^{k+1}) - \Psi(u)\right) \leq 2\langle x^{k+1} - x^k, u - x^k\rangle + 2\gamma\langle F(x^k) - \widetilde{g}^k, x^k - u\rangle$$
$$+ 2\langle x^{k+1} - x^k + \gamma\widetilde{g}^k, x^k - x^{k+1}\rangle$$
$$= \|x^{k+1} - x^k\|^2 + \|x^k - u\|^2 - \|x^{k+1} - u\|^2$$
$$+ 2\gamma\langle F(x^k) - \widetilde{g}^k, x^k - u\rangle$$
$$- 2\|x^{k+1} - x^k\|^2 + 2\gamma\langle\widetilde{g}^k, x^k - x^{k+1}\rangle$$
$$= \|x^k - u\|^2 - \|x^{k+1} - u\|^2 - \|x^{k+1} - x^k\|^2$$
$$+ 2\gamma\langle F(x^k) - \widetilde{g}^k, x^k - u\rangle$$
$$+ 2\gamma\langle F(u), x^k - x^{k+1}\rangle$$
$$+ 2\gamma\langle\widetilde{g}^k - F(u), x^k - x^{k+1}\rangle$$
$$\leq \|x^k - u\|^2 - \|x^{k+1} - u\|^2$$
$$+ 2\gamma\langle F(x^k) - \widetilde{g}^k, x^k - u\rangle$$
$$+ 2\gamma\langle F(u), x^k - x^{k+1}\rangle + \gamma^2\|\widetilde{g}^k - F(u)\|^2,$$

where in the last step we apply $2\gamma\langle \widetilde{g}^k - F(u), x^k - x^{k+1}\rangle \leq \gamma^2\|\widetilde{g}^k - F(u)\|^2 + \|x^k - x^{k+1}\|^2$. Adding $2\gamma\langle F(u), x^{k+1} - u\rangle - 2\gamma\langle F(x^k), x^k - u\rangle$ to the both sides, we derive

$$
\begin{aligned}
2\gamma\left(\langle F(u), x^{k+1} - u\rangle + \Psi(x^{k+1}) - \Psi(u)\right) &\leq \|x^k - u\|^2 - \|x^{k+1} - u\|^2 \\
&\quad + 2\gamma\langle F(u) - \widetilde{g}^k, x^k - u\rangle + \gamma^2\|\widetilde{g}^k - F(u)\|^2 \\
&= \|x^k - u\|^2 - \|x^{k+1} - u\|^2 \\
&\quad - 2\gamma\langle F(x^k) - F(u), x^k - u\rangle + \gamma^2\|\widetilde{g}^k - F(u)\|^2 \\
&\quad + 2\gamma\langle F(x^k) - \widetilde{g}^k, x^k - u\rangle \\
&\overset{(36)}{\leq} \|x^k - u\|^2 - \|x^{k+1} - u\|^2 \\
&\quad - \frac{2\gamma}{\ell}\|F(x^k) - F(u)\|^2 + 2\gamma^2\|F(x^k) - F(u)\|^2 \\
&\quad + 2\gamma\langle F(x^k) - \widetilde{g}^k, x^k - u\rangle + 2\gamma^2\|F(x^k) - \widetilde{g}^k\|^2 \\
&\leq \|x^k - u\|^2 - \|x^{k+1} - u\|^2 \\
&\quad + 2\gamma\langle \omega_k, x^k - u\rangle + 2\gamma^2\|\omega_k\|^2.
\end{aligned}
$$

Next, we sum up the above inequality for $k = 0, 1, \ldots, K-1$ and divide both sides by $2\gamma K$:

$$
\begin{aligned}
\frac{1}{K}\sum_{k=0}^{K-1}\left(\langle F(u), x^{k+1} - u\rangle + \Psi(x^{k+1}) - \Psi(u)\right) &\leq \frac{\|x^0 - u\|^2 - \|x^K - u\|^2}{2\gamma K} + \frac{\gamma}{K}\sum_{k=0}^{K-1}\|\omega_k\|^2 \\
&\quad + \frac{1}{K}\sum_{k=0}^{K-1}\langle x^k - u, \omega_k\rangle,
\end{aligned}
$$

To finish the proof, we need to use Jensen's inequality $\Psi\left(\frac{1}{K}\sum_{k=0}^{K-1} x^{k+1}\right) \leq \frac{1}{K}\sum_{k=0}^{K-1}\Psi(x^{k+1})$:

$$
\begin{aligned}
\langle F(u), x_{\mathrm{avg}}^K - u\rangle + \Psi(x_{\mathrm{avg}}^K) - \Psi(u) &\leq \frac{\|x^0 - u\|^2 - \|x^K - u\|^2}{2\gamma K} + \frac{\gamma}{K}\sum_{k=0}^{K-1}\|\omega_k\|^2 \\
&\quad + \frac{1}{K}\sum_{k=0}^{K-1}\langle x^k - u, \omega_k\rangle,
\end{aligned}
$$

where $x_{\mathrm{avg}}^K = \frac{1}{K}\sum_{k=0}^{K-1} x^{k+1}$. □

**Theorem G.1.** *Let Assumptions 7, 9, and 10 hold for $Q = B_{3\sqrt{V}}(x^*)$, where $V \geq \|x^0 - x^*\|^2 + \frac{25600\gamma^2 \ln^2 \frac{48n(K+1)}{\beta}}{n^2}\sum_{i=1}^n \|F_i(x^*)\|^2$, and*

$$
0 < \gamma \leq \min\left\{\frac{1}{480\ell \ln \frac{48n(K+1)}{\beta}}, \frac{\sqrt{V} n^{\frac{\alpha-1}{\alpha}}}{(86400)^{\frac{1}{\alpha}}(K+1)^{\frac{1}{\alpha}}\sigma \ln^{\frac{\alpha-1}{\alpha}} \frac{48n(K+1)}{\beta}}\right\}, \quad (199)
$$

$$
\lambda_k \equiv \lambda = \frac{n\sqrt{V}}{40\gamma \ln \frac{48n(K+1)}{\beta}}, \quad (200)
$$

*for some $K \geq 0$ and $\beta \in (0, 1]$. Then, after $K$ iterations the iterates produced by* DProx-clipped-SGDA-shift *with probability at least $1 - \beta$ satisfy*

$$
\mathrm{Gap}_{\sqrt{V}}(x_{avg}^{K+1}) \leq \frac{4V}{\gamma(K+1)} \quad \text{and} \quad \{x^k\}_{k=0}^{K+1} \subseteq B_{3\sqrt{V}}(x^*), \quad (201)
$$

*where $x_{avg}^{K+1}$ is defined in (197). In particular, when $\gamma$ equals the minimum from (199), then the iterates produced by* DProx-clipped-SGDA-shift *after $K$ iterations with probability at least $1 - \beta$ satisfy*

$$
\mathrm{Gap}_{\sqrt{V}}(x_{avg}^{K+1}) = \mathcal{O}\left(\max\left\{\frac{\ell V \ln \frac{nK}{\beta}}{K}, \frac{\sigma\sqrt{V}\ln^{\frac{\alpha-1}{\alpha}}\frac{nK}{\beta}}{n^{\frac{\alpha-1}{\alpha}}K^{\frac{\alpha-1}{\alpha}}}\right\}\right), \quad (202)
$$

*meaning that to achieve* $\mathtt{Gap}_{\sqrt{V}}(x_{\text{avg}}^{K+1}) \leq \varepsilon$ *with probability at least* $1 - \beta$ DProx-clipped-SGDA-shift *requires*

$$K = \mathcal{O}\left(\frac{\ell V}{\varepsilon}\ln\frac{n\ell V}{\varepsilon\beta}, \frac{1}{n}\left(\frac{\sigma\sqrt{V}}{\varepsilon}\right)^{\frac{\alpha}{\alpha-1}}\ln\left(\frac{1}{\beta}\left(\frac{\sigma\sqrt{V}}{\varepsilon}\right)^{\frac{\alpha}{\alpha-1}}\right)\right) \quad \text{iterations/oracle calls.} \quad (203)$$

*Proof.* The key idea behind the proof is similar to the one used in (Gorbunov et al., 2022a; Sadiev et al., 2023): we prove by induction that the iterates do not leave some ball and the sums decrease as $1/K+1$. To formulate the statement rigorously, we introduce probability event $E_k$ for each $k = 0, 1, \ldots, K+1$ as follows: inequalities

$$\underbrace{\max_{u \in B_{\sqrt{V}}(x^*)}\left\{\|x^0 - u\|^2 + 2\gamma\sum_{l=0}^{t-1}\langle x^l - u, \omega_l\rangle + 2\gamma^2\sum_{l=0}^{t-1}\|\omega_l\|^2\right\}}_{A_t} \leq 8V, \quad (204)$$

$$\left\|\gamma\sum_{l=0}^{t-1}\omega_l\right\| \leq \sqrt{V}, \quad (205)$$

$$\left\|\gamma\sum_{i=1}^{r-1}\omega_{i,t-1}^u\right\| \leq \frac{\sqrt{V}}{2} \quad (206)$$

hold for $t = 0, 1, \ldots, k$ and $r = 1, 2, \ldots, n$ simultaneously, where

$$\omega_l = \omega_l^u + \omega_l^b, \quad (207)$$

$$\omega_l^u \stackrel{\text{def}}{=} \frac{1}{n}\sum_{i=1}^n \omega_{i,l}^u, \quad \omega_l^b \stackrel{\text{def}}{=} \frac{1}{n}\sum_{i=1}^n \omega_{i,l}^b, \quad (208)$$

$$\omega_{i,l}^u \stackrel{\text{def}}{=} \mathbb{E}_{\xi_i^l}\left[\widetilde{g}_i^l\right] - \widetilde{g}_i^l, \quad \omega_{i,l}^b \stackrel{\text{def}}{=} F_i(x^l) - \mathbb{E}_{\xi_i^l}\left[\widetilde{g}_i^l\right] \quad \forall\, i \in [n]. \quad (209)$$

We will prove by induction that $\mathbb{P}\{E_k\} \geq 1 - k\beta/(K+1)$ for all $k = 0, 1, \ldots, K+1$. The base of induction follows immediately: for all $u \in B_{\sqrt{V}}(x^*)$ we have $\|x^0 - u\|^2 \leq 2\|x^0 - x^*\|^2 + 2\|x^* - u\|^2 \leq 4V < 8V$ and for $k = 0$ we have $\|\gamma\sum_{l=0}^{k-1}\omega_l\| = 0$. Next, we assume that the statement holds for $k = T-1 \leq K$, i.e., $\mathbb{P}\{E_{T-1}\} \geq 1 - (T-1)\beta/(K+1)$. Let us show that it also holds for $k = T$, i.e., $\mathbb{P}\{E_T\} \geq 1 - T\beta/(K+1)$.

To proceed, we need to show that $E_{T-1}$ implies $\|x^t - x^*\| \leq 3\sqrt{V}$ for all $t = 0, 1, \ldots, T-1$. We will use the induction argument as well. The base is already proven. Next, we assume that $\|x^t - x^*\| \leq 3\sqrt{V}$ for all $t = 0, 1, \ldots, t'$ for some $t' < T-1$. This means that $x^t \in B_{3\sqrt{V}}(x^*)$ for $t = 0, 1, \ldots, t'$ and we can apply Lemma G.1: $E_{T-1}$ implies

$$\max_{u \in B_{\sqrt{V}}(x^*)}\left\{2\gamma(t'+1)\left(\langle F(u), x_{\text{avg}}^{t'+1} - u\rangle + \Psi(x_{\text{avg}}^{t'+1}) - \Psi(u)\right) + \|x^{t'+1} - u\|^2\right\}$$

$$\leq \max_{u \in B_{\sqrt{V}}(x^*)}\left\{\|x^0 - u\|^2 + 2\gamma\sum_{l=0}^t \langle x^l - u, \omega_l\rangle\right\}$$

$$+ 2\gamma^2\sum_{l=0}^t \|\omega_l\|^2$$

$$\stackrel{(204)}{\leq} 8V.$$

that gives

$$\|x^{t'+1} - x^*\|^2 \leq \max_{u \in B_{\sqrt{V}}(x^*)}\left\{2\gamma(t'+1)\left(\langle F(u), x_{\text{avg}}^{t'} - u\rangle + \Psi(x_{\text{avg}}^{t'}) - \Psi(u)\right) + \|x^{t'+1} - u\|^2\right\}$$

$$\leq 8V.$$

That is, we showed that $E_{T-1}$ implies $\|x^t - x^*\| \leq 3\sqrt{V}$ and

$$\max_{u \in B_{\sqrt{V}}(x^*)} \left\{ 2\gamma t \left( \langle F(u), \widetilde{x}_{\text{avg}}^t - u \rangle + \Psi(\widetilde{x}_{\text{avg}}^t) - \Psi(u) \right) + \|x^{t+1} - u\|^2 \right\} \leq 8V \qquad (210)$$

for all $t = 0, 1, \ldots, T - 1$. Before we proceed, we introduce a new notation:

$$\eta_t = \begin{cases} x^t - x^*, & \text{if } \|x^t - x^*\| \leq 3\sqrt{V}, \\ 0, & \text{otherwise,} \end{cases}$$

for all $t = 0, 1, \ldots, T - 1$. Random vectors $\{\eta_t\}_{t=0}^T$ are bounded almost surely:

$$\|\eta_t\| \leq 3\sqrt{V} \qquad (211)$$

for all $t = 0, 1, \ldots, T$. In addition, $\eta_t = x^t - x^*$ follows from $E_{T-1}$ for all $t = 0, 1, \ldots, T$ and, thus, $E_{T-1}$ implies

$$A_T \overset{(204)}{=} \max_{u \in B_{\sqrt{V}}(x^*)} \left\{ \|x^0 - u\|^2 + 2\gamma \sum_{l=0}^{T-1} \langle x^* - u, \omega_l \rangle \right\} + 2\gamma \sum_{l=0}^{T-1} \langle x^l - x^*, \omega_l \rangle + 2\gamma^2 \sum_{l=0}^{T-1} \|\omega_l\|^2$$

$$\leq 4V + 2\gamma \max_{u \in B_{\sqrt{V}}(x^*)} \left\{ \left\langle x^* - u, \sum_{l=0}^{T-1} \omega_l \right\rangle \right\} + 2\gamma \sum_{l=0}^{T-1} \langle \eta_l, \omega_l \rangle + 2\gamma^2 \sum_{l=0}^{T-1} \|\omega_l\|^2$$

$$= 4V + 2\gamma\sqrt{V} \left\| \sum_{l=0}^{T-1} \omega_l \right\| + 2\gamma \sum_{l=0}^{T-1} \langle \eta_l, \omega_l \rangle + 2\gamma^2 \sum_{l=0}^{T-1} \|\omega_l\|^2.$$

Using the notation from (207)-(209), we can rewrite $\|\omega_l\|^2$ as

$$\|\omega_l\|^2 \leq 2\|\omega_l^u\|^2 + 2\|\omega_l^b\|^2 = \frac{2}{n} \left\| \sum_{i=1}^n \omega_{i,l}^u \right\|^2 + 2\|\omega_l^b\|^2$$

$$= \frac{2}{n^2} \sum_{i=1}^n \|\omega_{i,l}^u\|^2 + \frac{4}{n^2} \sum_{j=2}^n \left\langle \sum_{i=1}^{j-1} \omega_{i,l}^u, \omega_{j,l}^u \right\rangle + 2\|\omega_l^b\|^2. \qquad (212)$$

Putting all together, we obtain that $E_{T-1}$ implies

$$A_T \leq 4V + 2\gamma\sqrt{V} \left\| \sum_{l=0}^{T-1} \omega_l \right\| + \underbrace{\frac{2\gamma}{n} \sum_{l=0}^{T-1} \sum_{i=1}^n \langle \eta_l, \omega_{i,l}^u \rangle}_{①} + \underbrace{2\gamma \sum_{l=0}^{T-1} \langle \eta_l, \omega_l^b \rangle}_{②}$$

$$+ \underbrace{\frac{4\gamma^2}{n^2} \sum_{l=0}^{T-1} \sum_{i=1}^n \mathbb{E}_{\xi_i^l} \left[ \|\omega_{i,l}^u\|^2 \right]}_{③} + \underbrace{\frac{4\gamma^2}{n^2} \sum_{l=0}^{T-1} \sum_{i=1}^n \left( \|\omega_{i,l}^u\|^2 - \mathbb{E}_{\xi_i^l} \left[ \|\omega_{i,l}^u\|^2 \right] \right)}_{④}$$

$$+ \underbrace{4\gamma^2 \sum_{l=0}^{T-1} \|\omega_l^b\|^2}_{⑤} + \underbrace{\frac{8\gamma^2}{n^2} \sum_{l=0}^{T-1} \sum_{j=2}^n \left\langle \sum_{i=1}^{j-1} \omega_{i,l}^u, \omega_{j,l}^u \right\rangle}_{⑥}. \qquad (213)$$

To finish the proof, it remains to estimate $2\gamma\sqrt{V} \left\| \sum_{l=0}^{T-1} \omega_l \right\|$, ①, ②, ③, ④, ⑤, ⑥ with high probability. More precisely, the goal is to prove that $2\gamma\sqrt{V} \left\| \sum_{l=0}^{T-1} \omega_l \right\| + ① + ② + ③ + ④ + ⑤ + ⑥ \leq 4V$ with high probability. Before we proceed, we need to derive several useful inequalities related to $\omega_{i,l}^u, \omega_l^b$. First of all, we have

$$\|\omega_{i,l}^u\| \leq 2\lambda \qquad (214)$$

by definition of the clipping operator. Next, probability event $E_{T-1}$ implies

$$\|F_i(x^l)\| \leq \|F_i(x^l) - F_i(x^*)\| + \|F_i(x^*)\| \leq \ell\|x^l - x^*\| + \sqrt{\sum_{i=1}^n \|F_i(x^*)\|^2}$$

$$\leq 3\ell\sqrt{V} + \frac{n\sqrt{V}}{160\gamma \ln \frac{48n(K+1)}{\beta}} \overset{(199)}{\leq} \frac{n\sqrt{V}}{80\gamma \ln \frac{48n(K+1)}{\beta}} \overset{(200)}{=} \frac{\lambda}{2} \qquad (215)$$

for $l = 0, 1, \ldots, T - 1$ and $i \in [n]$. Therefore, Lemma B.2 and $E_{T-1}$ imply

$$\left\| \omega_l^b \right\| \le \frac{1}{n} \sum_{i=1}^{n} \left\| \omega_{i,l}^b \right\| \le \frac{2^\alpha \sigma^\alpha}{\lambda^{\alpha-1}}, \tag{216}$$

$$\mathbb{E}_{\xi_i^l} \left[ \left\| \omega_{i,l}^u \right\|^2 \right] \le 18\lambda^{2-\alpha} \sigma^\alpha, \tag{217}$$

for all $l = 0, 1, \ldots, T - 1$ and $i \in [n]$.

**Upper bound for ①.**  To estimate this sum, we will use Bernstein's inequality. The summands have conditional expectations equal to zero:

$$\mathbb{E}_{\xi_i^l} \left[ \frac{2\gamma}{n} \langle \eta_l, \omega_{i,l}^u \rangle \right] = \frac{2\gamma}{n} \left\langle \eta_l, \mathbb{E}_{\xi_i^l} [\omega_{i,l}^u] \right\rangle = 0.$$

Moreover, for all $l = 0, \ldots, T - 1$ random vectors $\{\omega_{i,l}^u\}_{i=1}^n$ are independent. Thus, sequence $\left\{ \frac{2\gamma}{n} \langle \eta_l, \omega_{i,l}^u \rangle \right\}_{l,i=0,1}^{T-1,n}$ is a martingale difference sequence. Next, the summands are bounded:

$$\left| \frac{2\gamma}{n} \langle \eta_l, \omega_{i,l}^u \rangle \right| \le \frac{2\gamma}{n} \|\eta_l\| \cdot \|\omega_{i,l}^u\| \stackrel{(211),(214)}{\le} \frac{12\gamma\sqrt{V}\lambda}{n} \stackrel{(200)}{\le} \frac{3V}{10 \ln \frac{48n(K+1)}{\beta}} \stackrel{\text{def}}{=} c. \tag{218}$$

Finally, conditional variances $\sigma_{i,l}^2 \stackrel{\text{def}}{=} \mathbb{E}_{\xi_i^l} \left[ \frac{4\gamma^2}{n^2} \langle \eta_l, \omega_{i,l}^u \rangle^2 \right]$ of the summands are bounded:

$$\sigma_{i,l}^2 \le \mathbb{E}_{\xi_i^l} \left[ \frac{4\gamma^2}{n^2} \|\eta_l\|^2 \cdot \|\omega_{i,l}^u\|^2 \right] \stackrel{(211)}{\le} \frac{36\gamma^2 V}{n^2} \mathbb{E}_{\xi_i^l} \left[ \|\omega_{i,l}^u\|^2 \right]. \tag{219}$$

Applying Bernstein's inequality (Lemma B.1) with $X_{i,l} = \frac{2\gamma}{n} \langle \eta_l, \omega_{i,l}^u \rangle$, constant $c$ defined in (218), $b = \frac{3V}{10}, G = \frac{3V^2}{200 \ln \frac{48n(K+1)}{\beta}}$, we get

$$\mathbb{P} \left\{ |①| > \frac{3V}{10} \text{ and } \sum_{l=0}^{T} \sum_{i=1}^{n} \sigma_{i,l}^2 \le \frac{3V^2}{200 \ln \frac{48n(K+1)}{\beta}} \right\} \le 2 \exp \left( -\frac{b^2}{2G + {}^{2cb}/{3}} \right) = \frac{\beta}{24n(K+1)}.$$

The above is equivalent to

$$\mathbb{P}\{E_①\} \ge 1 - \frac{\beta}{24n(K+1)}, \text{ for } E_① = \left\{ \text{either} \quad \sum_{l=0}^{T} \sum_{i=1}^{n} \sigma_{i,l}^2 > \frac{3V^2}{200 \ln \frac{48n(K+1)}{\beta}} \quad \text{or} \quad |①| \le \frac{3V}{10} \right\}. \tag{220}$$

Moreover, $E_{T-1}$ implies

$$\begin{aligned}
\sum_{l=0}^{T} \sum_{i=1}^{n} \sigma_{i,l}^2 \quad &\stackrel{(219)}{\le} \quad \frac{36\gamma^2 V}{n^2} \sum_{l=0}^{T} \sum_{i=1}^{n} \mathbb{E}_{\xi_i^l} \left[ \|\omega_{i,l}^u\|^2 \right] \\
&\stackrel{(217), T \le K+1}{\le} \quad \frac{648\gamma^2 V \sigma^\alpha (K+1) \lambda^{2-\alpha}}{n} \\
&\stackrel{(200)}{\le} \quad \frac{648\gamma^\alpha \sqrt{V}^{4-\alpha} \sigma^\alpha (K+1) \ln^{\alpha-2} \frac{48n(K+1)}{\beta}}{40^{2-\alpha} n^{\alpha-1}} \\
&\stackrel{(199)}{\le} \quad \frac{3V^2}{200 \ln \frac{48n(K+1)}{\beta}}.
\end{aligned} \tag{221}$$

**Upper bound for ②.**  Probability event $E_{T-1}$ implies

$$\begin{aligned}
② \quad &\le \quad 2\gamma \sum_{l=0}^{T} \|\eta_l\| \cdot \|\omega_l^b\| \stackrel{(211),(216),T \le K+1}{\le} 6 \cdot 2^\alpha \gamma \sqrt{V} (K+1) \frac{\sigma^\alpha}{\lambda^{\alpha-1}} \\
&\stackrel{(200)}{=} \quad \frac{6 \cdot 40^{\alpha-1} \cdot 2^\alpha}{n^{\alpha-1}} \gamma^\alpha \sigma^\alpha \sqrt{V}^{2-\alpha} (K+1) \ln^{\alpha-1} \left( \frac{48n(K+1)}{\beta} \right) \stackrel{(199)}{\le} \frac{3V}{100}.
\end{aligned} \tag{222}$$

**Upper bound for ③.** Probability event $E_{T-1}$ implies

$$
③ = \frac{4\gamma^2}{n^2} \sum_{l=0}^{T} \sum_{i=1}^{n} \mathbb{E}_{\xi_i^l} \left[ \|\omega_{i,l}^u\|^2 \right] \overset{(217), T \leq K+1}{\leq} \frac{72\gamma^2 \lambda^{2-\alpha} \sigma^\alpha (K+1)}{n}
$$

$$
\overset{(200)}{\leq} \frac{72}{40^{2-\alpha} n^{\alpha-1}} \gamma^\alpha \sqrt{V}^{2-\alpha} \sigma^\alpha (K+1) \ln^{\alpha-2} \left( \frac{48n(K+1)}{\beta} \right) \overset{(199)}{\leq} \frac{3V}{100}. \tag{223}
$$

**Upper bound for ④.** To estimate this sum, we will use Bernstein's inequality. The summands have conditional expectations equal to zero:

$$
\frac{4\gamma^2}{n^2} \mathbb{E}_{\xi_i^l} \left[ \|\omega_{i,l}^u\|^2 - \mathbb{E}_{\xi_i^l} \left[ \|\omega_{i,l}^u\|^2 \right] \right] = 0.
$$

Moreover, for all $l = 0, \ldots, T-1$ random vectors $\{\omega_{i,l}^u\}_{i=1}^n$ are independent. Thus, sequence $\left\{ \frac{4\gamma^2}{n^2} \left( \|\omega_{i,l}^u\|^2 - \mathbb{E}_{\xi_i^l} \left[ \|\omega_{i,l}^u\|^2 \right] \right) \right\}_{l,i=0,1}^{T-1,n}$ is a martingale difference sequence. Next, the summands are bounded:

$$
\frac{4\gamma^2}{n^2} \left| \|\omega_{i,l}^u\|^2 - \mathbb{E}_{\xi_i^l} \left[ \|\omega_{i,l}^u\|^2 \right] \right| \leq \frac{4\gamma^2}{n^2} \left( \|\omega_{i,l}^u\|^2 + \mathbb{E}_{\xi_i^l} \left[ \|\omega_{i,l}^u\|^2 \right] \right) \overset{(214)}{\leq} \frac{32\gamma^2 \lambda^2}{n^2}
$$

$$
\overset{(200)}{\leq} \frac{V}{20 \ln^2 \frac{48n(K+1)}{\beta}} \leq \frac{V}{10 \ln \frac{48n(K+1)}{\beta}} \overset{\text{def}}{=} c. \tag{224}
$$

Finally, conditional variances

$$
\widetilde{\sigma}_{i,l}^2 \overset{\text{def}}{=} \frac{16\gamma^4}{n^2} \mathbb{E}_{\xi_i^l} \left[ \left( \|\omega_{i,l}^u\|^2 - \mathbb{E}_{\xi_i^l} \left[ \|\omega_{i,l}^u\|^2 \right] \right)^2 \right]
$$

of the summands are bounded:

$$
\widetilde{\sigma}_{i,l}^2 \overset{(224)}{\leq} \frac{\gamma^2 V}{5n^2 \ln^2 \frac{48n(K+1)}{\beta}} \mathbb{E}_{\xi_i^l} \left[ \left| \|\omega_{i,l}^u\|^2 - \mathbb{E}_{\xi_i^l} \left[ \|\omega_{i,l}^u\|^2 \right] \right| \right] \leq \frac{2\gamma^2 V}{5n^2 \ln^2 \frac{48n(K+1)}{\beta}} \mathbb{E}_{\xi_i^l} \left[ \|\omega_{i,l}^u\|^2 \right]. \tag{225}
$$

Applying Bernstein's inequality (Lemma B.1) with $X_{i,l} = \frac{4\gamma^2}{n^2} \left( \|\omega_{i,l}^u\|^2 - \mathbb{E}_{\xi_i^l}[\|\omega_{i,l}^u\|^2] \right)$, constant $c$ defined in (224), $b = \frac{V}{10}$, $G = \frac{V^2}{600 \ln \frac{48n(K+1)}{\beta}}$, we get

$$
\mathbb{P} \left\{ |④| > \frac{V}{10} \text{ and } \sum_{t=0}^{T} \sum_{i=1}^{n} \widetilde{\sigma}_{i,t}^2 \leq \frac{V^2}{600 \ln \frac{48n(K+1)}{\beta}} \right\} \leq 2 \exp \left( -\frac{b^2}{2G + 2cb/3} \right) = \frac{\beta}{24n(K+1)}.
$$

The above is equivalent to

$$
\mathbb{P}\{E_④\} \geq 1 - \frac{\beta}{24n(K+1)}, \text{ for } E_④ = \left\{ \text{either} \quad \sum_{l=0}^{T} \sum_{i=1}^{n} \widetilde{\sigma}_{i,l}^2 > \frac{V^2}{600 \ln \frac{48n(K+1)}{\beta}} \quad \text{or} \quad |④| \leq \frac{V}{10} \right\}. \tag{226}
$$

Moreover, $E_{T-1}$ implies

$$
\sum_{l=0}^{T} \sum_{i=1}^{n} \widetilde{\sigma}_{i,l}^2 \overset{(225)}{\leq} \frac{2\gamma^2 V}{5n^2 \ln^2 \frac{48n(K+1)}{\beta}} \sum_{l=0}^{T} \sum_{i=1}^{n} \mathbb{E}_{\xi_i^l} \left[ \|\omega_{i,l}^u\|^2 \right] \overset{(217), T \leq K+1}{\leq} \frac{36\gamma^2 V(K+1)}{5n \ln \frac{48n(K+1)}{\beta}} \lambda^{2-\alpha} \sigma^\alpha
$$

$$
\overset{(200)}{\leq} \frac{9 \cdot 40^\alpha \sqrt{2}^\alpha}{2000 n^{\alpha-1}} \gamma^\alpha \sqrt{V}^{4-\alpha} (K+1) \sigma^\alpha \ln^{\alpha-4} \frac{48n(K+1)}{\beta}
$$

$$
\overset{(199)}{\leq} \frac{V^2}{600 \ln \frac{48n(K+1)}{\beta}}. \tag{227}
$$

**Upper bound for ⑤.** Probability event $E_{T-1}$ implies

$$
\begin{aligned}
⑤ \;=\; & 4\gamma^2 \sum_{l=0}^{T} \|\omega_l^b\|^2 \overset{(216),T\leq K+1}{\leq} 2^{2\alpha+2}\gamma^2(K+1)\frac{\sigma^{2\alpha}}{\lambda^{2\alpha-2}} \\
& \overset{(200)}{=} \frac{12800^\alpha}{800}\gamma^{2\alpha}(K+1)\frac{\sigma^{2\alpha}}{n^{2\alpha-2}\sqrt{V}^{2\alpha-2}}\ln^{2\alpha-2}\frac{48n(K+1)}{\beta} \\
& \overset{(199)}{\leq} \frac{V}{10}.
\end{aligned}
\tag{228}
$$

**Upper bound for ⑥.** This sum requires more refined analysis. We introduce new vectors:

$$
\delta_j^l = \begin{cases} \frac{\gamma}{n}\sum_{i=1}^{j-1}\omega_{i,l}^u, & \text{if } \left\|\frac{\gamma}{n}\sum_{i=1}^{j-1}\omega_{i,l}^u\right\| \leq \frac{\sqrt{V}}{2}, \\ 0, & \text{otherwise,} \end{cases}
\tag{229}
$$

for all $j \in [n]$ and $l = 0, \ldots, T-1$. Then, by definition

$$
\|\delta_j^l\| \leq \frac{\sqrt{V}}{2}
\tag{230}
$$

and

$$
⑥ \;=\; \underbrace{\frac{8\gamma}{n}\sum_{l=0}^{T-1}\sum_{j=2}^{n}\left\langle \delta_j^l, \omega_{j,l}^u\right\rangle}_{⑥'} + \frac{8\gamma}{n}\sum_{l=0}^{T-1}\sum_{j=2}^{n}\left\langle \frac{\gamma}{n}\sum_{i=1}^{j-1}\omega_{i,l}^u - \delta_j^l, \omega_{j,l}^u\right\rangle.
\tag{231}
$$

We also note here that $E_{T-1}$ implies

$$
\frac{8\gamma}{n}\sum_{l=0}^{T-1}\sum_{j=2}^{n}\left\langle \frac{\gamma}{n}\sum_{i=1}^{j-1}\omega_{i,l}^u - \delta_j^l, \omega_{j,l}^u\right\rangle \;=\; \frac{8\gamma}{n}\sum_{j=2}^{n}\left\langle \frac{\gamma}{n}\sum_{i=1}^{j-1}\omega_{i,l}^u - \delta_j^l, \omega_{j,l}^u\right\rangle.
\tag{232}
$$

**Upper bound for ⑥′.** To estimate this sum, we will use Bernstein's inequality. The summands have conditional expectations equal to zero:

$$
\mathbb{E}_{\xi_j^l}\left[\frac{8\gamma}{n}\left\langle \delta_j^l, \omega_{j,l}^u\right\rangle\right] = \frac{8\gamma}{n}\left\langle \delta_j^l, \mathbb{E}_{\xi_j^l}[\omega_{j,l}^u]\right\rangle = 0.
$$

Moreover, for all $l = 0, \ldots, T-1$ random vectors $\{\omega_{i,l}^u\}_{i=1}^n$ are independent. Thus, sequence $\left\{\frac{8\gamma}{n}\left\langle \delta_j^l, \omega_{j,l}^u\right\rangle\right\}_{l,j=0,2}^{T-1,n}$ is a martingale difference sequence. Next, the summands are bounded:

$$
\left|\frac{8\gamma}{n}\left\langle \delta_j^l, \omega_{j,l}^u\right\rangle\right| \leq \frac{8\gamma}{n}\left\|\delta_j^l\right\| \cdot \left\|\omega_{j,l}^u\right\| \overset{(230),(214)}{\leq} \frac{8\gamma}{n}\cdot\frac{\sqrt{V}}{2}\cdot 2\lambda \leq \frac{V}{5\ln\frac{48n(K+1)}{\beta}} \overset{\text{def}}{=} c.
\tag{233}
$$

Finally, conditional variances $(\sigma_{j,l}')^2 \overset{\text{def}}{=} \mathbb{E}_{\xi_j^l}\left[\frac{64\gamma^2}{n^2}\left\langle \delta_j^l, \omega_{j,l}^u\right\rangle^2\right]$ of the summands are bounded:

$$
(\sigma_{j,l}')^2 \leq \mathbb{E}_{\xi_j^l}\left[\frac{64\gamma^2}{n^2}\|\delta_j^l\|^2 \cdot \|\omega_{j,l}^u\|^2\right] \overset{(230)}{\leq} \frac{16\gamma^2 V}{n^2}\mathbb{E}_{\xi_j^l}\left[\|\omega_{j,l}^u\|^2\right].
\tag{234}
$$

Applying Bernstein's inequality (Lemma B.1) with $X_{i,l} = \frac{8\gamma}{n}\left\langle \delta_j^l, \omega_{j,l}^u\right\rangle$, constant $c$ defined in (233), $b = \frac{V}{5}$, $G = \frac{V^2}{150\ln\frac{48n(K+1)}{\beta}}$, we get

$$
\mathbb{P}\left\{|⑥'| > \frac{V}{5} \text{ and } \sum_{l=0}^{T-1}\sum_{j=2}^{n}(\sigma_{i,l}')^2 \leq \frac{V^2}{150\ln\frac{48n(K+1)}{\beta}}\right\} \leq 2\exp\left(-\frac{b^2}{2G + \frac{2cb}{3}}\right) = \frac{\beta}{24n(K+1)}.
$$

The above is equivalent to

$$\mathbb{P}\{E_{\text{⑥}'}\} \geq 1 - \frac{\beta}{24n(K+1)}, \quad \text{for } E_{\text{⑥}'} = \left\{ \text{either } \sum_{l=0}^{T-1} \sum_{j=2}^{n} (\sigma'_{i,l})^2 > \frac{V^2}{150 \ln \frac{48n(K+1)}{\beta}} \text{ or } |\text{⑥}'| \leq \frac{V}{5} \right\}.$$

(235)

Moreover, $E_{T-1}$ implies

$$\sum_{l=0}^{T-1} \sum_{j=2}^{n} (\sigma'_{j,l})^2 \overset{(234)}{\leq} \frac{16\gamma^2 V}{n^2} \sum_{l=0}^{T-1} \sum_{j=2}^{n} \mathbb{E}_{\xi_j^l} \left[ \|\omega_{j,l}^u\|^2 \right] \overset{(217),T \leq K+1}{\leq} \frac{288(K+1)\gamma^2 V \lambda^{2-\alpha} \sigma^{\alpha}}{n}$$

$$\overset{(200)}{\leq} \frac{288(K+1)\gamma^{\alpha} \sigma^{\alpha} V^{2-\frac{\alpha}{2}}}{40^{2-\alpha} \sqrt{2}^{2-\alpha} n^{\alpha-1} \ln^{2-\alpha} \frac{48n(K+1)}{\beta}} \overset{(199)}{\leq} \frac{V^2}{150 \ln \frac{48n(K+1)}{\beta}}.$$

(236)

**Upper bound for** $2\gamma\sqrt{V} \left\| \sum_{l=0}^{T-1} \omega_t \right\|$. We introduce new random vectors:

$$\zeta_l = \begin{cases} \gamma \sum_{r=0}^{l-1} \omega_r, & \text{if } \left\| \gamma \sum_{r=0}^{l-1} \omega_r \right\| \leq \sqrt{V}, \\ 0, & \text{otherwise} \end{cases}$$

for $l = 1, 2, \ldots, T-1$. With probability 1 we have

$$\|\zeta_l\| \leq \sqrt{V}.$$

(237)

Using this and (205), we obtain that $E_{T-1}$ implies

$$\gamma \left\| \sum_{l=0}^{T-1} \omega_l \right\| = \sqrt{\gamma^2 \left\| \sum_{l=0}^{T-1} \omega_l \right\|^2}$$

$$= \sqrt{\gamma^2 \sum_{l=0}^{T-1} \|\omega_l\|^2 + 2\gamma \sum_{l=0}^{T-1} \left\langle \gamma \sum_{r=0}^{l-1} \omega_r, \omega_l \right\rangle}$$

$$= \sqrt{\gamma^2 \sum_{l=0}^{T-1} \|\omega_l\|^2 + 2\gamma \sum_{l=0}^{T-1} \langle \zeta_l, \omega_l \rangle}$$

$$\overset{(213)}{\leq} \sqrt{\frac{1}{4} (\text{③} + \text{④} + \text{⑤} + \text{⑥}) + \underbrace{\frac{2\gamma}{n} \sum_{l=0}^{T-1} \sum_{i=1}^{n} \langle \zeta_l, \omega_{i,l}^u \rangle}_{\text{⑦}} + \underbrace{2\gamma \sum_{l=0}^{T-1} \langle \zeta_l, \omega_l^b \rangle}_{\text{⑧}}}.$$

(238)

**Upper bound for ⑦.** To estimate this sum, we will use Bernstein's inequality. The summands have conditional expectations equal to zero:

$$\mathbb{E}_{\xi_i^l} \left[ \frac{2\gamma}{n} \langle \zeta_l, \omega_{i,l}^u \rangle \right] = \frac{2\gamma}{n} \left\langle \zeta_l, \mathbb{E}_{\xi_i^l} [\omega_{i,l}^u] \right\rangle = 0.$$

Moreover, for all $l = 0, \ldots, T-1$ random vectors $\{\omega_{i,l}^u\}_{i=1}^n$ are independent. Thus, sequence $\left\{ \frac{2\gamma}{n} \langle \zeta_l, \omega_{i,l}^u \rangle \right\}_{l,i=0,1}^{T-1,n}$ is a martingale difference sequence. Next, the summands are bounded:

$$\left| \frac{2\gamma}{n} \langle \zeta_l, \omega_{i,l}^u \rangle \right| \leq \frac{2\gamma}{n} \|\zeta_l\| \cdot \|\omega_{i,l}^u\| \overset{(237),(214)}{\leq} \frac{4\gamma}{n} R\lambda \overset{(200)}{\leq} \frac{V}{5 \ln \frac{48n(K+1)}{\beta}} \overset{\text{def}}{=} c.$$

(239)

Finally, conditional variances $\widehat{\sigma}_{i,l}^2 \overset{\text{def}}{=} \mathbb{E}_{\xi_i^l} \left[ \frac{4\gamma^2}{n^2} \langle \zeta_l, \omega_{i,l}^u \rangle^2 \right]$ of the summands are bounded:

$$\widehat{\sigma}_{i,l}^2 \le \mathbb{E}_{\xi_i^l} \left[ \frac{4\gamma^2}{n^2} \|\zeta_l\|^2 \cdot \|\omega_{i,l}^u\|^2 \right] \overset{(237)}{\le} \frac{4\gamma^2}{n^2} V \mathbb{E}_{\xi_i^l} \left[ \|\omega_{i,l}^u\|^2 \right]. \tag{240}$$

Applying Bernstein's inequality (Lemma B.1) with $X_{i,l} = \frac{2\gamma}{n} \langle \zeta_l, \omega_{i,l}^u \rangle$, constant $c$ defined in (239), $b = \frac{V}{5}$, $G = \frac{V^2}{150 \ln \frac{48n(K+1)}{\beta}}$, we get

$$\mathbb{P}\left\{ |\text{⑦}| > \frac{V}{5} \text{ and } \sum_{l=0}^T \sum_{i=1}^n \widehat{\sigma}_{i,l}^2 \le \frac{V^2}{150 \ln \frac{48n(K+1)}{\beta}} \right\} \le 2\exp\left( -\frac{b^2}{2G + 2cb/3} \right) = \frac{\beta}{24n(K+1)}.$$

The above is equivalent to

$$\mathbb{P}\{E_{\text{⑦}}\} \ge 1 - \frac{\beta}{24n(K+1)} \text{ for } E_{\text{⑦}} = \left\{ \text{either} \quad \sum_{l=0}^T \sum_{i=1}^n \widehat{\sigma}_{i,l}^2 > \frac{V^2}{150 \ln \frac{48n(K+1)}{\beta}} \quad \text{or} \quad |\text{⑦}| \le \frac{V}{5} \right\}. \tag{241}$$

Moreover, $E_{T-1}$ implies

$$\begin{aligned}
\sum_{l=0}^T \sum_{i=1}^n \widehat{\sigma}_{i,l}^2 &\overset{(240)}{\le} \frac{4\gamma^2}{n^2} V \sum_{l=0}^T \mathbb{E}_{\xi^l} \left[ \|\omega_l^u\|^2 \right] \\
&\overset{(217), T \le K+1}{\le} \frac{72\gamma^2 V \sigma^\alpha (K+1)\lambda^{2-\alpha}}{n} \\
&\overset{(200)}{\le} \frac{9 \cdot 20^\alpha \sqrt{2}^\alpha}{100 \cdot n^{\alpha-1}} \gamma^\alpha R^{4-\alpha} \sigma^\alpha (K+1) \ln^{\alpha-2} \frac{48n(K+1)}{\beta} \\
&\overset{(199)}{\le} \frac{V^2}{150 \ln \frac{48n(K+1)}{\beta}}.
\end{aligned} \tag{242}$$

**Upper bound for ⑧.** Probability event $E_{T-1}$ implies

$$\begin{aligned}
\text{⑧} &\le 2\gamma \sum_{l=0}^T \|\zeta_l\| \cdot \|\omega_l^b\| \overset{(211),(216), T \le K+1}{\le} 2 \cdot 2^\alpha \gamma R(K+1) \frac{\sigma^\alpha}{\lambda^{\alpha-1}} \\
&\overset{(200)}{=} \frac{40^\alpha \sqrt{2}^\alpha}{n^{\alpha-1} 10\sqrt{2}} \gamma^\alpha \sigma^\alpha R^{2-\alpha} (K+1) \ln^{\alpha-1} \frac{48n(K+1)}{\beta} \overset{(199)}{\le} \frac{V}{5}.
\end{aligned} \tag{243}$$

That is, we derive the upper bounds for $2\gamma\sqrt{V} \left\| \sum_{l=0}^{T-1} \omega_l \right\|, \text{①}, \text{②}, \text{③}, \text{④}, \text{⑤}, \text{⑥}$. More precisely, $E_{T-1}$ implies

$$A_T \overset{(213)}{\le} 4V + 2\gamma\sqrt{V} \left\| \sum_{l=0}^{T-1} \omega_l \right\| + \text{①} + \text{②} + \text{③} + \text{④} + \text{⑤} + \text{⑥},$$

$$\text{⑥} \overset{(231)}{=} \text{⑥}' + \frac{8\gamma}{n} \sum_{j=2}^n \left\langle \frac{\gamma}{n} \sum_{i=1}^{j-1} \omega_{i,T-1}^u - \delta_j^{T-1}, \omega_{j,T-1}^u \right\rangle,$$

$$2\gamma\sqrt{V} \left\| \sum_{l=0}^{T-1} \omega_l \right\| \overset{(238)}{\le} 2\sqrt{V} \sqrt{\frac{1}{4} (\text{③} + \text{④} + \text{⑤} + \text{⑥}) + \text{⑦} + \text{⑧}},$$

$$\text{②} \overset{(222)}{\le} \frac{3V}{100}, \quad \text{③} \overset{(223)}{\le} \frac{3V}{100}, \quad \text{⑤} \overset{(228)}{\le} \frac{V}{10}, \quad \text{⑧} \overset{(243)}{\le} \frac{V}{5},$$

$$\sum_{l=0}^{T-1} \sum_{i=1}^n \sigma_{i,l}^2 \overset{(221)}{\le} \frac{3V^2}{200 \ln \frac{48n(K+1)}{\beta}}, \quad \sum_{l=0}^{T-1} \sum_{i=1}^n \widetilde{\sigma}_{i,l}^2 \overset{(227)}{\le} \frac{V^2}{600 \ln \frac{48n(K+1)}{\beta}},$$

$$\sum_{l=0}^{T-1} \sum_{i=1}^n \widehat{\sigma}_{i,l}^2 \overset{(242)}{\le} \frac{V^2}{150 \ln \frac{48n(K+1)}{\beta}}, \quad \sum_{l=0}^{T-1} \sum_{i=1}^n (\sigma_{j,l}')^2 \overset{(236)}{\le} \frac{V^2}{150 \ln \frac{48n(K+1)}{\beta}}.$$

In addition, we also establish (see (220), (226), (243), (235) and our induction assumption)

$$\mathbb{P}\{E_{T-1}\} \geq 1 - \frac{(T-1)\beta}{K+1},$$

$$\mathbb{P}\{E_①\} \geq 1 - \frac{\beta}{24n(K+1)}, \quad \mathbb{P}\{E_④\} \geq 1 - \frac{\beta}{24n(K+1)},$$

$$\mathbb{P}\{E_{⑥'}\} \geq 1 - \frac{\beta}{24n(K+1)}, \quad \mathbb{P}\{E_⑦\} \geq 1 - \frac{\beta}{24n(K+1)},$$

where

$$E_① = \left\{ \text{either} \quad \sum_{l=0}^{T-1}\sum_{i=1}^{n} \sigma_{i,l}^2 > \frac{3V^2}{200\ln\frac{48n(K+1)}{\beta}} \quad \text{or} \quad |①| \leq \frac{3V}{10} \right\},$$

$$E_④ = \left\{ \text{either} \quad \sum_{l=0}^{T-1}\sum_{i=1}^{n} \widetilde{\sigma}_{i,l}^2 > \frac{V^2}{600\ln\frac{48n(K+1)}{\beta}} \quad \text{or} \quad |④| \leq \frac{V}{10} \right\},$$

$$E_{⑥'} = \left\{ \text{either} \quad \sum_{l=0}^{T-1}\sum_{j=2}^{n} (\sigma_{i,l}')^2 > \frac{V^2}{150\ln\frac{48n(K+1)}{\beta}} \text{ or } |⑥'| \leq \frac{V}{5} \right\},$$

$$E_⑦ = \left\{ \text{either} \quad \sum_{l=0}^{T-1}\sum_{i=1}^{n} \widehat{\sigma}_{i,l}^2 > \frac{V^2}{150\ln\frac{48n(K+1)}{\beta}} \quad \text{or} \quad |⑦| \leq \frac{V}{5} \right\}.$$

Therefore, probability event $E_{T-1} \cap E_① \cap E_④ \cap E_{⑥'} \cap E_⑦$ implies

$$\left\| \gamma\sum_{l=0}^{T-1}\omega_l \right\| \leq \sqrt{\frac{1}{4}\left(\frac{3V}{10} + \frac{V}{10} + \frac{V}{10} + \frac{V}{5}\right) + \frac{V}{5} + \frac{V}{5} + \frac{2\gamma}{n}\sum_{j=2}^{n}\left\langle \frac{\gamma}{n}\sum_{i=1}^{j-1}\omega_{i,T-1}^u - \delta_j^{T-1}, \omega_{j,T-1}^u \right\rangle}$$

$$\leq \sqrt{V} + \sqrt{\frac{2\gamma}{n}\sum_{j=2}^{n}\left\langle \frac{\gamma}{n}\sum_{i=1}^{j-1}\omega_{i,T-1}^u - \delta_j^{T-1}, \omega_{j,T-1}^u \right\rangle}, \tag{244}$$

$$A_T \leq 4V + 2V + 2\sqrt{V}\sqrt{\frac{2\gamma}{n}\sum_{j=2}^{n}\left\langle \frac{\gamma}{n}\sum_{i=1}^{j-1}\omega_{i,T-1}^u - \delta_j^{T-1}, \omega_{j,T-1}^u \right\rangle}$$

$$+ \frac{3V}{10} + \frac{3V}{100} + \frac{3V}{100} + \frac{V}{10} + \frac{V}{5} + \frac{V}{5}$$

$$+ \frac{8\gamma}{n}\sum_{j=2}^{n}\left\langle \frac{\gamma}{n}\sum_{i=1}^{j-1}\omega_{i,T-1}^u - \delta_j^{T-1}, \omega_{j,T-1}^u \right\rangle$$

$$\leq 8V + 2\sqrt{V}\sqrt{\frac{2\gamma}{n}\sum_{j=2}^{n}\left\langle \frac{\gamma}{n}\sum_{i=1}^{j-1}\omega_{i,T-1}^u - \delta_j^{T-1}, \omega_{j,T-1}^u \right\rangle}$$

$$+ \frac{8\gamma}{n}\sum_{j=2}^{n}\left\langle \frac{\gamma}{n}\sum_{i=1}^{j-1}\omega_{i,T-1}^u - \delta_j^{T-1}, \omega_{j,T-1}^u \right\rangle. \tag{245}$$

In the final part of the proof, we will show that $\frac{\gamma}{n}\sum_{i=1}^{j-1}\omega_{i,T-1}^u = \delta_j^{T-1}$ with high probability. In particular, we consider probability event $\widetilde{E}_{T-1,j}$ defined as follows: inequalities

$$\left\| \frac{\gamma}{n}\sum_{i=1}^{r-1}\omega_{i,T-1}^u \right\| \leq \frac{\sqrt{V}}{2} \tag{246}$$

hold for $r = 2, \ldots, j$ simultaneously. We want to show that $\mathbb{P}\{E_{T-1} \cap \widetilde{E}_{T-1,j}\} \geq 1 - \frac{(T-1)\beta}{K+1} - \frac{j\beta}{8n(K+1)}$ for all $j = 2, \ldots, n$. For $j = 2$ the statement is trivial since

$$\left\| \frac{\gamma}{n} \omega_{1,T-1}^u \right\| \overset{(214)}{\leq} \frac{2\gamma\lambda}{n} \leq \frac{\sqrt{V}}{2}.$$

Next, we assume that the statement holds for some $j = m - 1 < n$, i.e., $\mathbb{P}\{E_{T-1} \cap \widetilde{E}_{T-1,m-1}\} \geq 1 - \frac{(T-1)\beta}{K+1} - \frac{(m-1)\beta}{8n(K+1)}$. Our goal is to prove that $\mathbb{P}\{E_{T-1} \cap \widetilde{E}_{T-1,m}\} \geq 1 - \frac{(T-1)\beta}{K+1} - \frac{m\beta}{8n(K+1)}$. First, we consider $\left\| \frac{\gamma}{n} \sum_{i=1}^{m-1} \omega_{i,T-1}^u \right\|$:

$$
\begin{aligned}
\left\| \frac{\gamma}{n} \sum_{i=1}^{m-1} \omega_{i,T-1}^u \right\| &= \sqrt{\frac{\gamma^2}{n^2} \left\| \sum_{i=1}^{m-1} \omega_{i,T-1}^u \right\|^2} \\
&= \sqrt{\frac{\gamma^2}{n^2} \sum_{i=1}^{m-1} \|\omega_{i,T-1}^u\|^2 + \frac{2\gamma}{n} \sum_{i=1}^{m-1} \left\langle \frac{\gamma}{n} \sum_{r=1}^{i-1} \omega_{r,T-1}^u, \omega_{i,T-1}^u \right\rangle} \\
&\leq \sqrt{\frac{\gamma^2}{n^2} \sum_{l=0}^{T-1} \sum_{i=1}^{m-1} \|\omega_{i,l}^u\|^2 + \frac{2\gamma}{n} \sum_{i=1}^{m-1} \left\langle \frac{\gamma}{n} \sum_{r=1}^{i-1} \omega_{r,T-1}^u, \omega_{i,T-1}^u \right\rangle}.
\end{aligned}
$$

Next, we introduce a new notation:

$$
\rho_{i,T-1} = \begin{cases} \frac{\gamma}{n} \sum_{r=1}^{i-1} \omega_{r,T-1}^u, & \text{if } \left\| \frac{\gamma}{n} \sum_{r=1}^{i-1} \omega_{r,T-1}^u \right\| \leq \frac{\sqrt{V}}{2}, \\ 0, & \text{otherwise} \end{cases}
$$

for $i = 1, \ldots, m - 1$. By definition, we have

$$\|\rho_{i,T-1}\| \leq \frac{\sqrt{V}}{2} \tag{247}$$

for $i = 1, \ldots, m - 1$. Moreover, $\widetilde{E}_{T-1,m-1}$ implies $\rho_{i,T-1} = \frac{\gamma}{n} \sum_{r=1}^{i-1} \omega_{r,T-1}^u$ for $i = 1, \ldots, m - 1$ and

$$\left\| \frac{\gamma}{n} \sum_{i=1}^{m-1} \omega_{i,l}^u \right\| \leq \sqrt{③ + ④ + ⑨},$$

where

$$⑨ = \frac{2\gamma}{n} \sum_{i=1}^{m-1} \left\langle \rho_{i,T-1}, \omega_{i,T-1}^u \right\rangle.$$

It remains to estimate ⑨.

**Upper bound for ⑨.** To estimate this sum, we will use Bernstein's inequality. The summands have conditional expectations equal to zero:

$$\mathbb{E}_{\xi_i^{T-1}} \left[ \frac{2\gamma}{n} \langle \rho_{i,T-1}, \omega_{i,T-1}^u \rangle \right] = \frac{2\gamma}{n} \left\langle \rho_{i,T-1}, \mathbb{E}_{\xi_i^{T-1}} [\omega_{i,T-1}^u] \right\rangle = 0$$

since random vectors $\{\omega_{i,T-1}^u\}_{i=1}^n$ are independent. Thus, sequence $\left\{ \frac{2\gamma}{n} \langle \rho_{i,T-1}, \omega_{i,T-1}^u \rangle \right\}_{i=1}^{m-1}$ is a martingale difference sequence. Next, the summands are bounded:

$$\left| \frac{2\gamma}{n} \langle \rho_{i,T-1}, \omega_{i,T-1}^u \rangle \right| \leq \frac{2\gamma}{n} \|\rho_{i,T-1}\| \cdot \|\omega_{i,T-1}^u\| \overset{(247),(214)}{\leq} \frac{\gamma}{n} \sqrt{V}\lambda \overset{(200)}{\leq} \frac{V}{20 \ln \frac{48n(K+1)}{\beta}} \overset{\text{def}}{=} c. \tag{248}$$

Finally, conditional variances $(\widehat{\sigma}'_{i,T-1})^2 \stackrel{\text{def}}{=} \mathbb{E}_{\xi_i^{T-1}}\left[\frac{4\gamma^2}{n^2}\langle\rho_{i,T-1},\omega_{i,T-1}^u\rangle^2\right]$ of the summands are bounded:

$$(\widehat{\sigma}'_{i,T-1})^2 \leq \mathbb{E}_{\xi_i^{T-1}}\left[\frac{4\gamma^2}{n^2}\|\rho_{i,T-1}\|^2\cdot\|\omega_{i,T-1}^u\|^2\right] \stackrel{(247)}{\leq} \frac{\gamma^2 V}{n^2}\mathbb{E}_{\xi_i^{T-1}}\left[\|\omega_{i,T-1}^u\|^2\right]. \tag{249}$$

Applying Bernstein's inequality (Lemma B.1) with $X_{T-1,i} = \frac{2\gamma}{n}\langle\rho_{i,T-1},\omega_{i,T-1}^u\rangle$, constant $c$ defined in (248), $b = \frac{V}{20}$, $G = \frac{V^2}{2400\ln\frac{48n(K+1)}{\beta}}$, we get

$$\mathbb{P}\left\{|\text{⑨}| > \frac{V}{20} \text{ and } \sum_{i=1}^{m-1}(\widehat{\sigma}'_{i,T-1})^2 \leq \frac{V^2}{2400\ln\frac{48n(K+1)}{\beta}}\right\} \leq 2\exp\left(-\frac{b^2}{2G + 2cb/3}\right) = \frac{\beta}{24n(K+1)}.$$

The above is equivalent to

$$\mathbb{P}\{E_{\text{⑨}}\} \geq 1 - \frac{\beta}{24n(K+1)}, \quad \text{for } E_{\text{⑨}} = \left\{\text{either } \sum_{i=1}^{m-1}(\widehat{\sigma}'_{i,T-1})^2 > \frac{V^2}{2400\ln\frac{48n(K+1)}{\beta}} \text{ or } |\text{⑨}| \leq \frac{V}{20}\right\}. \tag{250}$$

Moreover, $E_{T-1}$ implies

$$\sum_{i=1}^{m-1}(\widehat{\sigma}'_{i,T-1})^2 \stackrel{(249)}{\leq} \frac{\gamma^2 V}{n^2}\sum_{i=1}^n\mathbb{E}_{\xi_i^l}\left[\|\omega_{i,l}^u\|^2\right] \stackrel{(217)}{\leq} \frac{18\gamma^2 V\lambda^{2-\alpha}\sigma^\alpha}{n}$$

$$\stackrel{(200)}{\leq} \frac{18\gamma^\alpha\sigma^\alpha V^{2-\frac{\alpha}{2}}}{40^{2-\alpha}n^{\alpha-1}\ln^{2-\alpha}\frac{48n(K+1)}{\beta}} \stackrel{(199)}{\leq} \frac{V^2}{2400\ln\frac{48n(K+1)}{\beta}}. \tag{251}$$

Putting all together we get that $E_{T-1}\cap\widetilde{E}_{T-1,m-1}$ implies

$$\left\|\frac{\gamma}{n}\sum_{i=1}^{m-1}\omega_{i,T-1}^u\right\| \leq \sqrt{\text{③}+\text{④}+\text{⑨}}, \quad \text{③} \stackrel{(223)}{\leq} \frac{V}{10},$$

$$\sum_{l=0}^{T-1}\sum_{i=1}^n\widetilde{\sigma}_{i,l}^2 \stackrel{(227)}{\leq} \frac{V^2}{600\ln\frac{48n(K+1)}{\beta}}, \quad \sum_{i=1}^{m-1}(\widehat{\sigma}'_{i,T-1})^2 \leq \frac{V^2}{2400\ln\frac{48n(K+1)}{\beta}}$$

In addition, we also establish (see (226), (250) and our induction assumption)

$$\mathbb{P}\{E_{T-1}\cap\widetilde{E}_{T-1,m-1}\} \geq 1 - \frac{(T-1)\beta}{K+1} - \frac{(m-1)\beta}{8n(K+1)},$$

$$\mathbb{P}\{E_{\text{④}}\} \geq 1 - \frac{\beta}{24n(K+1)}, \quad \mathbb{P}\{E_{\text{⑨}}\} \geq 1 - \frac{\beta}{24n(K+1)},$$

where

$$E_{\text{④}} = \left\{\text{either } \sum_{l=0}^{T-1}\sum_{i=1}^n\widetilde{\sigma}_{i,l}^2 > \frac{V^2}{600\ln\frac{48n(K+1)}{\beta}} \text{ or } |\text{④}| \leq \frac{V}{10}\right\},$$

$$E_{\text{⑨}} = \left\{\text{either } \sum_{i=1}^{m-1}(\widehat{\sigma}'_{i,T-1})^2 > \frac{V^2}{2400\ln\frac{48n(K+1)}{\beta}} \text{ or } |\text{⑩}| \leq \frac{V}{20}\right\}.$$

Therefore, probability event $E_{T-1}\cap\widetilde{E}_{T-1,m-1}\cap E_{\text{④}}\cap E_{\text{⑨}}$ implies

$$\left\|\frac{\gamma}{n}\sum_{i=1}^{m-1}\omega_{i,l}^u\right\| \leq \sqrt{\frac{V}{10}+\frac{V}{10}+\frac{V}{20}} = \frac{\sqrt{V}}{2}.$$

This implies $\widetilde{E}_{T-1,m}$ and

$$\begin{aligned}
\mathbb{P}\{E_{T-1}\cap\widetilde{E}_{T-1,m}\} &\geq \mathbb{P}\{E_{T-1}\cap\widetilde{E}_{T-1,m-1}\cap E_{\text{④}}\cap E_{\text{⑨}}\} \\
&= 1 - \mathbb{P}\left\{\overline{E_{T-1}\cap\widetilde{E}_{T-1,m-1}}\cup\overline{E}_{\text{④}}\cup\overline{E}_{\text{⑨}}\right\} \\
&\geq 1 - \frac{(T-1)\beta}{K+1} - \frac{m\beta}{8n(K+1)}.
\end{aligned}$$

Therefore, for all $m = 2, \ldots, n$ the statement holds and, in particular, $\mathbb{P}\{E_{T-1} \cap \widetilde{E}_{T-1,n}\} \geq 1 - \frac{(T-1)\beta}{K+1} - \frac{\beta}{8(K+1)}$. Taking into account (244) and (245), we conclude that $E_{T-1} \cap \widetilde{E}_{T-1,n}$ implies

$$\left\| \gamma \sum_{l=0}^{T-1} \omega_l \right\| \leq \sqrt{V}, \quad A_T \leq 8V,$$

which is equivalent to (204) and (205) for $t = T$. Moreover,

$$
\begin{aligned}
\mathbb{P}\{E_T\} &\geq \mathbb{P}\left\{ E_{T-1} \cap \widetilde{E}_{T-1,n} \cap E_① \cap E_④ \cap E_{⑥'} \cap E_⑦ \cap E_⑨ \right\} \\
&= 1 - \mathbb{P}\left\{ \overline{E_{T-1} \cap \widetilde{E}_{T-1,n}} \cup \overline{E}_① \cup \overline{E}_④ \cup \overline{E}_{⑥'} \cup \overline{E}_⑦ \cup \overline{E}_⑨ \right\} \\
&= 1 - \frac{(T-1)\beta}{K+1} - \frac{\beta}{8(K+1)} - 5 \cdot \frac{\beta}{8(K+1)} \geq 1 - \frac{T\beta}{K+1}.
\end{aligned}
$$

In other words, we showed that $\mathbb{P}\{E_k\} \geq 1 - \frac{k\beta}{(K+1)}$ for all $k = 0, 1, \ldots, K+1$. For $k = K+1$ we have that with probability at least $1 - \beta$

$$\mathrm{Gap}_{\sqrt{V}}(x_{\mathrm{avg}}^{K+1}) \overset{(210)}{\leq} \frac{4V}{\gamma(K+1)}.$$

Finally, if

$$\gamma = \min\left\{ \frac{1}{480\ell \ln \frac{48n(K+1)}{\beta}}, \left( \frac{1}{86400} \right)^{\frac{1}{\alpha}} \cdot \frac{\sqrt{V} n^{\frac{\alpha-1}{\alpha}}}{(K+1)^{\frac{1}{\alpha}} \sigma \ln^{\frac{\alpha-1}{\alpha}} \frac{48n(K+1)}{\beta}} \right\}$$

then with probability at least $1 - \beta$

$$
\begin{aligned}
\mathrm{Gap}_{\sqrt{V}}(x_{\mathrm{avg}}^{K+1}) &\leq \frac{4V}{\gamma(K+1)} = \max\left\{ \frac{480\ell V \ln \frac{48n(K+1)}{\beta}}{K+1}, \left( \frac{86400}{1} \right)^{\frac{1}{\alpha}} \cdot \frac{4\sigma\sqrt{V} \ln^{\frac{\alpha-1}{\alpha}} \frac{48n(K+1)}{\beta}}{n^{\frac{\alpha-1}{\alpha}} (K+1)^{\frac{\alpha-1}{\alpha}}} \right\} \\
&= \mathcal{O}\left( \max\left\{ \frac{\ell\sqrt{V} \ln \frac{nK}{\beta}}{K}, \frac{\sigma\sqrt{V} \ln^{\frac{\alpha-1}{\alpha}} \frac{K}{\beta}}{n^{\frac{\alpha-1}{\alpha}} K^{\frac{\alpha-1}{\alpha}}} \right\} \right).
\end{aligned}
$$

To get $\mathrm{Gap}_R(x_{\mathrm{avg}}^{K+1}) \leq \varepsilon$ with probability at least $1 - \beta$ it is sufficient to choose $K$ such that both terms in the maximum above are $\mathcal{O}(\varepsilon)$. This leads to

$$K = \mathcal{O}\left( \frac{\ell V}{\varepsilon} \ln \frac{n\ell V}{\varepsilon\beta}, \frac{1}{n} \left( \frac{\sigma\sqrt{V}}{\varepsilon} \right)^{\frac{\alpha}{\alpha-1}} \ln\left( \frac{1}{\beta} \left( \frac{\sigma\sqrt{V}}{\varepsilon} \right)^{\frac{\alpha}{\alpha-1}} \right) \right)$$

that concludes the proof. $\qquad\square$

### G.2 QUASI-STRONGLY MONOTONE CASE

**Lemma G.2.** *Let Assumptions 8, 9 hold for $Q = B_{\sqrt{2V}}(x^*)$, where $V \geq \|x^0 - x^*\| + \frac{9000000\gamma^2 \ln^2\left( \frac{48n(K+1)}{\beta} \right)}{n^2} \sum_{i=1}^{n} \|F_i(x^*)\|^2$, and $0 < \gamma \leq \frac{1}{\ell + 18000000\nu \ln^2\left( \frac{48n(K+1)}{\beta} \right)\ell/n}$, $\nu \leq \frac{1}{18000000 \ln^2\left( \frac{48n(K+1)}{\beta} \right)}$. If $x^k$ lies in $B_{\sqrt{2V}}(x^*)$ for all $k = 0, 1, \ldots, K$ for some $K \geq 0$, then the iterates produced by DProx-clipped-SGDA-shift satisfy*

$$
\begin{aligned}
V_{K+1} &\leq (1 - \gamma\mu)^{K+1} V_0 + \frac{2\gamma}{n} \sum_{k=0}^{K} \sum_{i=1}^{n} (1 - \gamma\mu)^{K-k} \langle x^k - x^* - \gamma(F(x^k) - h^*), \omega_{i,k} \rangle \\
&\quad + \frac{\gamma^2}{n^2} \sum_{k=0}^{K} \sum_{i=1}^{n} (1 - \gamma\mu)^{K-k} \|\omega_{i,k}\|^2 + \gamma^2 \sum_{k=0}^{K} (1 - \gamma\mu)^{K-k} \|\omega_k\|^2, \quad (252)
\end{aligned}
$$

*where* $V_k = \|x^k - x^*\|^2 + \frac{9000000\gamma^2 \ln^2\left(\frac{48n(K+1)}{\beta}\right)}{n^2} \sum_{i=1}^{n} \|h_i^k - h_i^*\|^2$, $h_i^* = F_i(x^*)$, *and* $\omega_k, \omega_k^u, \omega_k^b, \omega_{k,i}^u, \omega_{k,i}^b$ *are defined in* (207)-(209).

*Proof.* Using the update rule of DProx-clipped-SGDA-shift and $\omega_k = F(x^k) - \widetilde{g}^k$ we obtain

$$
\begin{aligned}
\|x^{k+1} - x^*\|^2 &= \|\operatorname{prox}_{\gamma\Psi}\left(x^k - \gamma\widetilde{g}^k\right) - \operatorname{prox}_{\gamma\Psi}\left(x^* - \gamma h^*\right)\|^2 \\
&\leq \|x^k - x^* - \gamma(\widetilde{g}^k - h^*)\|^2 \\
&= \|x^k - x^*\|^2 - 2\gamma\langle x^k - x^*, \widetilde{g}^k - h^*\rangle + \gamma^2\|\widetilde{g}^k - h^*\|^2 \\
&= \|x^k - x^*\|^2 - 2\gamma\langle x^k - x^*, F(x^k) - h^*\rangle - 2\gamma^2\langle F(x^k) - h^*, \omega_k\rangle \\
&\quad + 2\gamma\langle x^k - x^*, \omega_k\rangle + \gamma^2\|F(x^k) - h^*\|^2 + \gamma^2\|\omega_k\|^2,
\end{aligned}
$$

Next, let us recall that

$$
h_i^{k+1} = h_i^k + \nu\hat{\Delta}_i^k, \quad \hat{\Delta}_i^k = \texttt{clip}\left(F_{\xi_i^k}(x^k) - h_i^k, \lambda_k\right), \quad \widetilde{g}_i^k = h_i^k + \hat{\Delta}_i^k, \quad \omega_{i,k} = F_i(x^k) - \widetilde{g}_i^k.
$$

Then, $\forall i \in [n]$ we have

$$
\begin{aligned}
\|h_i^{k+1} - h_i^*\|^2 &= \|h_i^k - h_i^* + \nu\hat{\Delta}_i^k\|^2 = \|h_i^k - h_i^*\|^2 + 2\nu\langle h_i^k - h_i^*, \hat{\Delta}_i^k\rangle + \nu^2\|\hat{\Delta}_i^k\|^2 \\
&= \|h_i^k - h_i^*\|^2 + 2\nu\langle h_i^k - h_i^*, \widetilde{g}_i^k - h_i^k\rangle + \nu^2\|\widetilde{g}_i^k - h_i^k\|^2 \\
&\overset{\nu\leq\frac{1}{2}}{\leq} \|h_i^k - h_i^*\|^2 + 2\nu\langle h_i^k - h_i^*, \widetilde{g}_i^k - h_i^k\rangle + \nu\|\widetilde{g}_i^k - h_i^k\|^2 \\
&= \|h_i^k - h_i^*\|^2 + \nu\langle \widetilde{g}_i^k - h_i^k, \widetilde{g}_i^k + h_i^k - 2h_i^*\rangle \\
&= (1-\nu)\|h_i^k - h_i^*\|^2 + \nu\|\widetilde{g}_i^k - h_i^*\|^2 \\
&\leq (1-\nu)\|h_i^k - h_i^*\|^2 + 2\nu\|\widetilde{g}_i^k - F_i(x^k)\|^2 + 2\nu\|F_i(x^k) - h_i^*\|^2 \\
&= (1-\nu)\|h_i^k - h_i^*\|^2 + 2\nu\|\omega_{i,k}\|^2 + 2\nu\|F_i(x^k) - h_i^*\|^2.
\end{aligned}
$$

Let us consider the following stepsize condition

$$
0 < \gamma \leq \frac{1}{\ell + \frac{18000000\nu \ln^2\left(\frac{48n(K+1)}{\beta}\right)\ell}{n}}. \tag{253}
$$

Lyapunov function

$$
V_k = \|x^k - x^*\|^2 + \frac{9000000\gamma^2 \ln^2\left(\frac{48n(K+1)}{\beta}\right)}{n^2} \sum_{i=1}^{n} \|h_i^k - h_i^*\|^2.
$$

$$
\begin{aligned}
V_{k+1} \quad \leq \quad & \|x^k - x^*\|^2 - 2\gamma\langle x^k - x^*, F(x^k) - h^*\rangle - 2\gamma^2\langle F(x^k) - h^*, \omega_k\rangle \\
& + 2\gamma\langle x^k - x^*, \omega_k\rangle + \gamma^2\|F(x^k) - h^*\|^2 + \gamma^2\|\omega_k\|^2 \\
& + \frac{9 \cdot 10^6 \gamma^2 \ln^2\left(\frac{48n(K+1)}{\beta}\right)}{n^2} \sum_{i=1}^n \Big[(1-\nu)\|h_i^k - h_i^*\|^2 + 2\nu\|\omega_{i,k}\|^2 + 2\nu\|F_i(x^k) - h_i^*\|^2\Big]
\end{aligned}
$$

$$
\begin{aligned}
\overset{(35)}{\leq} \quad & \|x^k - x^*\|^2 + (1-\nu)\frac{9 \cdot 10^6 \gamma^2 \ln^2\left(\frac{48n(K+1)}{\beta}\right)}{n^2} \sum_{i=1}^n \|h_i^k - h_i^*\|^2 \\
& - 2\gamma\left(1 - \frac{\gamma\ell}{2} - \frac{\nu n}{\gamma} \cdot \frac{9 \cdot 10^6 \gamma^2 \ln^2\left(\frac{48n(K+1)}{\beta}\right)}{n^2}\ell_{\max}\right)\langle x^k - x^*, F(x^k) - h^*\rangle \\
& + \frac{2\gamma}{n}\sum_{i=1}^n\langle x^k - x^* - \gamma(F(x^k) - h^*), \omega_{i,k}\rangle + \gamma^2\|\omega_k\|^2 + \frac{\gamma^2}{n^2}\sum_{i=1}^n\|\omega_{i,k}\|^2
\end{aligned}
$$

$$
\begin{aligned}
\overset{(253)}{\leq} \quad & \|x^k - x^*\|^2 + (1-\nu)\frac{9 \cdot 10^6 \gamma^2 \ln^2\left(\frac{48n(K+1)}{\beta}\right)}{n^2} \sum_{i=1}^n \|h_i^k - h_i^*\|^2 \\
& - \gamma\langle x^k - x^*, F(x^k) - h^*\rangle \\
& + \frac{2\gamma}{n}\sum_{i=1}^n\langle x^k - x^* - \gamma(F(x^k) - h^*), \omega_{i,k}\rangle + \gamma^2\|\omega_k\|^2 + \frac{\gamma^2}{n^2}\sum_{i=1}^n\|\omega_{i,k}\|^2
\end{aligned}
$$

$$
\begin{aligned}
\overset{(34)}{\leq} \quad & (1-\gamma\mu)\|x^k - x^*\|^2 + (1-\nu)\frac{9 \cdot 10^6 \gamma^2 \ln^2\left(\frac{48n(K+1)}{\beta}\right)}{n^2} \sum_{i=1}^n \|h_i^k - h_i^*\|^2 \\
& + \frac{2\gamma}{n}\sum_{i=1}^n\langle x^k - x^* - \gamma(F(x^k) - h^*), \omega_{i,k}\rangle + \gamma^2\|\omega_k\|^2 + \frac{\gamma^2}{n^2}\sum_{i=1}^n\|\omega_{i,k}\|^2
\end{aligned}
$$

$$
\begin{aligned}
\overset{\gamma \leq \frac{\nu}{\mu}}{\leq} \quad & (1-\gamma\mu)V_k + \frac{2\gamma}{n}\sum_{i=1}^n\langle x^k - x^* - \gamma(F(x^k) - h^*), \omega_{i,k}\rangle \\
& + \gamma^2\|\omega_k\|^2 + \frac{\gamma^2}{n^2}\sum_{i=1}^n\|\omega_{i,k}\|^2.
\end{aligned}
$$

Unrolling the recurrence, we obtain (45). $\qquad\square$

**Theorem G.2.** *Let Assumptions 8, 9, hold for $Q = B_{\sqrt{2V}}(x^*)$, where $V \geq \|x^0 - x^*\| + \frac{9000000\gamma^2 \ln^2\left(\frac{48n(K+1)}{\beta}\right)}{n^2}\sum_{i=1}^n \|F_i(x^*)\|^2$, and $R \geq \|x^0 - x^*\|$,*

$$
0 < \gamma \leq \min\left\{\frac{1}{4096\ell\ln\frac{48n(K+1)}{\beta}}, \frac{\sqrt{n}R}{3000\zeta_*\ln\frac{48n(K+1)}{\beta}}, \frac{\ln(B_K)}{\mu(K+1)}\right\}, \tag{254}
$$

$$
B_K = \max\left\{2, \left(\frac{\sqrt{2}}{3456}\right)^{\frac{2}{\alpha}} \cdot \frac{(K+1)^{\frac{2(\alpha-1)}{\alpha}}\mu^2 V n^{\frac{2(\alpha-1)}{\alpha}}}{\sigma^2\ln^{\frac{2(\alpha-1)}{\alpha}}\left(\frac{48n(K+1)}{\beta}\right)\ln^2(B_K)}\right\} \tag{255}
$$

$$
= \mathcal{O}\left(\max\left\{2, \frac{K^{\frac{2(\alpha-1)}{\alpha}}\mu^2 V n^{\frac{2(\alpha-1)}{\alpha}}}{\sigma^2\ln^{\frac{2(\alpha-1)}{\alpha}}\left(\frac{nK}{\beta}\right)\ln^2\left(\max\left\{2, \frac{K^{\frac{2(\alpha-1)}{\alpha}}\mu^2 V n^{\frac{2(\alpha-1)}{\alpha}}}{\sigma^2\ln^{\frac{2(\alpha-1)}{\alpha}}\left(\frac{nK}{\beta}\right)}\right\}\right)}\right\}\right) \tag{256}
$$

$$
\lambda_k = \frac{n \cdot \exp(-\gamma\mu(1 + k/2))\sqrt{V}}{256\sqrt{2}\gamma\ln\frac{48n(K+1)}{\beta}}, \tag{257}
$$

*for some $K \geq 0$ and $\beta \in (0, 1]$. Then, after $K$ iterations the iterates produced by* DProx-clipped-SGDA-shift *with probability at least $1 - \beta$ satisfy*

$$V_{K+1} \leq 2 \exp(-\gamma\mu(K+1))V, \tag{258}$$

*where $V_k = \|x^k - x^*\|^2 + \frac{9000000\gamma^2 \ln^2\left(\frac{48n(K+1)}{\beta}\right)}{n^2} \sum\limits_{i=1}^{n} \|h_i^k - h_i^*\|^2$, $h_i^* = F_i(x^*)$. In particular, $V \leq 2R^2$, and when $\gamma$ equals the minimum from (254), then the iterates produced by* Dprox-clipped-SGDA-shift *after $K$ iterations with probability at least $1 - \beta$ satisfy*

$$V_K = \mathcal{O}\left(\max\left\{R^2 \exp\left(-\frac{\mu K}{\ell \ln \frac{nK}{\beta}}\right), R^2 \exp\left(-\frac{\mu\sqrt{n}RK}{\zeta_* \ln \frac{nK}{\beta}}\right), \frac{\sigma^2 \ln^{\frac{2(\alpha-1)}{\alpha}}\left(\frac{nK}{\beta}\right) \ln^2 B_K}{K^{\frac{2(\alpha-1)}{\alpha}} \mu^2 n^{\frac{2(\alpha-1)}{\alpha}}}\right\}\right), \tag{259}$$

*meaning that to achieve $V_K \leq \varepsilon$ with probability at least $1 - \beta$* DProx-clipped-SGDA-shift *requires*

$$K = \mathcal{O}\left(\max\left\{\frac{\ell}{\mu} \ln\left(\frac{R^2}{\varepsilon}\right) \ln\left(\frac{n\ell}{\mu\beta} \ln\frac{R^2}{\varepsilon}\right), \frac{\zeta_*}{\sqrt{n}R\mu} \ln\left(\frac{R^2}{\varepsilon}\right) \ln\left(\frac{\sqrt{n}\zeta_*}{R\mu\beta} \ln\frac{R^2}{\varepsilon}\right), \right.\right.$$

$$\left.\left. \frac{1}{n}\left(\frac{\sigma^2}{\mu^2\varepsilon}\right)^{\frac{\alpha}{2(\alpha-1)}} \ln\left(\frac{1}{\beta}\left(\frac{\sigma^2}{\mu^2\varepsilon}\right)^{\frac{\alpha}{2(\alpha-1)}}\right) \ln^{\frac{\alpha}{\alpha-1}}(B_\varepsilon)\right\}\right) \tag{260}$$

*iterations/oracle calls, where*

$$B_\varepsilon = \max\left\{2, \frac{2R^2}{\varepsilon \ln\left(\frac{1}{\beta}\left(\frac{\sigma^2}{\mu^2\varepsilon}\right)^{\frac{\alpha}{2(\alpha-1)}}\right)}\right\}.$$

*Proof.* The Lyapunov function has the following form

$$V_k = \|x^k - x^*\|^2 + \frac{9000000\gamma^2 \ln^2\left(\frac{48n(K+1)}{\beta}\right)}{n^2} \sum_{i=1}^{n} \|h_i^k - h_i^*\|^2.$$

Similar to previous results, our proof is induction-based. To formulate the statement rigorously, we introduce probability event $E_k$ for each $k = 0, 1, \ldots, K+1$ as follows: inequalities

$$V_t \leq 2 \exp(-\gamma\mu t)V \tag{261}$$

$$\left\|\frac{\gamma}{n}\sum_{i=1}^{r-1} \omega_{i,t-1}^u\right\| \leq \exp\left(-\frac{\gamma\mu(t-1)}{2}\right)\frac{\sqrt{V}}{2} \tag{262}$$

hold for $t = 0, 1, \ldots, k$ and $r = 1, 2, \ldots, n$ simultaneously. We will prove by induction that $\mathbb{P}\{E_k\} \geq 1 - {}^{k\beta}/(K+1)$ for all $k = 0, 1, \ldots, K+1$. The base of induction follows immediately by the definition of $V$. Next, we assume that the statement holds for $k = T - 1 \leq K$, i.e., $\mathbb{P}\{E_{T-1}\} \geq 1 - {}^{(T-1)\beta}/(K+1)$. Let us show that it also holds for $k = T$, i.e., $\mathbb{P}\{E_T\} \geq 1 - {}^{T\beta}/(K+1)$.

Similarly to the monotone case, one can show that due to our choice of the clipping level, we have that $E_{T-1}$ implies $x^t \in B_{\sqrt{2n}\sqrt{V}}(x^*)$ for $t = 0, \ldots, T - 1$. Indeed, for $t = 0, 1, \ldots, T - 1$ inequality (261) gives $x^t \in B_{\sqrt{2V}}(x^*)$. This means that we can apply Lemma G.2: $E_{T-1}$ implies

$$\begin{aligned}
V_T &\leq (1 - \gamma\mu)^T V + \frac{2\gamma}{n}\sum_{t=0}^{T-1}\sum_{i=1}^{n}(1 - \gamma\mu)^{T-1-t}\langle x^t - x^* - \gamma(F(x^t) - h^*), \omega_{i,t}\rangle \\
&\quad + \frac{\gamma^2}{n^2}\sum_{t=0}^{T-1}\sum_{i=1}^{n}(1 - \gamma\mu)^{T-1-t}\|\omega_{i,t}\|^2 + \gamma^2\sum_{t=0}^{T-1}(1 - \gamma\mu)^{T-1-t}\|\omega_t\|^2.
\end{aligned}$$

Before we proceed, we introduce a new notation:

$$\xi_t = \begin{cases} x^t - x^* - \gamma(F(x^t) - h^*), & \text{if } \|x^t - x^* - \gamma(F(x^t) - h^*)\| \leq 2\sqrt{2}\exp(-\gamma\mu t/2)\sqrt{V}, \\ 0, & \text{otherwise,} \end{cases}$$
(263)

for $t = 0, 1, \ldots, T$. Random vectors $\{\xi_t\}_{t=0}^T$ are bounded almost surely:

$$\|\xi_t\| \leq 2\sqrt{2}\exp(-\gamma\mu t/2)\sqrt{V}$$
(264)

for all $t = 0, 1, \ldots, T$. In addition, $\xi_t = x^t - x^* - \gamma(F(x^t) - h^*)$ follows from $E_{T-1}$ for all $t = 0, 1, \ldots, T$ and, thus, $E_{T-1}$ implies

$$V_T \leq \exp(-\gamma\mu T)V + \underbrace{\frac{2\gamma}{n}\sum_{t=0}^{T-1}\sum_{i=1}^{n}(1-\gamma\mu)^{T-1-t}\langle\xi_t, \omega_{i,t}^u\rangle}_{①} + \underbrace{\frac{2\gamma}{n}\sum_{t=0}^{T-1}\sum_{i=1}^{n}(1-\gamma\mu)^{T-1-t}\langle\xi_t, \omega_{i,t}^b\rangle}_{②}$$

$$+ \underbrace{\frac{4\gamma^2}{n^2}\sum_{t=0}^{T-1}\sum_{i=1}^{n}(1-\gamma\mu)^{T-1-t}\left[\|\omega_{i,t}^u\|^2 - \mathbb{E}_{\xi^t}[\|\omega_{i,t}^u\|^2]\right]}_{③}$$

$$+ \underbrace{\frac{4\gamma^2}{n^2}\sum_{t=0}^{T-1}\sum_{i=1}^{n}(1-\gamma\mu)^{T-1-t}\mathbb{E}_{\xi^t}[\|\omega_{i,t}^u\|^2]}_{④}$$

$$+ \underbrace{\frac{4\gamma^2}{n}\sum_{t=0}^{T-1}\sum_{i=1}^{n}(1-\gamma\mu)^{T-1-t}\|\omega_{i,t}^b\|^2}_{⑤}$$

$$+ \underbrace{\frac{4\gamma^2}{n^2}\sum_{t=0}^{T-1}\sum_{j=1}^{n}(1-\gamma\mu)^{T-1-t}\left\langle\sum_{i=1}^{j-1}\omega_{i,t}^u, \omega_{j,t}^u\right\rangle}_{⑥}.$$
(265)

To derive high-probability bounds for ①, ②, ③, ④, ⑤, ⑥ we need to establish several useful inequalities related to $\omega_{i,t}^u, \omega_{i,t}^b$. First, by definition of clipping

$$\|\omega_{i,t}^u\| \leq 2\lambda_t.$$
(266)

Next, we notice that $E_{T-1}$ implies

$$\|F_i(x^t) - h_i^t\| \leq \|F_i(x^t) - h_i^*\| + \|h_i^t - h_i^*\| \overset{(35)}{\leq} \ell\|x^t - x^*\| + \sqrt{\sum_{i=1}^{n}\|h_i^t - h_i^*\|^2}$$

$$\leq \left(\ell + \frac{n}{3000\gamma\ln\left(\frac{48n(K+1)}{\beta}\right)}\right)\sqrt{V_t}$$

$$\overset{(261)}{\leq} \sqrt{2}\left(\ell + \frac{n}{3000\gamma\ln\left(\frac{48n(K+1)}{\beta}\right)}\right)\exp(-\gamma\mu t/2)\sqrt{V} \overset{(254),(257)}{\leq} \frac{\lambda_t}{2}. \quad (267)$$

for $t = 0, 1, \ldots, T-1$ and $i \in [n]$. Therefore, one can apply Lemma B.2 and get

$$\|\omega_t^b\| \leq \frac{1}{n}\sum_{i=1}^{n}\|\omega_{i,t}^b\| \leq \frac{2^\alpha\sigma^\alpha}{\lambda_t^{\alpha-1}},$$
(268)

$$\mathbb{E}_{\xi_i^t}\left[\|\omega_{i,t}^u\|^2\right] \leq 18\lambda_t^{2-\alpha}\sigma^\alpha,$$
(269)

for all $t = 0, 1, \ldots, T-1$ and $i \in [n]$. In addition, we require the following condition

$$\nu \leq \frac{1}{18000000\ln^2\left(\frac{48n(K+1)}{\beta}\right)}.$$
(270)

**Upper bound for ①.** To estimate this sum, we will use Bernstein's inequality. The summands have conditional expectations equal to zero:

$$\mathbb{E}_{\xi_i^l}\left[\frac{2\gamma}{n}(1-\gamma\mu)^{T-1-l}\langle\xi_t,\omega_{i,l}^u\rangle\right] = \frac{2\gamma}{n}\exp\left(-\gamma\mu(T-1-l)\right)\left\langle\xi_l,\mathbb{E}_{\xi_i^l}[\omega_{i,l}^u]\right\rangle = 0.$$

Moreover, for all $l = 0,\ldots,T-1$ random vectors $\{\omega_{i,l}^u\}_{i=1}^n$ are independent. Thus, sequence $\left\{\frac{2\gamma}{n}\exp\left(-\gamma\mu(T-1-l)\right)\langle\xi_l,\omega_{i,l}^u\rangle\right\}_{l,i=0,1}^{T-1,n}$ is a martingale difference sequence. Next, the summands are bounded:

$$\left|\frac{2\gamma}{n}\exp(-\gamma\mu(T-1-l))\langle\xi_l,\omega_{i,l}^u\rangle\right| \quad\leq\quad \frac{2\gamma}{n}\exp(-\gamma\mu(T-1-l))\|\xi_l\|\cdot\|\omega_{i,l}^u\|$$

$$\overset{(264),(266)}{\leq}\quad \frac{8\sqrt{2}}{n}\gamma\exp(-\gamma\mu(T-1-{}^l\!/_2))\sqrt{V}\lambda_l$$

$$\overset{(257)}{\leq}\quad \frac{\exp(-\gamma\mu T)V}{8\ln\frac{48n(K+1)}{\beta}} \overset{\text{def}}{=} c. \tag{271}$$

Finally, conditional variances $\sigma_{i,l}^2 \overset{\text{def}}{=} \mathbb{E}_{\xi_i^l}\left[\frac{4\gamma^2}{n^2}\exp\left(-2\gamma\mu(T-1-l)\right)\langle\xi_l,\omega_{i,l}^u\rangle^2\right]$ of the summands are bounded:

$$\sigma_{i,l}^2 \quad\leq\quad \mathbb{E}_{\xi_i^l}\left[\frac{4\gamma^2}{n^2}\exp(-2\gamma\mu(T-1-l))\|\xi_l\|^2\cdot\|\omega_{i,l}^u\|^2\right]$$

$$\overset{(264)}{\leq}\quad \frac{32\gamma^2}{n^2}\exp(-\gamma\mu(2T-2-l))V\mathbb{E}_{\xi_i^l}\left[\|\omega_{i,l}^u\|^2\right]. \tag{272}$$

Applying Bernstein's inequality (Lemma B.1) with $X_{i,l} = \frac{2\gamma}{n}\exp\left(-\gamma\mu(T-1-l)\right)\langle\xi_l,\omega_{i,l}^u\rangle$, constant $c$ defined in (271), $b = \frac{\exp(-\gamma\mu T)V}{8}$, $G = \frac{\exp(-2\gamma\mu T)V^2}{384\ln\frac{48n(K+1)}{\beta}}$, we get

$$\mathbb{P}\left\{|①| > \frac{\exp(-\gamma\mu T)V}{8} \text{ and } \sum_{l=0}^{T-1}\sum_{i=1}^n\sigma_{i,l}^2 \leq \frac{\exp(-2\gamma\mu T)V^2}{384\ln\frac{48n(K+1)}{\beta}}\right\} \leq 2\exp\left(-\frac{b^2}{2G+{}^{2cb}\!/_3}\right) = \frac{\beta}{24n(K+1)}.$$

The above is equivalent to $\mathbb{P}\{E_①\} \geq 1 - \frac{\beta}{24n(K+1)}$ for

$$E_① = \left\{\text{either}\quad \sum_{l=0}^{T-1}\sum_{i=1}^n\sigma_{i,l}^2 > \frac{\exp(-2\gamma\mu T)V^2}{384\ln\frac{48n(K+1)}{\beta}} \quad\text{or}\quad |①| \leq \frac{\exp(-\gamma\mu T)V}{8}\right\}. \tag{273}$$

Moreover, $E_{T-1}$ implies

$$\sum_{l=0}^{T-1}\sum_{i=1}^n\sigma_{i,l}^2 \quad\overset{(272)}{\leq}\quad \frac{32\gamma^2}{n}\exp(-2\gamma\mu(T-1))V\sum_{l=0}^{T-1}\frac{\mathbb{E}_{\xi_i^l}\left[\|\omega_{i,l}^u\|^2\right]}{\exp(-\gamma\mu l)}$$

$$\overset{(269),T\leq K+1}{\leq}\quad \frac{576\gamma^2}{n}\exp(-2\gamma\mu(T-1)V\sigma^\alpha\sum_{l=0}^K\frac{\lambda_l^{2-\alpha}}{\exp(-\gamma\mu l)}$$

$$\overset{(257)}{\leq}\quad \frac{9(64\sqrt{2})^\alpha}{\sqrt{2}}\frac{\gamma^\alpha\exp(-2\gamma\mu(T-1)\sqrt{V}^{4-\alpha}\sigma^\alpha(K+1)\exp(\frac{\gamma\mu\alpha K}{2})}{n^{\alpha-1}\ln^{2-\alpha}\frac{48n(K+1)}{\beta}}$$

$$\overset{(254)}{\leq}\quad \frac{\exp(-2\gamma\mu T)V^2}{384\ln\frac{48n(K+1)}{\beta}}. \tag{274}$$

**Upper bound for ②.** Probability event $E_{T-1}$ implies

$$
\begin{aligned}
② \quad &\leq \quad \frac{2\gamma}{n}\exp(-\gamma\mu(T-1))\sum_{l=0}^{T-1}\sum_{i=1}^{n}\frac{\|\xi_l\|\cdot\|\omega_{i,l}^b\|}{\exp(-\gamma\mu l)} \\
&\overset{(264),(268)}{\leq} \quad 2^{2+\alpha}\sqrt{2}\gamma\exp(-\gamma\mu(T-1))\sqrt{V}\sigma^\alpha\sum_{l=0}^{T-1}\frac{1}{\lambda_l^{\alpha-1}\exp(-\gamma\mu l/2)} \\
&\overset{(257),T\leq K+1}{\leq} \quad \frac{(128\sqrt{2})^\alpha}{16}\cdot\frac{\gamma^\alpha\sigma^\alpha\exp(-\gamma\mu(T-1))(K+1)\exp\left(\frac{\gamma\mu\alpha K}{2}\right)\exp(\gamma\mu\alpha)\ln^{\alpha-1}\frac{48n(K+1)}{\beta}}{n^{\alpha-1}\sqrt{V}^{\alpha-2}} \\
&\overset{(254)}{\leq} \quad \frac{\exp(-\gamma\mu T)V}{8}.
\end{aligned}
\tag{275}
$$

**Upper bound for ③.** To estimate this sum, we will use Bernstein's inequality. The summands have conditional expectations equal to zero:

$$
\mathbb{E}_{\xi_i^l}\left[\frac{4\gamma^2}{n^2}(1-\gamma\mu)^{T-1-l}\left[\|\omega_{i,l}^u\|^2-\mathbb{E}_{\xi_i^l}\left[\|\omega_{i,l}^u\|^2\right]\right]\right]=0.
$$

Moreover, for all $l=0,\ldots,T-1$ random vectors $\{\omega_{i,l}^u\}_{i=1}^n$ are independent. Thus, sequence $\left\{\frac{4\gamma^2}{n^2}\exp\left(-\gamma\mu(T-1-l)\right)\left(\|\omega_{i,l}^u\|^2-\mathbb{E}_{\xi_i^l}\left[\|\omega_{i,l}^u\|^2\right]\right)\right\}_{l,i=0,1}^{T-1,n}$ is a martingale difference sequence. Next, the summands are bounded:

$$
\begin{aligned}
\frac{4\gamma^2}{n^2}(1-\gamma\mu)^{T-1-l}\left|\|\omega_{i,l}^u\|^2-\mathbb{E}_{\xi_i^l}\left[\|\omega_{i,l}^u\|^2\right]\right| \quad &\overset{(266)}{\leq} \quad \frac{32\gamma^2\lambda_l^2}{n^2}\frac{\exp(-\gamma\mu T)}{\exp(-\gamma\mu(1+l))} \\
&\overset{(257)}{\leq} \quad \frac{\exp(-\gamma\mu(T+1))V}{256\ln^2\frac{48n(K+1)}{\beta}} \\
&\leq \quad \frac{\exp(-\gamma\mu T)V}{8\ln\frac{48n(K+1)}{\beta}}\overset{\text{def}}{=}c.
\end{aligned}
\tag{276}
$$

Finally, conditional variances

$$
\widetilde{\sigma}_{i,l}^2\overset{\text{def}}{=}\mathbb{E}_{\xi_i^l}\left[\frac{16\gamma^4}{n^4}(1-\gamma\mu)^{2T-2-2l}\left|\|\omega_{i,l}^u\|^2-\mathbb{E}_{\xi_i^l}\left[\|\omega_{i,l}^u\|^2\right]\right|^2\right]
$$

of the summands are bounded:

$$
\begin{aligned}
\widetilde{\sigma}_{i,l}^2\quad &\overset{(276)}{\leq} \quad \frac{4\gamma^2\exp(-2\gamma\mu T)V}{8n^2\exp(-\gamma\mu(1+l))\ln\frac{48n(K+1)}{\beta}}\mathbb{E}_{\xi_i^l}\left[\left|\|\omega_{i,l}^u\|^2-\mathbb{E}_{\xi_i^l}\left[\|\omega_{i,l}^u\|^2\right]\right|\right] \\
&\leq \quad \frac{\gamma^2\exp(-2\gamma\mu T)V}{n^2\exp(-\gamma\mu(1+l))\ln\frac{48n(K+1)}{\beta}}\mathbb{E}_{\xi_i^l}\left[\|\omega_{i,l}^u\|^2\right].
\end{aligned}
\tag{277}
$$

Applying Bernstein's inequality (Lemma B.1) with $X_{i,l} = \frac{4\gamma^2}{n^2}(1-\gamma\mu)^{T-1-l}\left[\|\omega_{i,l}^u\|^2-\mathbb{E}_{\xi_i^l}\left[\|\omega_{i,l}^u\|^2\right]\right]$, constant $c$ defined in (276), $b = \frac{\exp(-\gamma\mu T)V}{8}$, $G = \frac{\exp(-2\gamma\mu T)V^2}{384\ln\frac{48n(K+1)}{\beta}}$, we get

$$
\mathbb{P}\left\{|③|>\frac{\exp(-\gamma\mu T)V}{8}\text{ and }\sum_{l=0}^{T-1}\sum_{i=1}^{n}\widetilde{\sigma}_{i,l}^2\leq\frac{\exp(-2\gamma\mu T)V^2}{384\ln\frac{48n(K+1)}{\beta}}\right\}\leq 2\exp\left(-\frac{b^2}{2G+2cb/3}\right)=\frac{\beta}{24n(K+1)}.
$$

The above is equivalent to $\mathbb{P}\{E_③\}\geq 1-\frac{\beta}{24n(K+1)}$ for

$$
E_③=\left\{\text{either}\quad\sum_{l=0}^{T-1}\sum_{i=1}^{n}\widetilde{\sigma}_{i,l}^2>\frac{\exp(-2\gamma\mu T)V^2}{384\ln\frac{48n(K+1)}{\beta}}\quad\text{or}\quad|③|\leq\frac{\exp(-\gamma\mu T)V}{8}\right\}.
\tag{278}
$$

Moreover, $E_{T-1}$ implies

$$
\begin{aligned}
\sum_{l=0}^{T-1}\sum_{i=1}^{n}\widetilde{\sigma}_{i,l}^2
&\overset{(277)}{\leq}
\frac{\gamma^2\exp(-\gamma\mu(2T-1))V}{n^2\ln\frac{48n(K+1)}{\beta}}\sum_{l=0}^{T-1}\sum_{i=1}^{n}\frac{\mathbb{E}_{\xi_i^l}\left[\|\omega_{i,l}^u\|^2\right]}{\exp(-\gamma\mu l)} \\[2mm]
&\overset{(269),T\leq K+1}{\leq}
\frac{18\gamma^2\exp(-\gamma\mu(2T-1))V\sigma^\alpha}{n\ln\frac{48n(K+1)}{\beta}}\sum_{l=0}^{K}\frac{\lambda_l^{2-\alpha}}{\exp(-\gamma\mu l)} \\[2mm]
&\overset{(257)}{\leq}
\frac{9(64\sqrt{2})^\alpha}{4096}\cdot\frac{\gamma^\alpha\exp(-\gamma\mu(2T-1))\sqrt{V}^{4-\alpha}\sigma^\alpha(K+1)\exp(\frac{\gamma\mu\alpha K}{2})}{n^{\alpha-1}\ln^{3-\alpha}\frac{48n(K+1)}{\beta}} \\[2mm]
&\overset{(254)}{\leq}
\frac{\exp(-2\gamma\mu T)V^2}{384\ln\frac{48n(K+1)}{\beta}}.
\end{aligned}
\tag{279}
$$

**Upper bound for ④.** Probability event $E_{T-1}$ implies

$$
\begin{aligned}
④ &=
\frac{4\gamma^2}{n^2}\sum_{l=0}^{T-1}\sum_{i=1}^{n}(1-\gamma\mu)^{T-1-l}\mathbb{E}_{\xi_i^l}[\|\omega_{i,l}^u\|^2] \\[2mm]
&\overset{(269)}{\leq}
\frac{72\gamma^2\exp(-\gamma\mu(T-1))\sigma^\alpha}{n}\sum_{l=0}^{T-1}\frac{\lambda_l^{2-\alpha}}{\exp(-\gamma\mu l)} \\[2mm]
&\overset{(257),T\leq K+1}{\leq}
\frac{9(64\sqrt{2})^\alpha}{1024}\cdot\frac{\gamma^\alpha\sqrt{V}^{2-\alpha}\exp(-\gamma\mu(T-1))\sigma^\alpha(K+1)\exp(\frac{\gamma\mu\alpha K}{2})}{n^{\alpha-1}\ln^{2-\alpha}\frac{48n(K+1)}{\beta}} \\[2mm]
&\overset{(254)}{\leq}
\frac{\exp(-\gamma\mu T)V}{8}.
\end{aligned}
\tag{280}
$$

**Upper bound for ⑤.** Probability event $E_{T-1}$ implies

$$
\begin{aligned}
⑤ &=
\frac{4\gamma^2}{n}\sum_{l=0}^{T-1}\sum_{i=1}^{n}(1-\gamma\mu)^{T-1-l}\|\omega_{i,l}^b\|^2 \\[2mm]
&\overset{(268)}{\leq}
4\cdot2^{2\alpha}\gamma^2\exp(-\gamma\mu(T-1))\sigma^{2\alpha}\sum_{l=0}^{T-1}\frac{1}{\lambda_l^{2\alpha-2}\exp(-\gamma\mu l)} \\[2mm]
&\overset{(257),T\leq K+1}{\leq}
\frac{(128\sqrt{2})^\alpha}{2048}\cdot\frac{\gamma^{2\alpha}\exp(-\gamma\mu(T-3))\sigma^{2\alpha}\ln^{2(\alpha-1)}\frac{48n(K+1)}{\beta}(K+1)\exp(\gamma\mu\alpha K)}{n^{2(\alpha-1)}V^{\alpha-1}} \\[2mm]
&\overset{(254)}{\leq}
\frac{\exp(-\gamma\mu T)V}{8}.
\end{aligned}
\tag{281}
$$

**Upper bounds for ⑥.** This sum requires more refined analysis. We introduce new vectors:

$$
\delta_j^l=\begin{cases}
\frac{\gamma}{n}\sum\limits_{i=1}^{j-1}\omega_{i,l}^u, & \text{if }\left\|\frac{\gamma}{n}\sum\limits_{i=1}^{j-1}\omega_{i,l}^u\right\|\leq\exp\left(-\frac{\gamma\mu l}{2}\right)\frac{\sqrt{V}}{2}, \\
0, & \text{otherwise,}
\end{cases}
\tag{282}
$$

for all $j\in[n]$ and $l=0,\dots,T-1$. Then, by definition

$$
\|\delta_j^l\|\leq\exp\left(-\frac{\gamma\mu l}{2}\right)\frac{\sqrt{V}}{2}
\tag{283}
$$

and

$$
\begin{aligned}
⑥ &=
\underbrace{\frac{4\gamma}{n}\sum_{l=0}^{T-1}\sum_{j=2}^{n}\exp\left(-\gamma\mu(T-1-t)\right)\left\langle\delta_j^l,\omega_{j,l}^u\right\rangle}_{⑥'} \\[2mm]
&\quad+\frac{4\gamma}{n}\sum_{l=0}^{T-1}\sum_{j=2}^{n}\exp\left(-\gamma\mu(T-1-t)\right)\left\langle\frac{\gamma}{n}\sum_{i=1}^{j-1}\omega_{i,l}^u-\delta_j^l,\omega_{j,l}^u\right\rangle.
\end{aligned}
\tag{284}
$$

We also note here that $E_{T-1}$ implies

$$\frac{4\gamma}{n} \sum_{l=0}^{T-1} \sum_{j=2}^{n} \exp\left(-\gamma\mu(T-1-t)\right) \left\langle \frac{\gamma}{n} \sum_{i=1}^{j-1} \omega_{i,l}^u - \delta_j^l, \omega_{j,l}^u \right\rangle$$

$$= \frac{4\gamma}{n} \sum_{j=2}^{n} \exp\left(-\gamma\mu(T-1-t)\right) \left\langle \frac{\gamma}{n} \sum_{i=1}^{j-1} \omega_{i,T-1}^u - \delta_j^{T-1}, \omega_{j,T-1}^u \right\rangle. \quad (285)$$

**Upper bound for ⑥′.** To estimate this sum, we will use Bernstein's inequality. The summands have conditional expectations equal to zero:

$$\mathbb{E}_{\xi_j^l}\left[ \frac{4\gamma}{n} \exp\left(-\gamma\mu(T-1-l)\right) \langle \delta_j^l, \omega_{j,l}^u \rangle \right] = \frac{4\gamma}{n} \exp\left(-\gamma\mu(T-1-l)\right) \left\langle \delta_j^l, \mathbb{E}_{\xi_j^l}[\omega_{j,l}^u] \right\rangle = 0.$$

Moreover, for all $l = 0, \ldots, T-1$ random vectors $\{\omega_{j,l}^u\}_{j=1}^n$ are independent. Thus, sequence $\left\{ \frac{4\gamma}{n} \exp\left(-\gamma\mu(T-1-l)\right) \langle \delta_j^l, \omega_{j,l}^u \rangle \right\}_{l,j=0,1}^{T-1,n}$ is a martingale difference sequence. Next, the summands are bounded:

$$\left| \frac{4\gamma}{n} \exp\left(-\gamma\mu(T-1-l)\right) \langle \delta_j^l, \omega_{j,l}^u \rangle \right| \leq \frac{4\gamma}{n} \exp\left(-\gamma\mu(T-1-l)\right) \|\delta_j^l\| \cdot \|\omega_{j,l}^u\|$$

$$\overset{(283),(266)}{\leq} \frac{4\sqrt{V}\gamma \exp\left(-\gamma\mu(T-1)\right)}{n} \exp\left(\frac{\gamma\mu l}{2}\right) \lambda_l$$

$$\overset{(257)}{=} \frac{\exp\left(-\gamma\mu T\right) V}{16\sqrt{2} \ln \frac{48n(K+1)}{\beta}}$$

$$\leq \frac{\exp\left(-\gamma\mu T\right) V}{8 \ln \frac{48n(K+1)}{\beta}} \overset{\text{def}}{=} c. \quad (286)$$

Finally, conditional variances $(\sigma_{j,l}')^2 \overset{\text{def}}{=} \mathbb{E}_{\xi_j^l}\left[ \frac{16\gamma^2}{n^2} \exp\left(-\gamma\mu(2T-2-2l)\right) \langle \delta_j^l, \omega_{j,l}^u \rangle^2 \right]$ of the summands are bounded:

$$(\sigma_{j,l}')^2 \leq \mathbb{E}_{\xi_j^l}\left[ \frac{16\gamma^2}{n^2} \exp\left(-\gamma\mu(2T-2-2l)\right) \|\delta_j^l\|^2 \cdot \|\omega_{j,l}^u\|^2 \right]$$

$$\overset{(283)}{\leq} \frac{4\gamma^2 V \exp\left(-\gamma\mu(2T-2-l)\right)}{n^2} \mathbb{E}_{\xi_j^l}\left[ \|\omega_{j,l}^u\|^2 \right]. \quad (287)$$

Applying Bernstein's inequality (Lemma B.1) with $X_{j,l} = \frac{4\gamma}{n} \exp\left(-\gamma\mu(T-1-l)\right) \langle \delta_j^l, \omega_{j,l}^u \rangle$, constant $c$ defined in (287), $b = \frac{\exp(-\gamma\mu T)V}{8}$, $G = \frac{\exp(-2\gamma\mu T)V^2}{384 \ln \frac{48n(K+1)}{\beta}}$, we get

$$\mathbb{P}\left\{ |⑥′| > \frac{\exp\left(-\gamma\mu T\right) V}{8} \text{ and } \sum_{l=0}^{T-1} \sum_{j=2}^{n} (\sigma_{j,l}')^2 \leq \frac{\exp\left(-2\gamma\mu T\right) V^2}{384 \ln \frac{48n(K+1)}{\beta}} \right\} \leq 2\exp\left(-\frac{b^2}{2G + 2cb/3}\right)$$

$$= \frac{\beta}{24n(K+1)}.$$

The above is equivalent to $\mathbb{P}\{E_{⑥′}\} \geq 1 - \frac{\beta}{24n(K+1)}$ for

$$E_{⑥′} = \left\{ \text{either } \sum_{l=0}^{T-1} \sum_{j=2}^{n} (\sigma_{j,l}')^2 > \frac{\exp\left(-2\gamma\mu T\right) V^2}{384 \ln \frac{48n(K+1)}{\beta}} \text{ or } |⑥′| \leq \frac{\exp\left(-\gamma\mu T\right) V}{8} \right\}. \quad (288)$$

Moreover, $E_{T-1}$ implies

$$
\begin{aligned}
\sum_{l=0}^{T-1}\sum_{j=2}^{n}(\sigma'_{j,l})^2 &\overset{(287)}{\leq} \frac{4\gamma^2 V \exp\left(-\gamma\mu\left(2T-2\right)\right)}{n^2} \sum_{l=0}^{T-1}\exp\left(\gamma\mu l\right)\sum_{i=1}^{n}\mathbb{E}_{\xi_i^l}\left[\|\omega_{i,l}^u\|^2\right]\\
&\overset{(269),T\leq K+1}{\leq} \frac{72\gamma^2 V \exp\left(-\gamma\mu\left(2T-2\right)\right)\sigma^\alpha}{n}\sum_{l=0}^{T-1}\exp\left(\gamma\mu l\right)\lambda_l^{2-\alpha}\\
&\overset{(257)}{\leq} \frac{72\gamma^\alpha V^{2-\frac{\alpha}{2}}\exp\left(-2\gamma\mu T\right)\sigma^\alpha}{(64\sqrt{2})^{2-\alpha}n^{\alpha-1}\ln^{2-\alpha}\frac{48n(K+1)}{\beta}}\sum_{l=0}^{T-1}\exp\left(\frac{\gamma\mu l\alpha}{2}\right)\\
&\leq \frac{72\gamma^\alpha V^{2-\frac{\alpha}{2}}\exp\left(-2\gamma\mu T\right)\sigma^\alpha(K+1)\exp\left(\frac{\gamma\mu K\alpha}{2}\right)}{(64\sqrt{2})^{2-\alpha}n^{\alpha-1}\ln^{2-\alpha}\frac{48n(K+1)}{\beta}}\\
&\overset{(254)}{\leq} \frac{\exp\left(-2\gamma\mu T\right)V^2}{384\ln\frac{48n(K+1)}{\beta}}.
\end{aligned}
\tag{289}
$$

That is, we derive the upper bounds for ①, ②, ③, ④, ⑤, ⑥. More precisely, $E_{T-1}$ implies

$$
V_T \overset{(265)}{\leq} \exp\left(\gamma\mu T\right)V + ① + ② + ③ + ④ + ⑤ + ⑥,
$$

$$
⑥ \overset{(284)}{=} ⑥' + \frac{4\gamma}{n}\sum_{j=2}^{n}\exp\left(-\gamma\mu(T-1-t)\right)\left\langle \frac{\gamma}{n}\sum_{i=1}^{j-1}\omega_{i,T-1}^u - \delta_j^{T-1}, \omega_{j,T-1}^u\right\rangle,
$$

$$
② \overset{(275)}{\leq} \frac{\exp\left(-\gamma\mu T\right)V}{8}, \quad ④ \overset{(280)}{\leq} \frac{\exp\left(-\gamma\mu T\right)V}{8},
$$

$$
⑤ \overset{(281)}{\leq} \frac{\exp\left(-\gamma\mu T\right)V}{8},
$$

$$
\sum_{l=0}^{T-1}\sum_{i=1}^{n}\sigma_{i,l}^2 \overset{(274)}{\leq} \frac{\exp\left(-2\gamma\mu T\right)V^2}{384\ln\frac{48n(K+1)}{\beta}}, \quad \sum_{l=0}^{T-1}\sum_{j=2}^{n}\widetilde{\sigma}_{j,l}^2 \overset{(279)}{\leq} \frac{\exp\left(-2\gamma\mu T\right)V^2}{384\ln\frac{48n(K+1)}{\beta}},
$$

$$
\sum_{l=0}^{T-1}\sum_{j=2}^{n}(\sigma'_{j,l})^2 \overset{(289)}{\leq} \frac{\exp\left(-2\gamma\mu T\right)V^2}{384\ln\frac{48n(K+1)}{\beta}}.
$$

In addition, we also establish (see (273), (278), (288), and our induction assumption)

$$
\mathbb{P}\{E_{T-1}\} \geq 1 - \frac{(T-1)\beta}{K+1},
$$

$$
\mathbb{P}\{E_①\} \geq 1 - \frac{\beta}{24n(K+1)}, \quad \mathbb{P}\{E_③\} \geq 1 - \frac{\beta}{24n(K+1)}, \quad \mathbb{P}\{E_{⑥'}\} \geq 1 - \frac{\beta}{24n(K+1)},
$$

where

$$
\begin{aligned}
E_① &= \left\{\text{either } \sum_{l=0}^{T-1}\sum_{i=1}^{n}\sigma_{i,l}^2 > \frac{\exp\left(-2\gamma\mu T\right)V^2}{384\ln\frac{48n(K+1)}{\beta}} \text{ or } |①| \leq \frac{\exp\left(-\gamma\mu T\right)V}{8}\right\},\\
E_③ &= \left\{\text{either } \sum_{l=0}^{T-1}\sum_{j=2}^{n}\widetilde{\sigma}_{j,l}^2 > \frac{\exp\left(-2\gamma\mu T\right)V^2}{384\ln\frac{48n(K+1)}{\beta}} \text{ or } |③| \leq \frac{\exp\left(-\gamma\mu T\right)V}{8}\right\},\\
E_{⑥'} &= \left\{\text{either } \sum_{l=0}^{T-1}\sum_{j=2}^{n}(\sigma'_{j,l})^2 > \frac{\exp\left(-2\gamma\mu T\right)V^2}{384\ln\frac{48n(K+1)}{\beta}} \text{ or } |⑥'| \leq \frac{\exp\left(-\gamma\mu T\right)V}{8}\right\}.
\end{aligned}
$$

Therefore, probability event $E_{T-1} \cap E_① \cap E_③ \cap E_{⑥'}$ implies

$$V_T \leq \exp(-\gamma\mu T) V \underbrace{\left(1 + \frac{1}{8} + \frac{1}{8} + \frac{1}{8} + \frac{1}{8} + \frac{1}{8} + \frac{1}{8}\right)}_{<2}$$

$$+ \frac{4\gamma}{n} \sum_{j=2}^{n} \exp(-\gamma\mu(T-1-t)) \left\langle \frac{\gamma}{n} \sum_{i=1}^{j-1} \omega_{i,T-1}^u - \delta_j^{T-1}, \omega_{j,T-1}^u \right\rangle. \quad (290)$$

To finish the proof, we need to show that $\frac{\gamma}{n} \sum_{i=1}^{j-1} \omega_{i,T-1}^u = \delta_j^{T-1}$ with high probability. In particular, we consider probability event $\widetilde{E}_{T-1,j}$ defined as follows: inequalities

$$\left\| \frac{\gamma}{n} \sum_{i=1}^{r-1} \omega_{i,T-1}^u \right\| \leq \exp\left(-\frac{\gamma\mu(T-1)}{2}\right) \frac{\sqrt{V}}{2}$$

hold for $r = 2, \ldots, j$ simultaneously. We want to show that $\mathbb{P}\{E_{T-1} \cap \widetilde{E}_{T-1,j}\} \geq 1 - \frac{(T-1)\beta}{K+1} - \frac{j\beta}{8n(K+1)}$ for all $j = 2, \ldots, n$. For $j = 2$ the statement is trivial since

$$\left\| \frac{\gamma}{n} \omega_{1,T-1}^u \right\| \overset{(266)}{\leq} \frac{2\gamma\lambda_{T-1}}{n} \leq \exp\left(-\frac{\gamma\mu(T-1)}{2}\right) \frac{\sqrt{V}}{2}.$$

Next, we assume that the statement holds for some $j = m - 1 < n$, i.e., $\mathbb{P}\{E_{T-1} \cap \widetilde{E}_{T-1,m-1}\} \geq 1 - \frac{(T-1)\beta}{K+1} - \frac{(m-1)\beta}{8n(K+1)}$. Our goal is to prove that $\mathbb{P}\{E_{T-1} \cap \widetilde{E}_{T-1,m}\} \geq 1 - \frac{(T-1)\beta}{K+1} - \frac{m\beta}{8n(K+1)}$. First, we consider $\left\| \frac{\gamma}{n} \sum_{i=1}^{m-1} \omega_{i,T-1}^u \right\|$:

$$\left\| \frac{\gamma}{n} \sum_{i=1}^{m-1} \omega_{i,T-1}^u \right\| = \sqrt{\frac{\gamma^2}{n^2} \left\| \sum_{i=1}^{m-1} \omega_{i,T-1}^u \right\|^2}$$

$$= \sqrt{\frac{\gamma^2}{n^2} \sum_{i=1}^{m-1} \|\omega_{i,T-1}^u\|^2 + \frac{2\gamma}{n} \sum_{i=1}^{m-1} \left\langle \frac{\gamma}{n} \sum_{r=1}^{i-1} \omega_{r,T-1}^u, \omega_{i,T-1}^u \right\rangle}$$

$$\leq \sqrt{\frac{\gamma^2}{n^2} \sum_{t=0}^{T-1} \exp(-\gamma\mu(T-1-t)) \sum_{i=1}^{m-1} \|\omega_{i,t}^u\|^2 + \frac{2\gamma}{n} \sum_{i=1}^{m-1} \left\langle \frac{\gamma}{n} \sum_{r=1}^{i-1} \omega_{r,T-1}^u, \omega_{i,T-1}^u \right\rangle}.$$

Next, we introduce a new notation:

$$\rho_{i,T-1}' = \begin{cases} \frac{\gamma}{n} \sum_{r=1}^{i-1} \omega_{r,T-1}^u, & \text{if } \left\| \frac{\gamma}{n} \sum_{r=1}^{i-1} \omega_{r,T-1}^u \right\| \leq \exp\left(-\frac{\gamma\mu(T-1)}{2}\right) \frac{\sqrt{V}}{2}, \\ 0, & \text{otherwise} \end{cases}$$

for $i = 1, \ldots, m - 1$. By definition, we have

$$\|\rho_{i,T-1}'\| \leq \exp\left(-\frac{\gamma\mu(T-1)}{2}\right) \frac{\sqrt{V}}{2} \quad (291)$$

for $i = 1, \ldots, m - 1$. Moreover, $\widetilde{E}_{T-1,m-1}$ implies $\rho_{i,T-1}' = \frac{\gamma}{n} \sum_{r=1}^{i-1} \omega_{r,T-1}^u$ for $i = 1, \ldots, m - 1$ and

$$\left\| \frac{\gamma}{n} \sum_{i=1}^{m-1} \omega_{i,l}^u \right\| \leq \sqrt{③ + ④ + ⑦},$$

where

$$⑦ = \frac{2\gamma}{n} \sum_{i=1}^{m-1} \left\langle \rho_{i,T-1}', \omega_{i,T-1}^u \right\rangle.$$

It remains to estimate ⑨.

**Upper bound for ⑦.** To estimate this sum, we will use Bernstein's inequality. The summands have conditional expectations equal to zero:

$$\mathbb{E}_{\xi_i^{T-1}}\left[\frac{2\gamma}{n}\langle\rho'_{i,T-1},\omega^u_{i,T-1}\rangle\right] = \frac{2\gamma}{n}\left\langle\rho'_{i,T-1},\mathbb{E}_{\xi_i^{T-1}}[\omega^u_{i,T-1}]\right\rangle = 0.$$

Thus, sequence $\left\{\frac{2\gamma}{n}\langle\rho'_{i,T-1},\omega^u_{i,T-1}\rangle\right\}_{i=1}^n$ is a martingale difference sequence. Next, the summands are bounded:

$$
\begin{aligned}
\left|\frac{2\gamma}{n}\langle\rho'_{i,T-1},\omega^u_{i,T-1}\rangle\right| &\leq & \frac{2\gamma}{n}\|\rho'_{i,T-1}\|\cdot\|\omega^u_{i,T-1}\| \\
&\overset{(291),(266)}{\leq} & \frac{2\sqrt{V}\gamma\exp\left(-\frac{\gamma\mu(T-1)}{2}\right)}{n}\lambda_{T-1} \\
&\overset{(257)}{=} & \frac{\exp\left(-\gamma\mu T\right)V}{32\sqrt{2}\ln\frac{48n(K+1)}{\beta}} \\
&\leq & \frac{\exp\left(-\gamma\mu T\right)V}{8\ln\frac{48n(K+1)}{\beta}}\overset{\text{def}}{=}c.
\end{aligned}
\tag{292}
$$

Finally, conditional variances $(\widetilde{\sigma}'_{i,T-1})^2\overset{\text{def}}{=}\mathbb{E}_{\xi_i^{T-1}}\left[\frac{4\gamma^2}{n^2}\langle\rho'_{i,T-1},\omega^u_{i,T-1}\rangle^2\right]$ of the summands are bounded:

$$
\begin{aligned}
(\widetilde{\sigma}'_{i,T-1})^2 &\leq & \mathbb{E}_{\xi_i^{T-1}}\left[\frac{4\gamma^2}{n^2}\|\rho'_{i,T-1}\|^2\cdot\|\omega^u_{i,T-1}\|^2\right] \\
&\overset{(291)}{\leq} & \frac{\gamma^2 V\exp\left(-\gamma\mu(T-1)\right)}{n^2}\mathbb{E}_{\xi_i^{T-1}}\left[\|\omega^u_{i,T-1}\|^2\right].
\end{aligned}
\tag{293}
$$

Applying Bernstein's inequality (Lemma B.1) with $X_i=\frac{2\gamma}{n}\langle\rho'_{i,T-1},\omega^u_{i,T-1}\rangle$, constant $c$ defined in (292), $b=\frac{\exp(-\gamma\mu T)V}{8}$, $G=\frac{\exp(-2\gamma\mu T)V^2}{384\ln\frac{48n(K+1)}{\beta}}$, we get

$$
\begin{aligned}
\mathbb{P}\left\{|⑦|>\frac{\exp\left(-\gamma\mu T\right)V}{8}\text{ and }\sum_{i=1}^n(\widetilde{\sigma}'_{i,T-1})^2\leq\frac{\exp\left(-2\gamma\mu T\right)V^2}{384\ln\frac{48n(K+1)}{\beta}}\right\} &\leq & 2\exp\left(-\frac{b^2}{2G+{}^{2cb}/_3}\right) \\
&= & \frac{\beta}{24n(K+1)}.
\end{aligned}
$$

The above is equivalent to $\mathbb{P}\{E_⑦\}\geq 1-\frac{\beta}{24n(K+1)}$ for

$$E_⑦=\left\{\text{either }\sum_{i=1}^n(\widetilde{\sigma}'_{i,T-1})^2>\frac{\exp\left(-2\gamma\mu T\right)V^2}{384\ln\frac{48n(K+1)}{\beta}}\text{ or }|⑦|>\frac{\exp\left(-\gamma\mu T\right)V}{8}\right\}.\tag{294}$$

Moreover, $E_{T-1}$ implies

$$
\begin{aligned}
\sum_{i=1}^n(\widetilde{\sigma}'_{i,T-1})^2 &\overset{(293)}{\leq} & \frac{\gamma^2 V\exp\left(-\gamma\mu(T-1)\right)}{n^2}\sum_{i=1}^n\mathbb{E}_{\xi_i^T-1}\left[\|\omega^u_{i,T-1}\|^2\right] \\
&\overset{(269)}{\leq} & \frac{18\gamma^2 V\exp\left(-\gamma\mu(T-1)\right)\sigma^\alpha}{n}\lambda_{T-1}^{2-\alpha} \\
&\overset{(257)}{\leq} & \frac{18\gamma^\alpha V^{2-\frac{\alpha}{2}}\exp\left(-\gamma\mu(T-1)\right)\sigma^\alpha}{(64\sqrt{2})^{2-\alpha}n^{\alpha-1}\ln^{2-\alpha}\frac{48n(K+1)}{\beta}}\exp\left(\frac{\gamma\mu(T-1)\alpha}{4}\right) \\
&\overset{T-1\leq K}{\leq} & \frac{18\gamma^\alpha V^{2-\frac{\alpha}{2}}\exp\left(-\gamma\mu(T-1)\right)\sigma^\alpha\exp\left(\frac{\gamma\mu K\alpha}{2}\right)}{(64\sqrt{2})^{2-\alpha}n^{\alpha-1}\ln^{2-\alpha}\frac{48n(K+1)}{\beta}} \\
&\overset{(254)}{\leq} & \frac{\exp\left(-2\gamma\mu T\right)V^2}{384\ln\frac{48n(K+1)}{\beta}}.
\end{aligned}
\tag{295}
$$

Putting all together we get that $E_{T-1} \cap \widetilde{E}_{T-1,m-1}$ implies

$$\left\| \frac{\gamma}{n} \sum_{i=1}^{m-1} \omega_{i,T-1}^u \right\| \leq \sqrt{③ + ④ + ⑦},$$

$$④ \overset{(280)}{\leq} \frac{\exp(-\gamma\mu T) V}{8}, \quad \sum_{l=0}^{T-1} \sum_{i=1}^{n} \widetilde{\sigma}_{i,l}^2 \overset{(279)}{\leq} \frac{\exp(-2\gamma\mu T) V}{384 \ln \frac{48n(K+1)}{\beta}},$$

$$\sum_{i=1}^{m-1} (\widetilde{\sigma}_{i,T-1}')^2 \leq \frac{\exp(-2\gamma\mu T) V^2}{384 \ln \frac{48n(K+1)}{\beta}}.$$

In addition, we also establish (see (278), (294) and our induction assumption)

$$\mathbb{P}\{E_{T-1} \cap \widetilde{E}_{T-1,m-1}\} \geq 1 - \frac{(T-1)\beta}{K+1} - \frac{(m-1)\beta}{8n(K+1)},$$

$$\mathbb{P}\{E_③\} \geq 1 - \frac{\beta}{24n(K+1)}, \quad \mathbb{P}\{E_⑦\} \geq 1 - \frac{\beta}{24n(K+1)}$$

where

$$E_③ = \left\{ \text{either } \sum_{l=0}^{T-1} \sum_{i=1}^{n} \widetilde{\sigma}_{i,l}^2 > \frac{\exp(-2\gamma\mu T) V^2}{384 \ln \frac{48n(K+1)}{\beta}} \text{ or } |③| \leq \frac{\exp(-\gamma\mu T) V}{8} \right\},$$

$$E_⑦ = \left\{ \text{either } \sum_{i=1}^{n} (\widetilde{\sigma}_{i,T-1}')^2 > \frac{\exp(-2\gamma\mu T) V^2}{384 \ln \frac{48n(K+1)}{\beta}} \text{ or } |⑦| > \frac{\exp(-\gamma\mu T) V}{8} \right\}.$$

Therefore, probability event $E_{T-1} \cap \widetilde{E}_{m-1} \cap E_③ \cap E_⑦$ implies

$$\left\| \frac{\gamma}{n} \sum_{i=1}^{m-1} \omega_{i,T-1}^u \right\| \leq \exp\left(-\frac{\gamma\mu(T-1)}{2}\right) \sqrt{V} \sqrt{\frac{1}{8} + \frac{1}{8}} \leq \frac{\exp\left(-\frac{\gamma\mu(T-1)}{2}\right) \sqrt{V}}{2}.$$

This implies $\widetilde{E}_{T-1,m}$ and

$$\begin{aligned}
\mathbb{P}\{E_{T-1} \cap \widetilde{E}_{T-1,m}\} &\geq \mathbb{P}\{E_{T-1} \cap \widetilde{E}_{T-1,m-1} \cap E_⑥ \cap E_{⑥'}\} \\
&= 1 - \mathbb{P}\left\{ \overline{E_{T-1} \cap \widetilde{E}_{T-1,m-1}} \cup \overline{E}_⑥ \cup \overline{E}_{⑥'} \right\} \\
&\geq 1 - \frac{(T-1)\beta}{K+1} - \frac{m\beta}{8n(K+1)}.
\end{aligned}$$

Therefore, for all $m = 2, \ldots, n$ the statement holds and, in particular, $\mathbb{P}\{E_{T-1} \cap \widetilde{E}_{T-1,n}\} \geq 1 - \frac{(T-1)\beta}{K+1} - \frac{\beta}{8(K+1)}$, i.e., (262) holds. Taking into account (290), we conclude that $E_{T-1} \cap \widetilde{E}_{T-1,n} \cap E_① \cap E_③ \cap E_⑤ \cap E_{⑧'}$ implies

$$V_T \leq 2 \exp(-\gamma\mu T) V$$

that is equivalent to (261) for $t = T$. Moreover,

$$\begin{aligned}
\mathbb{P}\{E_T\} &\geq \mathbb{P}\left\{ E_{T-1} \cap \widetilde{E}_{T-1,n} \cap E_① \cap E_③ \cap E_{⑥'} \right\} \\
&= 1 - \mathbb{P}\left\{ \overline{E_{T-1} \cap \widetilde{E}_n} \cup \overline{E}_① \cup \overline{E}_③ \cup \overline{E}_{⑥'} \right\} \\
&= 1 - \frac{(T-1)\beta}{K+1} - \frac{\beta}{8(K+1)} - 3 \cdot \frac{\beta}{24n(K+1)} \geq 1 - \frac{T\beta}{K+1}.
\end{aligned}$$

In other words, we showed that $\mathbb{P}\{E_k\} \geq 1 - {}^{k\beta}/(K+1)$ for all $k = 0, 1, \ldots, K+1$. For $k = K+1$ we have that with probability at least $1 - \beta$

$$\|x^{K+1} - x^*\|^2 \leq 2 \exp(-\gamma\mu(K+1)) V.$$

Finally, if

$$\gamma = \min\left\{\frac{1}{4096\ell\ln\frac{48n(K+1)}{\beta}}, \frac{\sqrt{n}R}{3000\zeta_*\ln\frac{48n(K+1)}{\beta}}, \frac{\ln(B_K)}{\mu(K+1)}\right\},$$

$$B_K = \max\left\{2, \left(\frac{\sqrt{2}}{3456}\right)^{\frac{2}{\alpha}}\cdot\frac{(K+1)^{\frac{2(\alpha-1)}{\alpha}}\mu^2 V n^{\frac{2(\alpha-1)}{\alpha}}}{\sigma^2\ln^{\frac{2(\alpha-1)}{\alpha}}\left(\frac{48n(K+1)}{\beta}\right)\ln^2(B_K)}\right\}$$

$$= \mathcal{O}\left(\max\left\{2, \frac{K^{\frac{2(\alpha-1)}{\alpha}}\mu^2 V n^{\frac{2(\alpha-1)}{\alpha}}}{\sigma^2\ln^{\frac{2(\alpha-1)}{\alpha}}\left(\frac{nK}{\beta}\right)\ln^2\left(\max\left\{2, \frac{K^{\frac{2(\alpha-1)}{\alpha}}\mu^2 V n^{\frac{2(\alpha-1)}{\alpha}}}{\sigma^2\ln^{\frac{2(\alpha-1)}{\alpha}}\left(\frac{nK}{\beta}\right)}\right\}\right)}\right\}\right)$$

then with probability at least $1-\beta$

$$\|x^{K+1}-x^*\|^2 \leq 2\exp(-\gamma\mu(K+1))V$$

$$= 2V\max\left\{\exp\left(-\frac{\mu(K+1)}{4096\ell\ln\frac{48n(K+1)}{\beta}}\right), \exp\left(-\frac{\mu\sqrt{n}RK}{3000\zeta_*\ln\frac{48nK}{\beta}}\right), \frac{1}{B_K}\right\}$$

$$= \mathcal{O}\left(\max\left\{R^2\exp\left(-\frac{\mu K}{\ell\ln\frac{nK}{\beta}}\right), R^2\exp\left(-\frac{\mu\sqrt{n}RK}{\zeta_*\ln\frac{nK}{\beta}}\right), \frac{\sigma^2\ln^2 B_K}{\ln^{\frac{2(1-\alpha)}{\alpha}}\left(\frac{nK}{\beta}\right)K^{\frac{2(\alpha-1)}{\alpha}}\mu^2 n^{\frac{2(\alpha-1)}{\alpha}}}\right\}\right).$$

To get $\|x^{K+1}-x^*\|^2 \leq \varepsilon$ with probability $\geq 1-\beta$, $K$ should be

$$K = \mathcal{O}\left(\max\left\{\frac{\ell}{\mu}\ln\left(\frac{R^2}{\varepsilon}\right)\ln\left(\frac{n\ell}{\mu\beta}\ln\frac{R^2}{\varepsilon}\right), \frac{\zeta_*}{\sqrt{n}R\mu}\ln\left(\frac{R^2}{\varepsilon}\right)\ln\left(\frac{\sqrt{n}\zeta_*}{R\mu\beta}\ln\frac{R^2}{\varepsilon}\right),\right.\right.$$

$$\left.\left.\frac{1}{n}\left(\frac{\sigma^2}{\mu^2\varepsilon}\right)^{\frac{\alpha}{2(\alpha-1)}}\ln\left(\frac{1}{\beta}\left(\frac{\sigma^2}{\mu^2\varepsilon}\right)^{\frac{\alpha}{2(\alpha-1)}}\right)\ln^{\frac{\alpha}{\alpha-1}}(B_\varepsilon)\right\}\right),$$

where

$$B_\varepsilon = \max\left\{2, \frac{2R^2}{\varepsilon\ln\left(\frac{1}{\beta}\left(\frac{\sigma^2}{\mu^2\varepsilon}\right)^{\frac{\alpha}{2(\alpha-1)}}\right)}\right\}.$$

This concludes the proof. □

# H  MISSING PROOFS FOR DProx-clipped-SEG-shift

In this section, we give the complete formulations of our results for DProx-clipped-SEG-shift and rigorous proofs. For the readers' convenience, the method's update rule is repeated below:

$$\widetilde{x}^k = \mathrm{prox}_{\gamma\Psi}\left(x^k - \gamma\widetilde{g}^k\right), \ \ \widetilde{g}^k = \frac{1}{n}\sum_{i=1}^n \widetilde{g}_i^k, \ \ \widetilde{g}_i^k = \widetilde{h}_i^k + \tilde{\Delta}_i^k, \ \ \widetilde{h}_i^{k+1} = \widetilde{h}_i^k + \nu\tilde{\Delta}_i^k,$$

$$x^{k+1} = \mathrm{prox}_{\gamma\Psi}\left(x^k - \gamma\widehat{g}^k\right), \ \ \widehat{g}^k = \frac{1}{n}\sum_{i=1}^n \widehat{g}_i^k, \ \ \widehat{g}_i^k = \widehat{h}_i^k + \hat{\Delta}_i^k, \ \ \widehat{h}_i^{k+1} = \widehat{h}_i^k + \nu\hat{\Delta}_i^k,$$

$$\tilde{\Delta}_i^k = \mathrm{clip}(F_{\xi_{1,i}^k}(x^k) - \widetilde{h}_i^k, \lambda_k), \ \ \hat{\Delta}_i^k = \mathrm{clip}(F_{\xi_{2,i}^k}(\widetilde{x}^k) - \widehat{h}_i^k, \lambda_k).$$

## H.1  MONOTONE CASE

The following lemma is the main "optimization" part of the analysis of DProx-clipped-SEG-shift.

**Lemma H.1.** *Let Assumptions 6 and 7 hold for $Q = B_{4n\sqrt{V}}(x^*)$, where $V \geq \|x^0 - x^*\|^2 + \frac{409600\gamma^2 \ln^2 \frac{48n(K+1)}{\beta}}{n^2}\sum_{i=1}^n \|F_i(x^*)\|^2$, and $0 < \gamma \leq \frac{1}{\sqrt{12}L}$. If $x^k$ and $\widetilde{x}^k$ lie in $B_{4n\sqrt{V}}(x^*)$ for all $k = 0, 1, \ldots, K$ for some $K \geq 0$, then for all $u \in B_{4n\sqrt{V}}(x^*)$ the iterates produced by DProx-clipped-SEG-shift satisfy*

$$\langle F(u), \widetilde{x}_{avg}^K - u\rangle + \Psi(\widetilde{x}_{avg}^K) - \Psi(u) \ \leq \ \frac{\|x^0 - u\|^2 - \|x^{K+1} - u\|^2}{2\gamma(K+1)}$$

$$+ \frac{\gamma}{K+1}\sum_{k=0}^K (3\|\omega_k\|^2 + 4\|\theta_k\|^2)$$

$$+ \frac{1}{K+1}\sum_{k=0}^K \langle\theta_k, x^k - u\rangle, \quad (296)$$

$$\widetilde{x}_{avg}^K \ \stackrel{def}{=} \ \frac{1}{K+1}\sum_{k=0}^K \widetilde{x}^k, \quad (297)$$

$$\theta_k \ \stackrel{def}{=} \ F(\widetilde{x}^k) - \widehat{g}^k, \quad (298)$$

$$\omega_k \ \stackrel{def}{=} \ F(x^k) - \widetilde{g}^k. \quad (299)$$

*Proof.* Since $\widetilde{x}^k = \mathrm{prox}_{\gamma\Psi}\left(x^k - \gamma\widetilde{g}^k\right)$ and $x^{k+1} = \mathrm{prox}_{\gamma\Psi}\left(x^k - \gamma\widehat{g}^k\right)$, we have $x^k - \gamma\widetilde{g}^k - \widetilde{x}^k \in \gamma\partial\Psi(\widetilde{x}^k)$ and $x^k - \gamma\widehat{g}^k - x^{k+1} \in \gamma\partial\Psi(x^{k+1})$. By definition of the subgradient, we have $\forall u \in \mathbb{R}^d$

$$\gamma\left(\Psi(\widetilde{x}^k) - \Psi(x^{k+1})\right) \ \leq \ \langle\widetilde{x}^k - x^k + \gamma\widetilde{g}^k, x^{k+1} - \widetilde{x}^k\rangle,$$

$$\gamma\left(\Psi(x^{k+1}) - \Psi(u)\right) \ \leq \ \langle x^{k+1} - x^k + \gamma\widehat{g}^k, u - x^{k+1}\rangle.$$

Summing up the above inequalities, we get

$$\gamma\left(\Psi(\widetilde{x}^k) - \Psi(u)\right) \ \leq \ \langle\widetilde{x}^k - x^k, x^{k+1} - \widetilde{x}^k\rangle + \langle x^{k+1} - x^k, u - x^{k+1}\rangle$$

$$+ \gamma\langle\widetilde{g}^k - \widehat{g}^k, x^{k+1} - \widetilde{x}^k\rangle + \gamma\langle\widehat{g}^k, u - \widetilde{x}^k\rangle. \quad (300)$$

Since

$$\langle\widetilde{x}^k - x^k, x^{k+1} - \widetilde{x}^k\rangle \ = \ \frac{1}{2}\|x^{k+1} - x^k\|^2 - \frac{1}{2}\|\widetilde{x}^k - x^k\|^2 - \frac{1}{2}\|x^{k+1} - \widetilde{x}^k\|^2,$$

$$\langle x^{k+1} - x^k, u - x^{k+1}\rangle \ = \ \frac{1}{2}\|x^k - u\|^2 - \frac{1}{2}\|x^{k+1} - x^k\|^2 - \frac{1}{2}\|x^{k+1} - u\|^2,$$

we can rewrite (300) as follows

$$\gamma\left(\langle F(\widetilde{x}^k), \widetilde{x}^k - u\rangle + \Psi(\widetilde{x}^k) - \Psi(u)\right) \ \leq \ \frac{1}{2}\|x^k - u\|^2 - \frac{1}{2}\|x^{k+1} - u\|^2 - \frac{1}{2}\|\widetilde{x}^k - x^k\|^2$$

$$- \frac{1}{2}\|x^{k+1} - \widetilde{x}^k\|^2 + \gamma\langle\widetilde{g}^k - \widehat{g}^k, x^{k+1} - \widetilde{x}^k\rangle$$

$$+ \gamma\langle\theta_k, \widetilde{x}^k - u\rangle. \quad (301)$$

Next, we upper-bound $\gamma\langle \widetilde{g}^k - \widehat{g}^k, x^{k+1} - \widetilde{x}^k\rangle$ using Young's inequality, stating that $\langle a, b\rangle \leq \frac{1}{2\eta}\|a\|^2 + \frac{\eta}{2}\|b\|^2$ for all $a, b \in \mathbb{R}^d$ and $\eta > 0$, and Jensen's inequality for the squared norm:

$$
\begin{aligned}
\gamma\langle \widetilde{g}^k - \widehat{g}^k, x^{k+1} - \widetilde{x}^k\rangle \;\leq\;& \gamma^2\|\widetilde{g}^k - \widehat{g}^k\|^2 + \frac{1}{4}\|x^{k+1} - \widetilde{x}^k\|^2 \\
=\;& \gamma^2\|F(x^k) - F(\widetilde{x}^k) - \omega_k + \theta_k\|^2 + \frac{1}{4}\|x^{k+1} - \widetilde{x}^k\|^2 \\
\leq\;& 3\gamma^2\|F(x^k) - F(\widetilde{x}^k)\|^2 + 3\gamma^2\|\omega_k\|^2 + 3\gamma^2\|\theta_k\|^2 + \frac{1}{4}\|x^{k+1} - \widetilde{x}^k\|^2 \\
\overset{(32)}{\leq}\;& 3\gamma^2 L^2\|x^k - \widetilde{x}^k\|^2 + 3\gamma^2\|\omega_k\|^2 + 3\gamma^2\|\theta_k\|^2 + \frac{1}{4}\|x^{k+1} - \widetilde{x}^k\|^2 \quad (302)
\end{aligned}
$$

Plugging (302) in (301), we derive for all $u \in \mathbb{R}^d$

$$
\begin{aligned}
\gamma\left(\langle F(\widetilde{x}^k), \widetilde{x}^k - u\rangle + \Psi(\widetilde{x}^k) - \Psi(u)\right) \;\leq\;& \frac{1}{2}\|x^k - u\|^2 - \frac{1}{2}\|x^{k+1} - u\|^2 \\
& - \frac{1}{2}\left(1 - 6\gamma^2 L^2\right)\|\widetilde{x}^k - x^k\|^2 - \frac{1}{4}\|x^{k+1} - \widetilde{x}^k\|^2 \\
& + 3\gamma^2\|\omega_k\|^2 + 3\gamma^2\|\theta_k\|^2 + \gamma\langle \theta_k, \widetilde{x}^k - u\rangle. \quad (303)
\end{aligned}
$$

We notice that the above inequality does not rely on monotonicity. Next, we apply monotonicity and get that for all $u \in B_{4n\sqrt{V}}(x^*)$:

$$
\begin{aligned}
\gamma\left(\langle F(u), \widetilde{x}^k - u\rangle + \Psi(\widetilde{x}^k) - \Psi(u)\right) \;\leq\;& \frac{1}{2}\|x^k - u\|^2 - \frac{1}{2}\|x^{k+1} - u\|^2 \\
& - \frac{1}{2}\left(1 - 6\gamma^2 L^2\right)\|\widetilde{x}^k - x^k\|^2 + \gamma\langle \theta^k, \widetilde{x}^k - x^k\rangle \\
& + 3\gamma^2\|\omega_k\|^2 + 3\gamma^2\|\theta_k\|^2 + \gamma\langle \theta_k, x^k - u\rangle \\
\leq\;& \frac{1}{2}\|x^k - u\|^2 - \frac{1}{2}\|x^{k+1} - u\|^2 \\
& - \frac{1}{2}\left(\frac{1}{2} - 6\gamma^2 L^2\right)\|\widetilde{x}^k - x^k\|^2 \\
& + 3\gamma^2\|\omega_k\|^2 + 4\gamma^2\|\theta_k\|^2 + \gamma\langle \theta_k, x^k - u\rangle,
\end{aligned}
$$

where in the last step we apply $\gamma\langle \theta^k, \widetilde{x}^k - x^k\rangle \leq \gamma^2\|\theta^k\|^2 + \frac{1}{4}\|\widetilde{x}^k - x^k\|^2$. Since $\gamma \leq 1/\sqrt{12}L$, we have

$$
\begin{aligned}
\gamma\left(\langle F(u), \widetilde{x}^k - u\rangle + \Psi(\widetilde{x}^k) - \Psi(u)\right) \;\leq\;& \frac{1}{2}\|x^k - u\|^2 - \frac{1}{2}\|x^{k+1} - u\|^2 \\
& + 3\gamma^2\|\omega_k\|^2 + 4\gamma^2\|\theta_k\|^2 + \gamma\langle \theta_k, x^k - u\rangle,
\end{aligned}
$$

Summing up the above inequalities for $k = 0, 1, \ldots, K$ and dividing both sides by $\gamma(K+1)$, we obtain

$$
\begin{aligned}
\frac{1}{K+1}\sum_{k=0}^{K+1}\left(\langle F(u), \widetilde{x}^k - u\rangle + \Psi(\widetilde{x}^k) - \Psi(u)\right) \;\leq\;& \frac{\|x^0 - u\|^2 - \|x^{K+1} - u\|^2}{2\gamma(K+1)} \\
& + \frac{\gamma}{K+1}\sum_{k=0}^{K+1}\left(3\|\omega_k\|^2 + 4\|\theta_k\|^2\right) \\
& + \frac{1}{K+1}\sum_{k=0}^{K+1}\langle \theta_k, x^k - u\rangle.
\end{aligned}
$$

Applying $\frac{1}{K+1}\sum_{i=1}^{n}\langle F(u), \widetilde{x}^k - u\rangle = \langle F(u), \widetilde{x}_{\text{avg}}^K - u\rangle$ and $\Psi(\widetilde{x}_{\text{avg}}^K) - \Psi(x^*) \leq \frac{1}{K+1}\sum_{i=1}^{n}\Psi(\widetilde{x}^k)$, we get the result. $\qquad\square$

Next, we proceed with the full statement of our main result for DProx-clipped-SEG-shift in the monotone case.

**Theorem H.1** (Case 2 from Theorem C.2). *Let Assumptions 1, 6 and 7 hold for $Q = B_{4n\sqrt{V}}(x^*)$, where $V \geq \|x^0 - x^*\|^2 + \frac{409600\gamma^2 \ln^2 \frac{48n(K+1)}{\beta}}{n^2} \sum_{i=1}^{n} \|F_i(x^*)\|^2$ and*

$$\gamma \leq \min\left\{ \frac{1}{1920L \ln \frac{48n(K+1)}{\beta}}, \frac{60^{\frac{2-\alpha}{\alpha}} \sqrt{V} n^{\frac{\alpha-1}{\alpha}}}{97200^{\frac{1}{\alpha}} (K+1)^{\frac{1}{\alpha}} \sigma \ln^{\frac{\alpha-1}{\alpha}} \frac{48n(K+1)}{\beta}} \right\}, \quad (304)$$

$$\lambda_k \equiv \lambda = \frac{n\sqrt{V}}{60\gamma \ln \frac{48n(K+1)}{\beta}}, \quad (305)$$

$$\nu = 0 \quad (306)$$

*for some $K \geq 1$ and $\beta \in (0,1]$. Then, after $K$ iterations of* DProx-clipped-SEG-shift*, the following inequality holds with probability at least $1 - \beta$:*

$$\mathrm{Gap}_{\sqrt{V}}(\widetilde{x}_{avg}^K) \leq \frac{9V}{2\gamma(K+1)} \quad and \quad \{x^k\}_{k=0}^{K+1} \subseteq B_{3\sqrt{V}}(x^*), \{\widetilde{x}^k\}_{k=0}^{K+1} \subseteq B_{4n\sqrt{V}}(x^*), \quad (307)$$

*where $\widetilde{x}_{avg}^K$ is defined in (297). In particular, when $\gamma$ equals the minimum from (304), then after $K$ iterations of* DProx-clipped-SEG-shift*, we have with probability at least $1 - \beta$*

$$\mathrm{Gap}_{\sqrt{V}}(\widetilde{x}_{avg}^K) = \mathcal{O}\left( \max\left\{ \frac{LV \ln \frac{nK}{\beta}}{K}, \frac{\sigma\sqrt{V} \ln^{\frac{\alpha-1}{\alpha}} \frac{nK}{\beta}}{n^{\frac{\alpha-1}{\alpha}} K^{\frac{\alpha-1}{\alpha}}} \right\} \right), \quad (308)$$

*i.e., to achieve $\mathrm{Gap}_{\sqrt{V}}(\widetilde{x}_{avg}^K) \leq \varepsilon$ with probability at least $1 - \beta$* DProx-clipped-SEG-shift *needs*

$$K = \mathcal{O}\left( \max\left\{ \frac{LV}{\varepsilon} \ln \frac{nLV}{\varepsilon\beta}, \frac{1}{n} \left( \frac{\sigma\sqrt{V}}{\varepsilon} \right)^{\frac{\alpha}{\alpha-1}} \ln \frac{\sigma\sqrt{V}}{\varepsilon\beta} \right\} \right) \quad (309)$$

*iterations/oracle calls per worker.*

*Proof.* The key idea behind the proof is similar to the one used in (Gorbunov et al., 2022a; Sadiev et al., 2023): we prove by induction that the iterates do not leave some ball and the sums decrease as $1/K+1$. To formulate the statement rigorously, we introduce probability event $E_k$ for each $k = 0, 1, \ldots, K+1$ as follows: inequalities

$$\underbrace{\max_{u \in B_{\sqrt{V}}(x^*)} \left\{ \|x^0 - u\|^2 + 2\gamma \sum_{l=0}^{t-1} \langle x^l - u, \theta_l \rangle + \gamma^2 \sum_{l=0}^{t-1} \left( 8\|\theta_l\|^2 + 6\|\omega_l\|^2 \right) \right\}}_{A_t} \leq 9V, \quad (310)$$

$$\left\| \gamma \sum_{l=0}^{t-1} \theta_l \right\| \leq \sqrt{V}, \quad (311)$$

$$\left\| \frac{\gamma}{n} \sum_{i=1}^{r-1} \theta_{i,t-1}^u \right\| \leq \frac{\sqrt{V}}{2}, \quad \left\| \frac{\gamma}{n} \sum_{i=1}^{r-1} \omega_{i,t-1}^u \right\| \leq \frac{\sqrt{V}}{2} \quad (312)$$

hold for $t = 0, 1, \ldots, k$ and $r = 1, 2, \ldots, n$ simultaneously, where

$$\theta_l = \theta_l^u + \theta_l^b, \quad \omega_l = \omega_l^u + \omega_l^b, \quad (313)$$

$$\theta_l^u \overset{\text{def}}{=} \frac{1}{n} \sum_{i=1}^{n} \theta_{i,l}^u, \quad \theta_l^b \overset{\text{def}}{=} \frac{1}{n} \sum_{i=1}^{n} \theta_{i,l}^b, \quad \omega_l^u \overset{\text{def}}{=} \frac{1}{n} \sum_{i=1}^{n} \omega_{i,l}^u, \quad \omega_l^b \overset{\text{def}}{=} \frac{1}{n} \sum_{i=1}^{n} \omega_{i,l}^b, \quad (314)$$

$$\theta_{i,l}^u \overset{\text{def}}{=} \mathbb{E}_{\xi_{2,i}^l} \left[ \widehat{g}_i^l \right] - \widehat{g}_i^l, \quad \theta_{i,l}^b \overset{\text{def}}{=} F_i(\widetilde{x}^l) - \mathbb{E}_{\xi_{2,i}^l} \left[ \widehat{g}_i^l \right] \quad \forall i \in [n], \quad (315)$$

$$\omega_{i,l}^u \overset{\text{def}}{=} \mathbb{E}_{\xi_{1,i}^l} \left[ \widetilde{g}_i^l \right] - \widetilde{g}_i^l, \quad \omega_{i,l}^b \overset{\text{def}}{=} F_i(x^l) - \mathbb{E}_{\xi_{1,i}^l} \left[ \widehat{g}_i^l \right] \quad \forall i \in [n]. \quad (316)$$

We will prove by induction that $\mathbb{P}\{E_k\} \geq 1 - k\beta/(K+1)$ for all $k = 0, 1, \ldots, K+1$. The base of induction follows immediately: for all $u \in B_{\sqrt{V}}(x^*)$ we have $\|x^0 - u\|^2 \leq 2\|x^0 - x^*\|^2 + 2\|x^* - u\|^2 \leq$

$4V < 9V$ and for $k = 0$ we have $\|\gamma \sum_{l=0}^{k-1} \theta_l\| = 0$, $\|\frac{\gamma}{n} \sum_{i=1}^{r-1} \theta_{i,k-1}^u\| = \|\frac{\gamma}{n} \sum_{i=1}^{r-1} \omega_{i,k-1}^u\| = 0$ since $\theta_{i,-1}^u = \omega_{i,-1}^u = 0$. Next, we assume that the statement holds for $k = T - 1 \leq K$, i.e., $\mathbb{P}\{E_{T-1}\} \geq 1 - (T-1)\beta/(K+1)$. Let us show that it also holds for $k = T$, i.e., $\mathbb{P}\{E_T\} \geq 1 - T\beta/(K+1)$.

To proceed, we need to show that $E_{T-1}$ implies $\|x^t - x^*\| \leq 3\sqrt{V}$ for all $t = 0, 1, \ldots, T$. We will use the induction argument as well. The base is already proven. Next, we assume that $\|x^t - x^*\| \leq 3\sqrt{V}$ for all $t = 0, 1, \ldots, t'$ for some $t' < T$. Then

$$\|F(x^*)\| = \sqrt{\|F(x^*)\|^2} \leq \sqrt{\sum_{i=1}^n \|F_i(x^*)\|^2} \leq \frac{n\sqrt{V}}{160\gamma \ln \frac{48n(K+1)}{\beta}} < \lambda \qquad (317)$$

and for $t = 0, 1, \ldots, t'$

$$\begin{aligned}
\|\widetilde{x}^t - x^*\| &= \|\operatorname{prox}_{\gamma\Psi}(x^t - \gamma\widetilde{g}^t) - \operatorname{prox}_{\gamma\Psi}(x^* - \gamma F(x^*))\| \\
&\leq \|x^t - x^* - \gamma(\widetilde{g}^t - F(x^*))\| \leq \|x^t - x^*\| + \gamma\|\widetilde{g}^t - F(x^*)\| \\
&\leq \|x^t - x^*\| + \gamma(\|\widetilde{g}^k\| + \|F(x^*)\|) \overset{(305),(317)}{\leq} 3\sqrt{V} + 2\gamma\lambda \leq 3\sqrt{V} + \frac{n\sqrt{V}}{30 \ln \frac{48(K+1)}{\beta}} \\
&\leq 4n\sqrt{V}. \qquad (318)
\end{aligned}$$

This means that $x^t, \widetilde{x}^t \in B_{4n\sqrt{V}}(x^*)$ for $t = 0, 1, \ldots, t'$ and we can apply Lemma H.1: $E_{T-1}$ implies

$$\begin{aligned}
\max_{B_{\sqrt{V}}(x^*)} &\left\{ 2\gamma(t'+1)\left(\langle F(u), \widetilde{x}_{\text{avg}}^{t'} - u\rangle + \Psi(\widetilde{x}_{\text{avg}}^{t'}) - \Psi(u)\right) + \|x^{t'+1} - u\|^2\right\} \\
&\leq \max_{B_{\sqrt{V}}(x^*)} \left\{ \|x^0 - u\|^2 + 2\gamma \sum_{l=0}^{t-1} \langle x^l - u, \theta_l\rangle\right\} \\
&\quad + \gamma^2 \sum_{l=0}^{t-1} \left(8\|\theta_l\|^2 + 6\|\omega_l\|^2\right) \\
&\overset{(310)}{\leq} 9V
\end{aligned}$$

that gives

$$\begin{aligned}
\|x^{t'+1} - x^*\|^2 &\leq \max_{B_{\sqrt{V}}(x^*)} \left\{ 2\gamma(t'+1)\left(\langle F(u), \widetilde{x}_{\text{avg}}^{t'} - u\rangle + \Psi(\widetilde{x}_{\text{avg}}^{t'}) - \Psi(u)\right) + \|x^{t'+1} - u\|^2\right\} \\
&\leq 9V.
\end{aligned}$$

That is, we showed that $E_{T-1}$ implies $\|x^t - x^*\| \leq 3\sqrt{V}$, $\|\widetilde{x}^t - x^*\| \leq 4n\sqrt{V}$ and

$$\max_{B_{\sqrt{V}}(x^*)} \left\{ 2\gamma(t+1)\left(\langle F(u), \widetilde{x}_{\text{avg}}^t - u\rangle + \Psi(\widetilde{x}_{\text{avg}}^t) - \Psi(u)\right) + \|x^{t+1} - u\|^2\right\} \leq 9V \qquad (319)$$

for all $t = 0, 1, \ldots, T$. Before we proceed, we introduce a new notation:

$$\eta_t = \begin{cases} x^t - x^*, & \text{if } \|x^t - x^*\| \leq 3\sqrt{V}, \\ 0, & \text{otherwise}, \end{cases}$$

for all $t = 0, 1, \ldots, T$. Random vectors $\{\eta_t\}_{t=0}^T$ are bounded almost surely:

$$\|\eta_t\| \leq 3\sqrt{V} \qquad (320)$$

for all $t = 0, 1, \ldots, T$. In addition, $\eta_t = x^t - x^*$ follows from $E_{T-1}$ for all $t = 0, 1, \ldots, T$ and, thus, $E_{T-1}$ implies

$$
\begin{aligned}
A_T \;&\overset{(310)}{=}\; \max_{u \in B_{\sqrt{V}}(x^*)} \left\{ \|x^0 - u\|^2 + 2\gamma \sum_{l=0}^{T-1} \langle x^* - u, \theta_l \rangle \right\} + 2\gamma \sum_{l=0}^{T-1} \langle x^l - x^*, \theta_l \rangle \\
&\qquad + \gamma^2 \sum_{l=0}^{T-1} \left( 8\|\theta_l\|^2 + 6\|\omega_l\|^2 \right) \\
&\leq\; 4V + 2\gamma \max_{u \in B_{\sqrt{V}}(x^*)} \left\{ \left\langle x^* - u, \sum_{l=0}^{T-1} \theta_l \right\rangle \right\} + 2\gamma \sum_{l=0}^{T-1} \langle \eta_l, \theta_l \rangle + \gamma^2 \sum_{l=0}^{T-1} \left( 8\|\theta_l\|^2 + 6\|\omega_l\|^2 \right) \\
&=\; 4V + 2\gamma\sqrt{V} \left\| \sum_{l=0}^{T-1} \theta_l \right\| + 2\gamma \sum_{l=0}^{T-1} \langle \eta_l, \theta_l \rangle + \gamma^2 \sum_{l=0}^{T-1} \left( 8\|\theta_l\|^2 + 6\|\omega_l\|^2 \right).
\end{aligned}
$$

Using the notation from (313)-(316), we can rewrite $\|\theta_l\|^2$ as

$$
\begin{aligned}
\|\theta_l\|^2 \;&\leq\; 2\|\theta_l^u\|^2 + 2\|\theta_l^b\|^2 = \frac{2}{n^2} \left\| \sum_{i=1}^n \theta_{i,l}^u \right\|^2 + 2\|\theta_l^b\|^2 \\
&=\; \frac{2}{n^2} \sum_{i=1}^n \|\theta_{i,l}^u\|^2 + \frac{4}{n^2} \sum_{j=2}^n \left\langle \sum_{i=1}^{j-1} \theta_{i,l}^u, \theta_{j,l}^u \right\rangle + 2\|\theta_l^b\|^2
\end{aligned} \tag{321}
$$

and, similarly, it holds for $\|\omega_l\|^2$. Putting all together, we obtain that $E_{T-1}$ implies

$$
\begin{aligned}
A_T \;\leq\;& 4V + 2\gamma\sqrt{V} \left\| \sum_{l=0}^{T-1} \theta_l \right\| + \underbrace{\frac{2\gamma}{n} \sum_{l=0}^{T-1} \sum_{i=1}^n \langle \eta_l, \theta_{i,l}^u \rangle}_{①} + \underbrace{2\gamma \sum_{l=0}^{T-1} \langle \eta_l, \theta_l^b \rangle}_{②} \\
&+ \underbrace{\frac{2\gamma^2}{n^2} \sum_{l=0}^{T-1} \sum_{i=1}^n \left( 8\mathbb{E}_{\xi_{2,i}^l}\left[ \|\theta_{i,l}^u\|^2 \right] + 6\mathbb{E}_{\xi_{1,i}^l}\left[ \|\omega_{i,l}^u\|^2 \right] \right)}_{③} \\
&+ \underbrace{\frac{2\gamma^2}{n^2} \sum_{l=0}^{T-1} \sum_{i=1}^n \left( 8\|\theta_{i,l}^u\|^2 + 6\|\omega_{i,l}^u\|^2 - 8\mathbb{E}_{\xi_{2,i}^l}\left[ \|\theta_{i,l}^u\|^2 \right] - 6\mathbb{E}_{\xi_{1,i}^l}\left[ \|\omega_{i,l}^u\|^2 \right] \right)}_{④} \\
&+ \underbrace{2\gamma^2 \sum_{l=0}^{T-1} \left( 8\|\theta_l^b\|^2 + 6\|\omega_l^b\|^2 \right)}_{⑤} + \underbrace{\frac{32\gamma^2}{n^2} \sum_{l=0}^{T-1} \sum_{j=2}^n \left\langle \sum_{i=1}^{j-1} \theta_{i,l}^u, \theta_{j,l}^u \right\rangle}_{⑥} \\
&+ \underbrace{\frac{24\gamma^2}{n^2} \sum_{l=0}^{T-1} \sum_{j=2}^n \left\langle \sum_{i=1}^{j-1} \omega_{i,l}^u, \omega_{j,l}^u \right\rangle}_{⑦}.
\end{aligned} \tag{322}
$$

To finish the proof, it remains to estimate $2\gamma\sqrt{V} \left\| \sum_{l=0}^{T-1} \theta_l \right\|$, ①, ②, ③, ④, ⑤, ⑥, ⑦ with high probability. More precisely, the goal is to prove that $2\gamma\sqrt{V} \left\| \sum_{l=0}^{T-1} \theta_l \right\| + ① + ② + ③ + ④ + ⑤ + ⑥ + ⑦ \leq 5V$ with high probability. Before we proceed, we need to derive several useful inequalities related to $\theta_{i,l}^u, \omega_{i,l}^u, \theta_l^b, \omega_l^b$. First of all, we have

$$
\|\theta_{i,l}^u\| \leq 2\lambda, \quad \|\omega_{i,l}^u\| \leq 2\lambda \tag{323}
$$

by definition of the clipping operator. Next, probability event $E_{T-1}$ implies

$$
\begin{aligned}
\|F_i(x^l)\| &\leq \|F_i(x^l) - F_i(x^*)\| + \|F_i(x^*)\| \overset{(32)}{\leq} L\|x^l - x^*\| + \sqrt{\sum_{i=1}^n \|F_i(x^*)\|^2} \\
&\leq 3L\sqrt{V} + \frac{n\sqrt{V}}{160\gamma \ln \frac{48n(K+1)}{\beta}} \overset{(304)}{\leq} \frac{n\sqrt{V}}{120\gamma \ln \frac{48n(K+1)}{\beta}} \overset{(305)}{=} \frac{\lambda}{2},
\end{aligned}
$$

$$
\begin{aligned}
\|F_i(\widetilde{x}^l)\| &\leq \|F_i(\widetilde{x}^l) - F_i(x^*)\| + \|F_i(x^*)\| \overset{(32)}{\leq} L\|\widetilde{x}^l - x^*\| + \sqrt{\sum_{i=1}^n \|F_i(x^*)\|^2} \\
&\leq 4Ln\sqrt{V} + \frac{n\sqrt{V}}{160\gamma \ln \frac{48n(K+1)}{\beta}} \overset{(304)}{\leq} \frac{n\sqrt{V}}{120\gamma \ln \frac{48n(K+1)}{\beta}} \overset{(305)}{=} \frac{\lambda}{2}
\end{aligned}
$$

for $l = 0, 1, \ldots, T - 1$ and $i \in [n]$. Therefore, Lemma B.2 and $E_{T-1}$ imply

$$
\left\|\theta_l^b\right\| \leq \frac{1}{n}\sum_{i=1}^n \|\theta_{i,l}^b\| \leq \frac{2^\alpha \sigma^\alpha}{\lambda^{\alpha-1}}, \quad \left\|\omega_l^b\right\| \leq \frac{1}{n}\sum_{i=1}^n \|\omega_{i,l}^b\| \leq \frac{2^\alpha \sigma^\alpha}{\lambda^{\alpha-1}}, \tag{324}
$$

$$
\mathbb{E}_{\xi_{2,i}^l}\left[\left\|\theta_{i,l}^u\right\|^2\right] \leq 18\lambda^{2-\alpha}\sigma^\alpha, \quad \mathbb{E}_{\xi_{1,i}^l}\left[\left\|\omega_{i,l}^u\right\|^2\right] \leq 18\lambda^{2-\alpha}\sigma^\alpha, \tag{325}
$$

for all $l = 0, 1, \ldots, T - 1$ and $i \in [n]$.

**Upper bound for ①.** To estimate this sum, we will use Bernstein's inequality. The summands have conditional expectations equal to zero:

$$
\mathbb{E}_{\xi_{2,i}^l}\left[\frac{2\gamma}{n}\langle \eta_l, \theta_{i,l}^u\rangle\right] = \frac{2\gamma}{n}\left\langle \eta_l, \mathbb{E}_{\xi_{2,i}^l}[\theta_{i,l}^u]\right\rangle = 0.
$$

Moreover, for all $l = 0, \ldots, T - 1$ random vectors $\{\theta_{i,l}^u\}_{i=1}^n$ are independent. Thus, sequence $\left\{\frac{2\gamma}{n}\langle \eta_l, \theta_{i,l}^u\rangle\right\}_{l,i=0,1}^{T-1,n}$ is a martingale difference sequence. Next, the summands are bounded:

$$
\left|\frac{2\gamma}{n}\langle \eta_l, \theta_{i,l}^u\rangle\right| \leq \frac{2\gamma}{n}\|\eta_l\| \cdot \|\theta_{i,l}^u\| \overset{(320),(323)}{\leq} \frac{12\gamma}{n}\sqrt{V}\lambda \overset{(305)}{\leq} \frac{3V}{10\ln\frac{48n(K+1)}{\beta}} \overset{\text{def}}{=} c. \tag{326}
$$

Finally, conditional variances $\sigma_{i,l}^2 \overset{\text{def}}{=} \mathbb{E}_{\xi_{2,i}^l}\left[\frac{4\gamma^2}{n^2}\langle \eta_l, \theta_{i,l}^u\rangle^2\right]$ of the summands are bounded:

$$
\sigma_{i,l}^2 \leq \mathbb{E}_{\xi_{2,i}^l}\left[\frac{4\gamma^2}{n^2}\|\eta_l\|^2 \cdot \|\theta_{i,l}^u\|^2\right] \overset{(320)}{\leq} \frac{36\gamma^2 V}{n^2}\mathbb{E}_{\xi_{2,i}^l}\left[\|\theta_{i,l}^u\|^2\right]. \tag{327}
$$

Applying Bernstein's inequality (Lemma B.1) with $X_{i,l} = \frac{2\gamma}{n}\langle \eta_l, \theta_{i,l}^u\rangle$, constant $c$ defined in (326), $b = \frac{3V}{10}$, $G = \frac{3V^2}{200\ln\frac{48n(K+1)}{\beta}}$, we get

$$
\mathbb{P}\left\{|①| > \frac{3V}{10} \text{ and } \sum_{l=0}^{T-1}\sum_{i=1}^n \sigma_{i,l}^2 \leq \frac{3V^2}{200\ln\frac{48n(K+1)}{\beta}}\right\} \leq 2\exp\left(-\frac{b^2}{2G + 2cb/3}\right) = \frac{\beta}{24n(K+1)}.
$$

The above is equivalent to

$$
\mathbb{P}\{E_①\} \geq 1 - \frac{\beta}{24n(K+1)}, \quad \text{for } E_① = \left\{\text{either } \sum_{l=0}^{T-1}\sum_{i=1}^n \sigma_{i,l}^2 > \frac{3V^2}{200\ln\frac{48n(K+1)}{\beta}} \text{ or } |①| \leq \frac{3V}{10}\right\}. \tag{328}
$$

Moreover, $E_{T-1}$ implies

$$
\begin{aligned}
\sum_{l=0}^{T-1}\sum_{i=1}^n \sigma_{i,l}^2 &\overset{(327)}{\leq} \frac{36\gamma^2 V}{n^2}\sum_{l=0}^{T-1}\sum_{i=1}^n \mathbb{E}_{\xi_{2,i}^l}\left[\|\theta_{i,l}^u\|^2\right] \overset{(325),T\leq K+1}{\leq} \frac{648(K+1)\gamma^2 V\lambda^{2-\alpha}\sigma^\alpha}{n} \\
&\overset{(305)}{\leq} \frac{648(K+1)\gamma^\alpha \sigma^\alpha V^{2-\frac{\alpha}{2}}}{60^{2-\alpha}n^{\alpha-1}\ln^{2-\alpha}\frac{48n(K+1)}{\beta}} \overset{(304)}{\leq} \frac{3V^2}{200\ln\frac{48n(K+1)}{\beta}}.
\end{aligned} \tag{329}
$$

**Upper bound for ②.** Probability event $E_{T-1}$ implies

$$② \quad \leq \quad 2\gamma \sum_{l=0}^{T-1} \|\eta_l\| \cdot \|\theta_l^b\| \overset{(320),(324),T\leq K+1}{\leq} \frac{6 \cdot 2^\alpha (K+1)\gamma\sqrt{V}\sigma^\alpha}{\lambda^{\alpha-1}}$$

$$\overset{(305)}{=} \quad \frac{6 \cdot 2^\alpha \cdot 60^{\alpha-1}(K+1)\gamma^\alpha\sigma^\alpha \ln^{\alpha-1}\frac{48n(K+1)}{\beta}}{n^{\alpha-1}V^{\frac{\alpha}{2}-1}} \overset{(304)}{\leq} \frac{3V}{100}. \tag{330}$$

**Upper bound for ③.** Probability event $E_{T-1}$ implies

$$\frac{16\gamma^2}{n^2} \sum_{l=0}^{T-1}\sum_{i=1}^{n} \mathbb{E}_{\xi_{2,i}^l}[\|\theta_{i,l}^u\|^2] \overset{(325),T\leq K+1}{\leq} \frac{288\gamma^2(K+1)\lambda^{2-\alpha}\sigma^\alpha}{n} \overset{(305)}{=} \frac{288\gamma^\alpha(K+1)\sigma^\alpha V^{1-\frac{\alpha}{2}}}{60^{2-\alpha}n^{\alpha-1}\ln^{2-\alpha}\frac{48n(K+1)}{\beta}}$$

$$\overset{(304)}{\leq} \quad \frac{3}{100}V, \tag{331}$$

$$\frac{12\gamma^2}{n^2} \sum_{l=0}^{T-1}\sum_{i=1}^{n} \mathbb{E}_{\xi_{1,i}^l}[\|\omega_{i,l}^u\|^2] \overset{(325),T\leq K+1}{\leq} \frac{216\gamma^2(K+1)\lambda^{2-\alpha}\sigma^\alpha}{n} \overset{(305)}{=} \frac{216\gamma^\alpha(K+1)\sigma^\alpha V^{1-\frac{\alpha}{2}}}{60^{2-\alpha}n^{\alpha-1}\ln^{2-\alpha}\frac{48n(K+1)}{\beta}}$$

$$\overset{(304)}{\leq} \quad \frac{1}{50}V, \tag{332}$$

$$③ \quad \overset{(331),(332)}{\leq} \quad \frac{1}{20}V. \tag{333}$$

**Upper bound for ④.** To estimate this sum, we will use Bernstein's inequality. The summands have conditional expectations equal to zero:

$$\frac{2\gamma^2}{n^2}\mathbb{E}_{\xi_{1,i}^l,\xi_{2,i}^l}\left[8\|\theta_{i,l}^u\|^2 + 6\|\omega_{i,l}^u\|^2 - 8\mathbb{E}_{\xi_{2,i}^l}\left[\|\theta_{i,l}^u\|^2\right] - 6\mathbb{E}_{\xi_{1,i}^l}\left[\|\omega_{i,l}^u\|^2\right]\right] = 0.$$

Moreover, for all $l = 0, \ldots, T-1$ random vectors $\{\theta_{i,l}^u\}_{i=1}^n$, $\{\omega_{i,l}^u\}_{i=1}^n$ are independent. Thus, sequence $\left\{\frac{2\gamma^2}{n^2}\left(8\|\theta_{i,l}^u\|^2 + 6\|\omega_{i,l}^u\|^2 - 8\mathbb{E}_{\xi_{2,i}^l}\left[\|\theta_{i,l}^u\|^2\right] - 6\mathbb{E}_{\xi_{1,i}^l}\left[\|\omega_{i,l}^u\|^2\right]\right)\right\}_{l,i=0,1}^{T-1,n}$ is a martingale difference sequence. Next, the summands are bounded:

$$\frac{2\gamma^2}{n^2}\left|8\|\theta_{i,l}^u\|^2 + 6\|\omega_{i,l}^u\|^2 - 8\mathbb{E}_{\xi_{2,i}^l}\left[\|\theta_{i,l}^u\|^2\right] - 6\mathbb{E}_{\xi_{1,i}^l}\left[\|\omega_{i,l}^u\|^2\right]\right|$$

$$\leq \frac{16\gamma^2}{n^2}\left(\|\theta_{i,l}^u\|^2 + \mathbb{E}_{\xi_{2,i}^l}\left[\|\theta_{i,l}^u\|^2\right]\right) + \frac{12\gamma^2}{n^2}\left(\|\omega_{i,l}^u\|^2 + \mathbb{E}_{\xi_{1,i}^l}\left[\|\omega_{i,l}^u\|^2\right]\right)$$

$$\overset{(323)}{\leq} \frac{224\gamma^2\lambda^2}{n^2}$$

$$\overset{(305)}{\leq} \frac{V}{6\ln\frac{48n(K+1)}{\beta}} \overset{\text{def}}{=} c. \tag{334}$$

Finally, conditional variances

$$\widetilde{\sigma}_{i,l}^2 \overset{\text{def}}{=} \frac{4\gamma^4}{n^4}\mathbb{E}_{\xi_{1,i}^l,\xi_{2,i}^l}\left[\left|8\|\theta_{i,l}^u\|^2 + 6\|\omega_{i,l}^u\|^2 - 8\mathbb{E}_{\xi_{2,i}^l}\left[\|\theta_{i,l}^u\|^2\right] - 6\mathbb{E}_{\xi_{1,i}^l}\left[\|\omega_{i,l}^u\|^2\right]\right|^2\right]$$

of the summands are bounded:

$$\widetilde{\sigma}_{i,l}^2 \overset{(334)}{\leq} \frac{\gamma^2 V}{3n^2\ln\frac{48n(K+1)}{\beta}}\mathbb{E}_{\xi_{1,i}^l,\xi_{2,i}^l}\left[\left|8\|\theta_{i,l}^u\|^2 + 6\|\omega_{i,l}^u\|^2 - 8\mathbb{E}_{\xi_{2,i}^l}\left[\|\theta_{i,l}^u\|^2\right] - 6\mathbb{E}_{\xi_{1,i}^l}\left[\|\omega_{i,l}^u\|^2\right]\right|\right]$$

$$\leq \frac{4\gamma^2 V}{3n^2\ln\frac{48n(K+1)}{\beta}}\mathbb{E}_{\xi_{1,i}^l,\xi_{2,i}^l}\left[4\|\theta_{i,l}^u\|^2 + 3\|\omega_{i,l}^u\|^2\right]. \tag{335}$$

Applying Bernstein's inequality (Lemma B.1) with $X_{i,l} = \frac{2\gamma^2}{n^2}\left(8\|\theta_{i,l}^u\|^2 + 6\|\omega_{i,l}^u\|^2 - 8\mathbb{E}_{\xi_{2,i}^l}\left[\|\theta_{i,l}^u\|^2\right] - 6\mathbb{E}_{\xi_{1,i}^l}\left[\|\omega_{i,l}^u\|^2\right]\right)$, constant $c$ defined in (334),

$b = \frac{V}{6}$, $G = \frac{V^2}{216 \ln \frac{48n(K+1)}{\beta}}$, we get

$$\mathbb{P}\left\{ |④| > \frac{V}{6} \text{ and } \sum_{l=0}^{T-1}\sum_{i=1}^{n} \widetilde{\sigma}_{i,l}^2 \leq \frac{V^2}{216 \ln \frac{48n(K+1)}{\beta}} \right\} \leq 2\exp\left( -\frac{b^2}{2G + {}^{2cb}/_3} \right) = \frac{\beta}{24n(K+1)}.$$

The above is equivalent to

$$\mathbb{P}\{E_④\} \geq 1 - \frac{\beta}{24n(K+1)}, \text{ for } E_④ = \left\{ \text{either } \sum_{l=0}^{T-1}\sum_{i=1}^{n} \widetilde{\sigma}_{i,l}^2 > \frac{V^2}{216 \ln \frac{48n(K+1)}{\beta}} \text{ or } |④| \leq \frac{V}{6} \right\}. \tag{336}$$

Moreover, $E_{T-1}$ implies

$$\sum_{l=0}^{T-1}\sum_{i=1}^{n} \widetilde{\sigma}_{i,l}^2 \overset{(335)}{\leq} \frac{4\gamma^2 V}{3n^2 \ln \frac{48n(K+1)}{\beta}} \sum_{l=0}^{T-1}\sum_{i=1}^{n} \mathbb{E}_{\xi_{1,i}^l,\xi_{2,i}^l}\left[ 4\|\theta_{i,l}^u\|^2 + 3\|\omega_{i,l}^u\|^2 \right]$$

$$\overset{(325),T\leq K+1}{\leq} \frac{168(K+1)\gamma^2 V \lambda^{2-\alpha}\sigma^\alpha}{n \ln \frac{48n(K+1)}{\beta}}$$

$$\overset{(305)}{\leq} \frac{168(K+1)\gamma^\alpha V^{2-\frac{\alpha}{2}}\sigma^\alpha}{60^{2-\alpha} n^{\alpha-1}\ln^{3-\alpha}\frac{48n(K+1)}{\beta}} \overset{(304)}{\leq} \frac{V^2}{216 \ln \frac{48n(K+1)}{\beta}}. \tag{337}$$

**Upper bound for ⑤.** Probability event $E_{T-1}$ implies

$$⑤ = 2\gamma^2 \sum_{l=0}^{T-1}\left( 8\|\theta_l^b\|^2 + 6\|\omega_l^b\|^2 \right) \overset{(324),T\leq K+1}{\leq} \frac{28 \cdot 2^{2\alpha}\gamma^2\sigma^{2\alpha}(K+1)}{\lambda^{2\alpha-2}}$$

$$\overset{(305)}{=} \frac{28 \cdot 2^{2\alpha} \cdot 60^{2\alpha-2}\gamma^{2\alpha}\sigma^{2\alpha}(K+1)\ln^{2\alpha-2}\frac{48n(K+1)}{\beta}}{n^{2\alpha-2}V^{\alpha-1}} \overset{(304)}{\leq} \frac{V}{6}. \tag{338}$$

**Upper bounds for ⑥ and ⑦.** These sums require more refined analysis. We introduce new vectors:

$$\zeta_j^l = \begin{cases} \frac{\gamma}{n}\sum_{i=1}^{j-1}\theta_{i,l}^u, & \text{if } \left\|\frac{\gamma}{n}\sum_{i=1}^{j-1}\theta_{i,l}^u\right\| \leq \frac{\sqrt{V}}{2}, \\ 0, & \text{otherwise}, \end{cases} \quad \delta_j^l = \begin{cases} \frac{\gamma}{n}\sum_{i=1}^{j-1}\omega_{i,l}^u, & \text{if } \left\|\frac{\gamma}{n}\sum_{i=1}^{j-1}\omega_{i,l}^u\right\| \leq \frac{\sqrt{V}}{2}, \\ 0, & \text{otherwise}, \end{cases} \tag{339}$$

for all $j \in [n]$ and $l = 0, \ldots, T-1$. Then, by definition

$$\|\zeta_j^l\| \leq \frac{\sqrt{V}}{2}, \quad \|\delta_j^l\| \leq \frac{\sqrt{V}}{2} \tag{340}$$

and

$$⑥ = \underbrace{\frac{32\gamma}{n}\sum_{l=0}^{T-1}\sum_{j=2}^{n}\langle \zeta_j^l, \theta_{j,l}^u\rangle}_{⑥'} + \frac{32\gamma}{n}\sum_{l=0}^{T-1}\sum_{j=2}^{n}\left\langle \frac{\gamma}{n}\sum_{i=1}^{j-1}\theta_{i,l}^u - \zeta_j^l, \theta_{j,l}^u\right\rangle, \tag{341}$$

$$⑦ = \underbrace{\frac{24\gamma}{n}\sum_{l=0}^{T-1}\sum_{j=2}^{n}\langle \delta_j^l, \omega_{j,l}^u\rangle}_{⑦'} + \frac{24\gamma}{n}\sum_{l=0}^{T-1}\sum_{j=2}^{n}\left\langle \frac{\gamma}{n}\sum_{i=1}^{j-1}\omega_{i,l}^u - \delta_j^l, \omega_{j,l}^u\right\rangle. \tag{342}$$

We also note here that $E_{T-1}$ implies

$$\frac{32\gamma}{n}\sum_{l=0}^{T-1}\sum_{j=2}^{n}\left\langle \frac{\gamma}{n}\sum_{i=1}^{j-1}\theta_{i,l}^u - \zeta_j^l, \theta_{j,l}^u\right\rangle = \frac{32\gamma}{n}\sum_{j=2}^{n}\left\langle \frac{\gamma}{n}\sum_{i=1}^{j-1}\theta_{i,T-1}^u - \zeta_j^{T-1}, \theta_{j,T-1}^u\right\rangle, \tag{343}$$

$$\frac{24\gamma}{n}\sum_{l=0}^{T-1}\sum_{j=2}^{n}\left\langle \frac{\gamma}{n}\sum_{i=1}^{j-1}\omega_{i,l}^u - \delta_j^l, \omega_{j,l}^u\right\rangle = \frac{24\gamma}{n}\sum_{j=2}^{n}\left\langle \frac{\gamma}{n}\sum_{i=1}^{j-1}\omega_{i,T-1}^u - \delta_j^{T-1}, \omega_{j,T-1}^u\right\rangle. \tag{344}$$

**Upper bound for ⑥′.** To estimate this sum, we will use Bernstein's inequality. The summands have conditional expectations equal to zero:

$$\mathbb{E}_{\xi_{2,j}^l}\left[\frac{32\gamma}{n}\left\langle\zeta_j^l,\theta_{j,l}^u\right\rangle\right] = \frac{32\gamma}{n}\left\langle\zeta_j^l,\mathbb{E}_{\xi_{2,j}^l}[\theta_{j,l}^u]\right\rangle = 0.$$

Moreover, for all $l = 0,\ldots,T-1$ random vectors $\{\theta_{i,l}^u\}_{i=1}^n$ are independent. Thus, sequence $\left\{\frac{32\gamma}{n}\left\langle\zeta_j^l,\theta_{j,l}^u\right\rangle\right\}_{l,j=0,2}^{T-1,n}$ is a martingale difference sequence. Next, the summands are bounded:

$$\left|\frac{32\gamma}{n}\left\langle\zeta_j^l,\theta_{j,l}^u\right\rangle\right| \leq \frac{32\gamma}{n}\left\|\zeta_j^l\right\|\cdot\left\|\theta_{j,l}^u\right\| \overset{(340),(323)}{\leq} \frac{32\gamma}{n}\cdot\frac{\sqrt{V}}{2}\cdot 2\lambda \overset{(305)}{\leq} \frac{4V}{5\ln\frac{48n(K+1)}{\beta}} \overset{\text{def}}{=} c. \quad (345)$$

Finally, conditional variances $\widehat{\sigma}_{j,l}^2 \overset{\text{def}}{=} \mathbb{E}_{\xi_{2,j}^l}\left[\frac{1024\gamma^2}{n^2}\langle\zeta_j^l,\theta_{j,l}^u\rangle^2\right]$ of the summands are bounded:

$$\widehat{\sigma}_{j,l}^2 \leq \mathbb{E}_{\xi_{2,j}^l}\left[\frac{1024\gamma^2}{n^2}\|\zeta_j^l\|^2\cdot\|\theta_{j,l}^u\|^2\right] \overset{(340)}{\leq} \frac{256\gamma^2 V}{n^2}\mathbb{E}_{\xi_{2,j}^l}\left[\|\theta_{j,l}^u\|^2\right]. \quad (346)$$

Applying Bernstein's inequality (Lemma B.1) with $X_{i,l} = \frac{32\gamma}{n}\left\langle\zeta_j^l,\theta_{j,l}^u\right\rangle$, constant $c$ defined in (345), $b = \frac{4V}{5}$, $G = \frac{8V^2}{75\ln\frac{48n(K+1)}{\beta}}$, we get

$$\mathbb{P}\left\{|⑥'| > \frac{4V}{5} \text{ and } \sum_{l=0}^{T-1}\sum_{j=2}^n\widehat{\sigma}_{i,l}^2 \leq \frac{8V^2}{75\ln\frac{48n(K+1)}{\beta}}\right\} \leq 2\exp\left(-\frac{b^2}{2G+2cb/3}\right) = \frac{\beta}{24n(K+1)}.$$

The above is equivalent to

$$\mathbb{P}\{E_{⑥'}\} \geq 1 - \frac{\beta}{24n(K+1)}, \quad \text{for } E_{⑥'} = \left\{\text{either } \sum_{l=0}^{T-1}\sum_{j=2}^n\widehat{\sigma}_{i,l}^2 > \frac{8V^2}{75\ln\frac{48n(K+1)}{\beta}} \text{ or } |⑥'| \leq \frac{4V}{5}\right\}. \quad (347)$$

Moreover, $E_{T-1}$ implies

$$\sum_{l=0}^{T-1}\sum_{j=2}^n\widehat{\sigma}_{j,l}^2 \overset{(346)}{\leq} \frac{256\gamma^2 V}{n^2}\sum_{l=0}^{T-1}\sum_{j=2}^n\mathbb{E}_{\xi_{2,j}^l}\left[\|\theta_{j,l}^u\|^2\right] \overset{(325),T\leq K+1}{\leq} \frac{4608(K+1)\gamma^2 V\lambda^{2-\alpha}\sigma^\alpha}{n}$$
$$\overset{(305)}{\leq} \frac{4608(K+1)\gamma^\alpha\sigma^\alpha V^{2-\frac{\alpha}{2}}}{40^{2-\alpha}n^{\alpha-1}\ln^{2-\alpha}\frac{48n(K+1)}{\beta}} \overset{(304)}{\leq} \frac{8V^2}{75\ln\frac{48n(K+1)}{\beta}}. \quad (348)$$

**Upper bound for ⑦′.** To estimate this sum, we will use Bernstein's inequality. The summands have conditional expectations equal to zero:

$$\mathbb{E}_{\xi_{1,j}^l}\left[\frac{24\gamma}{n}\left\langle\delta_j^l,\omega_{j,l}^u\right\rangle\right] = \frac{24\gamma}{n}\left\langle\delta_j^l,\mathbb{E}_{\xi_{1,j}^l}[\omega_{j,l}^u]\right\rangle = 0.$$

Moreover, for all $l = 0,\ldots,T-1$ random vectors $\{\omega_{i,l}^u\}_{i=1}^n$ are independent. Thus, sequence $\left\{\frac{24\gamma}{n}\left\langle\delta_j^l,\omega_{j,l}^u\right\rangle\right\}_{l,j=0,2}^{T-1,n}$ is a martingale difference sequence. Next, the summands are bounded:

$$\left|\frac{24\gamma}{n}\left\langle\delta_j^l,\omega_{j,l}^u\right\rangle\right| \leq \frac{24\gamma}{n}\left\|\delta_j^l\right\|\cdot\left\|\omega_{j,l}^u\right\| \overset{(340),(323)}{\leq} \frac{24\gamma}{n}\cdot\frac{\sqrt{V}}{2}\cdot 2\lambda \overset{(305)}{\leq} \frac{3V}{5\ln\frac{48n(K+1)}{\beta}} \overset{\text{def}}{=} c. \quad (349)$$

Finally, conditional variances $(\sigma_{j,l}')^2 \overset{\text{def}}{=} \mathbb{E}_{\xi_{1,j}^l}\left[\frac{576\gamma^2}{n^2}\langle\delta_j^l,\omega_{j,l}^u\rangle^2\right]$ of the summands are bounded:

$$(\sigma_{j,l}')^2 \leq \mathbb{E}_{\xi_{1,j}^l}\left[\frac{576\gamma^2}{n^2}\|\delta_j^l\|^2\cdot\|\omega_{j,l}^u\|^2\right] \overset{(340)}{\leq} \frac{144\gamma^2 V}{n^2}\mathbb{E}_{\xi_{1,j}^l}\left[\|\omega_{j,l}^u\|^2\right]. \quad (350)$$

Applying Bernstein's inequality (Lemma B.1) with $X_{i,l} = \frac{24\gamma}{n}\left\langle \delta_j^l, \omega_{j,l}^u \right\rangle$, constant $c$ defined in (349), $b = \frac{3V}{5}$, $G = \frac{3V^2}{50\ln\frac{48n(K+1)}{\beta}}$, we get

$$\mathbb{P}\left\{ |\mathcal{D}'| > \frac{3V}{5} \text{ and } \sum_{l=0}^{T-1}\sum_{j=2}^{n}(\sigma'_{i,l})^2 \leq \frac{3V^2}{50\ln\frac{48n(K+1)}{\beta}} \right\} \leq 2\exp\left( -\frac{b^2}{2G + \frac{2cb}{3}} \right) = \frac{\beta}{24n(K+1)}.$$

The above is equivalent to

$$\mathbb{P}\{E_{\mathcal{D}'}\} \geq 1 - \frac{\beta}{24n(K+1)}, \quad \text{for } E_{\mathcal{D}'} = \left\{ \text{either } \sum_{l=0}^{T-1}\sum_{j=2}^{n}(\sigma'_{i,l})^2 > \frac{3V^2}{50\ln\frac{48n(K+1)}{\beta}} \text{ or } |\mathcal{D}'| \leq \frac{3V}{5} \right\}.$$

$$(351)$$

Moreover, $E_{T-1}$ implies

$$\sum_{l=0}^{T-1}\sum_{j=2}^{n}(\sigma'_{j,l})^2 \overset{(350)}{\leq} \frac{144\gamma^2 V}{n^2}\sum_{l=0}^{T-1}\sum_{j=2}^{n}\mathbb{E}_{\xi_{1,j}^l}\left[\|\omega_{j,l}^u\|^2\right] \overset{(325),T\leq K+1}{\leq} \frac{2592(K+1)\gamma^2 V\lambda^{2-\alpha}\sigma^\alpha}{n}$$

$$\overset{(305)}{\leq} \frac{2592(K+1)\gamma^\alpha\sigma^\alpha V^{2-\frac{\alpha}{2}}}{60^{2-\alpha}n^{\alpha-1}\ln^{2-\alpha}\frac{48n(K+1)}{\beta}} \overset{(304)}{\leq} \frac{3V^2}{50\ln\frac{48n(K+1)}{\beta}}. \tag{352}$$

**Upper bound for $2\gamma\sqrt{V}\left\|\sum_{l=0}^{T-1}\theta_l\right\|$.** We introduce new random vectors:

$$\eta'_l = \begin{cases} \gamma\sum\limits_{r=0}^{l-1}\theta_r, & \text{if } \left\|\gamma\sum\limits_{r=0}^{l-1}\theta_r\right\| \leq \sqrt{V}, \\ 0, & \text{otherwise} \end{cases}$$

for $l = 1, 2, \ldots, T-1$. With probability 1 we have

$$\|\zeta'_l\| \leq \sqrt{V}. \tag{353}$$

Using this and (311), we obtain that $E_{T-1}$ implies

$$\begin{aligned}
2\gamma\sqrt{V}\left\|\sum_{l=0}^{T-1}\theta_l\right\| &= 2\sqrt{V}\sqrt{\gamma^2\left\|\sum_{l=0}^{T-1}\theta_l\right\|^2} \\
&= 2\sqrt{V}\sqrt{\gamma^2\sum_{l=0}^{T-1}\|\theta_l\|^2 + 2\gamma\sum_{l=0}^{T-1}\left\langle \gamma\sum_{r=0}^{l-1}\theta_r, \theta_l \right\rangle} \\
&= 2\sqrt{V}\sqrt{\gamma^2\sum_{l=0}^{T-1}\|\theta_l\|^2 + 2\gamma\sum_{l=0}^{T-1}\langle\zeta'_l, \theta_l\rangle} \\
&\overset{(315)}{\leq} 2\sqrt{V}\sqrt{\frac{③+④+⑤+⑥}{8} + \underbrace{\frac{2\gamma}{n}\sum_{l=0}^{T-1}\sum_{i=1}^{n}\langle\zeta'_l, \theta_{i,l}^u\rangle}_{⑧} + \underbrace{2\gamma\sum_{l=0}^{T-1}\langle\zeta'_l, \theta_l^b\rangle}_{⑨}} 
\end{aligned} \tag{354}$$

**Upper bound for ⑧.** To estimate this sum, we will use Bernstein's inequality. The summands have conditional expectations equal to zero:

$$\mathbb{E}_{\xi_{2,i}^l}\left[\frac{2\gamma}{n}\langle\zeta'_l, \theta_{i,l}^u\rangle\right] = \frac{2\gamma}{n}\left\langle \zeta'_l, \mathbb{E}_{\xi_{2,i}^l}[\theta_{i,l}^u] \right\rangle = 0.$$

Moreover, for all $l = 0, \ldots, T-1$ random vectors $\{\theta_{i,l}^u\}_{i=1}^n$ are independent. Thus, sequence $\left\{\frac{2\gamma}{n}\langle\zeta'_l, \theta_{i,l}^u\rangle\right\}_{l,i=0,1}^{T-1,n}$ is a martingale difference sequence. Next, the summands are bounded:

$$\left|\frac{2\gamma}{n}\langle\zeta'_l, \theta_{i,l}^u\rangle\right| \leq \frac{2\gamma}{n}\|\zeta'_l\|\cdot\|\theta_{i,l}^u\| \overset{(353),(323)}{\leq} \frac{4\gamma}{n}\sqrt{V}\lambda \overset{(305)}{\leq} \frac{V}{10\ln\frac{48n(K+1)}{\beta}} \overset{\text{def}}{=} c. \tag{355}$$

Finally, conditional variances $(\widetilde{\sigma}'_{i,l})^2 \overset{\text{def}}{=} \mathbb{E}_{\xi^l_{2,i}}\left[\frac{4\gamma^2}{n^2}\langle \zeta'_l, \theta^u_{i,l}\rangle^2\right]$ of the summands are bounded:

$$(\widetilde{\sigma}'_{i,l})^2 \le \mathbb{E}_{\xi^l_{2,i}}\left[\frac{4\gamma^2}{n^2}\|\zeta'_l\|^2 \cdot \|\theta^u_{i,l}\|^2\right] \overset{(353)}{\le} \frac{4\gamma^2 V}{n^2}\mathbb{E}_{\xi^l_{2,i}}\left[\|\theta^u_{i,l}\|^2\right]. \tag{356}$$

Applying Bernstein's inequality (Lemma B.1) with $X_{i,l} = \frac{2\gamma}{n}\langle \zeta'_l, \theta^u_{i,l}\rangle$, constant $c$ defined in (355), $b = \frac{V}{10}, G = \frac{V^2}{600\ln\frac{48n(K+1)}{\beta}}$, we get

$$\mathbb{P}\left\{|\text{⑧}| > \frac{V}{10} \text{ and } \sum_{l=0}^{T-1}\sum_{i=1}^{n}(\widetilde{\sigma}'_{i,l})^2 \le \frac{V^2}{600\ln\frac{48n(K+1)}{\beta}}\right\} \le 2\exp\left(-\frac{b^2}{2G + \nicefrac{2cb}{3}}\right) = \frac{\beta}{24n(K+1)}.$$

The above is equivalent to

$$\mathbb{P}\{E_\text{⑧}\} \ge 1 - \frac{\beta}{24n(K+1)}, \quad \text{for } E_\text{⑧} = \left\{\text{either } \sum_{l=0}^{T-1}\sum_{i=1}^{n}(\widetilde{\sigma}'_{i,l})^2 > \frac{V^2}{600\ln\frac{48n(K+1)}{\beta}} \text{ or } |\text{⑧}| \le \frac{V}{10}\right\}. \tag{357}$$

Moreover, $E_{T-1}$ implies

$$\sum_{l=0}^{T-1}\sum_{i=1}^{n}(\widetilde{\sigma}'_{i,l})^2 \overset{(356)}{\le} \frac{4\gamma^2 V}{n^2}\sum_{l=0}^{T-1}\sum_{i=1}^{n}\mathbb{E}_{\xi^l_{2,i}}\left[\|\theta^u_{i,l}\|^2\right] \overset{(325),T\le K+1}{\le} \frac{72(K+1)\gamma^2 V\lambda^{2-\alpha}\sigma^\alpha}{n}$$

$$\overset{(305)}{\le} \frac{72(K+1)\gamma^\alpha\sigma^\alpha V^{2-\frac{\alpha}{2}}}{60^{2-\alpha}n^{\alpha-1}\ln^{2-\alpha}\frac{48n(K+1)}{\beta}} \overset{(304)}{\le} \frac{V^2}{600\ln\frac{48n(K+1)}{\beta}}. \tag{358}$$

**Upper bound for ⑨.** Probability event $E_{T-1}$ implies

$$\text{⑨} \le 2\gamma\sum_{l=0}^{T-1}\|\zeta'_l\| \cdot \|\theta^b_l\| \overset{(353),(324),T\le K+1}{\le} \frac{2 \cdot 2^\alpha(K+1)\gamma\sqrt{V}\sigma^\alpha}{\lambda^{\alpha-1}}$$

$$\overset{(305)}{=} \frac{2 \cdot 2^\alpha \cdot 60^{\alpha-1}(K+1)\gamma^\alpha\sigma^\alpha\ln^{\alpha-1}\frac{48n(K+1)}{\beta}}{V^{\frac{\alpha}{2}-1}} \overset{(304)}{\le} \frac{V}{100}. \tag{359}$$

That is, we derive the upper bounds for $2\gamma\sqrt{V}\left\|\sum_{l=0}^{T-1}\theta_l\right\|, \text{①},\text{②},\text{③},\text{④},\text{⑤},\text{⑥},\text{⑦}$. More precisely, $E_{T-1}$ implies

$$A_T \overset{(322)}{\le} 4V + 2\gamma\sqrt{V}\left\|\sum_{l=0}^{T-1}\theta_l\right\| + \text{①} + \text{②} + \text{③} + \text{④} + \text{⑤} + \text{⑥} + \text{⑦},$$

$$\text{⑥} \overset{(341)}{=} \text{⑥}' + \frac{32\gamma}{n}\sum_{j=2}^{n}\left\langle\frac{\gamma}{n}\sum_{i=1}^{j-1}\theta^u_{i,T-1} - \zeta^{T-1}_j, \theta^u_{j,T-1}\right\rangle,$$

$$\text{⑦} \overset{(342)}{=} \text{⑦}' + \frac{24\gamma}{n}\sum_{j=2}^{n}\left\langle\frac{\gamma}{n}\sum_{i=1}^{j-1}\omega^u_{i,T-1} - \delta^{T-1}_j, \omega^u_{j,T-1}\right\rangle,$$

$$2\gamma\sqrt{V}\left\|\sum_{l=0}^{T-1}\theta_l\right\| \overset{(354)}{\le} 2\sqrt{V}\sqrt{\frac{\text{③} + \text{④} + \text{⑤} + \text{⑥}}{8} + \text{⑧} + \text{⑨}},$$

$$\text{②} \overset{(330)}{\le} \frac{3V}{100}, \quad \text{③} \overset{(333)}{\le} \frac{V}{20}, \quad \text{⑤} \overset{(338)}{\le} \frac{V}{6}, \quad \text{⑨} \overset{(359)}{\le} \frac{V}{100},$$

$$\sum_{l=0}^{T-1}\sum_{i=1}^{n}\sigma^2_{i,l} \overset{(329)}{\le} \frac{3V^2}{200\ln\frac{48n(K+1)}{\beta}}, \quad \sum_{l=0}^{T-1}\sum_{i=1}^{n}\widetilde{\sigma}^2_{i,l} \overset{(337)}{\le} \frac{V^2}{216\ln\frac{48n(K+1)}{\beta}}, \quad \sum_{l=0}^{T-1}\sum_{j=2}^{n}\widehat{\sigma}^2_{j,l} \overset{(348)}{\le} \frac{8V^2}{75\ln\frac{48n(K+1)}{\beta}},$$

$$\sum_{l=0}^{T-1}\sum_{j=2}^{n}(\sigma'_{j,l})^2 \overset{(352)}{\le} \frac{3V^2}{50\ln\frac{48n(K+1)}{\beta}}, \quad \sum_{l=0}^{T-1}\sum_{i=1}^{n}(\widetilde{\sigma}'_{i,l})^2 \le \frac{V^2}{600\ln\frac{48n(K+1)}{\beta}}.$$

In addition, we also establish (see (328), (336), (347), (351), (357) and our induction assumption)

$$\mathbb{P}\{E_{T-1}\} \geq 1 - \frac{(T-1)\beta}{K+1},$$

$$\mathbb{P}\{E_①\} \geq 1 - \frac{\beta}{24n(K+1)}, \quad \mathbb{P}\{E_④\} \geq 1 - \frac{\beta}{24n(K+1)}, \quad \mathbb{P}\{E_{⑥'}\} \geq 1 - \frac{\beta}{24n(K+1)},$$

$$\mathbb{P}\{E_{⑦'}\} \geq 1 - \frac{\beta}{24n(K+1)}, \quad \mathbb{P}\{E_⑧\} \geq 1 - \frac{\beta}{24n(K+1)},$$

where

$$
\begin{aligned}
E_① &= \left\{ \text{either} \sum_{l=0}^{T-1}\sum_{i=1}^{n} \sigma_{i,l}^2 > \frac{3V^2}{200\ln\frac{48n(K+1)}{\beta}} \text{ or } |①| \leq \frac{3V}{10} \right\}, \\
E_④ &= \left\{ \text{either} \sum_{l=0}^{T-1}\sum_{i=1}^{n} \widetilde{\sigma}_{i,l}^2 > \frac{V^2}{216\ln\frac{48n(K+1)}{\beta}} \text{ or } |④| \leq \frac{V}{6} \right\}, \\
E_{⑥'} &= \left\{ \text{either} \sum_{l=0}^{T-1}\sum_{j=2}^{n} \widehat{\sigma}_{i,l}^2 > \frac{8V^2}{75\ln\frac{48n(K+1)}{\beta}} \text{ or } |⑥'| \leq \frac{4V}{5} \right\}, \\
E_{⑦'} &= \left\{ \text{either} \sum_{l=0}^{T-1}\sum_{j=2}^{n} (\sigma_{i,l}')^2 > \frac{3V^2}{50\ln\frac{48n(K+1)}{\beta}} \text{ or } |⑦'| \leq \frac{3V}{5} \right\} \\
E_⑧ &= \left\{ \text{either} \sum_{l=0}^{T-1}\sum_{i=1}^{n} (\widetilde{\sigma}_{i,l}')^2 > \frac{V^2}{600\ln\frac{48n(K+1)}{\beta}} \text{ or } |⑧| \leq \frac{V}{10} \right\}
\end{aligned}
$$

Therefore, probability event $E_{T-1} \cap E_① \cap E_④ \cap E_{⑥'} \cap E_{⑦'} \cap E_⑧$ implies

$$
\begin{aligned}
\left\| \gamma \sum_{l=0}^{T-1} \theta_l \right\| &\leq \sqrt{\frac{1}{8}\left(\frac{V}{20} + \frac{V}{6} + \frac{V}{6} + \frac{4V}{5}\right) + \frac{V}{10} + \frac{V}{100}} \\
&\quad + \sqrt{\frac{32\gamma}{n}\sum_{j=2}^{n}\left\langle \frac{\gamma}{n}\sum_{i=1}^{j-1}\theta_{i,T-1}^u - \zeta_j^{T-1}, \theta_{j,T-1}^u \right\rangle} \\
&\leq \sqrt{V} + \sqrt{\frac{32\gamma}{n}\sum_{j=2}^{n}\left\langle \frac{\gamma}{n}\sum_{i=1}^{j-1}\theta_{i,T-1}^u - \zeta_j^{T-1}, \theta_{j,T-1}^u \right\rangle}, \qquad (360)
\end{aligned}
$$

and

$$
\begin{aligned}
A_T \quad \leq \quad & 4V + 2V + 2\sqrt{V}\sqrt{\frac{32\gamma}{n}\sum_{j=2}^{n}\left\langle\frac{\gamma}{n}\sum_{i=1}^{j-1}\theta_{i,T-1}^{u} - \zeta_j^{T-1}, \theta_{j,T-1}^{u}\right\rangle} \\
& + \frac{3V}{10} + \frac{3V}{100} + \frac{V}{20} + \frac{V}{6} + \frac{V}{6} + \frac{4V}{5} + \frac{3V}{5} \\
& + \frac{32\gamma}{n}\sum_{j=2}^{n}\left\langle\frac{\gamma}{n}\sum_{i=1}^{j-1}\theta_{i,T-1}^{u} - \zeta_j^{T-1}, \theta_{j,T-1}^{u}\right\rangle \\
& + \frac{24\gamma}{n}\sum_{j=2}^{n}\left\langle\frac{\gamma}{n}\sum_{i=1}^{j-1}\omega_{i,T-1}^{u} - \delta_j^{T-1}, \omega_{j,T-1}^{u}\right\rangle \\
\leq \quad & 9V + 2\sqrt{V}\sqrt{\frac{32\gamma}{n}\sum_{j=2}^{n}\left\langle\frac{\gamma}{n}\sum_{i=1}^{j-1}\theta_{i,T-1}^{u} - \zeta_j^{T-1}, \theta_{j,T-1}^{u}\right\rangle} \\
& + \frac{32\gamma}{n}\sum_{j=2}^{n}\left\langle\frac{\gamma}{n}\sum_{i=1}^{j-1}\theta_{i,T-1}^{u} - \zeta_j^{T-1}, \theta_{j,T-1}^{u}\right\rangle \\
& + \frac{24\gamma}{n}\sum_{j=2}^{n}\left\langle\frac{\gamma}{n}\sum_{i=1}^{j-1}\omega_{i,T-1}^{u} - \delta_j^{T-1}, \omega_{j,T-1}^{u}\right\rangle,
\end{aligned}
\tag{361}
$$

In the final part of the proof, we will show that $\frac{\gamma}{n}\sum_{i=1}^{j-1}\theta_{i,T-1}^{u} = \zeta_j^{T-1}$ and $\frac{\gamma}{n}\sum_{i=1}^{j-1}\omega_{i,T-1}^{u} = \delta_j^{T-1}$ with high probability. In particular, we consider probability event $\widetilde{E}_{T-1,j}$ defined as follows: inequalities

$$
\left\|\frac{\gamma}{n}\sum_{i=1}^{r-1}\theta_{i,T-1}^{u}\right\| \leq \frac{\sqrt{V}}{2}, \quad \left\|\frac{\gamma}{n}\sum_{i=1}^{r-1}\omega_{i,T-1}^{u}\right\| \leq \frac{\sqrt{V}}{2}
$$

hold for $r = 2, \ldots, j$ simultaneously. We want to show that $\mathbb{P}\{E_{T-1} \cap \widetilde{E}_{T-1,j}\} \geq 1 - \frac{(T-1)\beta}{K+1} - \frac{j\beta}{8n(K+1)}$ for all $j = 2, \ldots, n$. For $j = 2$ the statement is trivial since

$$
\left\|\frac{\gamma}{n}\theta_{1,T-1}^{u}\right\| \overset{(323)}{\leq} \frac{2\gamma\lambda}{n} \leq \frac{\sqrt{V}}{2}, \quad \left\|\frac{\gamma}{n}\omega_{1,T-1}^{u}\right\| \overset{(323)}{\leq} \frac{2\gamma\lambda}{n} \leq \frac{\sqrt{V}}{2}.
$$

Next, we assume that the statement holds for some $j = m - 1 < n$, i.e., $\mathbb{P}\{E_{T-1} \cap \widetilde{E}_{T-1,m-1}\} \geq 1 - \frac{(T-1)\beta}{K+1} - \frac{(m-1)\beta}{8n(K+1)}$. Our goal is to prove that $\mathbb{P}\{E_{T-1} \cap \widetilde{E}_{T-1,m}\} \geq 1 - \frac{(T-1)\beta}{K+1} - \frac{m\beta}{8n(K+1)}$. First, we consider $\left\|\frac{\gamma}{n}\sum_{i=1}^{m-1}\theta_{i,T-1}^{u}\right\|$:

$$
\begin{aligned}
\left\|\frac{\gamma}{n}\sum_{i=1}^{m-1}\theta_{i,T-1}^{u}\right\| \quad &= \quad \sqrt{\frac{\gamma^2}{n^2}\left\|\sum_{i=1}^{m-1}\theta_{i,T-1}^{u}\right\|^2} \\
&= \quad \sqrt{\frac{\gamma^2}{n^2}\sum_{i=1}^{m-1}\|\theta_{i,T-1}^{u}\|^2 + \frac{2\gamma}{n}\sum_{i=1}^{m-1}\left\langle\frac{\gamma}{n}\sum_{r=1}^{i-1}\theta_{r,T-1}^{u}, \theta_{i,T-1}^{u}\right\rangle} \\
&\leq \quad \sqrt{\frac{\gamma^2}{n^2}\sum_{l=0}^{T-1}\sum_{i=1}^{m-1}\|\theta_{i,l}^{u}\|^2 + \frac{2\gamma}{n}\sum_{i=1}^{m-1}\left\langle\frac{\gamma}{n}\sum_{r=1}^{i-1}\theta_{r,T-1}^{u}, \theta_{i,T-1}^{u}\right\rangle}.
\end{aligned}
$$

Similarly, we have

$$
\begin{aligned}
\left\| \frac{\gamma}{n} \sum_{i=1}^{m-1} \omega_{i,T-1}^u \right\| &= \sqrt{ \frac{\gamma^2}{n^2} \left\| \sum_{i=1}^{m-1} \omega_{i,T-1}^u \right\|^2 } \\
&= \sqrt{ \frac{\gamma^2}{n^2} \sum_{i=1}^{m-1} \| \omega_{i,T-1}^u \|^2 + \frac{2\gamma}{n} \sum_{i=1}^{m-1} \left\langle \frac{\gamma}{n} \sum_{r=1}^{i-1} \omega_{r,T-1}^u, \omega_{i,T-1}^u \right\rangle } \\
&\leq \sqrt{ \frac{\gamma^2}{n^2} \sum_{l=0}^{T-1} \sum_{i=1}^{m-1} \| \omega_{i,l}^u \|^2 + \frac{2\gamma}{n} \sum_{i=1}^{m-1} \left\langle \frac{\gamma}{n} \sum_{r=1}^{i-1} \omega_{r,T-1}^u, \omega_{i,T-1}^u \right\rangle }.
\end{aligned}
$$

Next, we introduce a new notation:

$$
\rho_{i,T-1} = \begin{cases} \frac{\gamma}{n} \sum_{r=1}^{i-1} \theta_{r,T-1}^u, & \text{if } \left\| \frac{\gamma}{n} \sum_{r=1}^{i-1} \theta_{r,T-1}^u \right\| \leq \frac{\sqrt{V}}{2}, \\ 0, & \text{otherwise} \end{cases},
$$

$$
\rho_{i,T-1}' = \begin{cases} \frac{\gamma}{n} \sum_{r=1}^{i-1} \omega_{r,T-1}^u, & \text{if } \left\| \frac{\gamma}{n} \sum_{r=1}^{i-1} \omega_{r,T-1}^u \right\| \leq \frac{\sqrt{V}}{2}, \\ 0, & \text{otherwise} \end{cases}
$$

for $i = 1, \ldots, m-1$. By definition, we have

$$
\| \rho_{i,T-1} \| \leq \frac{\sqrt{V}}{2}, \quad \| \rho_{i,T-1}' \| \leq \frac{\sqrt{V}}{2} \tag{362}
$$

for $i = 1, \ldots, m-1$. Moreover, $\widetilde{E}_{T-1,m-1}$ implies $\rho_{i,T-1} = \frac{\gamma}{n} \sum_{r=1}^{i-1} \theta_{r,T-1}^u, \rho_{i,T-1}' = \frac{\gamma}{n} \sum_{r=1}^{i-1} \omega_{r,T-1}^u$ for $i = 1, \ldots, m-1$ and

$$
\begin{aligned}
\left\| \frac{\gamma}{n} \sum_{i=1}^{m-1} \theta_{i,l}^u \right\| &\leq \sqrt{③ + ④ + ⑩}, \\
\left\| \frac{\gamma}{n} \sum_{i=1}^{m-1} \omega_{i,l}^u \right\| &\leq \sqrt{③ + ④ + ⑩'},
\end{aligned}
$$

where

$$
⑩ = \frac{2\gamma}{n} \sum_{i=1}^{m-1} \left\langle \rho_{i,T-1}, \theta_{i,T-1}^u \right\rangle, \quad ⑩' = \frac{2\gamma}{n} \sum_{i=1}^{m-1} \left\langle \rho_{i,T-1}', \omega_{i,T-1}^u \right\rangle.
$$

It remains to estimate ⑩ and ⑩'.

**Upper bound for ⑩.** To estimate this sum, we will use Bernstein's inequality. The summands have conditional expectations equal to zero:

$$
\mathbb{E}_{\xi_{2,i}^{T-1}} \left[ \frac{2\gamma}{n} \langle \rho_{i,T-1}, \theta_{i,T-1}^u \rangle \right] = \frac{2\gamma}{n} \left\langle \rho_{i,T-1}, \mathbb{E}_{\xi_{2,i}^{T-1}} [\theta_{i,T-1}^u] \right\rangle = 0,
$$

since random vectors $\{ \theta_{i,T-1}^u \}_{i=1}^n$ are independent. Thus, sequence $\left\{ \frac{2\gamma}{n} \langle \rho_{i,T-1}, \theta_{i,T-1}^u \rangle \right\}_{i=1}^{m-1}$ is a martingale difference sequence. Next, the summands are bounded:

$$
\left| \frac{2\gamma}{n} \langle \rho_{i,T-1}, \theta_{i,T-1}^u \rangle \right| \leq \frac{2\gamma}{n} \| \rho_{i,T-1} \| \cdot \| \theta_{i,T-1}^u \| \overset{(362),(323)}{\leq} \frac{2\gamma}{n} \sqrt{V} \lambda \overset{(305)}{=} \frac{V}{30 \ln \frac{48n(K+1)}{\beta}} \overset{\text{def}}{=} c. \tag{363}
$$

Finally, conditional variances $(\widehat{\sigma}_{i,T-1}')^2 \overset{\text{def}}{=} \mathbb{E}_{\xi_{2,i}^{T-1}} \left[ \frac{4\gamma^2}{n^2} \langle \rho_{i,T-1}, \theta_{i,T-1}^u \rangle^2 \right]$ of the summands are bounded:

$$
(\widehat{\sigma}_{i,T-1}')^2 \leq \mathbb{E}_{\xi_{2,i}^{T-1}} \left[ \frac{4\gamma^2}{n^2} \| \rho_{i,T-1} \|^2 \cdot \| \theta_{i,T-1}^u \|^2 \right] \overset{(360)}{\leq} \frac{\gamma^2 V}{n^2} \mathbb{E}_{\xi_{2,i}^{T-1}} \left[ \| \theta_{i,T-1}^u \|^2 \right]. \tag{364}
$$

Applying Bernstein's inequality (Lemma B.1) with $X_i = \frac{2\gamma}{n}\langle \rho_{i,T-1}, \theta^u_{i,T-1}\rangle$, constant $c$ defined in (363), $b = \frac{V}{30}$, $G = \frac{V^2}{5400\ln\frac{48n(K+1)}{\beta}}$, we get

$$\mathbb{P}\left\{|\text{⑩}| > \frac{V}{30} \text{ and } \sum_{i=1}^{m-1}(\widehat{\sigma}'_{i,T-1})^2 \leq \frac{V^2}{5400\ln\frac{48n(K+1)}{\beta}}\right\} \leq 2\exp\left(-\frac{b^2}{2G + 2cb/3}\right) = \frac{\beta}{24n(K+1)}.$$

The above is equivalent to

$$\mathbb{P}\{E_{\text{⑩}}\} \geq 1 - \frac{\beta}{24n(K+1)}, \quad \text{for } E_{\text{⑩}} = \left\{\text{either } \sum_{i=1}^{m-1}(\widehat{\sigma}'_{i,T-1})^2 > \frac{V^2}{5400\ln\frac{48n(K+1)}{\beta}} \text{ or } |\text{⑩}| \leq \frac{V}{30}\right\}.$$
(365)

Moreover, $E_{T-1}$ implies

$$\sum_{i=1}^{m-1}(\widehat{\sigma}'_{i,T-1})^2 \overset{(364)}{\leq} \frac{\gamma^2 V}{n^2}\sum_{i=1}^{n}\mathbb{E}_{\xi^{T-1}_{2,i}}\left[\|\theta^u_{i,T-1}\|^2\right] \overset{(325)}{\leq} \frac{18\gamma^2 V\lambda^{2-\alpha}\sigma^\alpha}{n}$$

$$\overset{(305)}{\leq} \frac{18\gamma^\alpha\sigma^\alpha V^{2-\frac{\alpha}{2}}}{60^{2-\alpha}n^{\alpha-1}\ln^{2-\alpha}\frac{48n(K+1)}{\beta}} \overset{(304)}{\leq} \frac{V^2}{5400\ln\frac{48n(K+1)}{\beta}}.$$
(366)

**Upper bound for $\text{⑩}'$.** To estimate this sum, we will use Bernstein's inequality. The summands have conditional expectations equal to zero:

$$\mathbb{E}_{\xi^{T-1}_{1,i}}\left[\frac{2\gamma}{n}\langle\rho'_{i,T-1},\omega^u_{i,T-1}\rangle\right] = \frac{2\gamma}{n}\left\langle\rho'_{i,T-1}, \mathbb{E}_{\xi^{T-1}_{1,i}}[\omega^u_{i,T-1}]\right\rangle = 0,$$

since random vectors $\{\omega^u_{i,T-1}\}_{i=1}^{n}$ are independent. Thus, sequence $\left\{\frac{2\gamma}{n}\langle\rho'_{i,T-1},\omega^u_{i,T-1}\rangle\right\}_{i=1}^{m-1}$ is a martingale difference sequence. Next, the summands are bounded:

$$\left|\frac{2\gamma}{n}\langle\rho'_{i,T-1},\omega^u_{i,T-1}\rangle\right| \leq \frac{2\gamma}{n}\|\rho'_{i,T-1}\|\cdot\|\omega^u_{i,T-1}\| \overset{(362),(323)}{\leq} \frac{2\gamma}{n}\sqrt{V}\lambda \overset{(305)}{=} \frac{V}{30\ln\frac{48n(K+1)}{\beta}} \overset{\text{def}}{=} c. (367)$$

Finally, conditional variances $(\widetilde{\sigma}'_{i,T-1})^2 \overset{\text{def}}{=} \mathbb{E}_{\xi^{T-1}_{1,i}}\left[\frac{4\gamma^2}{n^2}\langle\rho'_{i,T-1},\omega^u_{i,T-1}\rangle^2\right]$ of the summands are bounded:

$$(\widetilde{\sigma}'_{i,T-1})^2 \leq \mathbb{E}_{\xi^{T-1}_{1,i}}\left[\frac{4\gamma^2}{n^2}\|\rho'_{i,T-1}\|^2\cdot\|\omega^u_{i,T-1}\|^2\right] \overset{(360)}{\leq} \frac{\gamma^2 V}{n^2}\mathbb{E}_{\xi^{T-1}_{1,i}}\left[\|\omega^u_{i,T-1}\|^2\right].$$
(368)

Applying Bernstein's inequality (Lemma B.1) with $X_i = \frac{2\gamma}{n}\langle\rho'_{i,T-1},\omega^u_{i,T-1}\rangle$, constant $c$ defined in (367), $b = \frac{V}{30}$, $G = \frac{V^2}{5400\ln\frac{48n(K+1)}{\beta}}$, we get

$$\mathbb{P}\left\{|\text{⑩}'| > \frac{V}{30} \text{ and } \sum_{i=1}^{m-1}(\widetilde{\sigma}'_{i,T-1})^2 \leq \frac{V^2}{5400\ln\frac{48n(K+1)}{\beta}}\right\} \leq 2\exp\left(-\frac{b^2}{2G + 2cb/3}\right) = \frac{\beta}{24n(K+1)}.$$

The above is equivalent to

$$\mathbb{P}\{E_{\text{⑩}'}\} \geq 1 - \frac{\beta}{24n(K+1)}, \quad \text{for } E_{\text{⑩}'} = \left\{\text{either } \sum_{i=1}^{m-1}(\widetilde{\sigma}'_{i,T-1})^2 > \frac{V^2}{5400\ln\frac{48n(K+1)}{\beta}} \text{ or } |\text{⑩}'| \leq \frac{V}{30}\right\}.$$
(369)

Moreover, $E_{T-1}$ implies

$$\sum_{i=1}^{m-1}(\widetilde{\sigma}'_{i,T-1})^2 \overset{(368)}{\leq} \frac{\gamma^2 V}{n^2}\sum_{i=1}^{n}\mathbb{E}_{\xi^{T-1}_{1,i}}\left[\|\omega^u_{i,T-1}\|^2\right] \overset{(325)}{\leq} \frac{18\gamma^2 V\lambda^{2-\alpha}\sigma^\alpha}{n}$$

$$\overset{(305)}{\leq} \frac{18\gamma^\alpha\sigma^\alpha V^{2-\frac{\alpha}{2}}}{60^{2-\alpha}n^{\alpha-1}\ln^{2-\alpha}\frac{48n(K+1)}{\beta}} \overset{(304)}{\leq} \frac{V^2}{5400\ln\frac{48n(K+1)}{\beta}}.$$
(370)

Putting all together we get that $E_{T-1} \cap \widetilde{E}_{T-1,m-1}$ implies

$$\left\| \frac{\gamma}{n} \sum_{i=1}^{m-1} \theta_{i,T-1}^u \right\| \leq \sqrt{③ + ④ + ⑩}, \quad \left\| \frac{\gamma}{n} \sum_{i=1}^{m-1} \omega_{i,T-1}^u \right\| \leq \sqrt{③ + ④ + ⑩'}, \quad ③ \overset{(333)}{\leq} \frac{V}{20},$$

$$\sum_{l=0}^{T-1} \sum_{i=1}^{n} \widetilde{\sigma}_{i,l}^2 \overset{(337)}{\leq} \frac{V^2}{216 \ln \frac{48n(K+1)}{\beta}}, \quad \sum_{i=1}^{m-1} (\widehat{\sigma}_{i,T-1}')^2 \leq \frac{V^2}{5400 \ln \frac{48n(K+1)}{\beta}},$$

$$\sum_{i=1}^{m-1} (\widetilde{\sigma}_{i,T-1}')^2 \leq \frac{V^2}{5400 \ln \frac{48n(K+1)}{\beta}}.$$

In addition, we also establish (see (336), (365), (369) and our induction assumption)

$$\mathbb{P}\{E_{T-1} \cap \widetilde{E}_{T-1,m-1}\} \geq 1 - \frac{(T-1)\beta}{K+1} - \frac{(m-1)\beta}{8n(K+1)},$$

$$\mathbb{P}\{E_{④}\} \geq 1 - \frac{\beta}{24n(K+1)}, \quad \mathbb{P}\{E_{⑩}\} \geq 1 - \frac{\beta}{24n(K+1)}, \quad \mathbb{P}\{E_{⑩'}\} \geq 1 - \frac{\beta}{24n(K+1)}$$

where

$$E_{④} = \left\{ \text{either } \sum_{l=0}^{T-1} \sum_{i=1}^{n} \widetilde{\sigma}_{i,l}^2 > \frac{V^2}{216 \ln \frac{48n(K+1)}{\beta}} \text{ or } |④| \leq \frac{V}{6} \right\},$$

$$E_{⑩} = \left\{ \text{either } \sum_{i=1}^{m-1} (\widehat{\sigma}_{i,l}')^2 > \frac{V^2}{5400 \ln \frac{48n(K+1)}{\beta}} \text{ or } |⑩| \leq \frac{V}{30} \right\},$$

$$E_{⑩'} = \left\{ \text{either } \sum_{i=1}^{m-1} (\widetilde{\sigma}_{i,T-1}')^2 > \frac{V^2}{5400 \ln \frac{48n(K+1)}{\beta}} \text{ or } |⑩'| \leq \frac{V}{30} \right\}$$

Therefore, probability event $E_{T-1} \cap \widetilde{E}_{T-1,m-1} \cap E_{④} \cap E_{⑩} \cap E_{⑩'}$ implies

$$\left\| \frac{\gamma}{n} \sum_{i=1}^{m-1} \theta_{i,T-1}^u \right\| \leq \sqrt{\frac{V}{20} + \frac{V}{6} + \frac{V}{30}} = \frac{\sqrt{V}}{2},$$

$$\left\| \frac{\gamma}{n} \sum_{i=1}^{m-1} \omega_{i,T-1}^u \right\| \leq \sqrt{\frac{V}{20} + \frac{V}{6} + \frac{V}{30}} = \frac{\sqrt{V}}{2}.$$

This implies $\widetilde{E}_{T-1,m}$ and

$$\begin{aligned}
\mathbb{P}\{E_{T-1} \cap \widetilde{E}_{T-1,m}\} &\geq \mathbb{P}\{E_{T-1} \cap \widetilde{E}_{T-1,m-1} \cap E_{④} \cap E_{⑩} \cap E_{⑩'}\} \\
&= 1 - \mathbb{P}\left\{ \overline{E_{T-1} \cap \widetilde{E}_{T-1,m-1}} \cup \overline{E}_{④} \cup \overline{E}_{⑩} \cup \overline{E}_{⑩'} \right\} \\
&\geq 1 - \frac{(T-1)\beta}{K+1} - \frac{m\beta}{8n(K+1)}.
\end{aligned}$$

Therefore, for all $m = 2, \ldots, n$ the statement holds and, in particular, $\mathbb{P}\{E_{T-1} \cap \widetilde{E}_{T-1,n}\} \geq 1 - \frac{(T-1)\beta}{K+1} - \frac{\beta}{8(K+1)}$. Taking into account (361), we conclude that $E_{T-1} \cap \widetilde{E}_{T-1,n} \cap E_{①} \cap E_{④} \cap E_{⑥'} \cap E_{⑦'} \cap E_{⑧}$ implies

$$\left\| \gamma \sum_{l=0}^{T-1} \theta_l \right\| \leq \sqrt{V}, \quad A_T \leq 9V$$

that is equivalent to (310) and (311) for $t = T$. Moreover,

$$\begin{aligned}
\mathbb{P}\{E_T\} &\geq \mathbb{P}\left\{ E_{T-1} \cap \widetilde{E}_{T-1,n} \cap E_{①} \cap E_{④} \cap E_{⑥'} \cap E_{⑦'} \cap E_{⑧} \right\} \\
&= 1 - \mathbb{P}\left\{ \overline{E_{T-1} \cap \widetilde{E}_n} \cup \overline{E}_{①} \cup \overline{E}_{④} \cup \overline{E}_{⑥'} \cup \overline{E}_{⑦'} \cup \overline{E}_{⑧} \right\} \\
&= 1 - \frac{(T-1)\beta}{K+1} - \frac{\beta}{8(K+1)} - 5 \cdot \frac{\beta}{24n(K+1)} = 1 - \frac{T\beta}{K+1}.
\end{aligned}$$

In other words, we showed that $\mathbb{P}\{E_k\} \geq 1 - {}^{k\beta}/_{(K+1)}$ for all $k = 0, 1, \ldots, K+1$. For $k = K+1$ we have that with probability at least $1 - \beta$

$$
\begin{aligned}
\mathrm{Gap}_{\sqrt{V}}(\widetilde{x}_{\mathrm{avg}}^K) &= \max_{B_{\sqrt{V}}(x^*)} \left\{ \langle F(u), \widetilde{x}_{\mathrm{avg}}^{t'} - u \rangle + \Psi(\widetilde{x}_{\mathrm{avg}}^{t'}) - \Psi(u) \right\} \\
&\leq \frac{1}{2\gamma(K+1)} \max_{B_{\sqrt{V}}(x^*)} \left\{ 2\gamma(t'+1) \left( \langle F(u), \widetilde{x}_{\mathrm{avg}}^{t'} - u \rangle + \Psi(\widetilde{x}_{\mathrm{avg}}^{t'}) - \Psi(u) \right) + \|x^{t'+1} - u\|^2 \right\} \\
&\overset{(319)}{\leq} \frac{9V}{2\gamma(K+1)}.
\end{aligned}
$$

Finally, if

$$
\gamma = \min \left\{ \frac{1}{1920 L \ln \frac{48n(K+1)}{\beta}}, \frac{60^{\frac{2-\alpha}{\alpha}} \sqrt{V} n^{\frac{\alpha-1}{\alpha}}}{97200^{\frac{1}{\alpha}} (K+1)^{\frac{1}{\alpha}} \sigma \ln^{\frac{\alpha-1}{\alpha}} \frac{48n(K+1)}{\beta}} \right\}
$$

then with probability at least $1 - \beta$

$$
\begin{aligned}
\mathrm{Gap}_{\sqrt{V}}(\widetilde{x}_{\mathrm{avg}}^K) &\leq \frac{9V}{2\gamma(K+1)} = \max \left\{ \frac{8640 L V \ln \frac{48n(K+1)}{\beta}}{K+1}, \frac{9 \cdot 60^{\frac{2-\alpha}{\alpha}} \cdot \sigma R \ln^{\frac{\alpha-1}{\alpha}} \frac{48n(K+1)}{\beta}}{2 \cdot 97200^{\frac{1}{\alpha}} n^{\frac{\alpha-1}{\alpha}} (K+1)^{\frac{\alpha-1}{\alpha}}} \right\} \\
&= \mathcal{O} \left( \max \left\{ \frac{LV \ln \frac{nK}{\beta}}{K}, \frac{\sigma \sqrt{V} \ln^{\frac{\alpha-1}{\alpha}} \frac{nK}{\beta}}{n^{\frac{\alpha-1}{\alpha}} K^{\frac{\alpha-1}{\alpha}}} \right\} \right).
\end{aligned}
$$

To get $\mathrm{Gap}_R(\widetilde{x}_{\mathrm{avg}}^K) \leq \varepsilon$ with probability $\geq 1 - \beta$, $K$ should be

$$
K = \mathcal{O} \left( \frac{LV}{\varepsilon} \ln \frac{nLV}{\varepsilon\beta}, \frac{1}{n} \left( \frac{\sigma\sqrt{V}}{\varepsilon} \right)^{\frac{\alpha}{\alpha-1}} \ln \frac{\sigma\sqrt{V}}{\varepsilon\beta} \right)
$$

that concludes the proof. $\qquad \square$

## H.2 QUASI-STRONGLY MONOTONE CASE

We start with the following lemma.

**Lemma H.2.** *Let Assumptions 6 and 8 hold for $Q = B_{4n\sqrt{V}}(x^*)$, where $V \geq \|x^0 - x^*\|^2 + \frac{36000000\gamma^2 \ln^2 \frac{48n(K+1)}{\beta}}{n^2} \sum_{i=1}^n \|F_i(x^*)\|^2$, $\nu = \gamma\mu$, and $0 < \gamma \leq \min \left\{ \frac{1}{6L}, \frac{\sqrt{n}}{15000 L \ln \frac{48n(K+1)}{\beta}}, \frac{1}{72000000\mu \ln^2 \frac{48n(K+1)}{\beta}} \right\}$. If $x^k$ and $\widetilde{x}^k$ lie in $B_{4n\sqrt{V}}(x^*)$ for all $k = 0, 1, \ldots, K$ for some $K \geq 0$, then the iterates produced by* DProx-clipped-SEG-shift *satisfy*

$$
\begin{aligned}
V_{K+1} &\leq \exp\left(-\frac{\gamma\mu}{2}(K+1)\right) V + 2\gamma \sum_{k=0}^K \exp\left(-\frac{\gamma\mu}{2}(K-k)\right) \langle \theta_k, x^k - x^* \rangle \\
&\quad + \frac{\gamma^2}{n^2} \sum_{k=0}^K \sum_{i=1}^n \exp\left(-\frac{\gamma\mu}{2}(K-k)\right) \left( 18\|\theta_{i,k}^u\|^2 + 14\|\omega_{i,k}^u\|^2 \right) \\
&\quad + \gamma^2 \sum_{k=0}^K \sum_{i=1}^n \exp\left(-\frac{\gamma\mu}{2}(K-k)\right) \left( 18\|\theta_{i,k}^b\|^2 + 14\|\omega_{i,k}^b\|^2 \right) \\
&\quad + \frac{32\gamma^2}{n^2} \sum_{k=0}^K \sum_{j=2}^n \exp\left(-\frac{\gamma\mu}{2}(K-k)\right) \left\langle \sum_{i=1}^{j-1} \theta_{i,k}^u, \theta_{j,k}^u \right\rangle \\
&\quad + \frac{24\gamma^2}{n^2} \sum_{k=0}^K \sum_{j=2}^n \exp\left(-\frac{\gamma\mu}{2}(K-k)\right) \left\langle \sum_{i=1}^{j-1} \omega_{i,k}^u, \omega_{j,k}^u \right\rangle, \qquad (371)
\end{aligned}
$$

*where $V_k = \|x^k - x^*\|^2 + \frac{36000000\gamma^2 \ln^2 \frac{48n(K+1)}{\beta}}{n^2} \sum_{i=1}^n \left( \|\widetilde{h}_i^k - F_i(x^*)\|^2 + \|\widehat{h}_i^k - F_i(x^*)\|^2 \right)$ and $\theta_k, \theta_{i,k}^u, \theta_{i,k}^b, \omega_k, \omega_{i,k}^u, \omega_{i,k}^b$ are defined in (298), (299), and (315)-(316)*

*Proof.* From (303) with $u = x^*$ we have

$$
\gamma \left( \langle F(\widetilde{x}^k), \widetilde{x}^k - x^* \rangle + \Psi(\widetilde{x}^k) - \Psi(x^*) \right) \leq \frac{1}{2}\|x^k - x^*\|^2 - \frac{1}{2}\|x^{k+1} - x^*\|^2
$$
$$
- \frac{1}{2}\left(1 - 6\gamma^2 L^2\right)\|\widetilde{x}^k - x^k\|^2 - \frac{1}{4}\|x^{k+1} - \widetilde{x}^k\|^2
$$
$$
+ 3\gamma^2\|\omega_k\|^2 + 3\gamma^2\|\theta_k\|^2 + \gamma\langle\theta_k, \widetilde{x}^k - x^*\rangle.
$$

Using quasi-strong monotonicity of $F$, the fact that $-F(x^*) \in \partial\Psi(x^*)$, and convexity of $\Psi(x^*)$, we derive

$$
2\gamma\mu\|\widetilde{x}^k - x^*\|^2 \leq 2\gamma\langle F(\widetilde{x}^k) - F(x^*), \widetilde{x}^k - x^*\rangle
$$
$$
\leq 2\gamma\left(\langle F(\widetilde{x}^k), \widetilde{x}^k - x^*\rangle + \Psi(\widetilde{x}^k) - \Psi(x^*)\right)
$$
$$
\leq \|x^k - x^*\|^2 - \|x^{k+1} - x^*\|^2 - \left(1 - 6\gamma^2 L^2\right)\|\widetilde{x}^k - x^k\|^2
$$
$$
- \frac{1}{2}\|x^{k+1} - \widetilde{x}^k\|^2 + 6\gamma^2\|\omega_k\|^2 + 6\gamma^2\|\theta_k\|^2 + 2\gamma\langle\theta_k, \widetilde{x}^k - x^*\rangle.
$$

Next, we apply $\|\widetilde{x}^k - x^*\|^2 \geq \frac{1}{2}\|x^k - x^*\|^2 - \|\widetilde{x}^k - x^k\|^2$ and rearrange the terms:

$$
\|x^{k+1} - x^*\|^2 \leq \left(1 - \frac{\gamma\mu}{2}\right)\|x^k - x^*\|^2 - \gamma\mu\|\widetilde{x}^k - x^*\|^2 - \left(1 - \gamma\mu - 6\gamma^2 L^2\right)\|\widetilde{x}^k - x^k\|^2
$$
$$
+ 6\gamma^2\|\omega_k\|^2 + 6\gamma^2\|\theta_k\|^2 + 2\gamma\langle\theta_k, \widetilde{x}^k - x^*\rangle.
$$

Since $2\gamma\langle\theta^k, \widetilde{x}^k - x^*\rangle = 2\gamma\langle\theta^k, x^k - x^*\rangle + 2\gamma\langle\theta^k, \widetilde{x}^k - x^k\rangle \leq 2\gamma^2\|\theta^k\|^2 + \frac{1}{2}\|\widetilde{x}^k - x^k\|^2$, we have

$$
\|x^{k+1} - x^*\|^2 \leq \left(1 - \frac{\gamma\mu}{2}\right)\|x^k - x^*\|^2 - \gamma\mu\|\widetilde{x}^k - x^*\|^2 - \left(\frac{1}{2} - \gamma\mu - 6\gamma^2 L^2\right)\|\widetilde{x}^k - x^k\|^2
$$
$$
+ 6\gamma^2\|\omega_k\|^2 + 8\gamma^2\|\theta_k\|^2 + 2\gamma\langle\theta_k, x^k - x^*\rangle. \tag{372}
$$

Now, we move on to the shifts: for all $i \in [n]$ (for convenience, we use the new notation: $h_i^* = F_i(x^*)$ for all $i \in [n]$)

$$
\|\widetilde{h}_i^{k+1} - h_i^*\|^2 = \|\widetilde{h}_i^k - h_i^*\|^2 + 2\nu\left\langle\tilde{\Delta}_i^k, \widetilde{h}_i^k - h_i^*\right\rangle + \nu^2\|\tilde{\Delta}_i^k\|^2
$$
$$
= \|\widetilde{h}_i^k - h_i^*\|^2 + 2\nu\left\langle\widetilde{g}_i^k - \widetilde{h}_i^k, \widetilde{h}_i^k - h_i^*\right\rangle + \nu^2\|\widetilde{g}_i^k - \widetilde{h}_i^k\|^2
$$
$$
\overset{\nu \leq 1}{\leq} \|\widetilde{h}_i^k - h_i^*\|^2 + 2\nu\left\langle\widetilde{g}_i^k - \widetilde{h}_i^k, \widetilde{h}_i^k - h_i^*\right\rangle + \nu\|\widetilde{g}_i^k - \widetilde{h}_i^k\|^2
$$
$$
= \|\widetilde{h}_i^k - h_i^*\|^2 + \nu\left\langle\widetilde{g}_i^k - \widetilde{h}_i^k, \widetilde{g}_i^k + \widetilde{h}_i^k - 2h_i^*\right\rangle
$$
$$
\leq (1-\nu)\|\widetilde{h}_i^k - h_i^*\|^2 + \nu\|\widetilde{g}_i^k - h_i^*\|^2
$$
$$
\leq (1-\nu)\|\widetilde{h}_i^k - h_i^*\|^2 + 2\nu\|\widetilde{g}_i^k - F_i(x^k)\|^2 + 2\nu\|F_i(x^k) - h_i^*\|^2
$$
$$
\leq (1-\nu)\|\widetilde{h}_i^k - h_i^*\|^2 + 2\nu\|\omega_{i,k}\|^2 + 2\nu\|F_i(x^k) - h_i^*\|^2
$$
$$
= (1-\gamma\mu)\|\widetilde{h}_i^k - h_i^*\|^2 + 2\gamma\mu\|\omega_{i,k}\|^2 + 2\gamma\mu\|F_i(x^k) - h_i^*\|^2
$$
$$
\overset{(32)}{\leq} (1-\gamma\mu)\|\widetilde{h}_i^k - h_i^*\|^2 + 2\gamma\mu\|\omega_{i,k}\|^2 + 2\gamma\mu L^2\|x^k - x^*\|^2
$$
$$
\leq (1-\gamma\mu)\|\widetilde{h}_i^k - h_i^*\|^2 + 2\gamma\mu\|\omega_{i,k}\|^2 + 4\gamma\mu L^2\|\widetilde{x}^k - x^*\|^2
$$
$$
+ 4\gamma\mu L^2\|\widetilde{x}^k - x^k\|^2 \tag{373}
$$

and, similarly,

$$
\|\widehat{h}_i^{k+1} - h_i^*\|^2 \leq (1-\gamma\mu)\|\widehat{h}_i^k - h_i^*\|^2 + 2\gamma\mu\|\theta_{i,k}\|^2 + 2\gamma\mu\|F_i(\widetilde{x}^k) - h_i^*\|^2
$$
$$
\overset{(32)}{\leq} (1-\gamma\mu)\|\widehat{h}_i^k - h_i^*\|^2 + 2\gamma\mu\|\theta_{i,k}\|^2 + 2\gamma\mu L^2\|\widetilde{x}^k - x^*\|^2. \tag{374}
$$

Summing up (372), (373), and (374), we derive

$$
\begin{aligned}
V_{k+1} \quad \leq \quad & \left(1 - \frac{\gamma\mu}{2}\right)\|x^k - x^*\|^2 \\
& + (1 - \gamma\mu)\frac{36 \cdot 10^6 \gamma^2 \ln^2 \frac{48n(K+1)}{\beta}}{n^2} \sum_{i=1}^{n}\left(\|\widetilde{h}_i^k - h_i^*\|^2 + \|\widehat{h}_i^k - h_i^*\|^2\right) \\
& - \left(\gamma\mu - \frac{216 \cdot 10^6 \gamma^3 \mu L^2 \ln^2 \frac{48n(K+1)}{\beta}}{n}\right)\|\widetilde{x}^k - x^*\|^2 \\
& - \left(\frac{1}{2} - \gamma\mu - 6\gamma^2 L^2 - \frac{144 \cdot 10^6 \gamma^3 \mu L^2 \ln^2 \frac{48n(K+1)}{\beta}}{n}\right)\|\widetilde{x}^k - x^k\|^2 \\
& + 6\gamma^2\|\omega_k\|^2 + 8\gamma^2\|\theta_k\|^2 + 2\gamma\langle\theta_k, x^k - x^*\rangle \\
& + \frac{72 \cdot 10^6 \gamma^3 \mu \ln^2 \frac{48n(K+1)}{\beta}}{n^2} \sum_{i=1}^{n}\left(\|\theta_{i,k}\|^2 + \|\omega_{i,k}\|^2\right) \\
\leq \quad & \left(1 - \frac{\gamma\mu}{2}\right) V_k + 6\gamma^2\|\omega_k\|^2 + 8\gamma^2\|\theta_k\|^2 + 2\gamma\langle\theta_k, x^k - x^*\rangle \\
& + \frac{\gamma^2}{n^2} \sum_{i=1}^{n}\left(\|\theta_{i,k}\|^2 + \|\omega_{i,k}\|^2\right) \\
\overset{(321),(313)}{\leq} \quad & \exp\left(-\frac{\gamma\mu}{2}\right) V_k + 2\gamma\langle\theta_k, x^k - x^*\rangle + \frac{\gamma^2}{n^2}\sum_{i=1}^{n}\left(18\|\theta_{i,k}^u\|^2 + 14\|\omega_{i,k}^u\|^2\right) \\
& + \gamma^2 \sum_{i=1}^{n}\left(18\|\theta_{i,k}^b\|^2 + 14\|\omega_{i,k}^b\|^2\right) + \frac{32\gamma^2}{n^2}\sum_{j=2}^{n}\left\langle\sum_{i=1}^{j-1}\theta_{i,k}^u, \theta_{j,k}^u\right\rangle \\
& + \frac{24\gamma^2}{n^2}\sum_{j=2}^{n}\left\langle\sum_{i=1}^{j-1}\omega_{i,k}^u, \omega_{j,k}^u\right\rangle.
\end{aligned}
$$

Unrolling the recurrence, we get the result. $\qquad\square$

Next, we proceed with the full statement of our main result for DProx-clipped-SEG-shift in the quasi-strongly monotone case.

**Theorem H.2** (Case 1 from Theorem C.2). *Let Assumptions 1, 6 and 8 hold for $Q = B_{3n\sqrt{V}}(x^*)$, where $V \geq \|x^0 - x^*\|^2 + \frac{360000000 \gamma^2 \ln^2 \frac{48n(K+1)}{\beta}}{n^2}\sum_{i=1}^{n}\|F_i(x^*)\|^2$ and*

$$
0 < \gamma \leq \min\left\{\frac{1}{72 \cdot 10^6 \mu \ln^2 \frac{48n(K+1)}{\beta}}, \frac{1}{6L}, \frac{\sqrt{n}}{15000 L \ln \frac{48n(K+1)}{\beta}}, \frac{2\ln(B_K)}{\mu(K+1)}\right\}, \tag{375}
$$

$$
B_K = \max\left\{2, \frac{n^{\frac{2(\alpha-1)}{\alpha}}(K+1)^{\frac{2(\alpha-1)}{\alpha}}\mu^2 V}{3110400^{\frac{2}{\alpha}}\sigma^2 \ln^{\frac{2(\alpha-1)}{\alpha}}\left(\frac{48n(K+1)}{\beta}\right)\ln^2(B_K)}\right\} \tag{376}
$$

$$
= \mathcal{O}\left(\max\left\{2, \frac{n^{\frac{2(\alpha-1)}{\alpha}}K^{\frac{2(\alpha-1)}{\alpha}}\mu^2 V}{\sigma^2 \ln^{\frac{2(\alpha-1)}{\alpha}}\left(\frac{nK}{\beta}\right)\ln^2\left(\max\left\{2, \frac{n^{\frac{2(\alpha-1)}{\alpha}}K^{\frac{2(\alpha-1)}{\alpha}}\mu^2 V}{\sigma^2 \ln^{\frac{2(\alpha-1)}{\alpha}}\left(\frac{nK}{\beta}\right)}\right\}\right)}\right\}\right) \tag{377}
$$

$$
\lambda_k = \frac{n\exp(-\gamma\mu(1 + k/4))\sqrt{V}}{300\gamma \ln \frac{48n(K+1)}{\beta}}, \tag{378}
$$

$$
\nu = \gamma\mu \tag{379}
$$

*for some $K \geq 1$ and $\beta \in (0, 1]$. Then, after $K$ iterations of* DProx-clipped-SEG-shift, *the following inequality holds with probability at least $1 - \beta$:*

$$\|x^{K+1} - x^*\|^2 \leq 2\exp\left(-\frac{\gamma\mu(K+1)}{2}\right)V. \tag{380}$$

*In particular, when $\gamma$ equals the minimum from (304), then after $K$ iterations of* DProx-clipped-SEG-shift, *we have with probability at least $1 - \beta$ that*

$$\begin{aligned}
\|x^{K+1} - x^*\|^2 &= \mathcal{O}\left(\max\left\{V\exp\left(-\frac{K}{\ln^2\frac{nK}{\beta}}\right), V\exp\left(-\frac{\mu K}{L}\right),\right.\right. \\
&\qquad\left.\left. V\exp\left(-\frac{\mu\sqrt{n}K}{L\ln\frac{nK}{\beta}}\right), \frac{\sigma^2\ln^{\frac{2(\alpha-1)}{\alpha}}\left(\frac{nK}{\beta}\right)\ln^2 B_K}{n^{\frac{2(\alpha-1)}{\alpha}}K^{\frac{2(\alpha-1)}{\alpha}}\mu^2}\right\}\right),
\end{aligned}$$

*i.e., to achieve $\|x^K - x^*\|^2 \leq \varepsilon$ with probability at least $1 - \beta$* DProx-clipped-SEG-shift *needs*

$$\begin{aligned}
K = \mathcal{O}\left(\max\left\{\left(\frac{L}{\sqrt{n}\mu} + \ln\left(\frac{nL}{\mu\beta}\ln\frac{V}{\varepsilon}\right)\right)\ln\left(\frac{V}{\varepsilon}\right)\ln\left(\frac{nL}{\mu\beta}\ln\frac{V}{\varepsilon}\right),\right.\right. \\
\left.\left. \frac{L}{\mu}\ln\left(\frac{V}{\varepsilon}\right), \frac{1}{n}\left(\frac{\sigma^2}{\mu^2\varepsilon}\right)^{\frac{\alpha}{2(\alpha-1)}}\ln\left(\frac{n}{\beta}\left(\frac{\sigma^2}{\mu^2\varepsilon}\right)^{\frac{\alpha}{2(\alpha-1)}}\right)\ln^{\frac{\alpha}{\alpha-1}}(B_\varepsilon)\right\}\right)
\end{aligned} \tag{381}$$

*iterations/oracle calls per worker, where*

$$B_\varepsilon = \max\left\{2, \frac{V}{\varepsilon\ln\left(\frac{1}{\beta}\left(\frac{\sigma^2}{\mu^2\varepsilon}\right)^{\frac{\alpha}{2(\alpha-1)}}\right)}\right\}.$$

*Proof.* Similar to previous results, our proof is induction-based. To formulate the statement rigorously, we introduce probability event $E_k$ for each $k = 0, 1, \ldots, K+1$ as follows: inequalities

$$V_t \leq 2\exp\left(-\frac{\gamma\mu t}{2}\right)V \tag{382}$$

$$\left\|\frac{\gamma}{n}\sum_{i=1}^{r-1}\theta_{i,t-1}^u\right\| \leq \exp\left(-\frac{\gamma\mu(t-1)}{4}\right)\frac{\sqrt{V}}{2}, \tag{383}$$

$$\left\|\frac{\gamma}{n}\sum_{i=1}^{r-1}\omega_{i,t-1}^u\right\| \leq \exp\left(-\frac{\gamma\mu(t-1)}{4}\right)\frac{\sqrt{V}}{2} \tag{384}$$

hold for $t = 0, 1, \ldots, k$ and $r = 1, 2, \ldots, n$ simultaneously. We will prove by induction that $\mathbb{P}\{E_k\} \geq 1 - k\beta/(K+1)$ for all $k = 0, 1, \ldots, K+1$. The base of induction follows immediately by the definition of $V$. Next, we assume that the statement holds for $k = T - 1 \leq K$, i.e., $\mathbb{P}\{E_{T-1}\} \geq 1 - (T-1)\beta/(K+1)$. Let us show that it also holds for $k = T$, i.e., $\mathbb{P}\{E_T\} \geq 1 - T\beta/(K+1)$.

Similarly to the monotone case, one can show that due to our choice of the clipping level, we have that $E_{T-1}$ implies $x^t, \widetilde{x}^t \in B_{4n\sqrt{V}}(x^*)$ for $t = 0, \ldots, T - 1$. Indeed, for $t = 0, 1, \ldots, T - 1$ inequality

(382) gives $x^t \in B_{2\sqrt{V}}(x^*)$. Next, for $\widetilde{x}^t$, $t = 0, \ldots, T-1$ event $E_{T-1}$ implies

$$
\begin{aligned}
\|\widetilde{x}^t - x^*\| &= \| \operatorname{prox}_{\gamma\Psi}(x^t - \gamma\widetilde{g}^t) - \operatorname{prox}_{\gamma\Psi}(x^* - \gamma F(x^*))\| \\
&\leq \|x^t - x^* - \gamma(\widetilde{g}^t - F(x^*))\| \\
&\leq \|x^t - x^*\| + \gamma \left\| \frac{1}{n}\sum_{i=1}^n (\widetilde{h}_i^t - F_i(x^*)) + \frac{1}{n}\sum_{i=1}^n \tilde{\Delta}_i^t \right\| \\
&\leq 2\sqrt{V} + \gamma\sqrt{\frac{1}{n}\sum_{i=1}^n \|\widetilde{h}_i^t - F_i(x^*)\|^2} + \frac{\gamma}{n}\sum_{i=1}^n \|\tilde{\Delta}_i^t\| \\
&\leq 2\sqrt{V} + \frac{\sqrt{n}}{6000\ln\frac{48n(K+1)}{\beta}}\sqrt{V_t} + \gamma\lambda_t \\
&\overset{(382)}{\leq} \left( 2 + \frac{20n + \sqrt{2n}}{6000\ln\frac{48n(K+1)}{\beta}} \right)\sqrt{V} \leq 4n\sqrt{V}.
\end{aligned}
$$

This means that we can apply Lemma H.2: $E_{T-1}$ implies

$$
\begin{aligned}
V_T &\leq \exp\left(-\frac{\gamma\mu}{2}T\right)V + 2\gamma\sum_{t=0}^{T-1}\exp\left(-\frac{\gamma\mu}{2}(T-1-t)\right)\langle\theta_t, x^t - x^*\rangle \\
&\quad + \frac{\gamma^2}{n^2}\sum_{t=0}^{T-1}\sum_{i=1}^n\exp\left(-\frac{\gamma\mu}{2}(T-1-t)\right)\left(18\|\theta_{i,t}^u\|^2 + 14\|\omega_{i,t}^u\|^2\right) \\
&\quad + \gamma^2\sum_{t=0}^{T-1}\sum_{i=1}^n\exp\left(-\frac{\gamma\mu}{2}(T-1-t)\right)\left(18\|\theta_{i,t}^b\|^2 + 14\|\omega_{i,t}^b\|^2\right) \\
&\quad + \frac{32\gamma^2}{n^2}\sum_{t=0}^{T-1}\sum_{j=2}^n\exp\left(-\frac{\gamma\mu}{2}(T-1-t)\right)\left\langle\sum_{i=1}^{j-1}\theta_{i,t}^u, \theta_{j,t}^u\right\rangle \\
&\quad + \frac{24\gamma^2}{n^2}\sum_{t=0}^{T-1}\sum_{j=2}^n\exp\left(-\frac{\gamma\mu}{2}(T-1-t)\right)\left\langle\sum_{i=1}^{j-1}\omega_{i,t}^u, \omega_{j,t}^u\right\rangle.
\end{aligned}
$$

Before we proceed, we introduce a new notation:

$$
\eta_t = \begin{cases} x^t - x^*, & \text{if } \|x^t - x^*\| \leq \exp\left(-\frac{\gamma\mu t}{4}\right)\sqrt{2V}, \\ 0, & \text{otherwise,} \end{cases}
$$

for all $t = 0, 1, \ldots, T$. Random vectors $\{\eta_t\}_{t=0}^T$ are bounded almost surely:

$$
\|\eta_t\| \leq \exp\left(-\frac{\gamma\mu t}{4}\right)\sqrt{2V} \tag{385}
$$

for all $t = 0, 1, \ldots, T$. In addition, $\eta_t = x^t - x^*$ follows from $E_{T-1}$ for all $t = 0, 1, \ldots, T$ and, thus, $E_{T-1}$ implies

$$
\begin{aligned}
V_T \;\leq\; & \exp\left(-\frac{\gamma\mu}{2}T\right) V + \underbrace{\frac{2\gamma}{n} \sum_{l=0}^{T-1} \sum_{i=1}^{n} \exp\left(-\frac{\gamma\mu}{2}(T-1-l)\right) \langle \theta_{i,l}^u, \eta_l \rangle}_{\textcircled{1}} \\
& + \underbrace{2\gamma \sum_{l=0}^{T-1} \exp\left(-\frac{\gamma\mu}{2}(T-1-l)\right) \langle \theta_l^b, \eta_l \rangle}_{\textcircled{2}} \\
& + \underbrace{\frac{\gamma^2}{n^2} \sum_{l=0}^{T-1} \sum_{i=1}^{n} \exp\left(-\frac{\gamma\mu}{2}(T-1-l)\right) \left(18 \mathbb{E}_{\xi_{2,i}^l}\left[\|\theta_{i,l}^u\|^2\right] + 14 \mathbb{E}_{\xi_{1,i}^l}\left[\|\omega_{i,l}^u\|^2\right]\right)}_{\textcircled{3}} \\
& + \underbrace{\frac{\gamma^2}{n^2} \sum_{l=0}^{T-1} \sum_{i=1}^{n} \exp\left(-\frac{\gamma\mu}{2}(T-1-l)\right) \left(18\|\theta_{i,l}^u\|^2 + 14\|\omega_{i,l}^u\|^2 - 18 \mathbb{E}_{\xi_{2,i}^l}\left[\|\theta_{i,l}^u\|^2\right] - 14 \mathbb{E}_{\xi_{1,i}^l}\left[\|\omega_{i,l}^u\|^2\right]\right)}_{\textcircled{4}} \\
& + \underbrace{\frac{\gamma^2}{n^2} \sum_{l=0}^{T-1} \sum_{i=1}^{n} \exp\left(-\frac{\gamma\mu}{2}(T-1-l)\right) \left(18\|\theta_{i,l}^b\|^2 + 14\|\omega_{i,l}^b\|^2\right)}_{\textcircled{5}} \\
& + \underbrace{\frac{32\gamma^2}{n^2} \sum_{l=0}^{T-1} \sum_{j=2}^{n} \exp\left(-\frac{\gamma\mu}{2}(T-1-l)\right) \left\langle \sum_{i=1}^{j-1} \theta_{i,l}^u, \theta_{j,l}^u \right\rangle}_{\textcircled{6}} \\
& + \underbrace{\frac{24\gamma^2}{n^2} \sum_{l=0}^{T-1} \sum_{j=2}^{n} \exp\left(-\frac{\gamma\mu}{2}(T-1-l)\right) \left\langle \sum_{i=1}^{j-1} \omega_{i,l}^u, \omega_{j,l}^u \right\rangle}_{\textcircled{7}}.
\end{aligned}
\tag{386}
$$

To derive high-probability bounds for $\textcircled{1}, \textcircled{2}, \textcircled{3}, \textcircled{4}, \textcircled{5}, \textcircled{6}, \textcircled{7}$ we need to establish several useful inequalities related to $\theta_{i,l}^u, \theta_{i,l}^b, \omega_{i,l}^u, \omega_{i,l}^b$. First, by definition of clipping

$$
\|\theta_{i,l}^u\| \leq 2\lambda_l, \quad \|\omega_{i,l}^u\| \leq 2\lambda_l.
\tag{387}
$$

Next, we notice that $E_{T-1}$ implies

$$
\begin{aligned}
\|F_i(x^l) - \widetilde{h}_i^l\| \;\leq\;\; & \|F_i(x^l) - F_i(x^*)\| + \|\widetilde{h}_i^l - F_i(x^*)\| \\
\overset{(32)}{\leq}\;\; & L\|x^l - x^*\| + \sqrt{\sum_{j=1}^{n} \|\widetilde{h}_i^l - F_i(x^*)\|^2} \\
\leq\;\; & L\sqrt{V_l} + \frac{n\sqrt{V_l}}{6000\gamma \ln \frac{48n(K+1)}{\beta}} \\
\overset{(382)}{\leq}\;\; & \sqrt{2}\left(L + \frac{n}{6000\gamma \ln \frac{48n(K+1)}{\beta}}\right) \exp\left(-\frac{\gamma\mu l}{4}\right) \sqrt{V} \overset{(375),(378)}{\leq} \frac{\lambda_l}{2}
\end{aligned}
$$

and

$$\|F_i(\widetilde{x}^l) - \widehat{h}_i^l\| \leq \|F_i(\widetilde{x}^l) - F_i(x^*)\| + \|\widehat{h}_i^l - F_i(x^*)\|$$

$$\overset{(32)}{\leq} L\|\widetilde{x}^l - x^*\| + \sqrt{\sum_{j=1}^{n} \|\widehat{h}_i^l - F_i(x^*)\|^2}$$

$$\leq L\|\operatorname{prox}_{\gamma\Psi}(x^l - \gamma\widetilde{g}^l) - \operatorname{prox}_{\gamma\Psi}(x^* - \gamma F(x^*))\| + \frac{n\sqrt{V_l}}{6000\gamma \ln \frac{48n(K+1)}{\beta}}$$

$$\leq L\|x^l - x^* - \gamma(\widetilde{g}^l - F(x^*))\| + \frac{n\sqrt{V_l}}{6000\gamma \ln \frac{48n(K+1)}{\beta}}$$

$$\leq L\|x^l - x^*\| + L\gamma \left\| \frac{1}{n}\sum_{i=1}^{n}(\widetilde{h}_i^l - F_i(x^*)) + \frac{1}{n}\sum_{i=1}^{n}\tilde{\Delta}_i^l \right\| + \frac{n\sqrt{V_l}}{6000\gamma \ln \frac{48n(K+1)}{\beta}}$$

$$\leq \left( L + \frac{n}{6000\gamma \ln \frac{48n(K+1)}{\beta}} \right)\sqrt{V_l} + L\gamma\sqrt{\frac{1}{n}\sum_{i=1}^{n}\|\widetilde{h}_i^l - F_i(x^*)\|^2} + \frac{L\gamma}{n}\sum_{i=1}^{n}\|\tilde{\Delta}_i^l\|$$

$$\leq \left( L + \frac{n + L\gamma\sqrt{n}}{6000\gamma \ln \frac{48n(K+1)}{\beta}} \right)\sqrt{V_l} + L\gamma\lambda_t$$

$$\overset{(382)}{\leq} \sqrt{2}\left( L + \frac{n + L\gamma\sqrt{n}}{6000\gamma \ln \frac{48n(K+1)}{\beta}} \right)\exp\left(-\frac{\gamma\mu l}{4}\right)\sqrt{V} + L\gamma\lambda_l \overset{(375),(378)}{\leq} \frac{\lambda_l}{2}$$

for $l = 0, 1, \ldots, T-1$ and $i \in [n]$. Therefore, one can apply Lemma B.2 and get

$$\|\theta_l^b\| \leq \frac{1}{n}\sum_{i=1}^{n}\|\theta_{i,l}^b\| \leq \frac{2^\alpha \sigma^\alpha}{\lambda_l^{\alpha-1}}, \quad \|\omega_l^b\| \leq \frac{1}{n}\sum_{i=1}^{n}\|\omega_{i,l}^b\| \leq \frac{2^\alpha \sigma^\alpha}{\lambda_l^{\alpha-1}}, \tag{388}$$

$$\mathbb{E}_{\xi_{2,i}^l}\left[\|\theta_{i,l}^u\|^2\right] \leq 18\lambda_l^{2-\alpha}\sigma^\alpha, \quad \mathbb{E}_{\xi_{1,i}^l}\left[\|\omega_{i,l}^u\|^2\right] \leq 18\lambda_l^{2-\alpha}\sigma^\alpha, \tag{389}$$

for all $l = 0, 1, \ldots, T-1$ and $i \in [n]$.

**Upper bound for ①.** To estimate this sum, we will use Bernstein's inequality. The summands have conditional expectations equal to zero:

$$\mathbb{E}_{\xi_{2,i}^l}\left[\frac{2\gamma}{n}\exp\left(-\frac{\gamma\mu}{2}(T-1-l)\right)\langle\eta_l, \theta_{i,l}^u\rangle\right] = \frac{2\gamma}{n}\exp\left(-\frac{\gamma\mu}{2}(T-1-l)\right)\left\langle\eta_l, \mathbb{E}_{\xi_{2,i}^l}[\theta_{i,l}^u]\right\rangle = 0.$$

Moreover, for all $l = 0, \ldots, T-1$ random vectors $\{\theta_{i,l}^u\}_{i=1}^n$ are independent. Thus, sequence $\left\{\frac{2\gamma}{n}\exp\left(-\frac{\gamma\mu}{2}(T-1-l)\right)\langle\eta_l, \theta_{i,l}^u\rangle\right\}_{l,i=0,1}^{T-1,n}$ is a martingale difference sequence. Next, the summands are bounded:

$$\left|\frac{2\gamma}{n}\exp\left(-\frac{\gamma\mu}{2}(T-1-l)\right)\langle\eta_l, \theta_{i,l}^u\rangle\right| \leq \frac{2\gamma}{n}\exp\left(-\frac{\gamma\mu}{2}(T-1-l)\right)\|\eta_l\|\cdot\|\theta_{i,l}^u\|$$

$$\overset{(385),(387)}{\leq} \frac{2\sqrt{2V}\gamma\exp\left(-\frac{\gamma\mu(T-1)}{2}\right)}{n}\exp\left(\frac{\gamma\mu l}{4}\right)\lambda_l$$

$$\overset{(305)}{=} \frac{\exp\left(-\frac{\gamma\mu T}{2}\right)V}{100\ln\frac{48n(K+1)}{\beta}} \overset{\text{def}}{=} c. \tag{390}$$

Finally, conditional variances $\sigma_{i,l}^2 \overset{\text{def}}{=} \mathbb{E}_{\xi_{2,i}^l}\left[\frac{4\gamma^2}{n^2}\exp\left(-\gamma\mu(T-1-l)\right)\langle\eta_l, \theta_{i,l}^u\rangle^2\right]$ of the summands are bounded:

$$\sigma_{i,l}^2 \leq \mathbb{E}_{\xi_{2,i}^l}\left[\frac{4\gamma^2}{n^2}\exp\left(-\gamma\mu(T-1-l)\right)\|\eta_l\|^2\cdot\|\theta_{i,l}^u\|^2\right]$$

$$\overset{(385)}{\leq} \frac{8\gamma^2 V\exp\left(-\gamma\mu\left(T-1-\frac{l}{2}\right)\right)}{n^2}\mathbb{E}_{\xi_{2,i}^l}\left[\|\theta_{i,l}^u\|^2\right]. \tag{391}$$

Applying Bernstein's inequality (Lemma B.1) with $X_{i,l} = \frac{2\gamma}{n} \exp\left(-\frac{\gamma\mu}{2}(T-1-l)\right) \langle \eta_l, \theta_{i,l}^u \rangle$, constant $c$ defined in (390), $b = \frac{\exp\left(-\frac{\gamma\mu T}{2}\right)V}{100}$, $G = \frac{\exp(-\gamma\mu T)V^2}{60000 \ln \frac{48n(K+1)}{\beta}}$, we get

$$\mathbb{P}\left\{ |①| > \frac{\exp\left(-\frac{\gamma\mu T}{2}\right)V}{100} \text{ and } \sum_{l=0}^{T-1}\sum_{i=1}^{n} \sigma_{i,l}^2 \leq \frac{\exp(-\gamma\mu T)V^2}{60000 \ln \frac{48n(K+1)}{\beta}} \right\} \leq 2\exp\left(-\frac{b^2}{2G + \frac{2cb}{3}}\right)$$

$$= \frac{\beta}{24n(K+1)}.$$

The above is equivalent to $\mathbb{P}\{E_①\} \geq 1 - \frac{\beta}{24n(K+1)}$ for

$$E_① = \left\{ \text{either } \sum_{l=0}^{T-1}\sum_{i=1}^{n} \sigma_{i,l}^2 > \frac{\exp(-\gamma\mu T)V^2}{60000 \ln \frac{48n(K+1)}{\beta}} \text{ or } |①| \leq \frac{\exp\left(-\frac{\gamma\mu T}{2}\right)V}{100} \right\}. \tag{392}$$

Moreover, $E_{T-1}$ implies

$$\sum_{l=0}^{T-1}\sum_{i=1}^{n} \sigma_{i,l}^2 \overset{(391)}{\leq} \frac{8\gamma^2 V \exp(-\gamma\mu(T-1))}{n^2} \sum_{l=0}^{T-1} \exp\left(\frac{\gamma\mu l}{2}\right) \sum_{i=1}^{n} \mathbb{E}_{\xi_{2,i}^l}\left[\|\theta_{i,l}^u\|^2\right]$$

$$\overset{(389),T\leq K+1}{\leq} \frac{144\gamma^2 V \exp(-\gamma\mu(T-1))\sigma^\alpha}{n} \sum_{l=0}^{T-1} \exp\left(\frac{\gamma\mu l}{2}\right) \lambda_l^{2-\alpha}$$

$$\overset{(378)}{\leq} \frac{144\gamma^\alpha V^{2-\frac{\alpha}{2}} \exp(-\gamma\mu T)\sigma^\alpha}{6000^{2-\alpha} n^{\alpha-1} \ln^{2-\alpha} \frac{48n(K+1)}{\beta}} \sum_{l=0}^{T-1} \exp\left(\frac{\gamma\mu l\alpha}{4}\right)$$

$$\leq \frac{144\gamma^\alpha V^{2-\frac{\alpha}{2}} \exp(-\gamma\mu T)\sigma^\alpha (K+1) \exp\left(\frac{\gamma\mu K\alpha}{4}\right)}{6000^{2-\alpha} n^{\alpha-1} \ln^{2-\alpha} \frac{48n(K+1)}{\beta}}$$

$$\overset{(375)}{\leq} \frac{\exp(-\gamma\mu T)V^2}{60000 \ln \frac{48n(K+1)}{\beta}}. \tag{393}$$

**Upper bound for ②.** Probability event $E_{T-1}$ implies

$$② \leq 2\gamma \sum_{l=0}^{T-1} \exp\left(-\frac{\gamma\mu(T-1-l)}{2}\right) \|\eta_l\| \cdot \|\theta_l^b\| \overset{(385),(388)}{\leq} \tag{394}$$

$$\leq 2^{\alpha+1}\gamma\sigma^\alpha\sqrt{2V} \exp\left(-\frac{\gamma\mu(T-1)}{2}\right) \sum_{l=0}^{T-1} \exp\left(\frac{\gamma\mu l}{4}\right) \frac{1}{\lambda_l^{\alpha-1}}$$

$$\overset{(378)}{\leq} \frac{2^{\alpha+1} \cdot 120^{\alpha-1} \exp\left(-\frac{\gamma\mu T}{2}\right)(K+1)\exp\left(\frac{\gamma\mu K\alpha}{4}\right)\gamma^\alpha\sigma^\alpha \ln^{\alpha-1} \frac{48n(K+1)}{\beta}}{n^{\alpha-1}V^{\frac{\alpha}{2}-1}}$$

$$\overset{(375)}{\leq} \frac{3\exp\left(-\frac{\gamma\mu T}{2}\right)V}{100}. \tag{395}$$

**Upper bound for ③.** Probability event $E_{T-1}$ implies

$$\frac{18\gamma^2}{n^2} \sum_{l=0}^{T-1}\sum_{i=1}^{n} \exp\left(-\frac{\gamma\mu(T-1-l)}{2}\right) \mathbb{E}_{\xi_{2,i}^l}\left[\|\theta_{i,l}^u\|^2\right] \overset{(393)}{\leq} \frac{\exp\left(-\frac{\gamma\mu T}{2}\right)V}{100}$$

and, similarly,

$$\frac{14\gamma^2}{n^2} \sum_{l=0}^{T-1}\sum_{i=1}^{n} \exp\left(-\frac{\gamma\mu(T-1-l)}{2}\right) \mathbb{E}_{\xi_{1,i}^l}\left[\|\omega_{i,l}^u\|^2\right] \overset{(393)}{\leq} \frac{\exp\left(-\frac{\gamma\mu T}{2}\right)V}{100}$$

that give

$$③ \leq \frac{\exp\left(-\frac{\gamma\mu T}{2}\right)V}{50}. \tag{396}$$

**Upper bound for ④.** To estimate this sum, we will use Bernstein's inequality. The summands have conditional expectations equal to zero:

$$\frac{2\gamma^2}{n^2}\mathbb{E}_{\xi_{1,i}^l,\xi_{2,i}^l}\left[18\|\theta_{i,l}^u\|^2 + 14\|\omega_{i,l}^u\|^2 - 18\mathbb{E}_{\xi_{2,i}^l}\left[\|\theta_{i,l}^u\|^2\right] - 14\mathbb{E}_{\xi_{1,i}^l}\left[\|\omega_{i,l}^u\|^2\right]\right] = 0.$$

Moreover, for all $l = 0,\ldots,T-1$ random vectors $\{\theta_{i,l}^u\}_{i=1}^n$, $\{\omega_{i,l}^u\}_{i=1}^n$ are independent. Thus, sequence $\left\{\frac{2\gamma^2}{n^2}\exp\left(-\frac{\gamma\mu}{2}(T-1-l)\right)\left(18\|\theta_{i,l}^u\|^2 + 14\|\omega_{i,l}^u\|^2 - 18\mathbb{E}_{\xi_{2,i}^l}\left[\|\theta_{i,l}^u\|^2\right] - 14\mathbb{E}_{\xi_{1,i}^l}\left[\|\omega_{i,l}^u\|^2\right]\right)\right\}_{l,i=0,1}^{T-1,n}$ is a martingale difference sequence. Next, the summands are bounded:

$$\frac{2\gamma^2}{n^2}\exp\left(-\frac{\gamma\mu}{2}(T-1-l)\right)\left|18\|\theta_{i,l}^u\|^2 + 14\|\omega_{i,l}^u\|^2 - 18\mathbb{E}_{\xi_{2,i}^l}\left[\|\theta_{i,l}^u\|^2\right] - 14\mathbb{E}_{\xi_{1,i}^l}\left[\|\omega_{i,l}^u\|^2\right]\right|$$

$$\overset{(387)}{\leq} \frac{256\exp\left(-\frac{\gamma\mu}{2}(T-1-l)\right)\gamma^2\lambda_l^2}{n^2}$$

$$\overset{(378)}{\leq} \frac{\exp\left(-\frac{\gamma\mu T}{2}\right)V}{6\ln\frac{48n(K+1)}{\beta}} \overset{\text{def}}{=} c. \tag{397}$$

Finally, conditional variances

$$\widetilde{\sigma}_{i,l}^2 \overset{\text{def}}{=} \frac{4\gamma^4}{n^4}\exp\left(-\gamma\mu(T-1-l)\right)\mathbb{E}_{\xi_{1,i}^l,\xi_{2,i}^l}\left[\left|18\|\theta_{i,l}^u\|^2 + 14\|\omega_{i,l}^u\|^2 - 18\mathbb{E}_{\xi_{2,i}^l}\left[\|\theta_{i,l}^u\|^2\right] - 14\mathbb{E}_{\xi_{1,i}^l}\left[\|\omega_{i,l}^u\|^2\right]\right|^2\right]$$

of the summands are bounded:

$$\widetilde{\sigma}_{i,l}^2 \overset{(397)}{\leq} \frac{\gamma^2\exp\left(-\frac{\gamma\mu}{2}(2T-1-l)\right)V}{3n^2\ln\frac{48n(K+1)}{\beta}}$$

$$\times \mathbb{E}_{\xi_{1,i}^l,\xi_{2,i}^l}\left[\left|18\|\theta_{i,l}^u\|^2 + 14\|\omega_{i,l}^u\|^2 - 18\mathbb{E}_{\xi_{2,i}^l}\left[\|\theta_{i,l}^u\|^2\right] - 14\mathbb{E}_{\xi_{1,i}^l}\left[\|\omega_{i,l}^u\|^2\right]\right|\right]$$

$$\leq \frac{4\gamma^2\exp\left(-\frac{\gamma\mu}{2}(2T-1-l)\right)V}{3n^2\ln\frac{48n(K+1)}{\beta}}\mathbb{E}_{\xi_{1,i}^l,\xi_{2,i}^l}\left[9\|\theta_{i,l}^u\|^2 + 7\|\omega_{i,l}^u\|^2\right]. \tag{398}$$

Applying Bernstein's inequality (Lemma B.1) with $X_{i,l} = \frac{2\gamma^2}{n^2}\exp\left(-\frac{\gamma\mu}{2}(T-1-l)\right)\left(18\|\theta_{i,l}^u\|^2 + 14\|\omega_{i,l}^u\|^2 - 18\mathbb{E}_{\xi_{2,i}^l}\left[\|\theta_{i,l}^u\|^2\right] - 14\mathbb{E}_{\xi_{1,i}^l}\left[\|\omega_{i,l}^u\|^2\right]\right)$, constant $c$ defined in (397), $b = \frac{\exp\left(-\frac{\gamma\mu T}{2}\right)V}{6}$, $G = \frac{\exp(-\gamma\mu T)V^2}{216\ln\frac{48n(K+1)}{\beta}}$, we get

$$\mathbb{P}\left\{|④| > \frac{\exp\left(-\frac{\gamma\mu T}{2}\right)V}{6} \text{ and } \sum_{l=0}^{T-1}\sum_{i=1}^n\widetilde{\sigma}_{i,l}^2 \leq \frac{\exp\left(-\gamma\mu T\right)V^2}{216\ln\frac{48n(K+1)}{\beta}}\right\} \leq 2\exp\left(-\frac{b^2}{2G + 2cb/3}\right)$$

$$= \frac{\beta}{24n(K+1)}.$$

The above is equivalent to $\mathbb{P}\{E_④\} \geq 1 - \frac{\beta}{24n(K+1)}$ for

$$E_④ = \left\{\text{either } \sum_{l=0}^{T-1}\sum_{i=1}^n\widetilde{\sigma}_{i,l}^2 > \frac{\exp\left(-\gamma\mu T\right)V^2}{216\ln\frac{48n(K+1)}{\beta}} \text{ or } |④| \leq \frac{\exp\left(-\frac{\gamma\mu T}{2}\right)V}{6}\right\}. \tag{399}$$

Moreover, $E_{T-1}$ implies

$$\sum_{l=0}^{T-1}\sum_{i=1}^n\widetilde{\sigma}_{i,l}^2 \overset{(398)}{\leq} \frac{4\exp\left(-\gamma\mu(T-\frac{1}{2})\right)\gamma^2V}{3n^2\ln\frac{48n(K+1)}{\beta}}\sum_{l=0}^{T-1}\exp\left(\frac{\gamma\mu l}{2}\right)\sum_{i=1}^n\mathbb{E}_{\xi_{1,i}^l,\xi_{2,i}^l}\left[9\|\theta_{i,l}^u\|^2 + 7\|\omega_{i,l}^u\|^2\right]$$

$$\overset{(393)}{\leq} \frac{\exp\left(-\gamma\mu T\right)V}{216\ln\frac{48n(K+1)}{\beta}}. \tag{400}$$

**Upper bound for ⑤.** Probability event $E_{T-1}$ implies

$$⑤ \overset{(388)}{\leq} 32\gamma^2 \sum_{l=0}^{T-1} \exp\left(-\frac{\gamma\mu}{2}(T-1-l)\right) \frac{2^{2\alpha}\sigma^{2\alpha}}{\lambda_l^{2\alpha-2}} \overset{(378),(375)}{\leq} \frac{\exp\left(-\frac{\gamma\mu T}{2}\right)V}{6}. \quad (401)$$

**Upper bounds for ⑥ and ⑦.** These sums require more refined analysis. We introduce new vectors:

$$\zeta_j^l = \begin{cases} \frac{\gamma}{n}\sum_{i=1}^{j-1}\theta_{i,l}^u, & \text{if } \left\|\frac{\gamma}{n}\sum_{i=1}^{j-1}\theta_{i,l}^u\right\| \leq \exp\left(-\frac{\gamma\mu l}{4}\right)\frac{\sqrt{V}}{2}, \\ 0, & \text{otherwise,} \end{cases} \quad (402)$$

$$\delta_j^l = \begin{cases} \frac{\gamma}{n}\sum_{i=1}^{j-1}\omega_{i,l}^u, & \text{if } \left\|\frac{\gamma}{n}\sum_{i=1}^{j-1}\omega_{i,l}^u\right\| \leq \exp\left(-\frac{\gamma\mu l}{4}\right)\frac{\sqrt{V}}{2}, \\ 0, & \text{otherwise,} \end{cases} \quad (403)$$

for all $j \in [n]$ and $l = 0, \ldots, T-1$. Then, by definition

$$\|\zeta_j^l\| \leq \exp\left(-\frac{\gamma\mu l}{4}\right)\frac{\sqrt{V}}{2}, \quad \|\delta_j^l\| \leq \exp\left(-\frac{\gamma\mu l}{4}\right)\frac{\sqrt{V}}{2} \quad (404)$$

and

$$⑥ = \underbrace{\frac{32\gamma}{n}\sum_{l=0}^{T-1}\sum_{j=2}^{n}\exp\left(-\frac{\gamma\mu}{2}(T-1-l)\right)\langle\zeta_j^l,\theta_{j,l}^u\rangle}_{⑥'}$$

$$+ \frac{32\gamma}{n}\sum_{l=0}^{T-1}\sum_{j=2}^{n}\exp\left(-\frac{\gamma\mu}{2}(T-1-l)\right)\left\langle\frac{\gamma}{n}\sum_{i=1}^{j-1}\theta_{i,l}^u - \zeta_j^l,\theta_{j,l}^u\right\rangle, \quad (405)$$

$$⑦ = \underbrace{\frac{24\gamma}{n}\sum_{l=0}^{T-1}\sum_{j=2}^{n}\exp\left(-\frac{\gamma\mu}{2}(T-1-l)\right)\langle\delta_j^l,\omega_{j,l}^u\rangle}_{⑦'}$$

$$+ \frac{24\gamma}{n}\sum_{l=0}^{T-1}\sum_{j=2}^{n}\exp\left(-\frac{\gamma\mu}{2}(T-1-l)\right)\left\langle\frac{\gamma}{n}\sum_{i=1}^{j-1}\omega_{i,l}^u - \delta_j^l,\omega_{j,l}^u\right\rangle. \quad (406)$$

We also note here that $E_{T-1}$ implies

$$\frac{32\gamma}{n}\sum_{l=0}^{T-1}\sum_{j=2}^{n}\exp\left(-\frac{\gamma\mu}{2}(T-1-l)\right)\left\langle\frac{\gamma}{n}\sum_{i=1}^{j-1}\theta_{i,l}^u - \zeta_j^l,\theta_{j,l}^u\right\rangle$$

$$= \frac{32\gamma}{n}\sum_{j=2}^{n}\left\langle\frac{\gamma}{n}\sum_{i=1}^{j-1}\theta_{i,T-1}^u - \zeta_j^{T-1},\theta_{j,T-1}^u\right\rangle, \quad (407)$$

$$\frac{24\gamma}{n}\sum_{l=0}^{T-1}\sum_{j=2}^{n}\exp\left(-\frac{\gamma\mu}{2}(T-1-l)\right)\left\langle\frac{\gamma}{n}\sum_{i=1}^{j-1}\omega_{i,l}^u - \delta_j^l,\omega_{j,l}^u\right\rangle$$

$$= \frac{24\gamma}{n}\sum_{j=2}^{n}\left\langle\frac{\gamma}{n}\sum_{i=1}^{j-1}\omega_{i,T-1}^u - \delta_j^{T-1},\omega_{j,T-1}^u\right\rangle. \quad (408)$$

**Upper bound for ⑥'.** To estimate this sum, we will use Bernstein's inequality. The summands have conditional expectations equal to zero:

$$\mathbb{E}_{\xi_{2,j}^l}\left[\frac{32\gamma}{n}\exp\left(-\frac{\gamma\mu}{2}(T-1-l)\right)\langle\zeta_j^l,\theta_{j,l}^u\rangle\right] = \frac{32\gamma}{n}\exp\left(-\frac{\gamma\mu}{2}(T-1-l)\right)\left\langle\zeta_j^l,\mathbb{E}_{\xi_{2,j}^l}[\theta_{j,l}^u]\right\rangle = 0.$$

Moreover, for all $l = 0, \ldots, T-1$ random vectors $\{\theta_{j,l}^u\}_{j=1}^n$ are independent. Thus, sequence $\left\{ \frac{32\gamma}{n} \exp\left(-\frac{\gamma\mu}{2}(T-1-l)\right) \langle \zeta_j^l, \theta_{j,l}^u \rangle \right\}_{l,j=0,1}^{T-1,n}$ is a martingale difference sequence. Next, the summands are bounded:

$$
\left| \frac{32\gamma}{n} \exp\left(-\frac{\gamma\mu}{2}(T-1-l)\right) \langle \zeta_j^l, \theta_{j,l}^u \rangle \right| \quad \leq \quad \frac{32\gamma}{n} \exp\left(-\frac{\gamma\mu}{2}(T-1-l)\right) \|\zeta_j^l\| \cdot \|\theta_{j,l}^u\|
$$

$$
\overset{(404),(387)}{\leq} \quad \frac{16\sqrt{V}\gamma \exp\left(-\frac{\gamma\mu(T-1)}{2}\right)}{n} \exp\left(\frac{\gamma\mu l}{4}\right) \lambda_l
$$

$$
\overset{(305)}{=} \quad \frac{\exp\left(-\frac{\gamma\mu T}{2}\right) V}{10 \ln \frac{48n(K+1)}{\beta}} \overset{\text{def}}{=} c. \tag{409}
$$

Finally, conditional variances $\widehat{\sigma}_{j,l}^2 \overset{\text{def}}{=} \mathbb{E}_{\xi_{2,j}^l} \left[ \frac{1024\gamma^2}{n^2} \exp\left(-\gamma\mu(T-1-l)\right) \langle \zeta_j^l, \theta_{j,l}^u \rangle^2 \right]$ of the summands are bounded:

$$
\widehat{\sigma}_{j,l}^2 \quad \leq \quad \mathbb{E}_{\xi_{2,j}^l} \left[ \frac{1024\gamma^2}{n^2} \exp\left(-\gamma\mu(T-1-l)\right) \|\zeta_j^l\|^2 \cdot \|\theta_{j,l}^u\|^2 \right]
$$

$$
\overset{(404)}{\leq} \quad \frac{256\gamma^2 V \exp\left(-\gamma\mu\left(T-1-\frac{l}{2}\right)\right)}{n^2} \mathbb{E}_{\xi_{2,j}^l} \left[ \|\theta_{j,l}^u\|^2 \right]. \tag{410}
$$

Applying Bernstein's inequality (Lemma B.1) with $X_{j,l} = \frac{32\gamma}{n} \exp\left(-\frac{\gamma\mu}{2}(T-1-l)\right) \langle \zeta_j^l, \theta_{j,l}^u \rangle$, constant $c$ defined in (409), $b = \frac{\exp\left(-\frac{\gamma\mu T}{2}\right) V}{10}$, $G = \frac{\exp(-\gamma\mu T)V^2}{600 \ln \frac{48n(K+1)}{\beta}}$, we get

$$
\mathbb{P}\left\{ |\textcircled{6}'| > \frac{\exp\left(-\frac{\gamma\mu T}{2}\right) V}{10} \text{ and } \sum_{l=0}^{T-1} \sum_{j=2}^n \widehat{\sigma}_{j,l}^2 \leq \frac{\exp\left(-\gamma\mu T\right) V^2}{600 \ln \frac{48n(K+1)}{\beta}} \right\} \quad \leq \quad 2\exp\left(-\frac{b^2}{2G + 2cb/3}\right)
$$

$$
= \quad \frac{\beta}{24n(K+1)}.
$$

The above is equivalent to $\mathbb{P}\{E_{\textcircled{6}'}\} \geq 1 - \frac{\beta}{24n(K+1)}$ for

$$
E_{\textcircled{6}'} = \left\{ \text{either } \sum_{l=0}^{T-1} \sum_{j=2}^n \widehat{\sigma}_{j,l}^2 > \frac{\exp\left(-\gamma\mu T\right) V^2}{600 \ln \frac{48n(K+1)}{\beta}} \text{ or } |\textcircled{6}'| \leq \frac{\exp\left(-\frac{\gamma\mu T}{2}\right) V}{10} \right\}. \tag{411}
$$

Moreover, $E_{T-1}$ implies

$$
\sum_{l=0}^{T-1} \sum_{j=2}^n \widehat{\sigma}_{j,l}^2 \quad \overset{(410)}{\leq} \quad \frac{256\gamma^2 V \exp\left(-\gamma\mu\left(T-1\right)\right)}{n^2} \sum_{l=0}^{T-1} \exp\left(\frac{\gamma\mu l}{2}\right) \sum_{i=1}^n \mathbb{E}_{\xi_{2,i}^l} \left[ \|\theta_{i,l}^u\|^2 \right]
$$

$$
\overset{(389), T \leq K+1}{\leq} \quad \frac{4608\gamma^2 V \exp\left(-\gamma\mu\left(T-1\right)\right) \sigma^\alpha}{n} \sum_{l=0}^{T-1} \exp\left(\frac{\gamma\mu l}{2}\right) \lambda_l^{2-\alpha}
$$

$$
\overset{(378)}{\leq} \quad \frac{4608\gamma^\alpha V^{2-\frac{\alpha}{2}} \exp\left(-\gamma\mu T\right) \sigma^\alpha}{300^{2-\alpha} n^{\alpha-1} \ln^{2-\alpha} \frac{48n(K+1)}{\beta}} \sum_{l=0}^{T-1} \exp\left(\frac{\gamma\mu l\alpha}{4}\right)
$$

$$
\leq \quad \frac{4608\gamma^\alpha V^{2-\frac{\alpha}{2}} \exp\left(-\gamma\mu T\right) \sigma^\alpha (K+1) \exp\left(\frac{\gamma\mu K\alpha}{4}\right)}{300^{2-\alpha} n^{\alpha-1} \ln^{2-\alpha} \frac{48n(K+1)}{\beta}}
$$

$$
\overset{(375)}{\leq} \quad \frac{\exp\left(-\gamma\mu T\right) V^2}{600 \ln \frac{48n(K+1)}{\beta}}. \tag{412}
$$

**Upper bound for $\mathcal{T}'$.** To estimate this sum, we will use Bernstein's inequality. The summands have conditional expectations equal to zero:

$$\mathbb{E}_{\xi_{1,j}^l}\left[\frac{24\gamma}{n}\exp\left(-\frac{\gamma\mu}{2}(T-1-l)\right)\langle\delta_j^l,\omega_{j,l}^u\rangle\right] = \frac{24\gamma}{n}\exp\left(-\frac{\gamma\mu}{2}(T-1-l)\right)\left\langle\delta_j^l,\mathbb{E}_{\xi_{1,j}^l}[\omega_{j,l}^u]\right\rangle = 0.$$

Moreover, for all $l = 0,\ldots,T-1$ random vectors $\{\omega_{j,l}^u\}_{j=1}^n$ are independent. Thus, sequence $\left\{\frac{24\gamma}{n}\exp\left(-\frac{\gamma\mu}{2}(T-1-l)\right)\langle\delta_j^l,\omega_{j,l}^u\rangle\right\}_{l,j=0,1}^{T-1,n}$ is a martingale difference sequence. Next, the summands are bounded:

$$\left|\frac{24\gamma}{n}\exp\left(-\frac{\gamma\mu}{2}(T-1-l)\right)\langle\delta_j^l,\omega_{j,l}^u\rangle\right| \leq \frac{24\gamma}{n}\exp\left(-\frac{\gamma\mu}{2}(T-1-l)\right)\|\delta_j^l\|\cdot\|\omega_{j,l}^u\|$$

$$\overset{(404),(387)}{\leq} \frac{12\sqrt{V}\gamma\exp\left(-\frac{\gamma\mu(T-1)}{2}\right)}{n}\exp\left(\frac{\gamma\mu l}{4}\right)\lambda_l$$

$$\overset{(305)}{=} \frac{\exp\left(-\frac{\gamma\mu T}{2}\right)V}{10\ln\frac{48n(K+1)}{\beta}} \overset{\text{def}}{=} c. \tag{413}$$

Finally, conditional variances $(\sigma_{j,l}')^2 \overset{\text{def}}{=} \mathbb{E}_{\xi_{1,j}^l}\left[\frac{576\gamma^2}{n^2}\exp(-\gamma\mu(T-1-l))\langle\delta_j^l,\omega_{j,l}^u\rangle^2\right]$ of the summands are bounded:

$$(\sigma_{j,l}')^2 \leq \mathbb{E}_{\xi_{1,j}^l}\left[\frac{576\gamma^2}{n^2}\exp(-\gamma\mu(T-1-l))\|\delta_j^l\|^2\cdot\|\omega_{j,l}^u\|^2\right]$$

$$\overset{(404)}{\leq} \frac{288\gamma^2 V\exp\left(-\gamma\mu\left(T-1-\frac{l}{2}\right)\right)}{n^2}\mathbb{E}_{\xi_{1,j}^l}\left[\|\omega_{j,l}^u\|^2\right]. \tag{414}$$

Applying Bernstein's inequality (Lemma B.1) with $X_{j,l} = \frac{24\gamma}{n}\exp\left(-\frac{\gamma\mu}{2}(T-1-l)\right)\langle\delta_j^l,\omega_{j,l}^u\rangle$, constant $c$ defined in (409), $b = \frac{\exp\left(-\frac{\gamma\mu T}{2}\right)V}{10}$, $G = \frac{\exp(-\gamma\mu T)V^2}{600\ln\frac{48n(K+1)}{\beta}}$, we get

$$\mathbb{P}\left\{|\mathcal{T}'| > \frac{\exp\left(-\frac{\gamma\mu T}{2}\right)V}{10} \text{ and } \sum_{l=0}^{T-1}\sum_{j=2}^n(\sigma_{j,l}')^2 \leq \frac{\exp(-\gamma\mu T)V^2}{600\ln\frac{48n(K+1)}{\beta}}\right\} \leq 2\exp\left(-\frac{b^2}{2G+\frac{2cb}{3}}\right)$$

$$= \frac{\beta}{24n(K+1)}.$$

The above is equivalent to $\mathbb{P}\{E_{\text{⑥}'}\} \geq 1 - \frac{\beta}{24n(K+1)}$ for

$$E_{\mathcal{T}'} = \left\{\text{either } \sum_{l=0}^{T-1}\sum_{j=2}^n(\sigma_{j,l}')^2 > \frac{\exp(-\gamma\mu T)V^2}{600\ln\frac{48n(K+1)}{\beta}} \text{ or } |\mathcal{T}'| \leq \frac{\exp\left(-\frac{\gamma\mu T}{2}\right)V}{10}\right\}. \tag{415}$$

Moreover, $E_{T-1}$ implies

$$\sum_{l=0}^{T-1}\sum_{j=2}^n(\sigma_{j,l}')^2 \overset{(414)}{\leq} \frac{288\gamma^2 V\exp\left(-\gamma\mu\left(T-1\right)\right)}{n^2}\sum_{l=0}^{T-1}\exp\left(\frac{\gamma\mu l}{2}\right)\sum_{i=1}^n\mathbb{E}_{\xi_{2,i}^l}\left[\|\theta_{i,l}^u\|^2\right]$$

$$\overset{(389),T\leq K+1}{\leq} \frac{5184\gamma^2 V\exp\left(-\gamma\mu\left(T-1\right)\right)\sigma^\alpha}{n}\sum_{l=0}^{T-1}\exp\left(\frac{\gamma\mu l}{2}\right)\lambda_l^{2-\alpha}$$

$$\overset{(378)}{\leq} \frac{5184\gamma^\alpha V^{2-\frac{\alpha}{2}}\exp\left(-\gamma\mu T\right)\sigma^\alpha}{300^{2-\alpha}n^{\alpha-1}\ln^{2-\alpha}\frac{48n(K+1)}{\beta}}\sum_{l=0}^{T-1}\exp\left(\frac{\gamma\mu l\alpha}{4}\right)$$

$$\leq \frac{5184\gamma^\alpha V^{2-\frac{\alpha}{2}}\exp\left(-\gamma\mu T\right)\sigma^\alpha(K+1)\exp\left(\frac{\gamma\mu K\alpha}{4}\right)}{300^{2-\alpha}n^{\alpha-1}\ln^{2-\alpha}\frac{48n(K+1)}{\beta}}$$

$$\overset{(375)}{\leq} \frac{\exp\left(-\gamma\mu T\right)V^2}{600\ln\frac{48n(K+1)}{\beta}}. \tag{416}$$

That is, we derive the upper bounds for ①, ②, ③, ④, ⑤, ⑥, ⑦. More precisely, $E_{T-1}$ implies

$$V_T \overset{(386)}{\leq} \exp\left(-\frac{\gamma\mu}{2}T\right)V + ① + ② + ③ + ④ + ⑤ + ⑥ + ⑦,$$

$$⑥ \overset{(405)}{=} ⑥' + \frac{32\gamma}{n}\sum_{j=2}^{n}\left\langle \frac{\gamma}{n}\sum_{i=1}^{j-1}\theta_{i,T-1}^{u} - \zeta_j^{T-1}, \theta_{j,T-1}^{u}\right\rangle,$$

$$⑦ \overset{(406)}{=} ⑦' + \frac{24\gamma}{n}\sum_{j=2}^{n}\left\langle \frac{\gamma}{n}\sum_{i=1}^{j-1}\omega_{i,T-1}^{u} - \delta_j^{T-1}, \omega_{j,T-1}^{u}\right\rangle,$$

$$② \overset{(395)}{\leq} \frac{3\exp\left(-\frac{\gamma\mu T}{2}\right)V}{100}, \quad ③ \overset{(396)}{\leq} \frac{\exp\left(-\frac{\gamma\mu T}{2}\right)V}{50}, \quad ⑤ \overset{(401)}{\leq} \frac{\exp\left(-\frac{\gamma\mu T}{2}\right)V}{6},$$

$$\sum_{l=0}^{T-1}\sum_{i=1}^{n}\sigma_{i,l}^2 \overset{(393)}{\leq} \frac{\exp\left(-\gamma\mu T\right)V^2}{60000\ln\frac{48n(K+1)}{\beta}}, \quad \sum_{l=0}^{T-1}\sum_{i=1}^{n}\widetilde{\sigma}_{i,l}^2 \overset{(400)}{\leq} \frac{\exp\left(-\gamma\mu T\right)V}{216\ln\frac{48n(K+1)}{\beta}},$$

$$\sum_{l=0}^{T-1}\sum_{j=2}^{n}\widehat{\sigma}_{j,l}^2 \overset{(412)}{\leq} \frac{\exp\left(-\gamma\mu T\right)V^2}{600\ln\frac{48n(K+1)}{\beta}}, \quad \sum_{l=0}^{T-1}\sum_{j=2}^{n}(\sigma_{j,l}')^2 \overset{(416)}{\leq} \frac{\exp\left(-\gamma\mu T\right)V^2}{600\ln\frac{48n(K+1)}{\beta}}.$$

In addition, we also establish (see (392), (392), (411), (411), and our induction assumption)

$$\mathbb{P}\{E_{T-1}\} \geq 1 - \frac{(T-1)\beta}{K+1}, \quad \mathbb{P}\{E_①\} \geq 1 - \frac{\beta}{24n(K+1)},$$

$$\mathbb{P}\{E_④\} \geq 1 - \frac{\beta}{24n(K+1)}, \quad \mathbb{P}\{E_{⑥'}\} \geq 1 - \frac{\beta}{24n(K+1)}, \quad \mathbb{P}\{E_{⑦'}\} \geq 1 - \frac{\beta}{24n(K+1)},$$

where

$$E_① = \left\{ \text{either } \sum_{l=0}^{T-1}\sum_{i=1}^{n}\sigma_{i,l}^2 > \frac{\exp\left(-\gamma\mu T\right)V^2}{60000\ln\frac{48n(K+1)}{\beta}} \text{ or } |①| \leq \frac{\exp\left(-\frac{\gamma\mu T}{2}\right)V}{100} \right\},$$

$$E_④ = \left\{ \text{either } \sum_{l=0}^{T-1}\sum_{i=1}^{n}\widetilde{\sigma}_{i,l}^2 > \frac{\exp\left(-\gamma\mu T\right)V^2}{216\ln\frac{48n(K+1)}{\beta}} \text{ or } |④| \leq \frac{\exp\left(-\frac{\gamma\mu T}{2}\right)V}{6} \right\},$$

$$E_{⑥'} = \left\{ \text{either } \sum_{l=0}^{T-1}\sum_{j=2}^{n}\widehat{\sigma}_{j,l}^2 > \frac{\exp\left(-\gamma\mu T\right)V^2}{600\ln\frac{48n(K+1)}{\beta}} \text{ or } |⑥'| \leq \frac{\exp\left(-\frac{\gamma\mu T}{2}\right)V}{10} \right\},$$

$$E_{⑦'} = \left\{ \text{either } \sum_{l=0}^{T-1}\sum_{j=2}^{n}(\sigma_{j,l}')^2 > \frac{\exp\left(-\gamma\mu T\right)V^2}{600\ln\frac{48n(K+1)}{\beta}} \text{ or } |⑦'| \leq \frac{\exp\left(-\frac{\gamma\mu T}{2}\right)V}{10} \right\}.$$

Therefore, probability event $E_{T-1} \cap E_① \cap E_④ \cap E_{⑥'} \cap E_{⑦'}$ implies

$$V_T \leq \exp\left(-\frac{\gamma\mu}{2}T\right)V\underbrace{\left(1 + \frac{1}{100} + \frac{3}{100} + \frac{1}{50} + \frac{1}{6} + \frac{1}{6} + \frac{1}{10} + \frac{1}{10}\right)}_{\leq 2}$$

$$+\frac{32\gamma}{n}\sum_{j=2}^{n}\left\langle \frac{\gamma}{n}\sum_{i=1}^{j-1}\theta_{i,T-1}^{u} - \zeta_j^{T-1}, \theta_{j,T-1}^{u}\right\rangle$$

$$+\frac{24\gamma}{n}\sum_{j=2}^{n}\left\langle \frac{\gamma}{n}\sum_{i=1}^{j-1}\omega_{i,T-1}^{u} - \delta_j^{T-1}, \omega_{j,T-1}^{u}\right\rangle. \tag{417}$$

To finish the proof, we need to show that $\frac{\gamma}{n}\sum_{i=1}^{j-1}\theta_{i,T-1}^{u} = \zeta_j^{T-1}$ and $\frac{\gamma}{n}\sum_{i=1}^{j-1}\omega_{i,T-1}^{u} = \delta_j^{T-1}$ with high probability. In particular, we consider probability event $\widetilde{E}_{T-1,j}$ defined as follows: inequalities

$$\left\|\frac{\gamma}{n}\sum_{i=1}^{r-1}\theta_{i,T-1}^{u}\right\| \le \exp\left(-\frac{\gamma\mu(T-1)}{4}\right)\frac{\sqrt{V}}{2}, \quad \left\|\frac{\gamma}{n}\sum_{i=1}^{r-1}\omega_{i,T-1}^{u}\right\| \le \exp\left(-\frac{\gamma\mu(T-1)}{4}\right)\frac{\sqrt{V}}{2}$$

hold for $r = 2,\ldots,j$ simultaneously. We want to show that $\mathbb{P}\{E_{T-1}\cap\widetilde{E}_{T-1,j}\} \ge 1 - \frac{(T-1)\beta}{K+1} - \frac{j\beta}{8n(K+1)}$ for all $j = 2,\ldots,n$. For $j = 2$ the statement is trivial since

$$\left\|\frac{\gamma}{n}\theta_{1,T-1}^{u}\right\| \overset{(387)}{\le} \frac{2\gamma\lambda_{T-1}}{n} \le \exp\left(-\frac{\gamma\mu(T-1)}{4}\right)\frac{\sqrt{V}}{2},$$

$$\left\|\frac{\gamma}{n}\omega_{1,T-1}^{u}\right\| \overset{(323)}{\le} \frac{2\gamma\lambda_{T-1}}{n} \le \exp\left(-\frac{\gamma\mu(T-1)}{4}\right)\frac{\sqrt{V}}{2}.$$

Next, we assume that the statement holds for some $j = m - 1 < n$, i.e., $\mathbb{P}\{E_{T-1}\cap\widetilde{E}_{T-1,m-1}\} \ge 1 - \frac{(T-1)\beta}{K+1} - \frac{(m-1)\beta}{8n(K+1)}$. Our goal is to prove that $\mathbb{P}\{E_{T-1}\cap\widetilde{E}_{T-1,m}\} \ge 1 - \frac{(T-1)\beta}{K+1} - \frac{m\beta}{8n(K+1)}$.

First, we consider $\left\|\frac{\gamma}{n}\sum_{i=1}^{m-1}\theta_{i,T-1}^{u}\right\|$:

$$\begin{aligned}
\left\|\frac{\gamma}{n}\sum_{i=1}^{m-1}\theta_{i,T-1}^{u}\right\| &= \sqrt{\frac{\gamma^2}{n^2}\left\|\sum_{i=1}^{m-1}\theta_{i,T-1}^{u}\right\|^2} \\
&= \sqrt{\frac{\gamma^2}{n^2}\sum_{i=1}^{m-1}\|\theta_{i,T-1}^{u}\|^2 + \frac{2\gamma}{n}\sum_{i=1}^{m-1}\left\langle\frac{\gamma}{n}\sum_{r=1}^{i-1}\theta_{r,T-1}^{u},\theta_{i,T-1}^{u}\right\rangle} \\
&\le \sqrt{\frac{\gamma^2}{n^2}\sum_{l=0}^{T-1}\exp\left(-\frac{\gamma\mu(T-1-l)}{2}\right)\sum_{i=1}^{m-1}\|\theta_{i,l}^{u}\|^2 + \frac{2\gamma}{n}\sum_{i=1}^{m-1}\left\langle\frac{\gamma}{n}\sum_{r=1}^{i-1}\theta_{r,T-1}^{u},\theta_{i,T-1}^{u}\right\rangle}.
\end{aligned}$$

Similarly, we have

$$\begin{aligned}
\left\|\frac{\gamma}{n}\sum_{i=1}^{m-1}\omega_{i,T-1}^{u}\right\| &= \sqrt{\frac{\gamma^2}{n^2}\left\|\sum_{i=1}^{m-1}\omega_{i,T-1}^{u}\right\|^2} \\
&= \sqrt{\frac{\gamma^2}{n^2}\sum_{i=1}^{m-1}\|\omega_{i,T-1}^{u}\|^2 + \frac{2\gamma}{n}\sum_{i=1}^{m-1}\left\langle\frac{\gamma}{n}\sum_{r=1}^{i-1}\omega_{r,T-1}^{u},\omega_{i,T-1}^{u}\right\rangle} \\
&\le \sqrt{\frac{\gamma^2}{n^2}\sum_{l=0}^{T-1}\exp\left(-\frac{\gamma\mu(T-1-l)}{2}\right)\sum_{i=1}^{m-1}\|\omega_{i,l}^{u}\|^2 + \frac{2\gamma}{n}\sum_{i=1}^{m-1}\left\langle\frac{\gamma}{n}\sum_{r=1}^{i-1}\omega_{r,T-1}^{u},\omega_{i,T-1}^{u}\right\rangle}.
\end{aligned}$$

Next, we introduce a new notation:

$$\rho_{i,T-1} = \begin{cases} \frac{\gamma}{n}\sum_{r=1}^{i-1}\theta_{r,T-1}^{u}, & \text{if } \left\|\frac{\gamma}{n}\sum_{r=1}^{i-1}\theta_{r,T-1}^{u}\right\| \le \exp\left(-\frac{\gamma\mu(T-1)}{4}\right)\frac{\sqrt{V}}{2}, \\ 0, & \text{otherwise} \end{cases}$$

$$\rho_{i,T-1}' = \begin{cases} \frac{\gamma}{n}\sum_{r=1}^{i-1}\omega_{r,T-1}^{u}, & \text{if } \left\|\frac{\gamma}{n}\sum_{r=1}^{i-1}\omega_{r,T-1}^{u}\right\| \le \exp\left(-\frac{\gamma\mu(T-1)}{4}\right)\frac{\sqrt{V}}{2}, \\ 0, & \text{otherwise} \end{cases}$$

for $i = 1,\ldots,m-1$. By definition, we have

$$\|\rho_{i,T-1}\| \le \exp\left(-\frac{\gamma\mu(T-1)}{4}\right)\frac{\sqrt{V}}{2}, \quad \|\rho_{i,T-1}'\| \le \exp\left(-\frac{\gamma\mu(T-1)}{4}\right)\frac{\sqrt{V}}{2} \tag{418}$$

for $i = 1, \ldots, m-1$. Moreover, $\widetilde{E}_{T-1,m-1}$ implies $\rho_{i,T-1} = \frac{\gamma}{n} \sum_{r=1}^{i-1} \theta_{r,T-1}^u, \rho'_{i,T-1} = \frac{\gamma}{n} \sum_{r=1}^{i-1} \omega_{r,T-1}^u$
for $i = 1, \ldots, m-1$ and

$$\left\| \frac{\gamma}{n} \sum_{i=1}^{m-1} \theta_{i,l}^u \right\| \leq \sqrt{③ + ④ + ⑧},$$

$$\left\| \frac{\gamma}{n} \sum_{i=1}^{m-1} \omega_{i,l}^u \right\| \leq \sqrt{③ + ④ + ⑧'},$$

where

$$⑧ = \frac{2\gamma}{n} \sum_{i=1}^{m-1} \left\langle \rho_{i,T-1}, \theta_{i,T-1}^u \right\rangle, \quad ⑧' = \frac{2\gamma}{n} \sum_{i=1}^{m-1} \left\langle \rho'_{i,T-1}, \omega_{i,T-1}^u \right\rangle.$$

It remains to estimate ⑧ and ⑧′.

**Upper bound for ⑧.** To estimate this sum, we will use Bernstein's inequality. The summands have conditional expectations equal to zero:

$$\mathbb{E}_{\xi_{2,i}^{T-1}} \left[ \frac{2\gamma}{n} \langle \rho_{i,T-1}, \theta_{i,T-1}^u \rangle \right] = \frac{2\gamma}{n} \left\langle \rho_{i,T-1}, \mathbb{E}_{\xi_{2,i}^{T-1}} [\theta_{i,T-1}^u] \right\rangle = 0.$$

Thus, sequence $\left\{ \frac{2\gamma}{n} \langle \rho_{i,T-1}, \theta_{i,T-1}^u \rangle \right\}_{i=1}^n$ is a martingale difference sequence. Next, the summands are bounded:

$$\left| \frac{2\gamma}{n} \langle \rho_{i,T-1}, \theta_{i,T-1}^u \rangle \right| \leq \frac{2\gamma}{n} \|\rho_{i,T-1}\| \cdot \|\theta_{i,T-1}^u\|$$

$$\overset{(418),(387)}{\leq} \frac{2\sqrt{V}\gamma \exp\left(-\frac{\gamma\mu(T-1)}{4}\right)}{n} \lambda_{T-1}$$

$$\overset{(305)}{\leq} \frac{\exp\left(-\frac{\gamma\mu(T-1)}{2}\right) V}{80 \ln \frac{48n(K+1)}{\beta}} \overset{\text{def}}{=} c. \tag{419}$$

Finally, conditional variances $(\widehat{\sigma}'_{i,T-1})^2 \overset{\text{def}}{=} \mathbb{E}_{\xi_{2,i}^{T-1}} \left[ \frac{4\gamma^2}{n^2} \langle \rho_{i,T-1}, \theta_{i,T-1}^u \rangle^2 \right]$ of the summands are bounded:

$$(\widehat{\sigma}'_{i,T-1})^2 \leq \mathbb{E}_{\xi_{2,i}^{T-1}} \left[ \frac{4\gamma^2}{n^2} \|\rho_{i,T-1}\|^2 \cdot \|\theta_{i,T-1}^u\|^2 \right]$$

$$\overset{(418)}{\leq} \frac{\gamma^2 V \exp\left(-\frac{\gamma\mu(T-1)}{2}\right)}{n^2} \mathbb{E}_{\xi_{2,i}^{T-1}} \left[ \|\theta_{i,T-1}^u\|^2 \right]. \tag{420}$$

Applying Bernstein's inequality (Lemma B.1) with $X_i = \frac{2\gamma}{n} \langle \rho_{i,T-1}, \theta_{i,T-1}^u \rangle$, constant $c$ defined in (419), $b = \frac{\exp\left(-\frac{\gamma\mu(T-1)}{2}\right)V}{80}$, $G = \frac{\exp(-\gamma\mu(T-1))V^2}{38400 \ln \frac{48n(K+1)}{\beta}}$, we get

$$\mathbb{P}\left\{ |⑧| > \frac{\exp\left(-\frac{\gamma\mu(T-1)}{2}\right) V}{80} \text{ and } \sum_{i=1}^n (\widehat{\sigma}'_{i,T-1})^2 \leq \frac{\exp\left(-\gamma\mu(T-1)\right) V^2}{38400 \ln \frac{48n(K+1)}{\beta}} \right\} \leq 2 \exp\left(-\frac{b^2}{2G + {^{2cb}/_3}}\right)$$

$$= \frac{\beta}{24n(K+1)}.$$

The above is equivalent to $\mathbb{P}\{E_⑧\} \geq 1 - \frac{\beta}{24n(K+1)}$ for

$$E_⑧ = \left\{ \text{either } \sum_{i=1}^n (\widehat{\sigma}'_{i,T-1})^2 > \frac{\exp\left(-\gamma\mu(T-1)\right) V^2}{38400 \ln \frac{48n(K+1)}{\beta}} \text{ or } |⑧| > \frac{\exp\left(-\frac{\gamma\mu(T-1)}{2}\right) V}{80} \right\}. \tag{421}$$

Moreover, $E_{T-1}$ implies

$$
\begin{aligned}
\sum_{i=1}^{n}(\widehat{\sigma}'_{i,T-1})^2 &\overset{(420)}{\leq} \frac{\gamma^2 V \exp\left(-\gamma\mu\left(T-1\right)\right)}{n^2} \sum_{i=1}^{n} \mathbb{E}_{\xi_{2,i}^{T}-1}\left[\|\theta_{i,T-1}^u\|^2\right] \\
&\overset{(389)}{\leq} \frac{18\gamma^2 V \exp\left(-\gamma\mu\left(T-1\right)\right)\sigma^\alpha}{n}\lambda_{T-1}^{2-\alpha} \\
&\overset{(378)}{\leq} \frac{18\gamma^\alpha V^{2-\frac{\alpha}{2}} \exp\left(-\gamma\mu(T-1)\right)\sigma^\alpha}{300^{2-\alpha} n^{\alpha-1} \ln^{2-\alpha}\frac{48n(K+1)}{\beta}}\exp\left(\frac{\gamma\mu(T-1)\alpha}{4}\right) \\
&\overset{T-1\leq K}{\leq} \frac{18\gamma^\alpha V^{2-\frac{\alpha}{2}} \exp\left(-\gamma\mu(T-1)\right)\sigma^\alpha \exp\left(\frac{\gamma\mu K\alpha}{4}\right)}{300^{2-\alpha} n^{\alpha-1} \ln^{2-\alpha}\frac{48n(K+1)}{\beta}} \\
&\overset{(375)}{\leq} \frac{\exp\left(-\gamma\mu(T-1)\right)V^2}{38400\ln\frac{48n(K+1)}{\beta}}.
\end{aligned}
\tag{422}
$$

**Upper bound for $\circledast'$.** To estimate this sum, we will use Bernstein's inequality. The summands have conditional expectations equal to zero:

$$
\mathbb{E}_{\xi_{1,i}^{T-1}}\left[\frac{2\gamma}{n}\langle\rho'_{i,T-1},\omega_{i,T-1}^u\rangle\right] = \frac{2\gamma}{n}\left\langle\rho'_{i,T-1},\mathbb{E}_{\xi_{1,i}^{T-1}}[\omega_{i,T-1}^u]\right\rangle = 0.
$$

Thus, sequence $\left\{\frac{2\gamma}{n}\langle\rho'_{i,T-1},\omega_{i,T-1}^u\rangle\right\}_{i=1}^{n}$ is a martingale difference sequence. Next, the summands are bounded:

$$
\begin{aligned}
\left|\frac{2\gamma}{n}\langle\rho'_{i,T-1},\omega_{i,T-1}^u\rangle\right| &\leq \frac{2\gamma}{n}\|\rho'_{i,T-1}\|\cdot\|\omega_{i,T-1}^u\| \\
&\overset{(418),(387)}{\leq} \frac{2\sqrt{V}\gamma\exp\left(-\frac{\gamma\mu(T-1)}{4}\right)}{n}\lambda_{T-1} \\
&\overset{(305)}{=} \frac{\exp\left(-\frac{\gamma\mu(T-1)}{2}\right)V}{80\ln\frac{48n(K+1)}{\beta}}\overset{\text{def}}{=} c.
\end{aligned}
\tag{423}
$$

Finally, conditional variances $(\widetilde{\sigma}'_{i,T-1})^2 \overset{\text{def}}{=} \mathbb{E}_{\xi_{1,i}^{T-1}}\left[\frac{4\gamma^2}{n^2}\langle\rho'_{i,T-1},\omega_{i,T-1}^u\rangle^2\right]$ of the summands are bounded:

$$
\begin{aligned}
(\widetilde{\sigma}'_{i,T-1})^2 &\leq \mathbb{E}_{\xi_{1,i}^{T-1}}\left[\frac{4\gamma^2}{n^2}\|\rho'_{i,T-1}\|^2\cdot\|\omega_{i,T-1}^u\|^2\right] \\
&\overset{(418)}{\leq} \frac{\gamma^2 V\exp\left(-\frac{\gamma\mu(T-1)}{2}\right)}{n^2}\mathbb{E}_{\xi_{1,i}^{T-1}}\left[\|\omega_{i,T-1}^u\|^2\right].
\end{aligned}
\tag{424}
$$

Applying Bernstein's inequality (Lemma B.1) with $X_i = \frac{2\gamma}{n}\langle\rho'_{i,T-1},\omega_{i,T-1}^u\rangle$, constant $c$ defined in (423), $b = \frac{\exp\left(-\frac{\gamma\mu(T-1)}{2}\right)V}{80}$, $G = \frac{\exp(-\gamma\mu(T-1))V^2}{38400\ln\frac{48n(K+1)}{\beta}}$, we get

$$
\begin{aligned}
\mathbb{P}\left\{|\circledast'| > \frac{\exp\left(-\frac{\gamma\mu(T-1)}{2}\right)V}{80} \text{ and } \sum_{i=1}^{n}(\widetilde{\sigma}'_{i,T-1})^2 \leq \frac{\exp\left(-\gamma\mu(T-1)\right)V^2}{38400\ln\frac{48n(K+1)}{\beta}}\right\} &\leq 2\exp\left(-\frac{b^2}{2G+2cb/3}\right) \\
&= \frac{\beta}{24n(K+1)}.
\end{aligned}
$$

The above is equivalent to $\mathbb{P}\{E_{\circledast'}\} \geq 1 - \frac{\beta}{24n(K+1)}$ for

$$
E_{\circledast} = \left\{\text{either } \sum_{i=1}^{n}(\widetilde{\sigma}'_{i,T-1})^2 > \frac{\exp\left(-\gamma\mu(T-1)\right)V^2}{38400\ln\frac{48n(K+1)}{\beta}} \text{ or } |\circledast'| > \frac{\exp\left(-\frac{\gamma\mu(T-1)}{2}\right)V}{80}\right\}.
\tag{425}
$$

Moreover, $E_{T-1}$ implies

$$
\begin{aligned}
\sum_{i=1}^{n}(\widetilde{\sigma}'_{i,T-1})^2 &\overset{(424)}{\leq} \frac{\gamma^2 V \exp\left(-\gamma\mu\left(T-1\right)\right)}{n^2} \sum_{i=1}^{n} \mathbb{E}_{\xi_{2,i}^T - 1}\left[\|\theta_{i,T-1}^u\|^2\right] \\
&\overset{(389)}{\leq} \frac{18\gamma^2 V \exp\left(-\gamma\mu\left(T-1\right)\right)\sigma^\alpha}{n}\lambda_{T-1}^{2-\alpha} \\
&\overset{(378)}{\leq} \frac{18\gamma^\alpha V^{2-\frac{\alpha}{2}}\exp\left(-\gamma\mu(T-1)\right)\sigma^\alpha}{300^{2-\alpha}n^{\alpha-1}\ln^{2-\alpha}\frac{48n(K+1)}{\beta}}\exp\left(\frac{\gamma\mu(T-1)\alpha}{4}\right) \\
&\overset{T-1\leq K}{\leq} \frac{18\gamma^\alpha V^{2-\frac{\alpha}{2}}\exp\left(-\gamma\mu(T-1)\right)\sigma^\alpha \exp\left(\frac{\gamma\mu K\alpha}{4}\right)}{300^{2-\alpha}n^{\alpha-1}\ln^{2-\alpha}\frac{48n(K+1)}{\beta}} \\
&\overset{(375)}{\leq} \frac{\exp\left(-\gamma\mu(T-1)\right)V^2}{38400\ln\frac{48n(K+1)}{\beta}}.
\end{aligned}
\tag{426}
$$

Putting all together we get that $E_{T-1} \cap \widetilde{E}_{T-1,m-1}$ implies

$$
\left\|\frac{\gamma}{n}\sum_{i=1}^{m-1}\theta_{i,T-1}^u\right\| \leq \sqrt{③+④+⑧}, \quad \left\|\frac{\gamma}{n}\sum_{i=1}^{m-1}\omega_{i,T-1}^u\right\| \leq \sqrt{③+④+⑧'}, \quad ③ \overset{(396)}{\leq} \frac{\exp\left(-\frac{\gamma\mu T}{2}\right)V}{50},
$$

$$
\sum_{l=0}^{T-1}\sum_{i=1}^{n}\widetilde{\sigma}_{i,l}^2 \overset{(400)}{\leq} \frac{\exp\left(-\gamma\mu T\right)V}{216\ln\frac{48n(K+1)}{\beta}}, \quad \sum_{i=1}^{m-1}(\widehat{\sigma}'_{i,T-1})^2 \leq \frac{\exp\left(-\gamma\mu(T-1)\right)V^2}{38400\ln\frac{48n(K+1)}{\beta}},
$$

$$
\sum_{i=1}^{m-1}(\widetilde{\sigma}'_{i,T-1})^2 \leq \frac{\exp\left(-\gamma\mu(T-1)\right)V^2}{38400\ln\frac{48n(K+1)}{\beta}}.
$$

In addition, we also establish (see (399), (421), (425) and our induction assumption)

$$
\mathbb{P}\{E_{T-1}\cap\widetilde{E}_{T-1,m-1}\} \geq 1 - \frac{(T-1)\beta}{K+1} - \frac{(m-1)\beta}{8n(K+1)},
$$

$$
\mathbb{P}\{E_④\} \geq 1 - \frac{\beta}{24n(K+1)}, \quad \mathbb{P}\{E_⑧\} \geq 1 - \frac{\beta}{24n(K+1)}, \quad \mathbb{P}\{E_{⑧'}\} \geq 1 - \frac{\beta}{24n(K+1)}
$$

where

$$
\begin{aligned}
E_④ &= \left\{\text{either } \sum_{l=0}^{T-1}\sum_{i=1}^{n}\widetilde{\sigma}_{i,l}^2 > \frac{\exp\left(-\gamma\mu T\right)V^2}{216\ln\frac{48n(K+1)}{\beta}} \text{ or } |④| \leq \frac{\exp\left(-\frac{\gamma\mu T}{2}\right)V}{6}\right\}, \\
E_⑧ &= \left\{\text{either } \sum_{i=1}^{n}(\widehat{\sigma}'_{i,T-1})^2 > \frac{\exp\left(-\gamma\mu(T-1)\right)V^2}{38400\ln\frac{48n(K+1)}{\beta}} \text{ or } |⑧| > \frac{\exp\left(-\frac{\gamma\mu(T-1)}{2}\right)V}{80}\right\}, \\
E_{⑧'} &= \left\{\text{either } \sum_{i=1}^{n}(\widetilde{\sigma}'_{i,T-1})^2 > \frac{\exp\left(-\gamma\mu(T-1)\right)V^2}{38400\ln\frac{48n(K+1)}{\beta}} \text{ or } |⑧'| > \frac{\exp\left(-\frac{\gamma\mu(T-1)}{2}\right)V}{80}\right\}.
\end{aligned}
$$

Therefore, probability event $E_{T-1}\cap\widetilde{E}_{m-1}\cap E_④\cap E_⑧\cap E_{⑧'}$ implies

$$
\left\|\frac{\gamma}{n}\sum_{i=1}^{m-1}\theta_{i,T-1}^u\right\| \leq \exp\left(-\frac{\gamma\mu(T-1)}{4}\right)\sqrt{V}\sqrt{\frac{1}{50}+\frac{1}{6}+\frac{1}{80}} \leq \frac{\exp\left(-\frac{\gamma\mu(T-1)}{4}\right)\sqrt{V}}{2},
$$

$$
\left\|\frac{\gamma}{n}\sum_{i=1}^{m-1}\omega_{i,T-1}^u\right\| \leq \exp\left(-\frac{\gamma\mu(T-1)}{4}\right)\sqrt{V}\sqrt{\frac{1}{50}+\frac{1}{6}+\frac{1}{80}} \leq \frac{\exp\left(-\frac{\gamma\mu(T-1)}{4}\right)\sqrt{V}}{2}.
$$

This implies $\widetilde{E}_{T-1,m}$ and

$$
\begin{aligned}
\mathbb{P}\{E_{T-1} \cap \widetilde{E}_{T-1,m}\} &\geq \mathbb{P}\{E_{T-1} \cap \widetilde{E}_{T-1,m-1} \cap E_④ \cap E_⑧ \cap E_{⑧'}\} \\
&= 1 - \mathbb{P}\left\{\overline{E_{T-1} \cap \widetilde{E}_{T-1,m-1}} \cup \overline{E}_④ \cup \overline{E}_⑧ \cup \overline{E}_{⑧'}\right\} \\
&\geq 1 - \frac{(T-1)\beta}{K+1} - \frac{m\beta}{8n(K+1)}.
\end{aligned}
$$

Therefore, for all $m = 2,\ldots,n$ the statement holds and, in particular, $\mathbb{P}\{E_{T-1} \cap \widetilde{E}_{T-1,n}\} \geq 1 - \frac{(T-1)\beta}{K+1} - \frac{\beta}{8(K+1)}$, i.e., (383) and (384) hold. Taking into account (417), we conclude that $E_{T-1} \cap \widetilde{E}_{T-1,n} \cap E_① \cap E_④ \cap E_{⑥'} \cap E_{⑦'} \cap E_⑧$ implies

$$
V_T \leq 2\exp\left(-\frac{\gamma\mu}{2}T\right) V
$$

that is equivalent to (382) for $t = T$. Moreover,

$$
\begin{aligned}
\mathbb{P}\{E_T\} &\geq \mathbb{P}\left\{E_{T-1} \cap \widetilde{E}_{T-1,n} \cap E_① \cap E_④ \cap E_{⑥'} \cap E_{⑦'}\right\} \\
&= 1 - \mathbb{P}\left\{\overline{E_{T-1} \cap \widetilde{E}_n} \cup \overline{E}_① \cup \overline{E}_④ \cup \overline{E}_{⑥'} \cup \overline{E}_{⑦'}\right\} \\
&= 1 - \frac{(T-1)\beta}{K+1} - \frac{\beta}{8(K+1)} - 4\cdot\frac{\beta}{24n(K+1)} = 1 - \frac{T\beta}{K+1}.
\end{aligned}
$$

In other words, we showed that $\mathbb{P}\{E_k\} \geq 1 - {k\beta}/{(K+1)}$ for all $k = 0, 1, \ldots, K+1$. For $k = K+1$ we have that with probability at least $1 - \beta$

$$
\|x^{K+1} - x^*\|^2 \leq V_{K+1} \leq 2\exp\left(-\frac{\gamma\mu(K+1)}{2}\right) V.
$$

Finally, if

$$
\gamma = \min\left\{\frac{1}{72\cdot 10^6\mu\ln^2\frac{48n(K+1)}{\beta}}, \frac{1}{6L}, \frac{\sqrt{n}}{15000L\ln\frac{48n(K+1)}{\beta}}, \frac{2\ln(B_K)}{\mu(K+1)}\right\},
$$

$$
\begin{aligned}
B_K &= \max\left\{2, \frac{n^{\frac{2(\alpha-1)}{\alpha}}(K+1)^{\frac{2(\alpha-1)}{\alpha}}\mu^2 V}{3110400^{\frac{2}{\alpha}}\sigma^2\ln^{\frac{2(\alpha-1)}{\alpha}}\left(\frac{48n(K+1)}{\beta}\right)\ln^2(B_K)}\right\} \\
&= \mathcal{O}\left(\max\left\{2, \frac{n^{\frac{2(\alpha-1)}{\alpha}}K^{\frac{2(\alpha-1)}{\alpha}}\mu^2 V}{\sigma^2\ln^{\frac{2(\alpha-1)}{\alpha}}\left(\frac{nK}{\beta}\right)\ln^2\left(\max\left\{2, \frac{n^{\frac{2(\alpha-1)}{\alpha}}K^{\frac{2(\alpha-1)}{\alpha}}\mu^2 V}{\sigma^2\ln^{\frac{2(\alpha-1)}{\alpha}}\left(\frac{nK}{\beta}\right)}\right\}\right)}\right\}\right)
\end{aligned}
$$

then with probability at least $1 - \beta$

$$
\begin{aligned}
\|x^{K+1} - x^*\|^2 &\leq 2\exp\left(-\frac{\gamma\mu(K+1)}{2}\right) V \\
&= 2V\max\left\{\exp\left(-\frac{K+1}{144\cdot 10^6\ln^2\frac{48n(K+1)}{\beta}}\right), \exp\left(-\frac{\mu(K+1)}{12L}\right),\right. \\
&\qquad\qquad\qquad\left.\exp\left(-\frac{\mu\sqrt{n}(K+1)}{30000L\ln\frac{48n(K+1)}{\beta}}\right), \frac{1}{B_K}\right\} \\
&= \mathcal{O}\left(\max\left\{V\exp\left(-\frac{K}{\ln^2\frac{nK}{\beta}}\right), V\exp\left(-\frac{\mu K}{L}\right),\right.\right. \\
&\qquad\qquad\qquad\left.\left. V\exp\left(-\frac{\mu\sqrt{n}K}{L\ln\frac{nK}{\beta}}\right), \frac{\sigma^2\ln^{\frac{2(\alpha-1)}{\alpha}}\left(\frac{nK}{\beta}\right)\ln^2 B_K}{n^{\frac{2(\alpha-1)}{\alpha}}K^{\frac{2(\alpha-1)}{\alpha}}\mu^2}\right\}\right).
\end{aligned}
$$

To get $\|x^{K+1} - x^*\|^2 \leq \varepsilon$ with probability $\geq 1 - \beta$, $K$ should be

$$K = \mathcal{O}\Bigg( \max\Bigg\{ \left( \frac{L}{\sqrt{n}\mu} + \ln\left( \frac{nL}{\mu\beta} \ln\frac{V}{\varepsilon} \right) \right) \ln\left( \frac{V}{\varepsilon} \right) \ln\left( \frac{nL}{\mu\beta} \ln\frac{V}{\varepsilon} \right),$$

$$\frac{L}{\mu} \ln\left( \frac{V}{\varepsilon} \right), \frac{1}{n} \left( \frac{\sigma^2}{\mu^2\varepsilon} \right)^{\frac{\alpha}{2(\alpha-1)}} \ln\left( \frac{n}{\beta} \left( \frac{\sigma^2}{\mu^2\varepsilon} \right)^{\frac{\alpha}{2(\alpha-1)}} \right) \ln^{\frac{\alpha}{\alpha-1}}(B_\varepsilon) \Bigg\} \Bigg),$$

where

$$B_\varepsilon = \max\Bigg\{ 2, \frac{V}{\varepsilon \ln\left( \frac{1}{\beta} \left( \frac{\sigma^2}{\mu^2\varepsilon} \right)^{\frac{\alpha}{2(\alpha-1)}} \right)} \Bigg\}.$$

$\square$

# I NUMERICAL EXPERIMENTS

In this section we provide numerical experiments for the following simple problem:

$$\min_{x \in B_r(\hat{x})} f(x), \tag{427}$$

where a radius $r = 1$, a central point $\hat{x} = (3, 3, \ldots, 3)^\top \in \mathbb{R}^{10}$, and $f(x) = \frac{1}{2}\|x\|^2$, $f_\xi(x) = \frac{1}{2}\|x\|^2 + \langle \xi, x \rangle$, where $\xi$ comes from the symmetric Levy $\alpha$-stable distribution $\alpha = \frac{3}{2}$. We use the following parameters: $\gamma = 0.001$, $x^0 = \hat{x} + r\frac{e}{\|e\|}$, where $e = (1, 1, \ldots, 1)^\top$. We tried three values of $\lambda$: 0.1, 0.01 and 0.001.

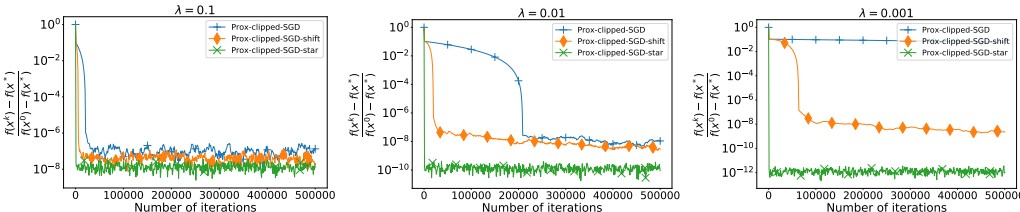

Figure 1: Comparison between performances of Prox-clipped-SGD, Prox-clipped-SGD-star, Prox-clipped-SGD-shift in solving problem (427) with fixed clipping level for each of them $\lambda \in \{0.1, 0.01, 0.001\}$.

In our numerical experiments (see Figure 1), we observe that the naïve Prox-clipped-SGD converges slower than Prox-clipped-SGD-star and Prox-clipped-SGD-shift. Moreover, when the clipping level is small Prox-clipped-SGD converges extremely slow, while Prox-clipped-SGD-shifts takes some time to learn the shift and then converges to much better accuracy. We also see that the smaller clipping level is, the better accuracy Prox-clipped-SGD-star achieves. For Prox-clipped-SGD-shift we observe the same phenomenon when we reduce $\lambda$ from 0.1 to 0.01 . We expect the improvement in the accuracy even further if we decrease the stepsizes $\gamma$ and $\nu$ .

