# OpenReview forum: "High-Probability Convergence for Composite and Distributed Stochastic Minimization and Variational Inequalities with Heavy-Tailed Noise"
_ICLR.cc/2024/Conference — Submitted to ICLR 2024_

### Official Review · Reviewer_sYPF · 2023-11-01

**Soundness:** 3 good
**Presentation:** 3 good
**Contribution:** 2 fair
**Rating:** 5
**Confidence:** 3

**Summary:**

This paper studies composite and distributed optimization as well as variational inequality problems. The paper proposes new stochastic gradient clipping methods with theoretical convergence bounds.

**Strengths:**

* The paper is technically sound. It provides theoretical analysis of the proposed methods. Proof sketches are presented to support the theoretical results. Related work and existing results on clipping methods are discussed in the paper
* The paper is well-written. The problem setting is clearly presented, followed by the proposed methods and theoretical analysis.

**Weaknesses:**

* The significance can be enhanced by clarifying the motivation and theoretical improvements of the new clipping method, such as providing theoretical analysis to illustrate the efficiency of this gradient update. The contribution can be strengthened by including a discussion that compares the derived bounds in this paper with existing gradient clipping results.
* The main contribution of the paper is the introduction of new optimization methods. It would be great to include experiments to validate the proposed methods and compare them with relevant algorithms for both the optimization and variational inequality problems considered in this paper.
* The motivation for considering both composite optimization and variational inequality problems is unclear. These two problems appear to be independent and weakly connected. The contribution of addressing two problems appears incremental.

**Questions:**

1. Regarding Assumption 1, would it be better to clarify that equations (4) and (5) apply to all i in [n]?
2. It would be helpful to include a discussion after each theorem to compare the bound with existing clipping method bounds.
3. How would the proposed algorithm perform in the centralized composite optimization problem? Would it yield better convergence bounds compared to existing bounds?

---

> ### Author Response · Authors · 2023-11-22
> **Response to Reviewer sYPF [Part 1/2]**
>
> We want to thank the reviewer once again for the feedback. We addressed all the concerns and comments raised by the reviewer below.
>
> >**The significance can be enhanced by clarifying the motivation and theoretical improvements of the new clipping method, such as providing theoretical analysis to illustrate the efficiency of this gradient update. The contribution can be strengthened by including a discussion that compares the derived bounds in this paper with existing gradient clipping results.**
>
> We provided the motivation behind the construction of the gradient updates in our methods in Section 2 (before Theorem 2.2). We also compared our results with the existing ones in Tables 1 and 2, and also provide discussions after each theorem that relate our results to the existing ones. We have expanded the discussions in the revised version.
>
> >**Numerical experiments.**
>
> We conducted  preliminary experiments showing the superiority of the shift idea over the naive approach, see the revised submission. In the experiments, we consider a simple problem: $\min_{x \in B_r(\hat{x})} f(x)$ where $r = 1$, $\hat{x} = (3,3,\dots,3)^{\top} \in \mathbb{R}^10$  and $f(x) = \frac{1}{2}\|x\|^2$, $f_{\xi}(x) = \frac{1}{2}\|x\|^2 +\langle \xi, x\rangle$, where $\xi$ comes from the symmetric Levy $\alpha$-stable distribution $\alpha = \frac{3}{2}$. We use the following parameters: $\gamma = 0.001$, $x^0 = \hat{x} + r \frac{e}{\|e\|}$, where $e = (1,1,\dots,1)^{\top}$. We tried three values of $\lambda$: $0.1$, $0.01$ and $0.001$.
>
> In our numerical experiments, we observe that the naive Prox-clipped-SGD converges slower than Prox-clipped-SGD-star and Prox-clipped-SGD-shift. Moreover, when the clipping level is small Prox-clipped-SGD converges extremely slow, while Prox-clipped-SGD-shifts takes some time to learn the shift and then converges to much better accuracy. We also see that the smaller clipping level is, the better accuracy Prox-clipped-SGD-star achieves. For Prox-clipped-SGD-shift we observe the same phenomenon when we reduce $\lambda$  from $0.1$ to $0.01$ . We expect the improvement in the accuracy even further if we decrease the stepsizes $\gamma$ and $\nu$ .
>
> The obtained experimental results are in good correspondence with our theoretical findings. Due to the time limitations, we managed to finish only these experiments. However, in the final version, we would like to add more experiments with large-scale problems as well as experiments with distributed methods. We also point out that our main contributions are mostly theoretical.
>
> >**The motivation for considering both composite optimization and variational inequality problems is unclear. These two problems appear to be independent and weakly connected. The contribution of addressing two problems appears incremental.**
>
> Minimization problems and variational inequalities are quite strongly related, though have important differences. For example, convex minimization of differentiable function $f(x)$ is equivalent to variational inequality with $F(x) = \nabla f(x)$. Therefore, one cannot say that these problems are unrelated. Moreover, under (star-)cocoercivity the analysis of SGDA-type methods share a lot of similarities with SGD-type methods, motivating to consider both problems together.
>
> However, minimization problems have a lot of differences with variational inequalities. For example, simple gradient method is not necessary convergent for monotone Lipschitz variational inequalities. This motivates us to consider Extragradient-type methods that are specific methods for variational inequalities (though can be applied to minimization problems as well). Next, acceleration is not possible in general for (strongly-) monotone variational inequalities. Therefore, to get a complete study in the case of minimization, it is not sufficient to consider only variational inequalities formulation.
>
> Finally, both problem types are important and appear in various applications. Therefore, we find it crucial to consider both minimization and variational inequalities problems. Such consideration provides a wider picture and more complete study of high-probability convergence of stochastic methods.
>
> >**Assumption 1.**
>
> We thank the reviewer for the suggestion. We clarified this assumption in the revised version.

---

> ### Author Response · Authors · 2023-11-22
> **Response to Reviewer sYPF [Part 2/2]**
>
> >**It would be helpful to include a discussion after each theorem to compare the bound with existing clipping method bounds.**
>
> We had such discussions in the original version of the paper and expanded them in the revised version.
>
> >**How would the proposed algorithm perform in the centralized composite optimization problem? Would it yield better convergence bounds compared to existing bounds?**
>
> To the best of our knowledge, the results for the distributed methods proposed in our work are the only existing ones under Assumption 2 (even if we take into account the in-expectation convergence results). In the special case of $\alpha = 2$, our results match the SOTA ones from [1,2] since parallelization with linear speed-up follows for free under the bounded variance assumption (if the clipping is applied after averaging as it should be in the parallelized version of methods from [1,2] to keep the analysis from [1,2] unchanged).
>
> ---
>
> References:
>
> [1] E. Gorbunov, M. Danilova, I. Shibaev, P. Dvurechensky, and A. Gasnikov. Near-optimal high probability complexity bounds for non-smooth stochastic optimization with heavy-tailed noise. arXiv preprint arXiv:2106.05958, 2021.
>
> [2] E. Gorbunov, M. Danilova, D. Dobre, P. Dvurechenskii, A. Gasnikov, and G. Gidel. Clipped stochastic methods for variational inequalities with heavy-tailed noise. NeurIPS 2022.

---

### Official Review · Reviewer_Da4b · 2023-11-03

**Soundness:** 3 good
**Presentation:** 2 fair
**Contribution:** 2 fair
**Rating:** 5
**Confidence:** 4

**Summary:**

Problem is to derive algorithms with high probability convergence rate guarantees with heavy-tailed noise and composite and distributed optimization and variational inequality. To go around the difficulty in the composite setting, the authors suggest using a technique involving clipping the difference between the stochastic gradients and a "shift" vector that is updated at every iteration to estimate the optimal gradient. The authors also consider the accelerated version of this method under both convexity or quasi-strong convexity assumptions. Similar results are also derived for variational inequality with assumptions such as star-cocoercivity, quasi strong monotonicity or monotonicity.

**Strengths:**

The paper is quite comprehensive, covering different problem templates such as convex or quasi strong convex, composite or distributed, moreover variational inequality with different assumptions. Heavy tailed noise and high probability guarantees are important avenues for research with potential impact in practice also due to the usage of clipped algorithms which also find use in practice. Since the research area is quite active, the table helps put the contributions in context. The developments of the ideas are well-motivated. For example, the authors show both naive or non-implementable approaches to highlight the difficulties and also point out the new techniques.

**Weaknesses:**

— Even though the paper takes the time to introduce the new ideas by the naive approach or non-implementable approach, since the paper is trying to do many things at the same time (distributed, convex, quasi strong convex), things get too complicated too quickly and makes it difficult for the reader to both understand and/or appreciate the novelties, new techniques or new results. For example, Theorem 2.2 is really difficult to unpack. Why not present a simpler version of the result in the main text and maybe also the crux of the analysis (which, from the presentation of the authors, is clipping the difference between shifts and the gradients) and how this helps. Right now, even though I see that the authors say including shifts and clipping differences is important, I cannot appreciate how this helps things in the analysis. I think it is better to have weaker/simpler results in the main text but convey the idea (both high level and technical) to the reader rather than trying to pack too many things into a small space.

Even though I think that the main idea for extension to composite case is well-explained (only on a high level, as I described above), what is the reason for this paper being able to extend to distributed setup but not the previous works? Since this is one of the main claims and focus of the paper, this is an important aspect.

For example, an important aspect of the analysis seems to be that the authors can provide a boundedness results for the iterates to go around the need to assume bounded domains. However, I cannot see explanations about this technique in the main paper which would help the reader get something out of the paper. Of course, this is developed in the earlier literature, but it is also used in this paper, so why not share this important point with the reader?

— Comparisons are quite confusing, especially with the work of Nguyen et al. 2023. Looking at the first section of the table, one can see that 3rd and 5th lines differ on following: 1. Nguyen et al. 2023 does not have distributed results (not clear if this is a fundamental drawback, i.e., why can't the results of Nguyen et al 2023 be extended to distributed setup in a straightforward way?) Second is the difference of R^2 vs V in the bounds. Even though it is difficult to unpack here too, after reading 9-line table description (a bit too long and probably would be better to have this clarification in the main text), I see that $R^2 \leq V$ since $V$ also includes terms with the norms of stochastic gradients at the solution. Then, this means that in terms of the constants, actually Nguyen et al. 2023 is better, is this correct? If so, it is better to state this in the paper for helping the reader understand both improvements in the paper and also the drawbacks compared to other works.

Also, the previous work by Nguyen et al. 2023 does not assume unique solution whereas this work does, why do we need this? The authors say in the footnote of page 3 that it is "for simplicity" but can we avoid it? What happens when we remove it? What does it affect in the analysis? Again, if this is a drawback compared to Nguyen et al. 2023 that seems more restrictive, this needs to be stated.

Another point is that for constrained setting, authors mention a couple of times the drawback of Nguyen et al. 2023 in requiring the gradient at the solution to be $0$, which is for sure an important drawback for a constrained result, on the other hand, depending on M and R, when Nguyen et al result applies, it is better, is this correct? Moreover, Nguyen et al. 2023 has the additional requirement for $\nabla f(x^*)=0$ but they don't seem to have the assumption of unique solution that the current submission has, how to compare these two (in the mere convex case) since they are both rather restrictive?

Also, skimming Nguyen et al. 2023, that paper seems to suggest that the approach also taken in this submission uses a union bound and resulting in a worse dependence in terms of the logarithmic terms. Since the authors do not put the dependence on the probability parameter in Theorem 2.2, it is difficult to compare, can the authors clarify? In my understanding, this drawback is appearing in the beginning of page 24, with an additional factor of $T$, is this the case? Can you please compare the dependence with log terms involving probability parameter compared to Nguyen et al. 2023?

— Organizations of the proofs are not very helpful for the reader. For example, Theorem D.1 has a 9 page proof and Theorem C.1 5-6 page proof, it is really difficult to follow for a reader. Please consider splitting to intermediate results and explaining the high level structure to help your reader.

— The paper is 115 pages. This, in a conference review cycle which gives to reviewers less than 20 days for 4-6 papers is a bit too much. Hence, I am not sure how suitable this paper is for a conference since there is almost no chance for reviewers to be able to do justice to this paper by checking the arguments. I can see that after page 55, the proofs for the variational inequality part starts and it is so long for the results that constitute 1 page in the main text making the presentation of the paper even worse. I appreciate that the authors are trying to avoid splitting too thin but maybe in this case, different papers are justified if the extension to variational inequality is significant enough. If not, the authors might decide to include this result only as a footnote and keep an "arXiv version" with full details. Submitting a version to a conference that can be reviewed is I think much better. This way, the authors will have enough space both in the main text and supplementary of the conference version to convey the main important ideas.


------------------------------------ After rebuttal update -------------------------------------------

I read the other reviews, and the authors' rebuttal for all reviews. Thanks the authors for your rebuttal and explanations. Good to see you can remove the assumption of unique solution. I see the nontrivial ideas such as clipping the gradient differences and the improvements it gives in terms of dimension independence and generality. As I wrote already in my review, I agree with authors that obtaining high probability guarantees is important and often more challenging than expectation guarantees. These are positive points about the paper. However, there are still some issues in the paper that stops me from recommending acceptance and/or championing this paper. For example, the paper talks about the importance of not having bounded domain assumptions, but Theorem 2.3 requires knowledge of $R$ such that $R \geq \| x^0 - x^*\|$. Without bounded domains, knowledge of $R$ requires knowing solution norm (I don't see how you can get the correct $R$ otherwise since one needs an upper bound on $\| x^0 - x^*\|$), which is not very realistic. So, the other way around without knowing $\|x^*\|$ is bounded domains, unless I am missing something. Even if I am missing something here, the authors should have justified these parameter choices well in the paper and/or rebuttal. Theorem 2.3 also requires knowing $\zeta_*$ depending on $\|\nabla f_i(x^*)\|^2$. This is also not very realistic. These aspects should have been explained better in the paper since these are not usual or desirable dependences that one has in optimization: I am sure the authors also know this.

Another aspect is that the authors mention the "sums in eq. (96) and eq. (120)" to justify their novelties, which I am not being against. However, their justifications in the rebuttal or in the appendix are not very convincing. Rebuttal says "these sums do not appear for non-distributed" which by itself is not sufficient since if these sums are similar to others, they can be bounded in a similar way. Appendix page 33 says upper bound for (6) "requires a more refined analysis" but there is no explanation what the difference compared to other sums is or what is the extra "refined" analysis is. I am definitely not trying to refute authors' claims about the novelty, I agree with them that this paper does have novel and new ideas. I am just stating that when you argue the novelties in the analysis, it is better to be more explicit.  I also emphasize that I totally agree with the authors' statement that "even a proper combination of existing techniques is not always obvious beforehand"

Another thing is that my suggestion was not to put all the VIP sections to the appendix. Instead, my suggestion was to make a reasonable length "conference version" of your work and if you think VIP sections are too similar to the minimization to justify another paper, you can have an arXiv version which is longer. Submitting a 119 page paper to a conference with short review cycle does not pose very realistic expectations from the review process.

Even though I am convinced that there is value in authors' ideas for the distributed extensions, the abovementioned issues keep me borderline and stop me from championing this paper unfortunately.

Last point (independent of my decision) is that I recommend the authors to post their response before the last day of the discussion deadline next time to give the reviewers more time to be able to interact with them (of course I understand this might not always be possible depending on the schedule of the authors, but just a suggestion). It is not very realistic to expect from the reviewers to be able to immediately read the rebuttal and reflect on the new information and respond to the authors. A better solution to this could be to separate the rebuttal-writing period and discussion period but given that this conference is not arranged that way, I would have preferred if the authors would adjust and  give the reviewers 1-2 days (at least) before the end of the discussion deadline if they wanted a more interactive discussion.

**Questions:**

Please see the questions above. I also summarize some of them below:

— What is the main reason making the extension of Nguyen et al. 2023 to distributed case non-straitghtforwrard and what is the main tool used in this submission for being able to handling distributed setup?

— What is the reason for assuming unique solution? Can it be avoided?

— With regards to Nguyen et al. 2023 how to compare different assumptions? This paper assumes unique solution and Nguyen et al 2023 assumes $\nabla f(x^*)=0$ in constrained case. Which one is more restrictive?

— Is it true that in terms of constants, Nguyen et al. 2023 has a better rate in the non-distributed setup?

---

> ### Author Response · Authors · 2023-11-22
> **Response to Reviewer Da4b [Part 1/3]**
>
> We want to thank the reviewer once again for a very detailed review.
>
> >**On non-triviality of the extension to the distributed case.**
>
> We thank the reviewer for an excellent question! To illustrate the difficulty of the extension to the distributed case and how we circumvent this, let us consider two straightforward extensions of clipped-SGD to the distributed setup. The first one has the following update rule:
>
> $$x^{k+1} = x^k - \gamma \text{clip}\left( \frac{1}{n}\sum\limits_{i=1}^n \nabla f_{\xi_i^k}(x^k), \lambda_k\right).$$
>
> Then, one can apply the existing results for clipped-SGD (e.g., from [1] or [2]) under bounded $\alpha$-th moment assumption, but in this case, one needs to estimate constant $\sigma$ for $\frac{1}{n}\sum\limits_{i=1}^n \nabla f_{\xi_i^k}(x^k)$. If individual stochastic gradients $\nabla f_{\xi_i^k}(x^k)$ satisfy Assumption 1 with parameter $\sigma$, then $\frac{1}{n}\sum\limits_{i=1}^n \nabla f_{\xi_i^k}(x^k)$ also satisfies Assumption 1 with parameter $\sigma$. However, if we use this estimate, we will not get $\frac{1}{n}$ dependence of the statistical term in the complexity bound. Instead, one can apply Lemma 7 from [3], implying that $\frac{1}{n}\sum\limits_{i=1}^n \nabla f_{\xi_i^k}(x^k)$ satisfies Assumption 1 with parameter $\frac{2^{2-\alpha}d^{\frac{1}{\alpha}-\frac{1}{2}}\sigma}{n^{\frac{\alpha-1}{\alpha}}}$, where $d$ is the dimension of the problem. This approach leads to $\frac{1}{n}$ dependence on $n$ of the statistical term in the complexity, but it gives an additional factor of $d^{\frac{1}{\alpha-1} - \frac{\alpha}{2(\alpha-1)}}$ in the same term, i.e., the final bound becomes dimension-dependent, which is problematic for large-scale problems. Moreover, even for small dimensional heavy-tailed problems with $d = 1000$ and $\alpha = \frac{7}{6}$, this factor is $1000^{6 - \frac{7}{3}} > 1000^3 = 10^9$, which is a huge number.
>
> The second straightforward extension of clipped-SGD to the distributed case has the following update rule:
>
> $$x^{k+1} = x^k - \gamma \frac{1}{n}\sum\limits_{i=1}^n \text{clip}\left(\nabla f_{\xi_i^k}(x^k), \lambda_k\right).$$
>
> In contrast to the first option, this version uses clipping and averaging in different order: it applies clipping first to the stochastic gradients on the workers and then averages the results from the workers. This method is a special case of our DProx-clipped-SGD-shift for $\Psi \equiv 0$ (unconstrained minimization), $\nu = 0$ and $h_1^0 = \ldots = h_n^0 = 0$ for all workers (no shifts). In particular, as we explain in the paper, for convex problems shifts are not needed for clipped-SGD to get our results. However, our proof significantly differs from the ones from [1, 2] even in this case: due to the distributed nature of the problem, we need to estimate new sums that do not have analogs in the non-distributed case. In particular, sum 6 from inequality (96) is completely new and requires a separate induction to estimate it (see the part of the proof after inequality (120)). This causes some extra technical challenges that do not appear in the non-distributed setup. We also point out that to achieve our results, we increased the clipping level $n$ times (see the motivation for this change in Appendix B after Lemma B.2). This change in the clipping level was not obvious beforehand. Moreover, this change also complicates the estimation of the sum 6 from inequality (96): if we apply Bernstein’s inequality directly to this sum, we will get that the summands are bounded by $\frac{8\gamma^2\lambda_l^2}{n} \sim n$ that leads to $n$-times worse bound than needed to achieve the desired result.
>
> Moreover, since in the non-distributed unconstrained case, the existing high-probability results for clipped-SGD under (quasi-)strong convexity assumption use a decreasing clipping level, it is natural to use a decreasing clipping level for the distributed case as well under the same assumption. However, this leads to another problem: when $\lambda$ is small enough, the method described above cannot converge with constant stepsize even without any stochasticity. Indeed, since $\frac{1}{n}\sum_{i=1}^n \text{clip}(\nabla f_i(x^\ast), \lambda) \neq 0$ in general for small enough $\lambda$, the method does not have a fixed-point property, i.e., $x^\ast \neq x^\ast - \gamma \frac{1}{n}\sum_{i=1}^n \text{clip}(\nabla f_i(x^\ast), \lambda)$ in general. This necessitates decreasing stepsize $\gamma$ to achieve a predefined accuracy of the solution and leads to worse rates of convergence. The same issue is present in the case of accelerated methods since known results for clipped accelerated methods also use decreasing clipping levels [1].
>
> We circumvent these issues by using properly constructed shifts $h_i^k$ that aim to learn $\nabla f_i(x^k)$. This idea was not presented in previous works on high-probability analysis and it is the key component of our methods.
>
> We incorporated the above remarks into the paper.

---

> ### Author Response · Authors · 2023-11-22
> **Response to Reviewer Da4b [Part 2/3]**
>
> >**Readability of the results and, in particular, Theorem 2.2.**
>
> We thank the reviewer for the suggestion. To improve readability, we have split Theorem 2.2 into two parts (quasi-strongly convex case and convex case). These results are simplifications of the full statements provided in the appendix. We also show these results in Table 1 that can be considered as a simplified summary of the results obtained for minimization problems.
> We also added more technical details regarding the role of shifts to the main part of the paper (the intuitive explanation was also expanded). We hope that in the revised version of our paper, Theorem 2.2 (and other theorems) and technical details are clearer than before. If Reviewer Da4b has any further suggestions about improving the readability, we will be happy to incorporate them into the paper.
>
> >**On the assumptions and boundedness of the iterates.**
>
> We have added to the revised version of the paper more explanations regarding our assumptions and more explanations on why it is sufficient to make our main assumtions only on a compact set. We emphasize that we do not need to assume the boundedness of the domain of the original problem.
>
> >**Different constants to the ones from the results by Nguen et al. (2023).**
>
> After an additional examination of the results from [2], we found out that the correct dependence on $R$ is a bit more involved than we presented in Table 1. In the revised version, we have corrected it. The second term in the complexity bound for Clipped-SMD from [2] remains unchanged, while the first term depends on $\hat R^2 = R\left(2R + \|\hat x - x^\ast\| + \frac{\eta \sigma + \|\hat g\|}{L}\right)$, where $\hat x$ is some point from the domain, $\eta > 0$, and $\hat g$ is such that $\mathbb{P}\lbrace \| \hat g - \nabla f(\hat x) \| > \eta \sigma \rbrace \leq \epsilon$ for some $\epsilon \in (0,1)$, i.e., $\hat g$ is an estimate of $\nabla f(\hat x)$. The authors of [2] do not provide a particular suggestion of how to choose $\hat x$ in the general case. One reasonable choice is $\hat x = x^0$. Then, according to [4], it is sufficient to take $\mathcal{O}(\log \frac{1}{\epsilon})$ samples and compute a geometric median of them. In this case, the constants in the bound from [2] become equivalent to our constants (up to logarithmic factors) if $\gamma$ is chosen smaller than $\frac{\|x^0 - x^\ast\|}{\|\nabla f(x^\ast)\|}$. We have clarified this part in the revised version.
>
> >**On the uniqueness of the solution.**
>
> We thank the reviewer for very good questions. In fact, we use the uniqueness of the solution only to make Assumptions~2, 4, 7, and 8 easier. We reformulated these assumptions to avoid the uniqueness of the solution in the revised version of the paper. Our proofs and results remain unchanged, and, in the definitions of constants $V$ and $M$ for DProx-clipped-SGD-shift and DProx-clipped-SSTM-shift, one can use the closest solution $x^\ast$ to the starting point $x^0$. We also emphasize that the authors of [2] do not consider quasi-strong convexity/quasi-convexity assumptions or refined smoothness assumptions like Assumption 2. Under standard smoothness and (strong) convexity assumptions, our proofs also do not require the uniqueness of the solution.
>
> >**Zero-gradient in the accelerated result from Nguen et al. (2023) and comparison with our results.**
>
> When $\nabla f(x^\ast) = 0$ and $n = 1$, constants $M$ and $R$ are identical. Therefore, our result recovers the one from [2] as a special case. Next, as we explain above, we do not require the uniqueness of the solution under standard convexity and smoothness and also do not require the uniqueness of the solution at all in the revised version of our submission. This means that our results for the accelerated method are strictly more general than the corresponding ones from [2].

---

> ### Author Response · Authors · 2023-11-22
> **Response to Reviewer Da4b [Part 3/3]**
>
> >**On the logarithmic factors.**
>
> Since in this paper, we do not focus on improving the dependencies on $\varepsilon$ and $\delta$ under the logarithms, we have slightly worse logarithmic factors than in the results from [2], where a very particular case of non-distributed convex optimization is considered. In the appendices of the *original submission*, we provide the complete formulations of all results (including explicit dependencies on the logarithmic factors). In particular, in the case of DProx-clipped-SGD-shift, for convex problems the logarithmic factor in the complexity bound is given in (83) and equals $\log\left(\frac{1}{\beta}\left(\frac{\sigma \sqrt{V}}{\varepsilon}\right)^{\frac{\alpha}{\alpha-1}}\right)$ in front of the leading term, while the corresponding factor in complexity bound from [2] equals $\log\frac{1}{\beta}$. In particular, for $\alpha = \frac{3}{2}$, $\varepsilon = 10^{-6}$, $\delta = 10^{-6}$, $\sigma = 10^3$, $V = 10^6$ our logarithmic factor is $\approx 97$, while the one from [2] is $\approx 14$, i.e., it is just about $7$ times less, which is a minor difference, though the improvement of logarithmic factors is an important theoretical contribution (requiring some important changes in the analysis). Moreover, the result of [2] applies only to one of the many settings where our results hold.
>
> >**On the structure of the proofs.**
>
> We have added a section to the main text that should help the readers to understand the proofs and navigate through them better.
>
> >**Paper length and results for variational inequalities.**
>
> Following the reviewer’s request, we have improved the readability of the paper as follows: we moved all the results and discussions on the methods for solving variational inequalities to the appendix. We use the extra space in the main part to better explain our proofs and techniques.
>
> ---
>
> References:
>
> [1] Sadiev et al. High-Probability Bounds for Stochastic Optimization and Variational Inequalities: the Case of Unbounded Variance. ICML 2023
>
> [2] Nguen et al. Improved convergence in high probability of clipped gradient methods with heavy tails. arXiv preprint arXiv:2304.01119. 2023.
>
> [3] Wang et al. Convergence rates of stochastic gradient descent under infinite noise variance. NeurIPS 2021.
>
> [4] Minsker, S. (2015). Geometric median and robust estimation in Banach spaces. Bernoulli, 21(4), 2308.

---

> > ### Comment · Reviewer_Da4b · 2023-11-23
> > **Acknowledgement**
> >
> > I am writing to acknowledge that I saw your rebuttal. However as you also know, last day of the discussion period (that your rebuttal was posted on) is a very busy one. Also, both your paper, my review and your rebuttal are on the longer side, therefore I could not get the time to process the information to provide a follow-up comment for you on time during the discussion period. I am writing this message to assure that I will read your rebuttal in detail, think about it, adjust my score if necessary and write a follow-up message accordingly.

---

> > > ### Author Response · Authors · 2023-11-23
> > > **Response to the comment Reviewer Da4b**
> > >
> > > We thank the reviewer for the comment and for letting us know that the reviewer will check our rebuttal.

---

### Official Review · Reviewer_LBP8 · 2023-11-05

**Soundness:** 4 excellent
**Presentation:** 4 excellent
**Contribution:** 2 fair
**Rating:** 5
**Confidence:** 3

**Summary:**

This paper studies the problem of composite and distributed stochastic optimization and variational inequalities. The author aims at giving high probability convergence bounds for the two problems.
The author considers the setting that the noise has bounded central $\alpha$-th moment and uses the method of gradient clipping for this condition of heavy-tailed noise.

**Strengths:**

Strengths:
(1) This paper is written very well and is well-organized.
(2) Two standard problems (distributed composite stochastic optimization and distributed composite VIPs) are studied and many convergence bounds are given. In the first case, they develop two stochastic methods for composite minimization problems – Proximal Clipped SGD with shifts (Prox-clipped-SGD-shift) and Proximal Clipped Similar Triangles Method with shifts (Prox-clipped-SSTM-shift). Instead of clipping stochastic gradients, these methods clip the difference between the stochastic gradient and the shifts that are updated on the fly. In the second case, they also apply the proposed trick to the methods for variational inequalities.

**Weaknesses:**

(1) High probability bounds do not bring some technical challenges than the in-expectation ones. After establishing the martingale difference sequences, the Freedman’s inequality is enough to give a high probability bound. In this line, there are numerous papers in recent years.
(2) The technical contribution is limited. The author uses gradient clipping to handle the $\alpha$-th moment, which is frequently used in recent papers. The $\alpha$-th moment condition is also widely studied. Combining the technique in gradient clipping and the technique in distributed composite minimization/distributed composite VIPs, most of the results in this paper are easily obtained.
(3) The paper is too long and more suitable for a review in the journal.
Refer to the following papers:
E. Gorbunov, M. Danilova, and A. Gasnikov. Stochastic optimization with heavy-tailed noise via accelerated gradient clipping. Advances in Neural Information Processing Systems, 33:15042– 344 15053, 2020a.
E. Gorbunov, F. Hanzely, and P. Richtárik. A unified theory of sgd: Variance reduction, sampling, quantization and coordinate descent. In International Conference on Artificial Intelligence and Statistics, pages 680–690. PMLR, 2020b.
E. Gorbunov, M. Danilova, I. Shibaev, P. Dvurechensky, and A. Gasnikov. Near-optimal high 349 probability complexity bounds for non-smooth stochastic optimization with heavy-tailed noise. arXiv preprint arXiv:2106.05958, 2021.
E. Gorbunov, M. Danilova, D. Dobre, P. Dvurechenskii, A. Gasnikov, and G. Gidel. Clipped stochastic methods for variational inequalities with heavy-tailed noise. Advances in Neural Information Processing Systems, 35:31319–31332, 2022.
A. Beznosikov, E. Gorbunov, H. Berard, and N. Loizou. Stochastic gradient descent-ascent: Unified theory and new efficient methods. pages 172–235, 2023.
K. Mishchenko, E. Gorbunov, M. Takác, and P. Richtárik. Distributed learning with compressed gradient differences. arXiv preprint arXiv:1901.09269, 2019.
A. Sadiev, M. Danilova, E. Gorbunov, S. Horváth, G. Gidel, P. Dvurechensky, A. Gasnikov, and P. Richtárik. High-probability bounds for stochastic optimization and variational inequalities: the case of unbounded variance. arXiv preprint arXiv:2302.00999, 2023.

**Questions:**

What are the technical innovations of this paper compared to existing works? This paper seems to be a simple combination of existing techniques in related works.

---

> ### Author Response · Authors · 2023-11-16
> **Response to Reviewer LBP8**
>
> Unfortunately, the submitted review is not about our paper. In particular, the reviewer claims
>
> > The article introduces a class of soft clipping schemes
>
> > The numerical experiments in this paper are beautiful which shows that soft-clipping algorithms may offer regularization benefits in cases where other algorithms tend to overfit, encouraging the use of soft-clipping algorithms and further research in the field
>
> > Especially for the symbols $\w_k(w)$ without interpretation in Corollary 2, it’s hard for readers to understand and what insight the corollary hopes to provide.
>
> > Is it reasonable to assume that $\sum_{k=1}^\infty \alpha_k^2 < \infty$ and $\sum_{k=1}^\infty \alpha_k = \infty$ as stated in Theorem 1?
>
> In fact, we do not introduce soft clipping schemes and do not study them in our work. In contrast, we use the standard gradient clipping and introduce new algorithms for composite and distributed problems and derive high-probability convergence results for them. Next, our paper does not have numerical experiments. We also do not have Corollary 2, symbols $\w_k(w)$, Theorem 1, and conditions $\sum_{k=1}^\infty \alpha_k^2 < \infty$ and $\sum_{k=1}^\infty \alpha_k = \infty$ mentioned in the review.
>
> **All of these aspects show that the review is written for a different paper and is unrelated to our work.**

---

> ### Author Response · Authors · 2023-11-22
> **Response to the completely new review by Reviewer LB8P [Part 1/2]**
>
> Although the reviewer silently replaced their review just two days before the deadline, we addressed all the comments by this reviewer. **Our general response and responses below justify that Reviewer LB8P provided a review of extremely low quality.**
>
> >**High probability bounds do not bring some technical challenges than the in-expectation ones. After establishing the martingale difference sequences, the Freedman’s inequality is enough to give a high probability bound. In this line, there are numerous papers in recent years.**
>
> We politely disagree with the reviewer. We have multiple solid arguments justifying that deriving high-probability bounds is, in general, more challenging than obtaining in-expectation ones.
>
> First of all, a large body of the optimization community has acknowledged the importance, interest, and difficulty to get high-probability convergence guarantees [1,2,3]. For instance, we can explicitly quote [1]:
>
> > *high probability guarantees are significantly harder to obtain and they hold in more limited settings with stronger assumptions on the problem settings and the stochastic noise distribution.*
>
> to illustrate our point that **high-probability convergence guarantees bring new challenges in contrast to in expectation proofs**. We now develop the reasons behind this statement.
>
> **Argument 1: proofs in expectation are usually simpler.** To prove in-expectation results, it is not needed to recursively upper-bound the distance to the solution/norm of the gradient of the objective: one can just take expectations and get rid of sums of inner product terms using unbiasedness (e.g., in the case of SGD / Accelerated SGD under bounded variance assumption, see [9]). Conversely, to apply Bernstein-Freedman, one needs either to assume that the problem is defined on a compact, which does not allow to consider an important class of unconstrained minimization problems, or to recursively bound the distance to the solution with high probability.
>
> **Argument 2: necessary additional assumption for convergence with high probability for standard methods.** In many previous works on SGD, the high-probability analysis requires an additional assumption in comparison to in-expectation analysis: sub-Gaussian noise assumption [5,6]. This is not an artifact of the analysis – **it is, in general, impossible to show good high-probability convergence for SGD; see Theorem 2.1 in [4]**
>
> **Argument 3: the large gap in time between in-expectation results and high-probability ones is evidence of non-triviality.** If high-probability analysis was that trivial to obtain, it would be done much earlier since Assumption 1.1 was known 40 years ago – in 1983, the first in-expectation results were already obtained in [7]. Next, in the non-convex case, the first in-expectation analysis appeared in December 2019 (the first version of [7]), but the first high-probability analysis appeared only in June 2021 [3] (then appeared in NeurIPS 2021; this fact indicates the importance of the problem for the community) and this analysis relies on the additional assumption and is also given for much more sophisticated method than clipped-SGD. If the extension was trivial, it would not require 1.5 years and many changes in the algorithm.

---

> ### Author Response · Authors · 2023-11-22
> **Response to the completely new review by Reviewer LB8P [Part 2/2]**
>
> >**The technical contribution is limited. The author uses gradient clipping to handle the $\alpha$-th moment, which is frequently used in recent papers. The $\alpha$-th moment condition is also widely studied. Combining the technique in gradient clipping and the technique in distributed composite minimization/distributed composite VIPs, most of the results in this paper are easily obtained.**
>
> >**What are the technical innovations of this paper compared to existing works? This paper seems to be a simple combination of existing techniques in related works**
>
> We politely disagree with the reviewer. Our contributions go far from just combining known techniques, though even a proper combination of existing techniques is not always obvious beforehand while the reviewer is trying to claim the opposite.
>
>
> Before we elaborate on the technical difficulties in the extensions, we would like to discuss the algorithms’ design. Having a goal in mind of proper extension of the results from [11] to the composite and distributed problems, we were trying to use “Occam’s razor” principle: we did not want to change the existing technique without the necessity of doing it. In particular, we did not want to change the existing algorithms just for the sake of developing new methods and having different analyses. For example, in our opinion, Prox-clipped-SGD-star has a minor but extremely important difference from just Prox-clipped-SGD. Due to this visually tiny algorithmic difference, the analysis of Prox-clipped-SGD-star in the *composite* case is almost the same as the analysis of clipped-SGD in the *unconstrained (non-composite)* case. We believe that this idea is non-trivial beforehand. Our claim is justified by the fact the previous works (like [12]) also study constrained problems but do not achieve similar results in the strongly convex case and do not get the acceleration without an extra restrictive assumption that $\nabla f(x^\ast) = 0$.
>
> Next, to get practical methods, we made an extra step forward and replaced $\nabla f(x^\ast)$ with a learnable shift $h^k$ and also generalized the method to the distributed setup. This is the moment when noticeable technical difficulties arise. First of all, we introduce a Lyapunov function to control not only the proximity of $x^k$ to $x^\ast$ but also the quality of shifts (see line 240). In the (quasi-) strongly (monotone) convex case, this leads to the need to estimate $\|\| h_i^{k+1} - h_i^\ast \|\|^2$ and also to apply Lemma B.2 we have to estimate $\|\| F_i(x^k) - h_i^k \|\|$ and connect it to the Lyapunov function and the clipping level (see inequality (200)). This leads to the non-trivial interplay between the coefficient in the second term in the Lyapunov function, the number of workers $n$, and the clipping level $\lambda_t$ that we also need to change (increase $n$ times) compared to the non-distributed case, see the discussion after Lemma B.3. Moreover, since the clipping level is $n$ times larger, we need to be very careful in the analysis of the sums like the ones appearing in (96), especially with sum (6) from inequality (96). We emphasize that they do not appear in the non-distributed case.
>
> Nevertheless, we circumvented all of these difficulties and managed to obtain a linear speed-up in the distributed case, and in the composite non-distributed case, we managed to obtain direct analogs of the results from [11] obtained for the unconstrained problems without any sacrifice in the complexities. That is, for the first time in the literature, we achieve $1/n$ dependence in the statistical term (the one depending on $\sigma$) of our complexity bounds under bounded central $\alpha$-th moment assumption. Let us elaborate on the non-triviality of achieving $1/n$ factor. When $\alpha = 2$, the result is standard since the variance of the average of independent random variables scales as $1/n$, and one can simply replace $\sigma^2$ with $\sigma^2/n$. This is a key property of mini-batching in stochastic optimization. However, there are no *dimension-independent* analogs of this result in the literature for the case of $\alpha < 2$. The only result we are aware of is given in Lemma 7 from [13], and it has an extra factor of $d^{1-\frac{\alpha}{2}}$, where $d$ is the dimension of the problem. For huge-scale problems, this factor can be large even for $\alpha \approx 3/2$. In contrast, our rates are dimension-independent.

---

> ### Author Response · Authors · 2023-11-22
> **References**
>
> [1] Liu, Zijian, et al. "High probability convergence of stochastic gradient methods." International Conference on Machine Learning. PMLR, 2023.
>
> [2] Li, Xiaoyu, and Francesco Orabona. "A high probability analysis of adaptive sgd with momentum." arXiv preprint arXiv:2007.14294 (2020).
>
> [3] Cutkosky, Ashok, and Harsh Mehta. "High-probability bounds for non-convex stochastic optimization with heavy tails." Advances in Neural Information Processing Systems 34 (2021)
>
> [4] Sadiev, Abdurakhmon, et al. "High-Probability Bounds for Stochastic Optimization and Variational Inequalities: the Case of Unbounded Variance." (2023).
>
> [5] Nemirovski, A., Juditsky, A., Lan, G., & Shapiro, A. (2009). Robust stochastic approximation approach to stochastic programming. SIAM Journal on optimization, 19(4), 1574-1609.
>
> [6] Ghadimi, S., & Lan, G. (2012). Optimal stochastic approximation algorithms for strongly convex stochastic composite optimization i: A generic algorithmic framework. SIAM Journal on Optimization, 22(4), 1469-1492.
>
> [7] Nemirovskij, A. S., & Yudin, D. B. (1983). Problem complexity and method efficiency in optimization.
>
> [8] Zhang, J., Karimireddy, S. P., Veit, A., Kim, S., Reddi, S., Kumar, S., & Sra, S. (2020). Why are adaptive methods good for attention models? Advances in Neural Information Processing Systems, 33, 15383-15393.
>
> [9] S. Ghadimi and G. Lan. Optimal stochastic approximation algorithms for strongly convex stochastic composite optimization i: A generic algorithmic framework. SIAM Journal on Optimization, 22(4), 1469-1492. 2012.
>
> [10]  S. Ghadimi and G. Lan. Optimal stochastic approximation algorithms for strongly convex stochastic composite optimization, II: shrinking procedures and optimal algorithms. SIAM Journal on Optimization, 23(4), 2061-2089. 2013.
>
> [11] Sadiev et al. High-Probability Bounds for Stochastic Optimization and Variational Inequalities: the Case of Unbounded Variance. ICML 2023.
>
> [12] Nguen et al. Improved convergence in high probability of clipped gradient methods with heavy tails. arXiv:2302.05437. 2023.
>
> [13] Wang et al. Convergence rates of stochastic gradient descent under infinite noise variance. NeurIPS 2021.

---

### Author Response · Authors · 2023-11-22
**General response to Reviewers**

We thank the reviewers for their feedback and time dedicated to the review of our paper. We appreciate that Reviewer Da4b found our work quite comprehensive, the topic we consider to be important and having potential practical impact, and the developments of the ideas to be well-motivated. We are grateful to Reviewer sYPF, who found our work to be technically solid and well-written.

The reviewers also raised several questions and concerns that we addressed in detail in our responses to each reviewer. Based on the reviewers’ comments, we incorporated the necessary changes into our paper and highlighted them using green color in the revised version of the paper. In particular, following the suggestion of Reviewer Da4b, we have moved the results on variational inequalities to appendix, added more discussions about the results for minimization, and provided a description of the general structure of the proofs. We believe this should improve the readability of our paper. Moreover, following the request by Reviewer sYPF, we added numerical experiments to our paper.

**We also would like to draw the attention of all Reviewers, Area Chairs, Senior Area Chairs, and Program Chairs that Reviewer LBP8 originally submitted the review for a different paper.** After we pointed out this issue in our response from November 16, this reviewer modified their review without any notification: one can see that the last version is from November 20 – just two days before the deadline for the rebuttal and discussion. We believe that this is inappropriate, dishonest behavior that shows complete disrespect to the authors and to the conference itself by Reviewer LBP8.

Moreover, the updated review by Reviewer LBP8 is of a very low quality. The reviewer makes very subjective and vague claims.

First, the reviewer claims that high-probability results are easy to obtain if in-expectation guarantees are known. As we explain in our response and in Section 3 of the revised paper, this is not true, and we believe the experts in stochastic optimization will agree with our arguments (we even quote some of them in our response). In our opinion, challenging the relevance of an entire line of research with results published in top-tier conferences without any precise argument is inappropriate for a scientific review.

Next, the reviewer claims that we simply combine the existing techniques and claims that “most of the results in the paper are easily obtained” via this combination. This a subjective, unjustified claim that deeply offenses our efforts. The reviewer essentially claims that our results are easy to obtain because the bounded $\alpha$-th moment condition, gradient clipping, and “the technique in distributed composite minimization/distributed composite VIPs” are known in the literature, and one just needs to combine them. **This is a very vague, non-scientific, and unjustified claim (we believe this violates the reviewers’ guidelines)**. In our response and our paper, we discuss multiple technical difficulties we aimed at when obtaining our results. Moreover, we believe that our idea is not as obvious beforehand as the reviewer is trying to claim (see our response to Reviewers Da4b and LBP8 and also pages 5-6 in our work).

---

### Meta-Review · Area_Chair_3EsV · 2023-12-06

**Metareview:**

The paper gives a comprehensive studies of high probability convergence for two standard settings, composite convex optimization and variational inequalities. The paper allows for heavy tailed noise, which necessitates clipping algorithms. The paper develops new results for several settings such as: 1) accelerated gradient descent when the constraint has a large (potentially unbounded) diameter whereas the previous results do not apply; 2) distributed optimization with high probability whereas previous works are only for the centralized setting or without heavy tailed noise.

On the other hand, the comprehensive results in the paper leads to an extremely lengthy manuscript. The reviewers feel that the key innovations are not communicated clearly enough to the reader, making it hard to know where the new ideas are and where the steps are a similar version of other results in previous work or in this same paper. The new more general result also has worse logarithmic factors than previous works. Last but not least, some assumptions on the results limit the scope of the novel settings. For example, the new clipping parameter depends on the distance to the optimal solution and the length of the gradient at the optimal solution, neither seems readily available in general.

**Justification For Why Not Higher Score:**

All reviewers consider the paper below the acceptance threshold. The complaints about the lengthy paper without sufficient explanation on the novel ideas and comparison with previous works are shared by several reviewers. The lack of justification for the various parameters in the clipping parameter is also an important issue.

**Justification For Why Not Lower Score:**

N/A

---

### Decision · Program_Chairs · 2024-01-16

Reject